# IN-CONTEXT ALGORITHM EMULATION IN FIXED-WEIGHT TRANSFORMERS

**Jerry Yao-Chieh Hu**[†‡*]  **Hude Liu**[*]  **Jennifer Yuntong Zhang**[§*]  **Han Liu**[†]

[†]Northwestern University    [‡]Ensemble AI    [§]University of Toronto

jhu@u.northwestern.edu, hudeliu0208@gmail.com,
jenniferyt.zhang@mail.utoronto.ca, hanliu@northwestern.edu

## ABSTRACT

We prove that a minimal Transformer with frozen weights emulates a broad class of algorithms by in-context prompting. We formalize two modes of in-context algorithm emulation. In the *task-specific mode*, for any continuously differentiable function $f : \mathbb{R} \to \mathbb{R}$, we construct a single-head softmax attention layer whose forward pass reproduces functions of the form $f(w^\top x - y)$ to arbitrary precision. This general template subsumes many popular machine learning algorithms (e.g., gradient descent, linear regression, ridge regression). In the *prompt-programmable mode*, we prove universality: a single fixed-weight two-layer softmax attention module emulates all algorithms from the task-specific class (i.e., each implementable by a single softmax attention) via only prompting. Our key idea is to construct prompts that encode an algorithm's parameters into token representations, creating sharp dot-product gaps that force the softmax attention to follow the intended computation. This construction requires no feed-forward layers and no parameter updates. All adaptation happens through the prompt alone. Numerical results corroborate our theory. These findings forge a direct link between in-context learning and algorithmic emulation, and offer a simple mechanism for large Transformers to serve as prompt-programmable interpreters of algorithms. They illuminate how GPT-style foundation models may swap algorithms via prompts alone, and establish a form of algorithmic universality in modern Transformer models.

## 1 INTRODUCTION

We show that a minimal Transformer architecture with frozen weights is capable of emulating a broad class of algorithms through prompt design alone. This stylized problem setting isolates the core of in-context computation and provides an analytic lens on fundamental questions in Transformer models: How do fixed-weight models execute diverse tasks from context alone? How does a prompt turn into an algorithmic procedure? How do prompt-encoded parameters and query-key routing realize task identification and stepwise execution? What minimal architectural ingredients suffice for general in-context capability? As foundation models rise to prominence in modern AI (Bommasani, 2021), these questions are central, since much of their practical utility comes from in-context learning (prompting) rather than explicit retraining (Brown et al., 2020; Liu et al., 2023). Against this backdrop, this work offers a rigorous basis for in-context *task* learning[1], supplies a simple mechanism for Transformers to act as prompt-programmable algorithm libraries, and shows how GPT-style models may swap algorithms via prompts alone, shedding light on their general-purpose capabilities.

Large Transformer models exhibit ability to adapt to a new task by conditioning on examples or instructions provided in the prompt without any gradient updates. This capability is known as In-Context Learning (ICL) (Min et al., 2022; Brown et al., 2020). Prior work on Transformer in-context

---

[*]Equal contribution. Part of the work done during JH's internship at Ensemble AI. Code is available at https://github.com/MAGICS-LAB/algo_emu. Latest version is at https://arxiv.org/abs/2508.17550.

[1]We use "task" to highlight algorithm-level adaptation (to diverse tasks), not mere pattern completion.

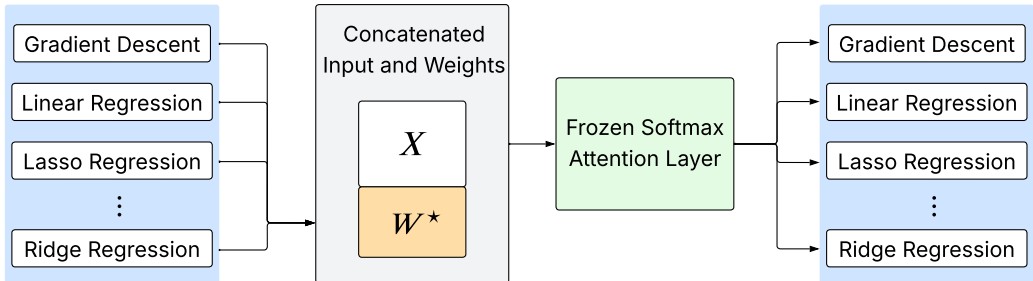

Figure 1: **Prompt-Programmable In-Context Algorithm Emulation Overview.** $X$ denotes the data input, and $W^\star$ encodes the instructions of the algorithm we aim to emulate. We show that even a 2-layer softmax attention module suffices to emulate a broad class of algorithms by changing prompt (Theorem 4.1), i.e., the $W^\star$ in the prompt. This separates *algorithm information* (in the prompt) from "model parameters" (frozen). By sending the algorithm-specific information (e.g., instructions + data) to a fixed-weight model, the *prompt* acts as the *program* and the *frozen transformer* as the *interpreter*. This makes the "weights-as-data" mechanism explicit and is the core mechanism of prompt-programmability: a minimal frozen Transformer serves as a modular interface in which swapping the prompt swaps the algorithm with no retraining.

learning falls into two strands. One trains models that learn in context for a specific function class (Garg et al., 2022; Akyürek et al., 2023; Li et al., 2023; Ahn et al., 2023; Zhang et al., 2024). The other hand-engineers Transformers to enact particular algorithms with fixed weights (Bai et al., 2023; Von Oswald et al., 2023; Wu et al., 2025). In particular, Bai et al. (2023) demonstrate that *task-specific* attention layers — attention mechanisms with weights designed for a given task — implement a variety of algorithms without gradient updates. For example, a single Transformer with fixed, task-tailored weights achieves near-optimal performance on algorithms such as least-squares/ridge/lasso regressions, and gradient descent (Bai et al., 2023; Wu et al., 2025), PCA (He et al., 2025), time series regression (Wu et al., 2026), Newton's method (Giannou et al., 2024; Fu et al., 2024), Bayesian network (Cao et al., 2025), causal learning (Nichani et al., 2024; Edelman et al., 2024). These results suggest that Transformers are capable of *in-context algorithm emulation*. Yet these approaches retrain per task or hard-wire per algorithm. They do not give a single fixed architecture that is prompt-programmable across many algorithms with explicit guarantees and minimal components.

To combat this, we advance this line of research by omitting the need for designing a new Transformer block for every algorithm. We propose a frozen Transformer architecture to emulate a library of attention-based algorithms in context without weight updates. We achieve this by embedding algorithm-specific information into input prompts. Specifically, we formalize two emulation modes, and establish explicit guarantees and constructive minimal designs for both. In the *task-specific* mode (Section 3), a dedicated attention module with fixed weights (single- or multi-head) executes one algorithm in context. In the *prompt-programmable* mode (Section 4), by contrast, a single Transformer module with fixed weights re-programs itself through different prompts to execute multiple algorithms on the fly. These constructions yield universality and minimality results for in-context algorithm emulation. Specifically, we demonstrate a minimalist model of internal algorithm swapping, where prompts serve as the context carrying algorithmic instructions.

**Contributions.** We establish a new form of in-context learning universality *for algorithm emulation*, limited to *attention-implementable algorithms*. Our contributions are three-fold:

- **Task-Specific Emulation of** $f(w^\top x - y)x$**.** A single-head, single-layer softmax attention with a linear map universally approximates functions of the form $f(w^\top x - y)x$ for any continuous $f$, with frozen weights and a suitable prompt. This general result subsumes, for example, computing per-sample gradients and performing gradient descent updates (by choosing $f$ as a loss derivative), as well as solving linear and ridge regression in one forward pass.

- **Constructive, Interpretable Prompt Design for Algorithm Emulation.** We give an explicit prompt design strategy that encodes the target task's parameters and induces large query-key margins so softmax follows the intended pattern, furnishing an interpretable, verifiable recipe for prompt-programming a fixed attention-only module.

- **A Simple Mechanism for Internal Algorithm Swapping of Transformer Models.** Changing only the prompt-encoded algorithm weights swaps the algorithm executed by the fixed attention-only module, without retraining. Theory (finite libraries) and experiments (e.g., Lasso, ridge, linear regression) confirm high-fidelity swapping. Altogether, these results shed light on the general-purpose capability of GPT-style Transformer models to select and swap internal routines via prompts (our formal proofs concern attention-only modules).

In conclusion, we show a minimalist transformer architecture serve as a general-purpose algorithm emulator in context through prompt design. Our findings contribute to a sharp theoretical foundation for viewing in-context learning as in-context algorithm emulation. They suggest that large pretrained softmax attention models (such as GPT-style Transformers) encode a library of algorithms, and swap among them based on prompts. This is achieved within a unified attention architecture and without any parameter updates. We believe this perspective opens new opportunities for understanding the emulation ability of Transformer models.

**Organization.** Section 2 presents ideas we build on. Section 3 presents illustrative examples of learning statistical models in-context with *task-specific* attention heads. Section 4 presents our main results. Appendix A presents our proof strategies. Section 5 presents numerical validations.

**Related Work.** Due to page limits, we defer related work discussions to Appendix B.

**Notations.** We denote the index set $\{1, \ldots, I\}$ by $[I]$. We use lowercase letters for vectors and uppercase letters for matrices. The vector $e_j^{(n)} \in \mathbb{R}^n$ denotes the one-hot vector with 1 in the $j$-th position and 0 elsewhere. We write $X \in \mathbb{R}^{d \times n}$ for the input sequence, where $d$ is the token dimension and $n$ is the sequence length. We denote the number of attention heads by $H$. We use $\| \cdot \|_\infty$ and $\| \cdot \|_2$ for the vector $\infty$-norm and 2-norm, respectively.

## 2 PRELIMINARIES: ATTENTION, IN-CONTEXT LEARNING AND EMULATION

**Softmax Attention.** We define a multi-layer self-attention layer with softmax activation as follows.

**Definition 2.1** (Softmax Attention Layer). For any input sequence $X \in \mathbb{R}^{d \times n}$, the multi-head attention output (with $H$ heads) is

$$\text{Attn}_m(X) = \sum_{h=1}^{H} \underbrace{W_V^{(h)} X}_{d_o \times n} \text{Softmax}(\underbrace{(W_K^{(h)} X)^\top W_Q^{(h)} X}_{n \times n}) \underbrace{W_O^{(h)}}_{n \times n_o} \in \mathbb{R}^{d_o \times n_o},$$

where $W_K^{(h)}, W_Q^{(h)} \in \mathbb{R}^{d_h \times d}$, $W_V^{(h)} \in \mathbb{R}^{d_o \times d}$, and $W_O^{(h)} \in \mathbb{R}^{n \times n_o}$ for $h \in [H]$. We use $\text{Attn}_s$ to denote *single-head* self-attention.

Following the notation of (Hu et al., 2025a), we pick non-identical dimensions for weight matrices $W_K, W_Q, W_V$ for generality of our analysis[2].

In the common $K := W_K X$, $Q := W_Q X$, $V := W_V X$ notation, a single-layer softmax attention takes a set of key vectors $K = \{k_1, \ldots, k_n\}$, value vectors $V = \{v_1, \ldots, v_n\}$, and a query vector $q$, to produce an output as a weighted sum of the value vectors. The weights on $v_i$ is $\text{Softmax}(k_i^\top q)$, emphasizing values whose keys are most similar to the query. That is, the softmax attention uses the query as a cue to retrieve the most relevant information from the values (via their keys).

**Linear Transformation Layer** $\text{Linear}(\cdot)$. Throughout this paper, we sometimes compose attention with an additional linear mapping for flexibility and simplicity of our proofs[3]. Such a linear transformation layer uses learned parameters to increase expressivity in attention-based constructions.

**Definition 2.2.** Let $Z = [z_1, \ldots, z_n] \in \mathbb{R}^{d \times n}$ be the input sequence with columns $z_i \in \mathbb{R}^d$. We use $\text{Linear} : \mathbb{R}^{d \times n} \to \mathbb{R}^{p \times m}$ (for some output length $m$) to denote *column-wise* linear affine maps. Each output column depends only on one input column, possibly with replication or an additive bias. We write $\text{Linear}$ when dimensions are clear (input/output shapes chosen to match attention).

This layer is a generic *column-wise affine* operator. It preprocesses the input to an attention mechanism or post-processes its output. For example, $\text{Attn}_s \circ \text{Linear}(Z)$ applies a per-token affine projection (optionally with replication, so $m \neq n$) before single-head attention. It subsumes the

---

[2]The right-multiplying $W_O^{(h)}$ in Definition 2.1 is nonstandard. Yet, this it not an issue. One can recover standard left-multiplying $W_O^{(h)}$ (i.e., $\text{Attn}_m(X) = \sum_{i=1}^{H} W_O^{(h)} V \text{Softmax}(K^\top V)$) by including two extra attention layers following (Hu et al., 2025a, Section G & Remark E.2). All our results still hold. We adopt this convention for simplicity of the proofs of Section 3. Our Section 4 does not rely on the presence of $W_O^{(h)}$.

[3]We remark that, one can replace all $\text{Linear}(\cdot)$ throughout this paper following (Hu et al., 2025a, Remark E.1) via extra input preprocessing. We use $\text{Linear}(\cdot)$ in this paper for simplicity of our proofs.

practical per-token linear layer as the special case $m = n$ with shared parameters and optional bias: $\text{Linear}(Z) = AZ + b\mathbb{1}_n^\top \in \mathbb{R}^{p \times n}$ with $A \in \mathbb{R}^{p \times d}$, $b \in \mathbb{R}^p$ and $\mathbb{1}_n$ the all-ones vector. In all cases, columns are processed independently (no cross-column mixing).

**In-Context Learning Setup.** In in-context learning, a fixed model (e.g., a pretrained Transformer) performs a new task without parameter updates. Formally, the model aims to approximate an unknown function $f : X \to Y$ given a few examples of $f$ in the input prompt. At inference, we provide $n$ exemplar pairs and a query $x_q$, and concatenate them into a single sequence

$$X := \begin{bmatrix} x_1 & x_2 & \cdots & x_n \\ y_1 & y_2 & \cdots & y_n \end{bmatrix} \in \mathbb{R}^{(d+1) \times n} \quad \text{and} \quad x_q \in \mathbb{R}^{d \times 1}. \tag{2.1}$$

Namely, the model receives $(X, x_q)$ as the input prompt. The goal of ICL is for the model, given input prompt $(X, x_q)$, to (i) infer $f$ from the exemplars and (ii) apply it to $x_q$ to predict $y_q = f(x_q)$. All the learning happens in the forward pass through the sequence $X$ in an implicit fashion.

**Task-Specific Attention.** Task-specific attention uses fixed parameters to carry out a particular task when the prompt follows the required structure (see (Bai et al., 2023) for examples.)

> **Definition 2.3.** An attention layer is *task-specific* if there exists a prompt family $\mathcal{P}$ such that, for any prompt $P \in \mathcal{P}$ constructed from task parameters/data, the attention's forward pass implements the task's mapping on the query token(s), with no parameter change.

In particular, we embed the task's defining transformations (e.g. a linear mapping corresponding to $f$ or part of $f$) into the attention weight matrices. Given a well-formed prompt of exemplar and query tokens, the attention selects and combines these tokens to compute the correct output. Effectively, this allows an attention layer to approximate diverse functions in context without weight updates.

**Terminology: Task-Specific vs. Prompt-Programmable In-Context Emulation.** In-context algorithm emulation refers to executing an algorithm through a forward pass without weight updates. The core contribution of this work is to formalize two in-context modes and study their scope:

- **Task-Specific In-Context Emulation**: for each algorithm $\mathcal{A}$, there exists an attention module (possibly multi-head) whose forward pass on a well-formed prompt implements $\mathcal{A}$ on the query token(s). Each algorithm therefore requires its own dedicated parameters.
- **Prompt-Programmable In-Context Emulation (via single frozen module)**: there exists a single attention module with fixed weights $\text{Attn}^\star$ such that, for every $\mathcal{A}$ in a target class, a suitable prompt $P_{\mathcal{A}}$ makes $\text{Attn}^\star$ implement $\mathcal{A}$ on the query token(s). All adaptation occurs through the prompt rather than through weight changes. Namely, one $\text{Attn}^\star$ implements a library of algorithms.

These modes are complementary: the first reflects the conventional dedicated-module view (e.g., (Bai et al., 2023)), while the second is stronger — one *fixed-weight attention module* emulates many algorithms via prompts (our contribution). In the remainder of the paper, Section 3 develops the task-specific case. Section 4 establishes the prompt-programmable case by showing how the latter *subsumes* the former via in-context simulation of task-specific modules.

## 3 TASK-SPECIFIC IN-CONTEXT ALGORITHM EMULATION

We present multiple examples demonstrating how softmax attention modules mimic behaviors of various learning algorithms including gradient descent and linear regression. We begin with a very general result showing that even a single-layer, single-head attention mechanism is a universal approximator for a broad class of functions defined in the prompt.

**In-Context Universal Approximation of** $f(w^\top x - y)x$**.** Let $x \in \mathbb{R}^d$, $y \in \mathbb{R}$, $w \in \mathbb{R}^d$, and let $f : \mathbb{R} \to \mathbb{R}$ be continuously differentiable. We consider functions of the form $f(w^\top x - y)x$, where $f$ acts on the residual $w^\top x - y$. This template is very general: many learning rules for linear models take this form, including many residual/gradient-style updates[4]. Hence $f(w^\top x - y)x$ subsumes a wide family of residual-driven updates central to machine learning. Thus, their in-context realization explains much of in-context learning. To this end, showing that attention is capable of emulating any continuous $f(w^\top x - y)x$ indicates a powerful and general capability. It means the attention module

---

[4]For example, $f(t) = t$ corresponds to the raw residual $(w^\top x - y)x$, $f(\cdot) = \nabla_w \ell(\cdot)$ corresponds to per-sample gradients $\nabla_w \ell(w^\top x - y)x$ linear regression or classification with loss $\ell(\cdot)$, and nonlinear $f$ (sigmoid, step, etc.) corresponds to perceptron updates or other error-correcting rules.

implements any continuous adjustment or mapping based on the prediction $w^\top x$ and the label $y$. The next theorem shows how a single-head attention approximates $[f(w^\top x_i - y_i)x_i]_{i=1}^n$ arbitrarily well.

---

**Theorem 3.1** (In-Context Emulation of $f(w^\top x - y)x$ with Single-Head Attention). Let $[L_{\min}, L_{\max}]$ be a bounded interval containing all values of $w^\top x - y$, and let

$$X := \begin{bmatrix} x_1 & x_2 & \cdots & x_n \\ y_1 & y_2 & \cdots & y_n \end{bmatrix} \in \mathbb{R}^{(d+1)\times n} \quad \text{and} \quad W := [w \quad w \quad \cdots \quad w] \in \mathbb{R}^{d \times n},$$

where $x_i \in \mathbb{R}^d$, $y_i \in \mathbb{R}$, and $w \in \mathbb{R}^d$ is the coefficient vector. Define the input as:

$$Z := \begin{bmatrix} x_1 & x_2 & \cdots & x_n \\ y_1 & y_2 & \cdots & y_n \\ w & w & \cdots & w \end{bmatrix} = \begin{bmatrix} X \\ W \end{bmatrix} \in \mathbb{R}^{(2d+1)\times n}. \tag{3.1}$$

Assume $\max\{\|X\|_\infty, \|W\|_\infty\} \le B$. For any continuously differentiable function $f : \mathbb{R} \to \mathbb{R}$ and any $\epsilon > 0$, there exists a single-head attention $\mathrm{Attn}_s$ with a linear layer $\mathrm{Linear}$ such that

$$\|\mathrm{Attn}_s \circ \mathrm{Linear}(Z) - \left[ f(w^\top x_1 - y_1)x_1 \quad \cdots \quad f(w^\top x_n - y_n)x_n \right]\|_\infty \le \epsilon, \quad \text{for any} \quad \epsilon > 0.$$

---

*Proof.* See Appendix D.1 for a detailed proof. □

Theorem 3.1 establishes that even the simplest softmax attention alone suffices to encode any continuous function of the form $f(w^\top x - y)x$ by incorporating weights in the prompt. A direct implication is by replacing $f$ with the derivatives of differentiable loss function as follows.

**Example 1: In-Context Emulation of Single-Step GD.** Building on Theorem 3.1, we show that a softmax attention layer emulates Gradient Descent (GD) in-context. First, we replace the continuous function $f(\cdot)$ in Theorem 3.1 with $\nabla\ell(\cdot)$, where $\ell : \mathbb{R} \to \mathbb{R}$ is any differentiable loss function. We show that the softmax attention emulates *per-sample gradients* in context.

---

**Corollary 3.1.1** (In-Context Emulation of Per-Sample Gradients). Let $\ell : \mathbb{R} \to \mathbb{R}$ be differentiable and $\ell' : \mathbb{R} \to \mathbb{R}$ for its scalar derivative, $\ell'(t) = \frac{d}{dt}\ell(t)$. For $z := w^\top x - y$ with $x \in \mathbb{R}^d$, $y \in \mathbb{R}$, $w \in \mathbb{R}^d$, denote $\nabla_w \ell(z) := \ell'(z)$. Set $f(\cdot) = \ell'(\cdot)$ in Theorem 3.1. With $Z = [X; W]$ as in (3.1), for any $\epsilon > 0$, there exist a single-head attention $\mathrm{Attn}_s(\cdot)$ and a linear map $\mathrm{Linear}(\cdot)$ such that,

$$\| \underbrace{\mathrm{Attn}_s \circ \mathrm{Linear}(Z)}_{=:\widehat{G} \; \textit{approximated} \text{ per-sample gradient matrix}} - \underbrace{[\ell'(w^\top x_1 - y_1)x_1, \cdots, \ell'(w^\top x_n - y_n)x_n]}_{=:G \; \textit{target} \text{ per-sample gradient matrix}} \|_\infty \le \epsilon.$$

---

Corollary 3.1.1 shows that a single-layer single-head softmax attention with a linear map approximates the individual (per-sample) gradient terms $\{\ell'(w^\top x_i - y_i)x_i\}_{i=1}^n$. Moreover, the layer outputs all per-sample gradient terms in parallel. Next, we extend Corollary 3.1.1 to show that a fixed attention layer implements the full gradient-descent update across all samples in-context.

Aggregating the per-sample gradients gives one GD step

$$\widehat{L}_n(w) := \frac{1}{n}\sum_{i=1}^n \ell(w^\top x_i - y_i), \quad \nabla\widehat{L}_n(w) = \frac{1}{n}\sum_{i=1}^n \ell'(w^\top x_i - y_i)x_i =: g.$$

From Corollary 3.1.1, let $\widehat{G}$ be the attention output and choose the readout $u := \frac{1}{n}\mathbf{1}_n \in \mathbb{R}^n$ (equivalently, right-multiply by $W_O = u$ in Definition 2.1). Define the attention estimate of the average gradient as $\widehat{g} := \widehat{G}u$. Then $\widehat{g} \approx g$, and the target update is $w_{\mathrm{GD}}^+ := w - \eta\nabla\widehat{L}_n(w)$. Feeding $w$ in the prompt and applying the same readout produces a single $d$-dimensional update vector from the layer. The next corollary states the precise approximation guarantee.

---

**Corollary 3.1.2** (In-Context Emulation of a Single GD Step). Let $\ell : \mathbb{R} \to \mathbb{R}$ be differentiable and define $\widehat{L}_n(w) := \frac{1}{n}\sum_{i=1}^n \ell(w^\top x_i - y_i)$. For any step size $\eta > 0$ and any $\epsilon > 0$, there exist a single-head attention $\mathrm{Attn}_s$ and a linear map $\mathrm{Linear}$ such that, with $Z = [X; W]$ as in (3.1), choosing the readout $u := \frac{1}{n}\mathbf{1}_n$ (equivalently, right-multiply by $W_O = u$ in Definition 2.1), we have

$$\widehat{w}_{\mathrm{GD}} := \underbrace{w + \overbrace{(\mathrm{Attn}_s \circ \mathrm{Linear}(Z))u}^{\widehat{G}u = \widehat{g}}}_{w - \eta\widehat{g}} \in \mathbb{R}^d \quad \text{and} \quad \|\widehat{w}_{\mathrm{GD}} - \underbrace{(w - \eta\nabla\widehat{L}_n(w))}_{w_{\mathrm{GD}}^+}\|_\infty \le \epsilon.$$

---

*Proof.* See Appendix D.2 for a detailed proof. □

Corollary 3.1.2 shows that a single-layer, single-head softmax attention with a linear map aggregates the per-sample gradients via the output projection. It produces a $d$-vector $\widehat{w}_{\text{GD}}$ that approximates the GD update $w_{\text{GD}}^+ = w - \eta \nabla \widehat{L}_n(w)$. Notably, each output column encodes a copy of $w$ together with a scaled per-sample gradient term. Averaging via the readout $u = \frac{1}{n}\mathbf{1}_n$ then recovers $w_{\text{GD}}^+$ up to $\epsilon$.

**Example 2: In-Context Emulation of Multi-Step GD.** We extend the single-step construction to show that a multi-layer softmax attention network emulates multi-step gradient descent. In particular, an $(L+1)$-layer transformer approximates $L$ steps of gradient descent.

Stack $(L+1)$ copies of the single-head layer from Corollary 3.1.2. At layer $t$ $(0 \le t < L)$, use the readout $u^{(t)} = \frac{1}{n}\mathbf{1}_n$ and the prompt $Z^{(t)} = [X; W^{(t)}]$ with $W^{(t)} := [\widehat{w}_{\text{GD}}^{(t)} \cdots \widehat{w}_{\text{GD}}^{(t)}]$. Define

$$\widehat{w}_{\text{GD}}^{(0)} := w, \quad \text{and} \quad \widehat{w}_{\text{GD}}^{(t+1)} := \text{Attn}_s \circ \text{Linear}(Z^{(t)})u^{(t)}.$$

For the target iterates, set $w_{\text{GD}}^{(0)} = w$ and $w_{\text{GD}}^{(t+1)} = w_{\text{GD}}^{(t)} - \eta \nabla \widehat{L}_n(w_{\text{GD}}^{(t)})$. By Corollary 3.1.2, Lemma D.4 and $\|\cdot\|_\infty \le \|\cdot\|_2$, we arrive

$$\|\widehat{w}_{\text{GD}}^{(t)} - w_{\text{GD}}^{(t)}\|_\infty \le t\epsilon, \quad t \in [L].$$

**Example 3: In-Context Emulation of Linear Regression.** We now present the construction for squared loss. We show that a multi-layer softmax attention emulates linear regression in-context.

---

**Corollary 3.1.3** (In-Context Emulation of Linear Regression). For any dataset $\{(x_i, y_i)\}_{i=1}^n$ with $x_i \in \mathbb{R}^d$, $y_i \in \mathbb{R}$ and any $\epsilon > 0$, there exists a depth-$L$ stack of single-head attention layers $\{\text{Attn}_s^l\}_{l=0}^{L-1}$, linear maps $\{\text{Linear}^l\}_{l=0}^{L-1}$, and a readout $u \in \mathbb{R}^n$ such that the iterates

$$\widehat{w}_{\text{linear}}^{l+1} := \widehat{w}_{\text{linear}}^l + (\text{Attn}_s^l \circ \text{Linear}^l(Z^l))u, \quad \ell = 0, \dots, L-1,$$

with $Z^l := [X; W^l]$ and $W^l := [\widehat{w}_{\text{linear}}^l, \dots, \widehat{w}_{\text{linear}}^l]$ as in (3.1) (for any fixed bounded $w^l$), satisfy

$$\|\widehat{w}_{\text{linear}}^L - w_{\text{linear}}\|_\infty \le \epsilon,$$

where $w_{\text{linear}} := \text{argmin}_{w \in \mathbb{R}^d} \frac{1}{2n}\sum_{i=1}^n (\langle w, x_i \rangle - y_i)^2$.

---

*Proof.* See Appendix D.3 for detailed proof. □

**Example 4: In-Context Emulation of Ridge Regression.** We add regularization term and show that a multi-layer softmax attention emulates ridge regression with $L_2$ penalty.

---

**Corollary 3.1.4** (In-Context Emulation of Ridge Regression). For any input-output pair $(x_i, y_i)$, where $x_i \in \mathbb{R}^d, y_i \in \mathbb{R}, i \in [n]$, and any $\epsilon > 0$, there exists a depth-$L$ stack of single-head attention layers $\{\text{Attn}_s^l\}_{l=0}^{L-1}$, linear maps $\{\text{Linear}^l\}_{l=0}^{L-1}$, and a readout $u \in \mathbb{R}^n$ such that the iterates

$$\widehat{w}_{\text{ridge}}^{l+1} := \widehat{w}_{\text{ridge}}^l + (\text{Attn}_s^l \circ \text{Linear}^l(Z^l))u, \quad \ell = 0, \dots, L-1,$$

with $Z^l := [X; W^l]$ and $W^l := [\widehat{w}_{\text{ridge}}^l, \dots, \widehat{w}_{\text{ridge}}^l]$ as in (3.1) (for any fixed bounded $w^l$), satisfy

$$\|\widehat{w}_{\text{ridge}}^L - w_{\text{ridge}}\|_\infty \le \epsilon,$$

where $w_{\text{ridge}} := \text{argmin}_{w \in \mathbb{R}^d} \frac{1}{2n}\sum_{i=1}^n (\langle w, x_i \rangle - y_i)^2 + \frac{\lambda}{2}\|w\|_2^2$ with regularization term $\lambda \ge 0$.

---

*Proof.* See Appendix D.4 for detailed proof. □

So far our constructions in Section 3 show that, given freedom to choose parameters per algorithm, attention modules emulate gradient descent, linear regression, ridge regression, and related updates in context. These results establish the expressive power of task-specific in-context emulation, akin to (Bai et al., 2023). In Section 4, we build on this foundation and prove a stronger universality: a single frozen attention module $\text{Attn}^\star$, via prompt programming, simulates all such task-specific modules.

## 4 PROMPT-PROGRAMMABLE IN-CONTEXT ALGORITHM EMULATION

This section presents our main results: softmax attention is capable of (i) emulating *task-specific attention heads* in-context (Section 4.1), (ii) emulating statistical models in-context (Section 4.2), and (iii) emulating any network (with linear projections) in-context (Section 4.3). Unlike Section 3 requiring a separate task-specific module for each algorithm, here we fix one frozen module $\text{Attn}^\star$ and

show that suitable prompts instruct it to emulate every algorithm in the target class. This establishes universality: one set of weights executes a library of algorithms through prompt programming.

## 4.1 IN-CONTEXT EMULATION OF ATTENTION

We first specify the input prompt with weight encoding.

---

**Definition 4.1** (Vectorization). For any matrix $X \in \mathbb{R}^{d_h \times d}$, we define $\underline{X} := \text{vec}(X) \in \mathbb{R}^{dd_h}$ such that $\underline{X}_{(i-1)d+j} = X_{i,j}$ for all $i \in [d_h]$ and $j \in [d]$.

---

**Definition 4.2** (Input Prompt of In-Context Emulation of Attention). Let $X \in \mathbb{R}^{d \times n}$ be the input sequence, and let $W_K, W_Q, W_V \in \mathbb{R}^{d_h \times d}$ be the weight matrices of the target attention head to be emulated. Define the vectorizations

$$\underline{W}_K := \text{vec}(W_K) \in \mathbb{R}^{dd_h}, \ \underline{W}_Q := \text{vec}(W_Q) \in \mathbb{R}^{dd_h}, \ \underline{W}_V := \text{vec}(W_V) \in \mathbb{R}^{dd_h},$$

and $w := [\underline{W}_K; \underline{W}_Q; \underline{W}_V] \in \mathbb{R}^{3dd_h}$, where $w$ is the concatenation of $\underline{W}_K, \underline{W}_Q, \underline{W}_V$. Finally, define the extended input $X_p$ for in-context emulation of the attention head specified by $W_K, W_Q, W_V$ as

$$X_p := \begin{bmatrix} X \\ W_{\text{in}} \\ I_n \end{bmatrix} \quad \text{with} \quad W_{\text{in}} := \begin{bmatrix} 0 \cdot w & 1 \cdot w & 2 \cdot w & \cdots & (n-1) \cdot w \\ w & w & w & \cdots & w \end{bmatrix} \in \mathbb{R}^{6dd_h \times n}.$$

---

In other words, $W_{\text{in}}$ is a $2 \times n$ block matrix whose $j$-th column consists of $j \cdot w \in \mathbb{R}^{dd_h}$ (in the first block-row) and $w \in \mathbb{R}^{dd_h}$ (in the second block-row), for $j = 0, 1, \ldots, n-1$. Appending $W_{\text{in}}$ as additional rows to $X$ produces the prompt $X_p$ that encodes the target weights.

Using this weight-encoding prompt, we now design a two-layer attention mechanism to reproduces the effect of the target attention head in-context.

---

**Theorem 4.1** (In-Context Emulation of Attention). Let $X \in \mathbb{R}^{d \times n}$ be an input sequence, and let $W_K, W_Q, W_V \in \mathbb{R}^{d_h \times d}$ be the weight matrices of the target attention head we wish to emulate in-context. Assume $\|W_K X\|_\infty, \|W_Q X\|_\infty, \|W_V X\|_\infty \leq B_{KQV}$ with $B_{KQV} > 0$. Then, for any $\epsilon > 0$, there exists a two-layer attention network — a multi-head attention layer $\text{Attn}_m$ followed by a single-head attention layer $\text{Attn}_s$ — such that

$$\| \underbrace{\text{Attn}_s \circ \text{Attn}_m(X_p)}_{\text{Emulator}} - \underbrace{W_V X \text{Softmax}_\beta((W_K X)^\top W_Q X)}_{\text{Target}} \|_\infty \leq \epsilon,$$

where $X_p$ is the prompt defined in Definition 4.2.

---

*Proof.* See Appendix A.1 for the proof sketch and Appendix D.5 for a detailed proof. □

**Remark 4.1** (Permutation Equivariance). Our construction keeps the permutation equivariance of attention in its approximation. This means changing the order of columns in $X$ results in an identical change in the order of the columns in $\text{Attn}_s \circ \text{Attn}_m(X_p)$.

We now provide an alternative formulation of the above result. In this variant, a single-head attention layer comes first, followed by a multi-head layer with sequence-wise linear projections.

---

**Theorem 4.2** (In-Context Emulation of Attention; Alternative Formulation). Let $X \in \mathbb{R}^{d \times n}$ be the input sequence, and let $W_K, W_Q, W_V \in \mathbb{R}^{n \times d}$ be the weight matrices of the target attention. Assume $B = \max\{\|X\|_\infty, \|W_K\|_\infty, \|W_Q\|_\infty, \|W_V\|_\infty\}$ and $\|W_K X\|_\infty, \|W_Q X\|_\infty, \|W_V X\|_\infty \leq B_{KQV}$ for $B_{KQV} \geq 0$. Then, for any $\epsilon > 0$, there exists a single-head attention layer $\text{Attn}_s$ followed by a multi-head attention layer with linear projections such that

$$\|\text{Attn}_s \circ (\sum_{j=1}^{3n} \text{Attn}_j \circ \text{Linear}_j(\begin{bmatrix} X \\ W_K^\top \\ W_Q^\top \\ W_V^\top \end{bmatrix})) - \underbrace{W_V X}_{n \times n} \text{Softmax}_\beta \underbrace{((W_K X)^\top W_Q X)}_{n \times n} \|_\infty \leq \epsilon.$$

---

*Proof.* See Appendix A.2 for the proof sketch and Appendix D.6 for a detailed proof. □

Theorems 4.1 and 4.2 allow us to approximate arbitrary target one-layer attention using another two-layer attention. This construction requires no feed-forward layers and no parameter updates. All

approximation happens through the prompt alone (by embedding target attention weights and input $X$ into the prompt).

**Discussion: Target Attention Approximation for Algorithm Emulation.** Theorems 4.1 and 4.2 present a general algorithm emulation result: a fixed-weight two-layer softmax attention module emulates *all* algorithms implementable by softmax attention via only prompting. For example, if we choose the input sequence $X \in \mathbb{R}^{d \times n}$ in Theorem 4.1 and Theorem 4.2 to be the $\text{Linear}(Z) \in \mathbb{R}^{2(2d+n+2) \times n(P+1)}$ in Theorem 3.1, then we are able to approximate all one-layer attentions implementing target algorithms of the $f(w^\top x - y)x$ class: Corollaries 3.1.1 to 3.1.4. Thus, we achieve in-context emulation of the entire class of algorithms expressible as $f(w^\top x - y)x$.

This provides a constructive toy model of fixed-weight transformer exhibiting *general-purpose ability* (i.e., *one* fixed-weight model for *many* tasks). Moreover, the construction is explicit, interpretable, and softmax-native. A few remarks are in order.

**Remark 4.2** (Differences between Theorems 4.1 and 4.2)**.** Theorems 4.1 and 4.2 both establish that a fixed multi-head attention network can approximate any given attention head in-context. We present two versions of the construction using different formulations and analytical techniques. In particular, Theorem 4.1 encodes the target algorithm into the token representations (keeping the sequence length fixed), whereas Theorem 4.2 achieves a similar effect by encoding the weights as additional tokens in the input sequence (keeping each token's dimension fixed).

**Remark 4.3.** Our constructions may contain non-standard choices, including encoding information along the embedding dimension and using $3n$ parallel attention heads. We emphasize that the methods apply to approximate a more realistic attention with far fewer hidden dimensions and number of heads in practice. Section 5 provides further details.

**Remark 4.4** (Comparison with Prior Work)**.** We remark that our results differ from prior work in three key aspects. First, we study the practical softmax attention rather than linear or ReLU attention (Bai et al., 2023; Von Oswald et al., 2023; Vladymyrov et al., 2024). Second, our results in Section 4 go beyond task-specific ICL and establish that fixed-weight Transformers are prompt-programmable (Bai et al., 2023; Wu et al., 2025; Li et al., 2025). Third, our results are constructive, providing concrete emulation examples in contrast to prior prompting expressivity (Wang et al., 2023; Furuya et al., 2024) or Turing-completeness results (Pérez et al., 2021; Giannou et al., 2023; Qiu et al., 2024). The closest works to ours are (Bai et al., 2023; Giannou et al., 2023):

- Bai et al. (2023) show that Transformers can execute several standard algorithms in-context. However, their constructions use ReLU-based attention and require a separate customized Transformer for each algorithm (i.e., task-specific). By contrast, we study Transformers with softmax attention, as used in practice, and give constructive results. We show that a single fixed attention module emulates a broad class of algorithms through prompt changes, with explicit mechanisms and guarantees. In particular, Section 3 establishes softmax-attention counterparts of their main results.

- Giannou et al. (2023) are closest in spirit, since they also use one fixed model for many tasks. However, their goal is arbitrary program execution via Turing completeness. They achieve this with a much heavier 13-layer construction with looping. Our scope is narrower but cleaner: we show that a fixed two-layer attention-only module emulates any algorithm implementable by a softmax attention layer. The prompt supplies the program, and the model executes it. While our results are also extensible to looped setting (i.e., Corollary 3.1.2), our focus is different: we use an attention-only, FFN-free model and analyze its algorithmic universality *constructively*. This yields a more transparent mechanism for algorithm emulation and a sharper theory of internal algorithm execution in softmax Transformers.

## 4.2 IN-CONTEXT EMULATION OF STATISTICAL METHODS

Theorem 4.2 shows that a frozen attention module approximates a target attention head by embedding the head's weights into its input prompt. We now leverage this idea to emulate a broader class of algorithms. In essence, we replace the embedded target attention weights with the parameters of an arbitrary statistical method that we aim to emulate. By the same principle, the fixed attention module then mimics the behavior of diverse statistical models within the in-context learning framework.

**Corollary 4.2.1** (In-Context Emulation of Statistical Methods)**.** Let $\mathcal{A}$ be the set of all algorithms implementable by a single-layer attention network in-context. For any finite collection of algorithms

$\{a_1, a_2, \ldots, a_k\} := \mathcal{A}_0 \subseteq \mathcal{A}$, there exists a two-layer attention network (a single-head layer $\text{Attn}_s$ composed with a multi-head layer $\text{Attn}_m$) such that for each $a \in \mathcal{A}_0$ in the collection

$$\|\sum_{j=1}^{3n} \text{Attn}_s \circ \text{Attn}_j \circ \text{Linear}_j \left( \begin{bmatrix} X \\ W^a \end{bmatrix} \right) - a(X)\|_\infty \leq \epsilon,$$

where $W^a$ is the $W$ defined as Definition 4.2 using $W_K^a, W_Q^a, W_V^a$.

*Proof.* See Appendix D.7 for detailed proof. □

We show that a fixed attention module emulates an arbitrary finite library of in-context algorithms by varying its prompt. This result highlights the flexibility of softmax attention: unlike prior work that requires re-training or fine-tuning of the model, here we provably achieve task-specific behavior by modifying the input prompt. In effect, a pretrained Transformer internalizes a small set of fundamental procedures and later deploys them, via prompting, across a wide range of data distributions. Since the number of distinct algorithms is far smaller than the number of possible datasets, a model that learns a handful of algorithms can leverage them to handle many different scenarios.

## 4.3 ATTENTION MAKES EVERY (LINEAR) NETWORK IN-CONTEXT

We now extend the above ideas to show that softmax attention emulates *any* network (comprised of linear transformations) in-context. Consider any layer of a neural network that applies a trainable linear map $x \to \Theta x$ with weight matrix $\Theta$. Our results imply that if $\Theta$ is provided as part of the input sequence, a fixed attention module is capable of approximating this transformation to arbitrary precision. Hence linear layers in standard architectures are replaceable with attention whose effective weights are encoded in the prompt rather than learned. This substitution turns the network into an in-context learner in place of, or alongside, conventional training.

**Remark 4.5** (In-Context Emulation of Linear Layers). For example, suppose a model contains a linear layer $f(x) = \Theta x$ with weight matrix $\Theta$. By including $\Theta$ (appropriately encoded) in the input as in our constructions above, a single softmax attention layer emulates $f(x)$ in-context to arbitrary precision. In other words, any trainable linear mapping in the original network is replicable with a prompt-programmable attention layer whose parameters are set by the input sequence. This enables the overall network to adjust that layer's behavior on-the-fly via prompts, rather than having to learn $\Theta$ through pre-training.

## 5 NUMERICAL STUDIES

This section provides numerical results to back up our theory. We validate two building blocks on synthetic data: (i) approximation of continuous functions (Section 5.1); and (ii) approximation of attention heads (Section 5.2). These studies quantify approximation error and its relation to model size and the number of heads.

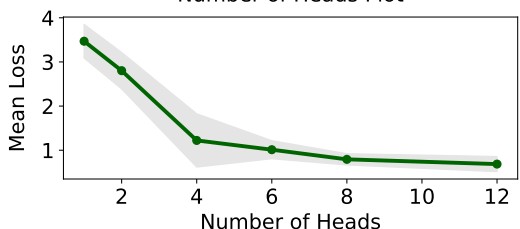

Number of Heads Plot

Figure 2: **Sensitivity of Attention Emulation to the Number of Heads.** We report loss (MSE) as the mean and one standard deviation (shaded region) over 10 random seed runs. We use synthetic data of 50000 data points with sequence length being 20 and input dimension being 24. We set batch size to be 32 and hidden dimension to be 48. Each multi-head model and the single-head softmax attention layer is trained for 50 epochs. The optimizer used is Adam with learning rate 0.001. We visualize the performance (MSE ± Std) for $1, 2, 4, 6, 8, 12$ heads.

### 5.1 PROOF-OF-CONCEPT EXPERIMENT ON THEOREM 3.1

**Objective: Verifying Attention Approximates** $f(w^\top x - y)x$**.** We investigate accuracy of softmax attention approximating $f(w^\top x - y)x$ by training a single-head softmax attention with linear connection.

**Data Generation.** We randomly generate $X \in \mathbb{R}^{n \times d}$ drawn from a normal distribution, $X \sim 10 \cdot N(0, 1) - 5$. We also generate weight matrix $W \in \mathbb{R}^{n \times d}$ and $y \in \mathbb{R}^n$, both randomly drawn from a standard normal distribution, $N(0, 1)$. Here, $n$ represents the sequence length and $d$ represents input dimension. The true label is $f(w^\top x - y)x$, where we choose $f(\cdot) = \tanh(\cdot)$.

**Model Architecture.** We train a single-head attention network to approximate $\tanh(w^\top x - y)x$. We first apply linear transformation to both $[X; y]$ and $W$. We then train the single-head attention model to approximate our target function as shown in the proof of Theorem 3.1.

**Results.** As shown in Figure 2, evaluated on Mean Square Error loss, the model approximates the target $\tanh(w^\top x - y)x$ with minimal error. This experiment proves our theory.

## 5.2 Proof-of-Concept Experiment on Emulating Attention Heads

**Objective: Verifying Approximation Rates.** We investigate the affect of the number of attention heads $H$ on the accuracy of softmax attention approximating softmax attention head.

**Data Generation.** We randomly generate a sequence of tokens $X = [x_1, x_2, \cdots, x_n] \in \mathbb{R}^{d \times n}$, where each entry $x_i$ is drawn independently from a normal distribution,

$$X \sim 2 \cdot N(0, 1) - 1.$$

We also generate weight matrices $K = W_K X^\top, Q = W_Q X^\top \in \mathbb{R}^{h \times n}$, and $V = W_V X^\top \in \mathbb{R}^{d \times n}$. Each parameter matrix is randomly drawn from a standard normal distribution, $N(0, 1)$. Here, $n, d, h$ represents the sequence length, token dimension, and hidden dimension, respectively. The true label $Y \in \mathbb{R}^{d \times n}$ results from applying a single-layer softmax attention mechanism on inputs $X, K, Q, V$.

**Model Architecture.** We train a multi-layer attention network to approximate softmax attention. We first train separate multi-head models to approximate $K$, $Q$, and $V$. Then, we use a single-head softmax attention layer to approximate softmax attention function as in the proof.

**Results.** As shown in Figure 2 and Table 1, the result validates our claim that a multi-head softmax attention mimics a target softmax attention head to arbitrary precision. Moreover, it demonstrates the convergence of multi-head softmax attention emulating softmax-based attention mapping as the number of heads increases. The approximation rate is in the trend of $O(1/H)$

Table 1: **Sensitivity to the Number of Heads.** Emulation MSE (mean $\pm$ std) for multi-head softmax attention with $1, 2, 4, 6, 8$, and $12$ heads.

| Heads | 1 | 2 | 4 | 6 | 8 | 12 |
|-------|-------|-------|-------|-------|-------|-------|
| MSE | 3.469 | 2.802 | 1.222 | 1.012 | 0.793 | 0.686 |
| Std | 0.381 | 0.413 | 0.603 | 0.204 | 0.127 | 0.171 |

where $H$ is the number of heads. The small and decreasing MSE error indicates that the simple softmax attention model approximates softmax attention head with stability.

**Additional Experiments.** Due to page limits, we defer several experimental results to Appendix C. These include simulations of statistical algorithms (Appendix C.1) and approximations of statistical models on real-world datasets where the model does not have access to the true algorithm weights (Appendix C.2). They further illustrate the approximation capabilities of Transformer in practice.

## 6 Discussion and Conclusion

We study *in-context algorithm emulation* in fixed-weight Transformers and formalize two modes: task-specific (Section 3) and prompt-programmable algorithm emulation (Section 4). For the former, we show that even a single-layer, single-head module suffices for emulating core families (of the form $f(w^\top x - y)x$) such as one-step gradient descent and linear/ridge regression, achieving architectural minimality (Theorem 3.1). For the latter, we show that a two-layer multi-head softmax attention module emulates a broad class of algorithms by embedding the algorithm's weights into the input prompt (Theorem 4.1). Altogether, a fixed softmax attention module becomes a prompt-programmable *library of algorithms*: weights remain frozen, and the prompt selects the routine.

**Mechanism.** The mechanism is constructive. By encoding target weights in the input and creating large dot-product margins, softmax attention routes along the intended computation without weight updates. Numerical studies support the theory: on synthetic data the model accurately approximates continuous maps of the form $f(\langle w, x \rangle - y)x$ and emulates attention heads. Approximation error decreases as the number of heads grows. On a real dataset (Ames Housing), the frozen module-driven by prompts rather than true algorithm weights-achieves low error against standard statistical models.

**Implications.** Our results tighten the link between in-context learning and algorithmic emulation. Viewing prompts as callable subroutines that select and configure algorithms within a frozen model, we draw three takeaways: (i) prompt engineering becomes interface design for algorithm selection, (ii) pretraining objectives *could*, in future work, be designed to encourage learning compact libraries of reusable procedures, and (iii) analyses of internal routing help clarify how foundation models select among algorithms. This lens explains the breadth of in-context generalization, guides prompt design, and motivates new pretraining objectives for more effective algorithm installation and utilization.

## ETHIC STATEMENT

This paper does not involve human subjects, personally identifiable data, or sensitive applications. We do not foresee direct ethical risks. We follow the ICLR Code of Ethics and affirm that all aspects of this research comply with the principles of fairness, transparency, and integrity.

## REPRODUCIBILITY STATEMENT

We ensure reproducibility on both theoretical and empirical fronts. For theory, we include all formal assumptions, definitions, and complete proofs in the appendix. For experiments, we describe model architectures, datasets, preprocessing steps, hyperparameters, and training details in the main text and appendix. Code and scripts are provided in the supplementary materials to replicate the empirical results.

## ACKNOWLEDGMENTS

JH thanks Alex Reneau, Yingyu Liang, Mimi Gallagher, Sara Sanchez, T.Y. Ball, Dino Feng and Andrew Chen for valuable conversations; Po-Chiao Lin for meticulous proofreading and pointing out typos; Po-Chiao Lin, Xiwen Zhang, Mingcheng Lu, Hong-Yu Chen, Maojiang Su, Weimin Wu, Zhao Song and Pin-Yu Chen for collaborations on related topics; the Red Maple Family for support. The authors also thank the anonymous reviewers and program chairs for constructive comments.

This work is in remembrance of Prof. Bi-Ming Tsai (February 9, 1965 - October 21, 2025). Her wisdom in health, cooking, and the art of thoughtful living continues to inspire JH.

JH is partially supported by Ensemble AI and Northwestern University's Walter P. Murphy Fellowship and Terminal Year Fellowship (Paul K. Richter Memorial Award). Han Liu is partially supported by NIH R01LM1372201, NSF AST-2421845, Simons Foundation MPS-AI-00010513, AbbVie , Dolby and Chan Zuckerberg Biohub Chicago Spoke Award. This research was supported in part through the computational resources and staff contributions provided for the Quest high performance computing facility at Northwestern University which is jointly supported by the Office of the Provost, the Office for Research, and Northwestern University Information Technology. The content is solely the responsibility of the authors and does not necessarily represent the official views of the funding agencies.

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

# Appendix

## IMPACT STATEMENT

We prove that a single frozen softmax attention head emulates a broad library of attention-implementable algorithms via prompt design, establishing pretrained Transformers as universal algorithm stores and reducing the need for task-specific fine-tuning. This sharpens the theoretical basis of in-context learning, offers a principled recipe for prompt engineering, and equips auditors with a clear test for hidden prompt-encoded behaviors, all without releasing new models or data. Therefore, the work advances foundational understanding, lowers compute and energy demands, and introduces minimal societal risk.

## LIMITATIONS AND FUTURE DIRECTION

Prompt length grows linearly with the weight dimension, which limits practicality. The proofs assume exact real-valued softmax and ignore token discretizations or numerical noise. Prompts are hand-crafted. Learning them automatically is open. Language and vision inputs are untested. Weight encoding happens along embedding dimension. The construction is not permutation invariant, but permutation equivariant of the input data. Lastly, we leave tighter constants, shorter prompts, extensions to deeper models and connection with model pretraining to future work.

## LLM USAGE DISCLOSURE

We used large language models (LLMs) to aid and polish writing, such as improving clarity, grammar, and conciseness. We also used LLMs for retrieval and discovery, for example exhausting literature to identify potential missing related work. All technical content, proofs, experiments, and results are original contributions by the authors.

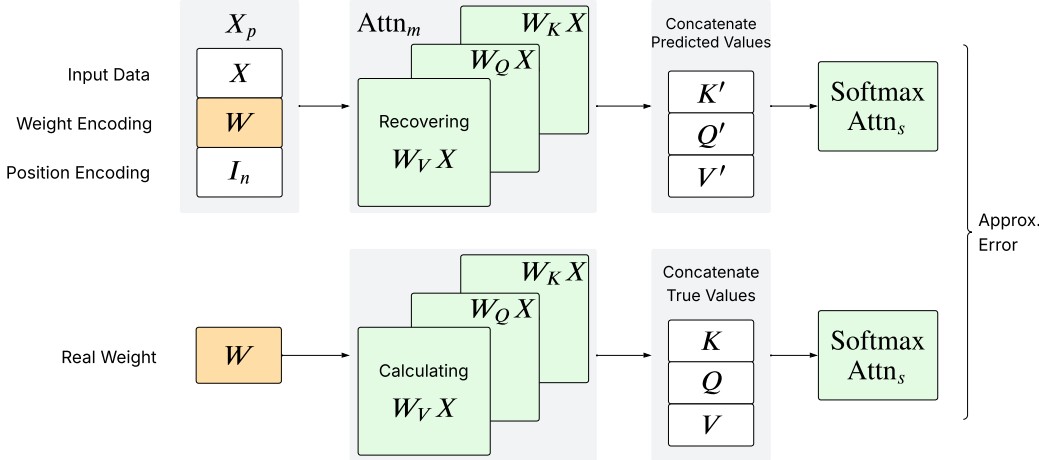

Figure 3: **Visualization of Proof Sketch for** Theorem 4.1. We visualize our proof technique. We combine input data, weight encoding, and position encoding into $X_p$ as input to the multi-head attention $\text{Attn}_m$ to recover approximate key, query, and value representations. We then compare the single-head attention $\text{Attn}_s$ outputs from approximate values with ground truth values to obtain approximation error.

## A  PROOF SKETCHES

We present our proof strategies here.

### A.1  PROOF SKETCH FOR THEOREM 4.1

We construct a two-layer Transformer (single-head layer $\text{Attn}_s$ followed by multi-head layer $\text{Attn}_m$) that replicates the target attention to within any error $\epsilon > 0$. Recall from Theorem 4.1:

$$\| \overbrace{\text{Attn}_s}^{\textbf{Step 3}} \circ \underbrace{\text{Attn}_m(\overbrace{X_p}^{\textbf{Step 1}})}_{\textbf{Step 2}} - \underbrace{W_V X \text{Softmax}((W_K X)^\top W_Q X)}_{\text{Target}} \|_\infty \leq \epsilon.$$

The high-level idea is: (**Step 1**) augment the input with a prompt encoding of the target weights $W_K, W_Q, W_V$, (**Step 2**) use groups of heads in $\text{Attn}_m$ to approximate the matrices $K := W_K X$, $Q := W_Q X$, $V := W_V X$ in-context (up to small error), and (**Step 3**) apply $\text{Attn}_s$ with fixed weights to assemble the $\text{Attn}_m$ output: the approximators $K', Q', V'$. We then argue in (**Step 4**) the approximation error can be made $< \epsilon$ via a stability bound on softmax attention.

**Step 1: In-Context Weight Encoding.**  We augment the input $X \in \mathbb{R}^{d \times n}$ by appending special tokens encoding the matrices $W_K, W_Q, W_V$. We denote the augmented input as $X_p$. This allows the transformer $\text{Attn}_s \circ \text{Attn}_m$ to "read" the relevant weight parameters in its attention heads.

Explicitly, we embed both the data sequence and the target head into the input (Definition 4.2):

$$X_p = \begin{bmatrix} X \\ W_{\text{in}} \\ I_n \end{bmatrix} \quad \text{with} \quad W_{\text{in}} := \begin{bmatrix} 0 \cdot w & 1 \cdot w & 2 \cdot w & \cdots & (n-1) \cdot w \\ w & w & w & \cdots & w \end{bmatrix},$$

where $w = [\underline{W}_K^\top, \underline{W}_Q^\top, \underline{W}_V^\top]^\top$ concatenates every column of $W_K^\top, W_Q^\top, W_V^\top$ following Definition 4.2. The block $I_n$ provides token-position codes used in our construction.

The idea of this augmentation is to emulate the target computation $W_V X \text{Softmax}((W_K X)^\top W_Q X)$ with the emulator: a two-layer transformer $\text{Attn}_s \circ \text{Attn}_m$. To achieve this, $\text{Attn}_s \circ \text{Attn}_m$ must access the information of $X, W_K, W_Q$ and $W_V$ in-context. The augmentation above encodes these parameters (target algorithm's specifications) in the prompt.

**Step 2: Multi-Head Decomposition for In-Context Recovery of $K, Q, V$.** We devote the first attention layer ($\text{Attn}_m$) to recovering the key, query, and value matrices that the target attention would compute. By definition, each row in $K, Q$, and $V$ takes the form: $k_i^\top X, q_i^\top X$ and $v_i^\top X$. Here $k_i^\top, q_i^\top$ and $v_i^\top$ are rows in $W_K, W_Q$ and $W_V$. Our goal is, for each data token $x_i$ (the $i$-th column of $X$), to approximate $k_j^\top x_i, q_j^\top x_i$, and $v_j^\top x_i$. To do this, we design $\text{Attn}_m$ to have a fixed number of heads partitioned into *three groups*, corresponding to $K, Q$, and $V$ respectively. Combining heads' outputs within each group yields approximations of $K, Q$, and $V$. Explicitly, in the first multi-head layer $\text{Attn}_m$, we split the heads so that:

- A group of heads jointly approximates $W_K X$. By (Hu et al., 2025a, Theorem 3.2), the heads in this group admit further subdivision into sub-groups. Each sub-group outputs a linear transformations of $X$, namely $k_i^\top X$ for rows $k_i^\top X$ of $K$.

- Another group of heads approximates $W_Q X$ in a similar manner.

- A final group approximates $W_V X$.

Concatenate or combine these head outputs so that the final embedding from $\text{Attn}_m(X_p)$ contains (up to small error) the blocks $[K; Q; V]$ for all positions in $X$.

Explicitly, for each row $k_j^\top$ of $W_K$ (and similarly for $q_j^\top$ and $v_j^\top$), we prepend the corresponding heads with a token-wise linear map $A(\cdot)$. $A(X_p)$ pulls out the target row (i.e., $k_j$) from $w$ and repeats it $n$ times. The resulting sub-prompt $A(X_p)$ has the form

$$\begin{bmatrix} & & X & \\ 0 \cdot k_j & 1 \cdot k_j & \cdots & (n-1) \cdot k_j \\ k_j & k_j & \cdots & k_j \\ & & I_n & \end{bmatrix},$$

so the corresponding softmax heads return $k_j^\top X$ up to any error $\epsilon_0$ by the truncated-linear interpolation theorem (Theorem D.1). With $H = \lceil 2(b-a)/((n-2)\epsilon_0) \rceil$ heads per sub-group, we cover all $d_h$ rows in $K$ (and similarly for $Q$ and $V$). Altogether, the $3N = 3d_h H$ heads satisfy

$$\underbrace{\| \sum_{j=1}^{d_h} \text{Attn}_j^K(X_p) - K \|_\infty \leq \epsilon_0}_{:=K'}, \; \underbrace{\| \sum_{j=d_h+1}^{2d_h} \text{Attn}_j^Q(X_p) - Q \|_\infty \leq \epsilon_0}_{:=Q'}, \; \underbrace{\| \sum_{j=2d_h+1}^{3d_h} \text{Attn}_j^V(X_p) - V \|_\infty \leq \epsilon_0}_{:V'}.$$

We collect these outputs column-wise into

$$\begin{bmatrix} K' \\ Q' \\ V' \end{bmatrix}, \quad \text{and} \quad \| \begin{bmatrix} K' \\ Q' \\ V' \end{bmatrix} - \begin{bmatrix} K \\ Q \\ V \end{bmatrix} \|_\infty \leq \epsilon_0.$$

**Step 3: Single-Head Assembly for Emulated Map.** We consider the second layer $\text{Attn}_s$ as a single-head attention with fixed weights chosen to "read" the $K', Q', V'$ triples from $Z := \text{Attn}_m(X_p)$ and perform the "emulated" attention mechanism. Explicitly, apply a single-head attention layer $\text{Attn}_s$ whose parameters are set to read off the $K, Q$, and $V$ sub-blocks in each token embedding:

$$\text{Attn}_s(Z) := W_V^{(s)} Z \text{Softmax}((W_K^{(s)} Z)^\top (W_Q^{(s)} Z)).$$

For $Z := \text{Attn}_m(X_p)$, we choose fixed weights

$$W_K^{(s)} = [0_{d_h \times 2d_h} \quad I_{d_h}], \quad W_Q^{(s)} = [I_{d_h} \quad 0_{d_h \times 2d_h}], \quad W_V^{(s)} = [0_{d_h \times d_h} \quad I_{d_h} \quad 0_{d_h \times d_h}],$$

so that

$$W_K^{(s)} Z \approx W_K X, \quad W_Q^{(s)} Z \approx W_Q X, \quad W_V^{(s)} Z \approx W_V X.$$

Hence,

$$\text{Attn}_s(Z) = \text{Attn}_s \circ \text{Attn}_m(X_p) = \text{Attn}_s(\begin{bmatrix} K' \\ Q' \\ V' \end{bmatrix}) = V'\text{Softmax}((K')^\top Q')$$

$$\approx W_V X \text{Softmax}((W_K X)^\top W_Q X).$$

To be precise, because $K', Q', V'$ differ from $K, Q, V$ by at most $\epsilon_0$ (in $\|\cdot\|_\infty$), a first-order perturbation argument for softmax (uniform Lipschitz in sup-norm) shows

$$\|\text{Attn}_s(\begin{bmatrix} K' \\ Q' \\ V' \end{bmatrix}) - \text{Attn}_s(\begin{bmatrix} K \\ Q \\ V \end{bmatrix})\| \leq \epsilon_0 + nB_{KQV}\epsilon_1,$$

where $B_{KQV}$ bounds $X, W_K, W_Q, W_V$ and $\epsilon_1 = O(\epsilon_0)$.

**Step 4: Error Bound.** Finally, we make the approximation arbitrarily precise. Because we are capable of making each head's linear approximation arbitrarily close, we ensure

$$\|\text{Attn}_s \circ \text{Attn}_m([X; W]) - W_V X \text{Softmax}((W_K X)^\top W_Q X)\|_\infty \leq \epsilon,$$

for any $\epsilon > 0$. This completes the construction, proving in-context emulation of the target attention. Please see Appendix D.5 for a detailed proof and Figure 3 for proof visualization.

## A.2 PROOF SKETCH FOR THEOREM 4.2

We outline how to emulate the desired attention step-by-step with a fixed two-layer transformer. Similar to Theorem 4.1 (and Theorem D.1), our construction ensures each token's representation in the intermediate layer carries an approximate copy of its key, query, and value vectors, which the final layer uses to perform the softmax attention. All necessary components (including the weight matrices $W_K, W_Q, W_V$ themselves) are encoded into the input, so the network's weights remain untrained and generic.

**Step 1: Encoding Weights into the Input.** Let $X \in \mathbb{R}^{d \times n}$ be the input tokens. Append a "weight encoding" matrix $W$ that contains the rows of $W_K, W_Q, W_V$ (the weight matrices of the target attention head). This forms an extended input $[X; W]$. The entries of $X$ and $W$ remain within a bounded range $[-B, B]$. This bound ensures that all inner products remain finite.

**Step 2: Multi-Head Approximation of $K, Q, V$.** The first layer has many heads. Partition them into three groups. One group approximates $K := W_K X$; one approximates $Q := W_Q X$; and one approximates $V := W_V X$. Then

- **Simulating Dot Products on a 1D Grid.** Consider a single entry $k_j^\top x_c$. All entries in $K, Q, V$ lie between $[-dB^2, dB^2]$, since the entries of $X$ and $W$ remain within a bounded range $[-B, B]$. We create grid points $L_0 < \cdots < L_P$ covering $[-dB^2, dB^2]$. We design the head's key and query so that the softmax assigns each grid point $L_i$ a weight based on its distance to $k_j^\top x_c$. We set the value vector to encode $L_i$. Thus, the head output for token $x_c$ approximates $k_j^\top x_c$. Fine grids reduce the error.

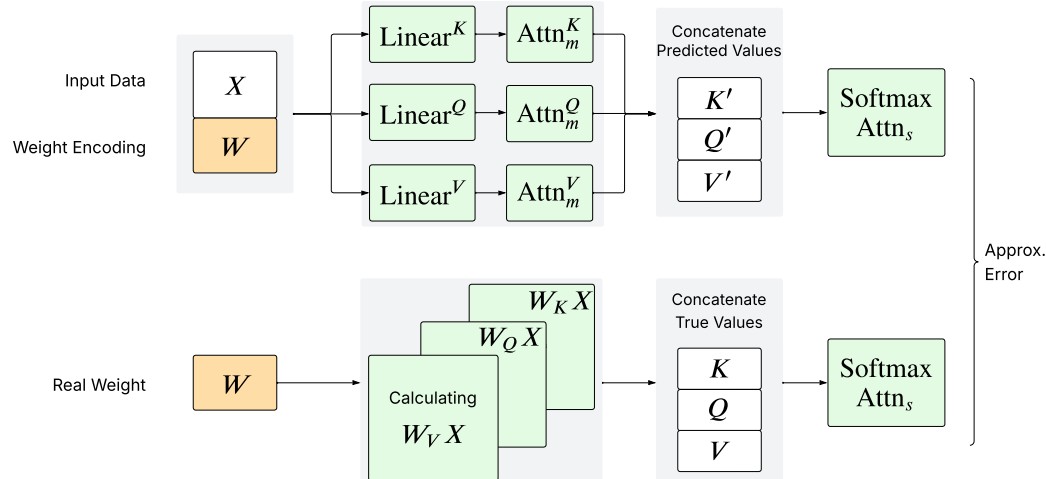

Figure 4: **Visualization of Proof Sketch for Theorem 4.2**. We visualize our proof technique. We combine input data and weight encoding as input. Each key, query, and value has a unique set of linear transformation of input (Linear) and multi-head attention ($\text{Attn}_m$). We feed the input into each set to attain the approximate key, query, and value representations, respectively. We then compare the single-head attention $\text{Attn}_s$ outputs from approximate values with ground truth values to obtain approximation error.

- **Reconstructing Full** $K, Q, V$. Repeat this idea for every entry in $K, Q, V$. Each row uses one head to approximate $k_j^\top X$, $q_j^\top X$, or $v_j^\top X$. Combine these approximations to obtain the matrices $K', Q', V'$. The sup norm

$$\| \begin{bmatrix} K' \\ Q' \\ V' \end{bmatrix} - \begin{bmatrix} K \\ Q \\ V \end{bmatrix} \|_\infty, \quad \text{can be made arbitrarily small.}$$

**Step 3: Single-Head Assembly of the Attention Output.** The second layer, $\text{Attn}_s$, has one head. We set its weight matrices $W_K^{(s)}, W_Q^{(s)}, W_V^{(s)}$ to pick out $K', Q', V'$ from each token's embedding. Then, $\text{Attn}_s$ computes

$$V'\text{Softmax}((K')^\top Q') \approx W_V X \text{Softmax}((W_K X)^\top W_Q X),$$

since $K' \approx K, Q' \approx Q$ and $V' \approx V$.

**Step 4: Error Bound.** Softmax and matrix multiplication are continuous. Small errors in $K', Q', V'$ cause a small error in the final output. By refining the grid (and using enough heads), we make the sup norm error below any $\epsilon > 0$. Please see Appendix D.6 for a detailed proof and Figure 4 for proof visualization.

## B    RELATED WORK

Our results diverge from prior findings on Transformer universality and in-context learning.

### B.1    CORE RELATED WORK

**Universal Approximation.**    Prior studies establish that Transformers approximate arbitrary sequence-to-sequence functions, but they do not address in-context learning and often assume complex architectures. For example, Yun et al. (2020) prove that deep multi-head Transformers with feed-forward layers are universal approximators of continuous sequence-to-sequence functions. Subsequent advances tighten this finding: Kajitsuka and Sato (2024); Hu et al. (2025b) show that even

a single-layer Transformer realizes any continuous sequence function. However, these results treat Transformers as parametric function approximators. The model requires re-training and re-prompting to adapt to a new target function instead of handling multiple tasks through context. In contrast, we prove that a minimal Transformer architecture, even a single-layer, single-head attention module with no feed-forward network, emulates a broad class of algorithms without weight updates by varying its prompt. This result achieves a new level of generality through context alone (i.e. prompt-based conditioning) despite a fixed minimalist model.

**In-Context Learning and Algorithm Emulation.** Another line of recent theory bridges Transformers with in-context learning by designing model components to carry out specific algorithms. For example, Bai et al. (2023) show that Transformers execute a broad range of standard algorithms in-context, but each algorithm requires a distinct, tailored attention head. In comparison, we extend this approach by showing that one fixed attention mechanism emulates any specialized attention head via prompt encoding. Rather than crafting a different attention module for each algorithm, a single frozen softmax-based attention layer takes its instructions from the prompt to perform all tasks in context. This minimal model thus becomes a unified and compact in-context algorithm emulator. It switches behaviors by changing only its input prompt, setting it apart from earlier approaches that required per-task reparameterization.

## B.2 BROADER DISCUSSION

**Universal Approximation and Expressivity of Transformers.** Transformers exhibit strong expressive power as sequence models. Recent theory shows even minimal Transformer architectures approximate broad classes of functions. Kajitsuka and Sato (2024); Hu et al. (2025b) prove a single-layer, single-head Transformer can memorize any finite dataset perfectly. Kajitsuka and Sato (2024) achieve this with low-rank attention matrices, while Hu et al. (2025b) use attention matrices of any rank. Adding two small feed-forward layers makes it a universal approximator for continuous sequence functions under permutation-equivariance. More recently, Hu et al. (2025a) show self-attention layers alone are universal approximators. Specifically, two attention-only layers approximate continuous sequence-to-sequence mappings, and even a single softmax-attention layer suffices for universal approximation. Similarly, Liu et al. (2025) also demonstrate that one single-head attention connected with linear transformations is sufficient to approximate any continuous function in $L_\infty$ norm. These results eliminate the need for feed-forward networks, improving on earlier constructions. Overall, these findings highlight the inherent expressiveness of minimal attention mechanisms.

**Transformers as In-Context Learners and Algorithm Emulators.** Large Transformers also learn in-context by conditioning on examples in their prompts, without updating weights (Brown et al., 2020). Recent work formally explains this by showing attention-based models implement standard learning algorithms internally. Bai et al. (2023) construct Transformer heads executing algorithms such as linear regression, ridge regression, Lasso, and gradient descent steps, achieving near-optimal predictions. Wu et al. (2025) further build Transformers explicitly simulating multiple gradient descent iterations for training deep neural networks, with provable convergence guarantees. Empirical and theoretical studies confirm Transformers internalize learning algorithms when meta-trained on task families. Garg et al. (2022) show meta-trained Transformers mimic classical algorithms, such as ordinary least squares regression, in-context. Similarly, Akyürek et al. (2023); Von Oswald et al. (2023); Zhang et al. (2024) analyze Transformers trained on linear regression tasks and demonstrate their outputs mimic gradient descent steps precisely. Overall, existing literature shows that sufficiently trained or carefully designed Transformers emulate step-by-step computations of standard algorithms through prompt conditioning.

**Prompt-Tuning.** Prompt-tuning adapts frozen models by learning a short continuous prefix (Lester et al., 2021; Li and Liang, 2021; Liu et al., 2022). It keeps backbone weights fixed and updates only prompt embeddings. Our setting is stricter: prompts are hand-designed, not learned, and we give exact approximation bounds. Thus we expose the theoretical limit of prompt control: a single frozen softmax head can mimic any task-specific head.

**Encoding Context Along Embedding Dimension.**    Recent work in in-context learning explores encoding and manipulating context in the embedding space rather than sequence dimension. For example, Liu et al. (2024) propose In-Context Vectors for steering the model's behavior by adding task-specific vectors along the embedding space. Zhuang et al. (2025) extend this idea by showing that manipulating embedding vectors such as interpolation makes in-context learning more controllable. Abernethy et al. (2024) showcase that appending additional information along the embedding dimension allows the model to perform sample-efficient in-context learning.

**Comparison to Our Work.**    The above results demonstrate the versatility of Transformer networks, but they require task-specific weights, training, or learned prompts. For instance, Bai et al. (2023) design a different task-specific head for each algorithm of interest, raising the question of whether a single fixed attention mechanism could instead serve as a universal emulator for any algorithm given the right prompt. Our work directly addresses this question. In contrast, we prove one fixed softmax head emulates any specialized head through prompt encoding alone. No additional weights or training are required. Even the simplest attention (one layer, one head) acts as a universal algorithm emulator when given the right prompt, shifting focus from architecture to prompt design.

## C    ADDITIONAL NUMERICAL STUDIES

We extend the synthetic validation to statistical algorithms (Appendix C.1) and include a real-world study (Appendix C.2). The frozen attention module emulates linear, ridge, and lasso on synthetic data. On the Ames Housing dataset, the model operates without access to true algorithm weights and achieves low approximation error. In addition, we validate Theorem 4.2 through handcrafted frozen attention weights and parameters as constructed in the proof (Appendix C.3).

### C.1    PROOF-OF-CONCEPT EXPERIMENT ON EMULATING STATISTICAL MODELS

**Objective: Emulation of Statistical Models.**    We investigate the accuracy of a frozen softmax attention approximating statistical models including Lasso, Ridge and linear regression by only varying the input prompts.

**Data Generation.**    We simulate an in-context dataset by randomly generating a sequence of input tokens $X = [x_1, x_2, \ldots, x_n] \in \mathbb{R}^{n \times d}$, where each $x_i$ is independently drawn from a scaled standard normal distribution,

$$x_i \sim 2 \cdot N(0, 1) - 1.$$

A task-specific prompt vector $w \in \mathbb{R}^{p \times 1}$ is sampled from $N(0, 1)$. In the case of Lasso, we randomly zero out entries in $w$ with probability $0.5$ to induce sparsity. We generate the output sequence $Y \in \mathbb{R}^{n \times 1}$ via a noisy linear projection: $Y = Xw + \epsilon$, where $\epsilon \sim N(0, \sigma^2)$ is Gaussian noise. For Ridge, we calculate weights using $(X^\top X + \lambda I_d)^{-1} X^\top Y$ with $\lambda = 5$.

**Model Architecture and Training.**    We use a mixture of statistical data to train a single-layer attention network with linear transformation. Each input sample consists of $X \in \mathbb{R}^{n \times d}$ and algorithm-specific prompt $w \in \mathbb{R}^p$. We replicate $w$ across the sequence length and concatenate it with $X$ along the feature dimension to obtain an augmented input $[X; w] \in \mathbb{R}^{n \times (d+p)}$. We pass it through a multi-head attention layer. We train the model for 300 epochs using the Adam optimizer with a learning rate of $0.001$. We use 6 attention heads, a hidden dimension of $48$, an input dimension of $24$, a batch size of $32$, and $50000$ synthetic samples. After training, we freeze the attention weights, resulting in a fixed softmax attention layer. We evaluate the frozen model on its ability to emulate various statistical algorithms using test data.

**Baseline Architecture.**    We train three separate attention models for Lasso, Ridge, and linear regression. That is, each attention model weights are adaptive to its corresponding algorithm. We use

these models as baselines for comparison with the frozen attention model we propose. All baseline models use the same hyper-parameters as the frozen model.

**Results.** As shown in Table 2, we compare mean MSE and standard deviation over 5 random seed runs for the frozen attention model against baseline for Lasso, Ridge, and linear regression on the synthetic data. The frozen attention model performs as well as the baseline models trained individually on each algorithm. It achieves lower MSE on Lasso and linear regression tasks compared to their corresponding baselines. It shows that a frozen attention mechanism generalizes across these tasks given task-specific prompts. Moreover, the frozen model exhibits lower variance across all tasks, suggesting increased stability and robustness. These results support our claim that a frozen softmax attention layer, when conditioned on task-specific prompts, emulates statistical algorithms in context without much performance degradation.

Table 2: **Comparison Between Baseline and Frozen Attention Layer on Synthetic Dataset.** We compare loss (MSE) as the mean and one standard deviation over 5 random seed runs for baseline vs. frozen model on different algorithms. We train on 50000 training data points evaluate on 10000 testing data points for each algorithm.

| Model | Lasso | Ridge Regression | Linear Regression |
|---|---|---|---|
| Baseline | 0.068±0.015 | 0.004±0.0003 | 0.147±0.067 |
| Frozen Attention | 0.059±0.001 | 0.071±0.0002 | 0.120±0.003 |

## C.2 REAL-WORLD EXPERIMENT ON EMULATING STATISTICAL MODELS

**Objective: Real-World Emulation of Statistical Models.** Building on Appendix C.1, we use real-world data to investigate the accuracy of a frozen softmax attention emulating algorithms.

**Data Collection and Processing.** We collect data from Ames Housing Dataset (De Cock, 2011). This dataset consists of 2930 observations and 79 features. We process the data by log-transforming the target variable, encoding categorical variables with one-hot vectors, replacing missing entries with median values, and standardizing numerical features. The resulting data consists of 262 features. We fit the processed data to Lasso, Ridge, and linear regression models to obtain algorithm weights as part of the input.

**Model Architecture and Training.** We use a mixture of statistical data to train a single-layer attention network with linear transformation. The input is passed through a multi-head attention layer with a linear transformation. We train the model for 300 epochs using the Adam optimizer with a learning rate of $0.001$. We use 8 attention heads, a hidden dimension of $524$, and a batch size of $32$. After training, we freeze the attention weights, resulting in a fixed softmax attention layer. The frozen model is then evaluated on its ability to emulate various statistical algorithms using test data. We train the baseline models the same way as the synthetic experiment.

Table 3: **Comparison Between Baseline and Frozen Attention Layer on Ames Housing Dataset.** We compare loss (MSE) as the mean and one standard deviation over 5 random seed runs for baseline vs. frozen model on different algorithms. We train on $80\%$ training data and evaluate on $20\%$ testing data for each algorithm.

| Model | Lasso | Ridge Regression | Linear Regression |
|---|---|---|---|
| Baseline | 0.0354±0.0000 | 0.0132±0.0000 | 0.0288±0.0000 |
| Frozen Attention | 0.0322±0.0000 | 0.0252±0.0000 | 0.0250±0.0000 |

**Results.** As shown in Table 3, we compare mean MSE and standard deviation over 5 random seed runs for the frozen attention model against baseline for Lasso, Ridge, and linear regression on Ames

Housing Data. The results shows the frozen attention model performs as well as the baseline models trained individually. We use an auxiliary network to approximate the required weight encoding. Our experiment validates that the mechanism works even when the exact weights are not supplied in real world scenarios.

### C.3 PROOF-OF-CONCEPT EXPERIMENT ON THEOREM 4.2

**Objective: Verifying Handcrafted Frozen Attention Approximates Attention.** We validate that the frozen attention prescribed in Theorem 4.2 approximates softmax attention with low error. In particular, we handcraft the weights as in the proof of Theorem 4.2.

**Data Generation.** We create a synthetic dataset. We randomly generate $X \in \mathbb{R}^{n \times d}$ drawn from a uniform distribution over $[-1, 1]$, $X \sim U(-1, 1)$. For each sample, we generate three weight matrices $W_K, W_Q, W_V \in \mathbb{R}^{n \times d}$ drawn from standard normal distribution $N(0, 1)$. We then compute $K = W_K X^\top, Q = W_Q X^\top, V = W_V X^\top \in \mathbb{R}^{n \times n}$. The true target attention output is therefore given by $Y = V \mathrm{Softmax}(K^\top Q) \in \mathbb{R}^{n \times n}$.

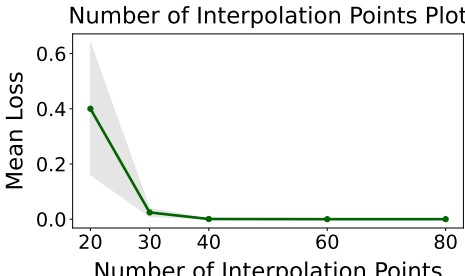

**Model Architecture.** Following the proof in Appendix D.6, we hard-wire the linear layer weights, attention weights, and interpolation points for the two-layer softmax attention module as our emulator. The model operates in a zero-shot, one-pass setting with no training or parameter updates.

Figure 5: **Sensitivity of Handcrafted Attention Emulation to the Number of Interpolation Points.** We report loss (MSE) as the mean and one standard deviation (shaded region) over 4 sample data points. Each data point has sequence length $n = 12$ and input dimension $d = 24$. We set softmax temperature $\beta = 2$. We visualize the performance (MSE $\pm$ Std) for $P = 20, 30, 40, 60, 80$.

**Results.** We report the results in Table 4 and Figure 5. We compare the MSE loss between the emulator output and the target attention output. Specifically, we fix the number of data points $n$, input dimension $d$, softmax temperature $\beta$, and number of samples for testing. We vary the number of interpolation points $P$. The result validates our claim that the handcrafted frozen attention approximates the target attention. Moreover, we show that as $P$ increases, the approximation error and standard deviation both further decrease.

Table 4: **Sensitivity to the Number of Interpolation Points.** We report MSE loss (mean$\pm$ std) between outputs of handcrafted frozen attention and target attention varying number of interpolation points $P$ over 4 samples. We choose $n = 12, d = 4, \beta = 2, \text{samples} = 4$ for evaluation.

| P | 20 | 30 | 40 | 60 | 80 |
|---|---|---|---|---|---|
| **Mean MSE** | $4.002 \times 10^{-1}$ | $2.442 \times 10^{-2}$ | $5.852 \times 10^{-4}$ | $5.770 \times 10^{-9}$ | $5.037 \times 10^{-14}$ |
| **Std** | $2.393 \times 10^{-1}$ | $1.451 \times 10^{-2}$ | $8.538 \times 10^{-5}$ | $1.994 \times 10^{-9}$ | $1.620 \times 10^{-14}$ |

## D PROOFS OF MAIN TEXT

To prepare our proofs, we state the following axillary definitions and lemmas.

**Definition D.1** (Truncated Linear Function). We define the truncated linear function as follows:

$$\text{Range}_{[a,b]}(x) = \begin{cases} a & x \le a, \\ x & a \le x \le b, \\ b & b \le x. \end{cases}$$

Intuitively, $\text{Range}_{[a,b]}(\cdot)$ is the part of a linear function whose value is in $[a,b]$.

We then define the interpolation points in $[a,b]$ that are used in later proofs.

**Definition D.2** (Interpolation). Let $[a,b] \subset \mathbb{R}$ be an interval with $a \le b$ and let $p \in \mathbb{N}^*$ be a positive integer. We define

$$\widetilde{L}_0^{[a,b]} := a, \quad \widetilde{L}_p^{[a,b]} := b, \quad \widetilde{L}_i^{[a,b]} := a + \frac{i}{p}(b-a), \quad i = [p-1].$$

Hence, $\widetilde{L}_0 < \widetilde{L}_1 < \cdots < \widetilde{L}_p$ forms a uniform partition of $[a,b]$. We also write

$$\Delta L := \widetilde{L}_i^{[a,b]} - \widetilde{L}_{i-1}^{[a,b]}, \quad i \in [p].$$

We often omit the superscript $[a,b]$ when the context is clear.

We also propose the following lemma to show Hardmax property that is capable of being approximated by Softmax.

**Lemma D.1** (Lemma F.1 in (Hu et al., 2025a): Approximating Hardmax with Finite-Temperature Softmax). Let $x = [x_1, x_2, \ldots, x_n] \in \mathbb{R}^n$, $\epsilon > 0$. Define $\text{Softmax}_\beta(\cdot)$ as

$$\text{Softmax}_\beta(x) := [\frac{\exp(\beta x_1)}{\sum_{j=1}^n \exp(\beta x_j)}, \cdots, \frac{\exp(\beta x_n)}{\sum_{j=1}^n \exp(\beta x_j)}].$$

The following statements hold:

- **Case of a Unique Largest Entry.** Assume $x_1 = \max_{i \in [n]} x_i$ is unique, and $x_2 = \max_{i \in [n] \setminus \{1\}} x_i$. Then, if $\beta \ge (\ln(n-1) - \ln(\epsilon))/(x_1 - x_2)$, we have

$$\left\| \text{Softmax}_\beta(x) - e_1 \right\|_\infty \le \epsilon,$$

  where $e_1 \in \mathbb{R}^n$ is the one-hot vector corresponding to to the maximal entry of $x$ (i.e., $x_1$.)
- **Case of Two Largest Entries (Tied or Separated by $\delta$).** Assume $x_1$ and $x_2$ are the first and second largest entries, respectively, with $\delta = x_1 - x_2 \ge 0$. Let $x_3$ be the third largest entry and is smaller than $x_1$ by a constant $\gamma > 0$ irrelevant to the input. Then, if $\beta \ge (\ln(n-2) - \ln \epsilon)/\gamma$, we have

$$\left\| \text{Softmax}_\beta(x) - \frac{1}{1+e^{-\beta\delta}} e_1 - \frac{e^{-\beta\delta}}{1+e^{-\beta\delta}} e_2 \right\|_\infty \le \epsilon.$$

The following technical lemma is used in the proof of Theorem D.1.

**Lemma D.2** (Refined Version of Lemma F.2 in (Hu et al., 2025a): Cases of All Heads in $\text{Attn}^H$). For $a \in [\widetilde{L}_0, \widetilde{L}_{H(n-2)}]$. For any $h \in [H]$, define three cases of the relationship between $a$ and $h$

- **Case 1:** $a \in [\widetilde{L}_{(h-1)(n-2)}, \widetilde{L}_{h(n-2)-1}]$,
- **Case 2:** $a \notin [\widetilde{L}_{(h-1)(n-2)-1}, \widetilde{L}_{h(n-2)}]$.
- **Case 3:** $a \in [\widetilde{L}_{(h-1)(n-2)-1}, \widetilde{L}_{(h-1)(n-2)}] \cup [\widetilde{L}_{h(n-2)-1}, \widetilde{L}_{h(n-2)}]$.

These cases includes all possible situation. Then for all $h$, only two cases exists

- $a$ falls in Case 1 for an $h$ and Case 2 for all others.
- $a$ falls in Case 3 for two adjacent $h$ and Case 2 for all others.

*Proof.* Because $a \in [\widetilde{L}_0, \widetilde{L}_{H(n-2)}]$ and

$$[\widetilde{L}_0, \widetilde{L}_{H(n-2)}] = \cup_{h=1}^{H} [\widetilde{L}_{(h-1)(n-2)}, \widetilde{L}_{h(n-2)}],$$

we have

$$a \in [\widetilde{L}_{(h_a-1)(n-2)}, \widetilde{L}_{h_a(n-2)}] \tag{D.1}$$

for an arbitrary $h_a$.

This leads to only two possible cases

- Case 1*: $a \in [\widetilde{L}_{(h_a-1)(n-2)}, \widetilde{L}_{h_a(n-2)-1}]$.

- Case 2*: $a \in [\widetilde{L}_{h_a(n-2)-1}, \widetilde{L}_{h_a(n-2)}]$.

**Case 1*:** $a \in [\widetilde{L}_{(h_a-1)(n-2)}, \widetilde{L}_{h_a(n-2)-1}]$. Because $a \in [\widetilde{L}_{(h_a-1)(n-2)}, \widetilde{L}_{h_a(n-2)-1}]$, for $h \neq h_a$, we have

$$\widetilde{L}_{h(n-2)-2}, \widetilde{L}_{h(n-2)} < \widetilde{L}_{(h_a-1)(n-2)}, \quad h < h_a$$
$$\widetilde{L}_{h(n-2)+1}, \widetilde{L}_{(h-1)(n-2)-1} \geq \widetilde{L}_{h_a(n-2)-1}, \quad h > h_a.$$

Thus

$$[\widetilde{L}_{(h_a-1)(n-2)}, \widetilde{L}_{h_a(n-2)-1}] \cap [\widetilde{L}_{(h-1)(n-2)-1}, \widetilde{L}_{h(n-2)}] = \emptyset$$
$$[\widetilde{L}_{(h_a-1)(n-2)}, \widetilde{L}_{h_a(n-2)-1}] \cap [\widetilde{L}_{(h-1)(n-2)-1}, \widetilde{L}_{h(n-2)}] = \emptyset$$

for all $h \neq h_a$.

This means that $a$ does not fall into Case 1 nor Case 3 for other $h \in [H]$. Thus $a$ has to fall into Case 2 for other $h$.

**Case 2*:** $a \in [\widetilde{L}_{(h_a-1)(n-2)}, \widetilde{L}_{(h_a-1)(n-2)+1}] \cup [\widetilde{L}_{h_a(n-2)-1}, \widetilde{L}_{h_a(n-2)}]$. Without loss of generality, assume $a$ to be in the left half $[\widetilde{L}_{(h_a-1)(n-2)}, \widetilde{L}_{(h_a-1)(n-2)+1}]$. Because

$$[\widetilde{L}_{(h_a-1)(n-2)}, \widetilde{L}_{(h_a-1)(n-2)+1}] = [\widetilde{L}_{(h_a-1)(n-2)-1}, \widetilde{L}_{(h_a-1)(n-2)}], \qquad (\text{Case 3 of } h_a - 1)$$
$$[\widetilde{L}_{(h_a-1)(n-2)}, \widetilde{L}_{(h_a-1)(n-2)+1}] = [\widetilde{L}_{(h_a-1)(n-2)-1}, \widetilde{L}_{(h_a-1)(n-2)}], \qquad (\text{Case 3 of } h_a)$$

this means $a$ falls into Case 3 for $h_a$ and $h_a - 1$.

This completes the proof. □

We are now ready to prove a refined version of (Hu et al., 2025a, Theorem 3.2).

**Theorem D.1** (Multi-Head Attention Approximate Truncated Linear Models In-Context). Let $X \in \mathbb{R}^{d \times n}$ be the input. Fix real numbers $a < b$, and let the truncation operator $\text{Range}_{[a,b]}(\cdot)$ follow Definition D.1. Let $w_s$ denote the linear coefficient and $t$ denote the bias of the in-context truncated linear model. Define $W_s$ as

$$W_s := \begin{bmatrix} 0 \cdot w_s & 1 \cdot w_s & \cdots & (n-1) \cdot w_s \\ w_s & w_s & \cdots & w_s \end{bmatrix} \in \mathbb{R}^{2d \times n}.$$

For a precision parameter $p > n$ with $\epsilon = O(1/p)$, number of head $H = p/(n-2)$ there exists a single-layer, $H$-head self-attention $\text{Attn}^H$ with a linear transformation $A : \mathbb{R}^{d \times n} \to \mathbb{R}^{(3d+n) \times n}$, such that $\text{Attn}^H \circ A : \mathbb{R}^{d \times n} \to \mathbb{R}^{d_o \times n}$ satisfies, for any $i \in [n]$,

$$\left\| \text{Attn}^H \circ A\left( \begin{bmatrix} X \\ W_s \\ t \cdot I_n \end{bmatrix} \right)_{:,i} - \text{Range}_{[a,b]}(w_s^\top x_i + t) e_{\widetilde{k}_i} \right\|_\infty \leq \underbrace{\max\{|a|, |b|\} \cdot \epsilon_0}_{\text{finite-}\beta\text{ softmax error}} + \underbrace{\frac{b - a}{(n-2)H}}_{\text{interpolation error}} .$$

Here $e_{\widetilde{k}_i}$ is the one-hot vector with a 1 at position $\widetilde{k}_i$-th index and 0 elsewhere, and

$$\widetilde{k}_i = G(k_i) \in [d_o], \quad \text{with} \quad k_i = \underset{k \in \{0, 1, \cdots, p-1\}}{\arg\min} (-2x_i^\top w_i - 2t_i + \widetilde{L}_0 + \widetilde{L}_k) \cdot k,$$

where $G : [p] \to [d_o]$ denotes any set-to-constant function sending each selected interpolation index $k_i$ into an appropriate integer $\widetilde{k}_i \in [d_o]$ for $i \in [n]$.

**Remark D.1.** We remark that, the same in-context approximation guarantee holds for bias-less truncated linear model (i.e., generalized ReLU) $\text{Range}_{[a,b]}(w_s^\top x_i) e_{\widetilde{k}_i}$.

*Proof.* Define $A : \mathbb{R}^{d \times n} \to \mathbb{R}^{(3d+n) \times n}$ for the input sequence $X$ as

$$A(X) := \underbrace{\begin{bmatrix} I_{3d+n} \\ 0_{n \times (3d+n)} \end{bmatrix}}_{\text{token-wise linear}} \underbrace{\begin{bmatrix} X \\ W_s \\ t \cdot I_n \end{bmatrix}}_{(3d+n) \times n} + \underbrace{\begin{bmatrix} 0_{(3d+n) \times n} \\ I_n \end{bmatrix}}_{\text{positional encoding}} = \underbrace{\begin{bmatrix} X \\ W_s \\ t \cdot I_n \\ I_n \end{bmatrix}}_{(3d+2n) \times n} .$$

Thus, $A$ is a token-wise linear layer augmented with positional encoding, as it applies a linear projection to each token and then adds a unique per-token bias.

Let $p$ be a precision parameter, without loss of generality, let it be divisible by $n - 2$ and denote $p/(n-2)$ as $H$.

Now we define the multi-head attention $\text{Attn}$ of $H$ heads. Denote $\ell_k := k(\widetilde{L}_k + \widetilde{L}_0)$ for $k \in [p]$ following Definition D.2. Let $c_k := -2k$. We denote the $h$-th head as $\text{Attn}_h$, and define the weight matrices as

$$W_K^{(h)} = -\beta \underbrace{\begin{bmatrix} 0_{d \times d} & -2I_d & -2[(h-1)(n-2)-1]I_d & 0_{d \times n} & 0 & 0 & \cdots & 0 \\ 0_{1 \times d} & 0_{1 \times d} & 0_{1 \times d} & 0_{1 \times n} & \ell_{(h-1)(n-2)-1} & \ell_{(h-1)(n-2)} & \cdots & \ell_{h(n-2)} \\ 0_{1 \times d} & 0_{1 \times d} & 0_{1 \times d} & 0_{1 \times n} & c_{(h-1)(n-2)-1} & c_{(h-1)(n-2)} & \cdots & c_{h(n-2)} \end{bmatrix}}_{(d+2) \times (3d+2n)},$$

$$W_Q^{(h)} = \underbrace{\begin{bmatrix} I_d & 0_{d \times 2d} & 0_{d \times n} & 0_{d \times n} \\ 0_{1 \times d} & 0_{1 \times 2d} & 0_{1 \times n} & 1_{1 \times n} \\ 0_{1 \times d} & 0_{1 \times 2d} & 1_{1 \times n} & 0_{1 \times n} \end{bmatrix}}_{(d+2) \times (3d+2n)},$$

$$W_V^{(h)} = \begin{bmatrix} 0_{d_o \times (3d+1)} & \widetilde{L}_{(h-1)(n-2)} e_{\widetilde{k}_{(h-1)(n-2)}} & \widetilde{L}_{(h-1)(n-2)+1} e_{\widetilde{k}_{(h-1)(n-2)+1}} & \cdots & \widetilde{L}_{h(n-2)-1} e_{\widetilde{k}_{h(n-2)-1}} & 0_{d_o} \end{bmatrix},$$

for every $h \in [H]$.

Here $\beta > 0$ is a coefficient we use to control the precision of our approximation. The attention reaches higher precision as $\beta$ gets larger.

Let With the construction of weights, we are also able to calculate the $K$, $Q$, $V$ matrices in $\text{Attn}$

$$K^{(h)} := W_K^{(h)} \cdot A(X)$$

$$= W_K^{(h)} \cdot \underbrace{\begin{bmatrix} X \\ W_s \\ t \cdot I_n \\ I_n \end{bmatrix}}_{(3d+2n) \times n}$$

$$= W_K^{(h)} \cdot \begin{bmatrix} x_1 & x_2 & \cdots & x_n \\ 0 \cdot w_s & 1 \cdot w_s & \cdots & (n-1) \cdot w_s \\ w_s & w_s & \cdots & w_s \\ t \cdot e_1^{(n)} & t \cdot e_2^{(n)} & \cdots & t \cdot e_n^{(n)} \\ e_1^{(n)} & e_2^{(n)} & \cdots & e_n^{(n)} \end{bmatrix}$$

$$= -\beta \cdot \underbrace{\begin{bmatrix} 0_{d \times d} & -2I_d & -2[(h-1)(n-2)-1]I_d & 0_{d \times n} & 0 & 0 & \cdots & 0 \\ 0_{1 \times d} & 0_{1 \times d} & 0_{1 \times d} & 0_{1 \times n} & \ell_{(h-1)(n-2)-1} & \ell_{(h-1)(n-2)} & \cdots & \ell_{h(n-2))} \\ 0_{1 \times d} & 0_{1 \times d} & 0_{1 \times d} & 0_{1 \times n} & c_{(h-1)(n-2)-1} & c_{(h-1)(n-2)} & \cdots & c_{h(n-2)} \end{bmatrix}}_{(d+2) \times (3d+2n)} \cdot$$

$$\underbrace{\begin{bmatrix} x_1 & x_2 & \cdots & x_n \\ 0 \cdot w_s & 1 \cdot w_s & \cdots & (n-1) \cdot w_s \\ w_s & w_s & \cdots & w_s \\ t \cdot e_1^{(n)} & t \cdot e_2^{(n)} & \cdots & t \cdot e_n^{(n)} \\ e_1^{(n)} & e_2^{(n)} & \cdots & e_n^{(n)} \end{bmatrix}}_{(3d+2n) \times n}$$

$$= -\beta \underbrace{\begin{bmatrix} -2[(h-1)(n-2)-1]w & \cdots & -2h(n-2)w \\ \ell_{(h-1)(n-2)-1} & \cdots & \ell_{h(n-2)} \\ c_{(h-1)(n-2)-1} & \cdots & c_{h(n-2)} \end{bmatrix}}_{(d+2) \times n}, \tag{D.2}$$

where the last equality comes from multiplying $X$ with 0, thus this is a extraction of non-zero entries in $W_K$.

For $Q$, we have

$$Q^{(h)} := W_Q^h \cdot A(X)$$

$$= \begin{bmatrix} I_d & 0_{d \times 2d} & 0_{d \times n} & 0_{d \times n} \\ 0_{1 \times d} & 0_{1 \times 2d} & 0_{1 \times n} & 1_{1 \times n} \\ 0_{1 \times d} & 0_{1 \times 2d} & 1_{1 \times n} & 0_{1 \times n} \end{bmatrix} \cdot \underbrace{\begin{bmatrix} X \\ W_s \\ t \cdot I_n \\ I_n \end{bmatrix}}_{(3d+2n) \times n}$$

$$= \underbrace{\begin{bmatrix} X \\ 1_{1 \times n} \\ t_{1 \times n} \end{bmatrix}}_{(d+1) \times n}. \tag{D.3}$$

For $V$, we have

$$V^{(h)} := W_V^{(h)} \cdot A(X)$$

$$= \underbrace{\begin{bmatrix} 0_{d_o \times 3d} & 0_{d_o \times n} & 0_{d_o \times 1} & \widetilde{L}_{(h-1)(n-2)}e_{\widetilde{k}_{(h-1)(n-2)}} & \cdots & \widetilde{L}_{h(n-2)-1}e_{\widetilde{k}_{h(n-2)-1}} & 0_{d_o \times 1} \end{bmatrix}}_{d_o \times (3d+2n)} \cdot \underbrace{\begin{bmatrix} X \\ W_s \\ t \cdot I_n \\ I_n \end{bmatrix}}_{(3d+2n) \times n}$$

$$= \begin{bmatrix} 0_{d_o} & \widetilde{L}_{(h-1)(n-2)}e_{\widetilde{k}_{(h-1)(n-2)}} & \widetilde{L}_{(h-1)(n-2)+1}e_{\widetilde{k}_{(h-1)(n-2)+1}} & \cdots & \widetilde{L}_{h(n-2)-1}e_{\widetilde{k}_{h(n-2)-1}} & 0_{d_o} \end{bmatrix}. $$

$$\underbrace{\phantom{xxxxxxxxxxxxxxxxxxxxxxxxxxxxxxxxxxxxxxxxxxxxxxxxxxxxxxxxxxxxxxxxxxxxxxxxxxxxx}}_{d_o \times n}$$

$$\text{(D.4)}$$

Given that all $\widetilde{k}_j$, where $j \in [p]$, share the same number in $[d_o]$, we denote this number by $k_G$.
Hence we rewrite $V^{(h)}$ as

$$V^{(h)} = \begin{bmatrix} 0_{d_o} & \widetilde{L}_{(h-1)(n-2)}e_{k_G} & \widetilde{L}_{(h-1)(n-2)+1}e_{k_G} & \cdots & \widetilde{L}_{h(n-2)-1}e_{k_G} & 0_{d_o} \end{bmatrix}.$$

We define $m_v$ as

$$m_v := \max\{|a|, |b|\}.$$

By the definition of $V^{(h)}$, we have

$$\|V\|_\infty \le \max_{i \in [P]}\{\widetilde{L}_i\} \le m_v. \tag{D.5}$$

**Remark D.2** (Intuition of the Construction of $V^{(h)}$). As previously mentioned, $\widetilde{L}_i$, for $i \in [p]$, are all the interpolation points. In this context, $V^{(h)}$ encompasses the $(n-2)$ elements of these interpolations (i.e., $(h-1)(n-2)$ to $h(n-2)-1$). Meanwhile, the value on the two ends of $V^h$ are both set to $0_{d_o}$, because we suppress the head and let it output $0$ when the input $X$ is not close enough to the interpolations of the head.

Now we are ready to calculate the output of each $\text{Attn}_h$

$$\begin{aligned}
&\text{Attn}_h(A(X)) \\
&= V^{(h)}\text{Softmax}((K^{(h)})^\top Q^{(h)}) \\
&= V^{(h)}\text{Softmax}\left(-\beta \begin{bmatrix} -2[(h-1)(n-2)-1]w & \cdots & -2h(n-2)w \\ \ell_{(h-1)(n-2)-1} & \cdots & \ell_{h(n-2)} \\ c_{(h-1)(n-2)-1} & \cdots & c_{h(n-2)} \end{bmatrix}^\top \begin{bmatrix} X \\ 1_{1\times n} \\ t_{1\times n} \end{bmatrix}\right),
\end{aligned}$$

where last line is by plug in (D.2) and (D.3). Note the $i$-th column of the attention score matrix (the Softmax nested expression) is equivalent to the following expressions

$$\begin{aligned}
&\text{Softmax}((K^{(h)})^\top Q^{(h)})_{:,i} \\
&= \text{Softmax}(-\beta \begin{bmatrix} -2[(h-1)(n-2)-1]w^\top & \ell_{(h-1)(n-2)-1} & c_{(h-1)(n-2)-1} \\ \vdots & \vdots & \vdots \\ -2h(n-2)w^\top & \ell_{h(n-2)} & c_{h(n-2)} \end{bmatrix} \cdot \begin{bmatrix} X \\ 1_{1\times n} \\ t_{1\times n} \end{bmatrix})_{:,i} \\
&= \text{Softmax}\left(-\beta \begin{bmatrix} -2[(h-1)(n-2)-1]w^\top x_i + \ell_{(h-1)(n-2)-1} + c_{(h-1)(n-2)-1}\cdot t \\ -2(h-1)(n-2)w^\top x_i + \ell_{(h-1)(n-2)} + c_{(h-1)(n-2)}\cdot t \\ \vdots \\ -2h(n-2)w^\top x_i + \ell_{h(n-2)} + c_{h(n-2)}\cdot t \end{bmatrix}\right) \\
&\qquad\qquad\qquad\qquad\qquad\qquad\qquad\qquad\qquad\qquad\qquad\qquad\qquad\qquad\text{(pick column } i) \\
&= \text{Softmax}\left(-\beta \begin{bmatrix} [(h-1)(n-2)-1](-2w^\top x_i + \widetilde{L}_{(h-1)(n-2)-1} + \widetilde{L}_0) - 2[(h-1)(n-2)-1]t \\ (h-1)(n-2)(-2w^\top x_i + \widetilde{L}_{(h-1)(n-2)} + \widetilde{L}_0) - 2(h-1)(n-2)t \\ \vdots \\ h(n-2)(-2w^\top x_i + \widetilde{L}_{h(n-2)} + \widetilde{L}_0) - 2h(n-2)t \end{bmatrix}\right)
\end{aligned}$$

$$= \text{Softmax} \left( -\frac{\beta}{\Delta L} \begin{bmatrix} (-2x_i^\top w - 2t + \widetilde{L}_0 + \widetilde{L}_{(h-1)(n-2)-1}) \cdot [(h-1)(n-2)-1]\Delta L \\ (-2x_i^\top w - 2t + \widetilde{L}_0 + \widetilde{L}_{(h-1)(n-2)}) \cdot (h-1)(n-2)\Delta L \\ \vdots \\ (-2x_i^\top w - 2t + \widetilde{L}_0 + \widetilde{L}_{h(n-2)}) \cdot h(n-2)\Delta L \end{bmatrix} \right)$$
(By mutiplying and dividing by $\Delta L$)

$$= \text{Softmax} \left( -\frac{\beta}{\Delta L} \begin{bmatrix} (-2x_i^\top w - 2t + \widetilde{L}_0 + \widetilde{L}_{(h-1)(n-2)-1}) \cdot (\widetilde{L}_{(h-1)(n-2)-1} - \widetilde{L}_0) \\ (-2x_i^\top w - 2t + \widetilde{L}_0 + \widetilde{L}_{(h-1)(n-2)}) \cdot (\widetilde{L}_{(h-1)(n-2)} - \widetilde{L}_0) \\ \vdots \\ (-2x_i^\top w - 2t + \widetilde{L}_0 + \widetilde{L}_{h(n-2)}) \cdot (\widetilde{L}_{h(n-2)} - \widetilde{L}_0) \end{bmatrix} \right)$$
(By $k\Delta L = \widetilde{L}_k - \widetilde{L}_0$)

$$= \text{Softmax} \left( -\frac{\beta}{\Delta L} \begin{bmatrix} (-2x_i^\top w - 2t) \cdot \widetilde{L}_{(h-1)(n-2)-1} + (\widetilde{L}_{(h-1)(n-2)-1})^2 + (x_i^\top w + t)^2 \\ (-2x_i^\top w - 2t) \cdot \widetilde{L}_{(h-1)(n-2)} + (\widetilde{L}_{(h-1)(n-2)})^2 + (x_i^\top w + t)^2 \\ \vdots \\ (-2x_i^\top w - 2t) \cdot \widetilde{L}_{h(n-2)} + (\widetilde{L}_{h(n-2)})^2 + (x_i^\top w + t)^2 \end{bmatrix} \right)$$

$$= \text{Softmax} \left( -\frac{\beta}{\Delta L} \begin{bmatrix} (x_i^\top w + t - \widetilde{L}_{(h-1)(n-2)-1})^2 \\ (x_i^\top w + t - \widetilde{L}_{(h-1)(n-2)})^2 \\ \vdots \\ (x_i^\top w + t - \widetilde{L}_{h(n-2)})^2 \end{bmatrix} \right). \tag{D.6}$$

Here, the second-last equality arises from the fact that the softmax function is shift-invariant, allowing us to subtract and add a constant across all coordinates. To be more precise, we first expand the product for $k$-th coordinate of the column vector

$$(-2x_i^\top w - 2t + \widetilde{L}_0 + \widetilde{L}_k)(\widetilde{L}_k - \widetilde{L}_0)$$
$$= (-2x_i^\top w - 2t)L_k + L_0 L_k + L_k^2 - (-2x_i^\top w - 2t)L_0 - L_0^2 - L_0 L_k$$
$$= (-2x_i^\top w - 2t)L_k + L_k^2 - \underbrace{(-2x_i^\top w - 2t)L_0 - L_0^2}_{\text{constant across the column vector}}.$$

Then, dropping the constant and adding another constant $(x_i^\top w + t)^2$ across all coordinates, the above equation becomes

$$(-2x_i^\top w - 2t)L_k + L_k^2 + (x_i^\top w + t)^2 = (x_i^\top w + t - L_k)^2.$$

Hence we finish the derivation of (D.6). Thus we have

$$\text{Attn}_h(A(X))_{:,i} = V^{(h)} \text{Softmax} \left( -\frac{\beta}{\Delta L} \begin{bmatrix} (x_i^\top w + t - \widetilde{L}_{(h-1)(n-2)-1})^2 \\ (x_i^\top w + t - \widetilde{L}_{(h-1)(n-2)})^2 \\ \vdots \\ (x_i^\top w + t - \widetilde{L}_{h(n-2)})^2 \end{bmatrix} \right). \tag{D.7}$$

For a specific $h$, we calculate the result of (D.7) column by column. Let $X_i$ denote any column (token) of the matrix $X$. We partition the situation at each column (token) into three distinct cases:

- **Case 1:** $w^\top X_i + t$ is strictly within the interpolation range of $\text{Attn}_h$ ($X \in [\widetilde{L}_{(h-1)(n-2)}, \widetilde{L}_{h(n-2)-1}]$). This excludes the following range at the edge of the interpolation range of

$$[\widetilde{L}_{(h-1)(n-2)-1}, \widetilde{L}_{(h-1)(n-2)}] \cup [\widetilde{L}_{h(n-2)-1}, \widetilde{L}_{h(n-2)}].$$

- **Case 2:** $w^\top X_i + t$ is not within the interpolation range of $\text{Attn}_h$:

$$w^\top X_i + t \notin [\widetilde{L}_{(h-1)(n-2)-1}, \widetilde{L}_{h(n-2)}].$$

- **Case 3:** $w^\top X_i + t$ is on the edge (region) of the interpolation range of $\text{Attn}_h$:

$$w^\top X_i + t \in [\widetilde{L}_{(h-1)(n-2)-1}, \widetilde{L}_{(h-1)(n-2)}] \cup [\widetilde{L}_{h(n-2)-1}, \widetilde{L}_{h(n-2)}].$$

Two remarks are in order.

**Remark D.3** (Cases of a Single Head Attention). The $H$ heads split the approximation of the truncated linear map across disjoint intervals. For head $h$,

$$\|\text{Attn}_h(X) - \text{Range}_{[a+\frac{b-a}{p}((h-1)(n-2)-1),a+\frac{b-a}{p}h(n-2)]}(X)\|_\infty \leq \epsilon_1,$$

where $\epsilon > 0$ is arbitrarily small.
With this understanding, $w^\top X_i + t$:

- **Case 1:** falls into the interior of the interpolation range of the $h$-th head $\text{Attn}_h$, denoted as $\text{Range}_{[a+(b-a)((h-1)(n-2)-1)/p,a+(b-a)h(n-2)/p]}$.
- **Case 2:** remains outside of the interpolation range of the $h$-th head $\text{Attn}_h$.
- **Case 3:** falls on the boundary of the interpolation range of the $h$-th head $\text{Attn}_h$.

**Remark D.4** (Cases of All Attention Heads). According to Lemma D.2, for all heads in $\text{Attn}^H$, there are two possible cases:

- **Case 1*:** $x$ falls into Case 1 for a head, and Case 2 for all other heads.

- **Case 2*:** $x$ falls into Case 3 for two heads with adjacent interpolation ranges, and Case 2 for other heads.

This also means that when Case 1 appears in $\text{Attn}^H$, the situation of all head in $\text{Attn}^H$ falls into Case 1*. And when Case 3 appears in $\text{Attn}^H$, the situation of all head in $\text{Attn}^H$ falls into Case 2*. Thus, We discuss Case 2* in the discussion of Case 3.

**Case 1:** $X_i \in [\widetilde{L}_{(h-1)(n-2)}, \widetilde{L}_{h(n-2)-1}]$. In this case, our goal is to demonstrate this attention head outputs a value close to $\text{Range}_{[a,b]}(w^\top X_i + t)$.

Let $\widetilde{L}_s$ and $\widetilde{L}_{s+1}$ be the two interpolants such that

$$w^\top X_i + t \in [\widetilde{L}_s, \widetilde{L}_{s+1}]. \tag{D.8}$$

Then, $s$ and $s+1$ are also the labels of the two largest entries in

$$-\frac{\beta}{\Delta L} \begin{bmatrix} (w^\top X_i + t - \widetilde{L}_{(h-1)(n-2)-1})^2 \\ (w^\top X_i + t - \widetilde{L}_{(h-1)(n-2)})^2 \\ \vdots \\ (w^\top X_i + t - \widetilde{L}_{h(n-2)})^2 \end{bmatrix},$$

since

$$\operatorname*{argmax}_{k\in\{(h-1)(n-2)-1,h(n-2)\}} -\frac{\beta}{\Delta L}(w^\top X_i + t - \widetilde{L}_k)^2$$

$$= \operatorname*{argmin}_{k\in\{(h-1)(n-2)-1,h(n-2)\}} (w^\top X_i + t - \widetilde{L}_k)^2$$

$$= \operatorname*{argmin}_{k\in\{(h-1)(n-2)-1,h(n-2)\}} |w^\top X_i + t - \widetilde{L}_k|.$$

We also note that the distance of $w^\top X_i + t$ to interpolants beside $\widetilde{L}_s$ and $\widetilde{L}_{s+1}$ differs from $w^\top X_i + t$ for at least $\widetilde{L}_s - \widetilde{L}_{s-1} = (b-a)/p$ or $\widetilde{L}_{s+1} - \widetilde{L}_s = (b-a)/p$.

This is equivalent to the occasion when $x_1 - x_3$ in Lemma D.1 is larger than

$$\max\left\{ \frac{\beta}{\Delta L}(w^\top X_i + t - \widetilde{L}_{s-1})^2 - (w^\top X_i + t - \widetilde{L}_s)^2, \frac{\beta}{\Delta L}(w^\top X_i + t - \widetilde{L}_{s+2})^2 - (w^\top X_i + t - \widetilde{L}_{s+1})^2 \right\}$$

$$\geq \frac{\beta}{\Delta L} \cdot (\frac{b-a}{p})^2,$$

which is invariant to $X_i$.

Thus according to Lemma D.1 and the fact that the $s$ and $s+1$ are the two largest entries in the $i$-th column of the attention score matrix, we have

$$\left\| \operatorname{Softmax}\left( -\frac{\beta}{\Delta L} \begin{bmatrix} (w^\top X_i + t - \widetilde{L}_{(h-1)(n-2)-1})^2 \\ (w^\top X_i + t - \widetilde{L}_{(h-1)(n-2)})^2 \\ \vdots \\ (w^\top X_i + t - \widetilde{L}_{h(n-2)})^2 \end{bmatrix} \right) - \frac{1}{1+e^{-\beta\delta}} \underbrace{e_s}_{n\times 1} - \frac{e^{-\beta\delta}}{1+e^{-\beta\delta}} \underbrace{e_{s+1}}_{n\times 1} \right\|_\infty \leq \epsilon_2,$$

for any $\epsilon_2 > 0$.

This yields that

$$\left\| V\operatorname{Softmax}\left( -\frac{\beta}{\Delta L} \begin{bmatrix} (w^\top X_i + t - \widetilde{L}_{(h-1)(n-2)-1})^2 \\ (w^\top X_i + t - \widetilde{L}_{(h-1)(n-2)})^2 \\ \vdots \\ (w^\top X_i + t - \widetilde{L}_{h(n-2)})^2 \end{bmatrix} \right) - V\frac{1}{1+e^{-\beta\delta}}e_s - V\frac{e^{-\beta\delta}}{1+e^{-\beta\delta}}e_{s+1} \right\|_\infty$$

$$\leq \left\| \operatorname{Softmax}\left( -\frac{\beta}{\Delta L} \begin{bmatrix} (w^\top X_i + t - \widetilde{L}_{(h-1)(n-2)-1})^2 \\ (w^\top X_i + t - \widetilde{L}_{(h-1)(n-2)})^2 \\ \vdots \\ (w^\top X_i + t - \widetilde{L}_{h(n-2)})^2 \end{bmatrix} \right) - \frac{1}{1+e^{-\beta\delta}}e_s - \frac{e^{-\beta\delta}}{1+e^{-\beta\delta}}e_{s+1} \right\|_\infty \cdot \|V\|_\infty$$

$$\leq \|V\|_\infty \epsilon_2.$$

This is equivalent to

$$\left\| V\operatorname{Softmax}(K^\top Q)_{:,i} - \frac{1}{1+e^{-\beta\delta}}\widetilde{L}_{(h-1)(n-2)+s-1}e_{k_G} - \frac{e^{-\beta\delta}}{1+e^{-\beta\delta}}\widetilde{L}_{(h-1)(n-2)+s}e_{k_G} \right\|_\infty$$

$$\leq \|V\|_\infty \cdot \epsilon_2 \qquad\qquad\qquad\qquad \text{(By } \|AB\| \leq \|A\| \cdot \|B\|)$$

$$\leq m_v \epsilon_2, \tag{D.9}$$

where the last line is by (D.5).

From (D.8), we derive that

$$
\begin{aligned}
&\|\frac{1}{1+e^{-\beta\delta}}\widetilde{L}_{(h-1)(n-2)+s-1} + \frac{e^{-\beta\delta}}{1+e^{-\beta\delta}}\widetilde{L}_{(h-1)(n-2)+s} - (w^\top X_i + t)e_{k_G}\|_\infty \\
&\leq \|\frac{1}{1+e^{-\beta\delta}}(\widetilde{L}_{(h-1)(n-2)+s-1} - (w^\top X_i + t)e_{k_G})\|_\infty + \|\frac{e^{-\beta\delta}}{1+e^{-\beta\delta}}(\widetilde{L}_{(h-1)(n-2)+s} - (w^\top X_i + t))\|_\infty \\
&\qquad\qquad\qquad\qquad\qquad \text{\small(By convex combination of } (w^\top X_i + t) \text{ and triangle inequality)} \\
&\leq \frac{1}{1+e^{-\beta\delta}} \cdot \frac{b-a}{p} + \frac{e^{-\beta\delta}}{1+e^{-\beta\delta}} \cdot \frac{b-a}{p} \qquad\qquad\qquad\qquad\qquad\qquad \text{\small(By (D.8))} \\
&= \frac{b-a}{p}. \qquad\qquad\qquad\qquad\qquad\qquad\qquad\qquad\qquad\qquad\qquad\qquad \text{(D.10)}
\end{aligned}
$$

Combing (D.9) and (D.10) yields

$$
\begin{aligned}
&\|V\text{Softmax}(K^\top Q)_{:,i} - (w^\top X_i + t)\|_\infty \\
&\leq \|V\text{Softmax}(K^\top Q)_{:,i} - \frac{1}{1+e^{-\beta\delta}}\widetilde{L}_{(h-1)(n-2)+s-1} - \frac{e^{-\beta\delta}}{1+e^{-\beta\delta}}\widetilde{L}_{(h-1)(n-2)+s}\|_\infty \\
&\quad + \|\frac{1}{1+e^{-\beta\delta}}\widetilde{L}_{(h-1)(n-2)+s-1} + \frac{e^{-\beta\delta}}{1+e^{-\beta\delta}}\widetilde{L}_{(h-1)(n-2)+s} - (w^\top X_i + t)e_{k_G}\|_\infty \\
&\qquad\qquad\qquad\qquad\qquad\qquad\qquad\qquad\qquad\qquad \text{\small(By triangle inequality)} \\
&\leq m_v\epsilon_2 + \frac{b-a}{p}, \qquad\qquad\qquad\qquad\qquad\qquad\qquad\qquad\qquad\qquad\qquad \text{(D.11)}
\end{aligned}
$$

where the first inequality comes from adding and subtracting the interpolation points' convex combination and then applying triangle inequality.

**Case 2:** $X \notin [\widetilde{L}_{(h-1)(n-2)-1}, \widetilde{L}_{h(n-2)}]$.   In this case, $X_i$ falls out of the range of interpolation covered by $\text{Attn}_h$.

Without loss of generality, suppose $w^\top X_i + t$ to lie left to the range of interpolation of $\text{Attn}_h$.

This yields that $\widetilde{L}_{(h-1)(n-2)-1}$ is the closest interpolant within $\text{Attn}_h$ to $w^\top X_i + t$. Furthermore, the second closest interpolant $\widetilde{L}_{(h-1)(n-2)}$ is at least further for at least $(b-a)/p$, which is a constant irrelevant to $X_i$

Then by Lemma D.1, we have

$$
\left\| \text{Softmax}\left( -\frac{\beta}{\Delta L} \begin{bmatrix} (w^\top X_i + t - \widetilde{L}_{(h-1)(n-2)-1})^2 \\ (w^\top X_i + t - \widetilde{L}_{(h-1)(n-2)})^2 \\ \vdots \\ (w^\top X_i + t - \widetilde{L}_{h(n-2)})^2 \end{bmatrix} \right) - \underbrace{e_1}_{n\times 1} \right\|_\infty \leq \epsilon_3,
$$

for any $\epsilon_3 > 0$.

This yields that

$$
\begin{aligned}
&\left\| V\text{Softmax}\left( -\frac{\beta}{\Delta L} \begin{bmatrix} (w^\top X_i + t - \widetilde{L}_{(h-1)(n-2)-1})^2 \\ (w^\top X_i + t - \widetilde{L}_{(h-1)(n-2)})^2 \\ \vdots \\ (w^\top X_i + t - \widetilde{L}_{h(n-2)})^2 \end{bmatrix} \right) - V\underbrace{e_1}_{n\times 1} \right\|_\infty \\
&\leq \|V\|_\infty \cdot \epsilon_3 \qquad\qquad\qquad\qquad\qquad\qquad\qquad\qquad\qquad \text{\small(By } \|AB\| \leq \|A\| \cdot \|B\|) \\
&\leq m_v\epsilon_3,
\end{aligned}
$$

where the last line is by (D.5).

This is equivalent to

$$\left\| V\text{Softmax}\left( -\frac{\beta}{\Delta L} \begin{bmatrix} (w^\top X_i + t - \widetilde{L}_{(h-1)(n-2)-1})^2 \\ (w^\top X_i + t - \widetilde{L}_{(h-1)(n-2)})^2 \\ \vdots \\ (w^\top X_i + t - \widetilde{L}_{h(n-2)})^2 \end{bmatrix} \right) - 0_{d_o} \right\|_\infty \leq m_v \epsilon_3. \tag{D.12}$$

**Case 1\*.**   According to Lemma D.2, when Case 1 occurs for one head in the $H$ heads of $\text{Attn}^H$, all other head will be in Case 2.

Combining with the result in Case 2, we have the output of all heads as

$$\begin{aligned} & \|\text{Attn}^H(A(X))_{:,i} - (w^\top X_i + t)e_{k_G}\|_\infty \\ =\; & \|\sum_{h_0 \in [H]/\{h\}} \text{Attn}_{h_0} \circ A(X)_{:,i}\|_\infty + \|\text{Attn}_h \circ A(X)_{:,i} - (w^\top X_i + t)e_{k_G}\|_\infty \\ =\; & (H-1)m_v\epsilon_3 + m_v\epsilon_2 + \frac{b-a}{p} && (\text{By (D.11) and (D.12)}) \\ =\; & (H-1)m_v\epsilon_3 + m_v\epsilon_2 + \frac{b-a}{H(n-2)}. \end{aligned}$$

Setting $\epsilon_2, \epsilon_3$ to be

$$\epsilon_2 = \frac{\epsilon_0}{2}$$
$$\epsilon_3 = \frac{\epsilon_0}{2(H-1)m}$$

yields the final result.

**Case 3 (and Case 2\*):** $X \in [\widetilde{L}_{(h-1)(n-2)-1}, \widetilde{L}_{(h-1)(n-2)}] \cup [\widetilde{L}_{h(n-2)-1}, \widetilde{L}_{h(n-2)}]$.   In this case, $w^\top X_i + t$ is the boundary of the interpolation range of $\text{Attn}_{h_0}$. By Lemma D.2, it should also fall on the boundary of a head with neighboring interpolation range. Without loss of generality, we set it to be $\text{Attn}_{h_0-1}$. Furthermore, Lemma D.2 indicates that $w^\top X_i + t$ should fall on no other interpolation range of any heads beside $\text{Attn}_{h_0}$ and $\text{Attn}_{h_0-1}$.

Combining this with case 2, we have

$$\begin{aligned} \text{Attn}^H(A(X))_{:,i} &= \sum_{h=1}^H \text{Attn}_h \circ A(X)_{:,i} \\ &\in [(-(H-2)m_v\epsilon_3 + \text{Attn}_{h_0} \circ A(X)_{:,i} + \text{Attn}_{h_0-1} \circ A(X)_{:,i}), \\ &\qquad ((H-2)m_v\epsilon_3 + \text{Attn}_{h_0} \circ A(X)_{:,i} + \text{Attn}_{h_0-1} \circ A(X)_{:,i})]. \\ &\qquad\qquad\qquad\qquad\qquad\qquad\qquad\qquad\qquad\qquad (\text{By (D.12)}) \end{aligned}$$

By Lemma D.1, let $\delta$ denote

$$\delta = \widetilde{L}_{(h-1)(n-2)+s} - (w^\top X_i + t)e_{k_G} - [\widetilde{L}_{(h-1)(n-2)+s} - (w^\top X_i + t)e_{k_G}],$$

we have

$$\|\text{Softmax}((K^{(h)})^\top Q^{(h)}) - (\frac{1}{1+e^{-\beta\delta}}e_1 + \frac{e^{-\beta\delta}}{1+e^{-\beta\delta}}e_2)\| \leq \epsilon_4,$$

and

$$\|\text{Softmax}((K^{(h-1)})^\top Q^{(h-1)}) - (\frac{1}{1+e^{-\beta\delta}}e_{n-1} + \frac{e^{-\beta\delta}}{1+e^{-\beta\delta}}e_n)\| \leq \epsilon_5,$$

for any $\epsilon_4, \epsilon_5 > 0$.

Thus we have

$$\|V^{(h)}\text{Softmax}((K^{(h)})^\top Q^{(h)}) + V^{(h-1)}\text{Softmax}((K^{(h-1)})^\top Q^{(h-1)})$$
$$- V(\frac{1}{1+e^{-\beta\delta}}e_1 + \frac{e^{-\beta\delta}}{1+e^{-\beta\delta}}e_2 + \frac{1}{1+e^{-\beta\delta}}e_{n-1} + \frac{e^{-\beta\delta}}{1+e^{-\beta\delta}}e_n)\|_\infty$$
$$\leq \|V\|_\infty(\epsilon_4 + \epsilon_5).$$

This is equivalent to

$$\|V^{(h)}\text{Softmax}((K^{(h)})^\top Q^{(h)}) + V^{(h-1)}\text{Softmax}((K^{(h-1)})^\top Q^{(h-1)})$$
$$- (\frac{1}{1+e^{-\beta\delta}}\cdot 0 + \frac{e^{-\beta\delta}}{1+e^{-\beta\delta}}e_{k_G}\widetilde{L}_{(h-1)(n-2)+s} + \frac{1}{1+e^{-\beta\delta}}e_{k_G}\widetilde{L}_{(h-1)(n-2)+s-1} + \frac{e^{-\beta\delta}}{1+e^{-\beta\delta}}e_{k_G})\cdot 0\|_\infty$$
$$\leq \|V\|_\infty \cdot (\epsilon_4 + \epsilon_5).$$

Thus we have

$$\|V^{(h)}\text{Softmax}((K^{(h)})^\top Q^{(h)}) + V^{(h-1)}\text{Softmax}((K^{(h-1)})^\top Q^{(h-1)})$$
$$- (\frac{e^{-\beta\delta}}{1+e^{-\beta\delta}}e_{k_G}\widetilde{L}_{(h-1)(n-2)+s} + \frac{1}{1+e^{-\beta\delta}}e_{k_G}\widetilde{L}_{(h-1)(n-2)+s-1})\|_\infty$$
$$\leq \|V\|_\infty(\epsilon_4 + \epsilon_5),$$

which implies

$$\|\sum_{h=1}^{H}\text{Attn}_h(A(X))_{:,i} - (\frac{e^{-\beta\delta}}{1+e^{-\beta\delta}}e_{k_G}\widetilde{L}_{(h-1)(n-2)+s} + \frac{1}{1+e^{-\beta\delta}}e_{k_G}\widetilde{L}_{(h-1)(n-2)+s-1})\|_\infty$$
$$\leq (H-2)m_v\epsilon_3 + \|V\|_\infty(\epsilon_4 + \epsilon_5). \tag{D.13}$$

Finally, since

$$\|\frac{e^{-\beta\delta}}{1+e^{-\beta\delta}}e_{k_G}\widetilde{L}_{(h-1)(n-2)+s} + \frac{1}{1+e^{-\beta\delta}}e_{k_G}\widetilde{L}_{(h-1)(n-2)+s-1} - (w^\top X_i + t)e_{k_G}\|_\infty \leq \frac{b-a}{p},$$
$$\text{(By (D.10))}$$

combining with (D.13), we have

$$\|\sum_{h=1}^{H}\text{Attn}_h(A(X))_{:,i} - (w^\top X_i + t)e_{k_G}\|_\infty$$
$$\leq \|\sum_{h=1}^{H}\text{Attn}_h(A(X))_{:,i} - (\frac{e^{-\beta\delta}}{1+e^{-\beta\delta}}e_{k_G}\widetilde{L}_{(h-1)(n-2)+s} + \frac{1}{1+e^{-\beta\delta}}e_{k_G}\widetilde{L}_{(h-1)(n-2)+s-1})\|_\infty$$
$$+ \|(\frac{e^{-\beta\delta}}{1+e^{-\beta\delta}}e_{k_G}\widetilde{L}_{(h-1)(n-2)+s} + \frac{1}{1+e^{-\beta\delta}}e_{k_G}\widetilde{L}_{(h-1)(n-2)+s-1}) - (w^\top X_i + t)e_{k_G}\|_\infty$$
$$\text{(By triangle inequality)}$$
$$\leq \frac{b-a}{p} + (H-2)m_v\epsilon_3 + \|V\|_\infty(\epsilon_4 + \epsilon_5)$$

$$\leq \frac{b-a}{H(n-2)} + (H-2)\max\{|a|,|b|\}\epsilon_3 + \max\{|a|,|b|\}(\epsilon_4 + \epsilon_5).$$

Setting $\epsilon_3, \epsilon_4, \epsilon_5$ to be

$$\epsilon_3 = \frac{\epsilon_0}{3(H-2)}$$
$$\epsilon_4 = \epsilon_5 = \frac{\epsilon_0}{3}$$

yields the final result.

This completes the proof. $\qquad\square$

**Lemma D.3** (Attention Prepended with Token-Wise Linear Transformation is Still a Transformer). For any attention $\mathrm{Attn}$ and any linear transformation $A$, $\mathrm{Attn} \circ A$ is still an attention.

*Proof.* We denote the transformation matrix of $A$ also as $M_A$. Denote the attention $\mathrm{Attn}$ as

$$\mathrm{Attn}(Z) := W_V Z \mathrm{Softmax}((W_K Z)^\top W_Q Z).$$

Then we have

$$\mathrm{Attn} \circ A(Z) = W_V M_A Z \mathrm{Softmax}((W_K M_A Z)^\top W_Q M_A Z).$$

It is a new attention with parameters $W_K M_A, W_Q M_A$ and $W_V M_A$. $\qquad\square$

**Lemma D.4** (Lemma 14 in (Bai et al., 2023): Composition of Error for Approximating Convex GD). Suppose $f : \mathbb{R}^d \to \mathbb{R}$ is a convex function. Let $w^\star \in \mathrm{argmin}_{w \in \mathbb{R}^d} f(w)$, $R \geq 2\|w^\star\|_2$, and assume that $\nabla f$ is $L_f$-smooth on $B_2^d(R)$. Let sequences $\{\widehat{w}^\ell\}_{\ell \geq 0} \subset \mathbb{R}^d$ and $\{w_{\mathrm{GD}}^\ell\}_{\ell \geq 0} \subset \mathbb{R}^d$ be given by $\widehat{w}^0 = w_{\mathrm{GD}}^0 = \mathbf{0}$,

$$\begin{cases} \widehat{w}^{\ell+1} = \widehat{w}^\ell - \eta \nabla f(\widehat{w}^\ell) + \epsilon^\ell, & \|\epsilon^\ell\|_2 \leq \epsilon, \\ w_{\mathrm{GD}}^{\ell+1} = w_{\mathrm{GD}}^\ell - \eta \nabla f(w_{\mathrm{GD}}^\ell), \end{cases}$$

for all $\ell \geq 0$. Then as long as $\eta \leq 2/L_f$, for any $0 \leq L \leq R/(2\epsilon)$, it holds that $\|\widehat{w}^L - w_{\mathrm{GD}}^L\|_2 \leq L\epsilon$ and $\|\widehat{w}^L\|_2 \leq \frac{R}{2} + L\epsilon \leq R$.

**Corollary D.1.1** (Corollary A.2 in (Bai et al., 2023): Gradient Descent for Smooth and Strongly Convex Function). Suppose $L : \mathbb{R}^d \to \mathbb{R}$ is a $\alpha$-strongly convex and $\beta$-smooth for som $0 < \alpha \leq \beta$. Then, the gradient descent iterates $w_{\mathrm{GD}}^{(t+1)} := w_{\mathrm{GD}}^t - \eta \nabla L(w_{\mathrm{GD}}^L)$ with learning rate $\eta = 1/\beta$ and initialization $w_{\mathrm{GD}}^0 \in \mathbb{R}^d$ satisfies for any $t \geq 1$,

$$\|w_{\mathrm{GD}}^t - w^\star\|_2^2 \leq \exp\left(-\frac{t}{\kappa}\right) \cdot \|w_{\mathrm{GD}}^0 - w^\star\|_2^2,$$

$$L(w_{\mathrm{GD}}^t) - L(w^\star) \leq \frac{\beta}{2} \exp\left(-\frac{t}{\kappa}\right) \cdot \|w_{\mathrm{GD}}^0 - w^\star\|_2^2,$$

where $\kappa := \beta/\alpha$ is the condition number of $L$, and $w^\star := \mathrm{argmin}_{w \in \mathbb{R}^d} L(w)$.

## D.1 PROOF OF THEOREM 3.1

**Definition D.3** (Interpolation Points). Define $P + 1$ interpolation points of the effective domain of $f$, i.e., the range of $w^\top x - y$, as

$$L_j := L_{\min} + \frac{j}{P}(L_{\max} - L_{\min}), \quad \text{for} \quad j \in 0, 1, \ldots, P,$$

where $[L_{\min}, L_{\max}]$ is a bounded interval containing all values of $w^\top x - y$.

**Theorem D.2** (In-Context Emulation of $f(w^\top x - y)x$ with Single-Head Attention; Theorem 3.1 Restate). Let $[L_{\min}, L_{\max}]$ be a bounded interval containing all values of $w^\top x - y$, and let

$$X := \begin{bmatrix} x_1 & x_2 & \cdots & x_n \\ y_1 & y_2 & \cdots & y_n \end{bmatrix} \in \mathbb{R}^{(d+1)\times n} \quad \text{and} \quad W := \begin{bmatrix} w & w & \cdots & w \end{bmatrix} \in \mathbb{R}^{d\times n},$$

where $x_i \in \mathbb{R}^d$, $y_i \in \mathbb{R}$, and $w \in \mathbb{R}^d$ is the coefficient vector. Define the input as:

$$Z := \begin{bmatrix} x_1 & x_2 & \cdots & x_n \\ y_1 & y_2 & \cdots & y_n \\ w & w & \cdots & w \end{bmatrix} = \begin{bmatrix} X \\ W \end{bmatrix} \in \mathbb{R}^{(2d+1)\times n}. \tag{D.14}$$

Assume $\max\{\|X\|_\infty, \|W\|_\infty\} \leq B$. For any continuously differentiable function $f : \mathbb{R} \to \mathbb{R}$ and any $\epsilon > 0$, there exists a single-head attention $\text{Attn}_s$ with a linear layer Linear such that

$$\|\text{Attn}_s \circ \text{Linear}(Z) - \begin{bmatrix} f(w^\top x_1 - y_1)x_1 & \cdots & f(w^\top x_n - y_n)x_n \end{bmatrix}\|_\infty \leq \epsilon, \quad \text{for any} \quad \epsilon > 0.$$

*Proof.* We define the linear transformation of $Z$ as a concatenation of two functions with some manual padding of zeros:

$$\text{Linear}(Z) := \underbrace{\begin{bmatrix} \text{Linear}_x(X) \\ \text{Linear}_w(W) & 0_{(2d+n+2)\times nP} \end{bmatrix}}_{2(2d+n+2)\times n(P+1)},$$

where we define $\text{Linear}_x \in \mathbb{R}^{(2d+n+2)\times n(P+1)}$ and $\text{Linear}_w \in \mathbb{R}^{(2d+n+2)\times n}$ as below.

We define $\text{Linear}_w$ as:

$$\text{Linear}_w(W) := \underbrace{\begin{bmatrix} I_d \\ 0_{(d+n+2)\times d} \end{bmatrix}}_{(2d+n+2)\times d} \underbrace{W}_{d\times n} + \begin{bmatrix} 0_{d\times n} \\ -1_{1\times n} \\ 0_{d\times n} \\ 1_{1\times n} \\ I_n \end{bmatrix} = \underbrace{\begin{bmatrix} W \\ -1_{1\times n} \\ 0_{d\times n} \\ 1_{1\times n} \\ I_n \end{bmatrix}}_{(2d+n+2)\times n}.$$
$$\underbrace{\phantom{XXXXXXXXX}}_{(2d+n+2)\times n}$$

We define $\text{Linear}_x$ as:

$$\text{Linear}_x(X)$$
$$:= \sum_{i=1}^n \underbrace{\begin{bmatrix} I_{d+1} \\ 0_{(d+1+n)\times(d+1)} \end{bmatrix}}_{(2d+n+2)\times(d+1)} \underbrace{X}_{(d+1)\times n} \underbrace{\begin{bmatrix} 0_{n\times(i-1)(P+1)} & 2L_0 e_i^{(n)} & 2L_1 e_i^{(n)} & \cdots & 2L_P e_i^{(n)} & 0_{n\times(n-i)(P+1)} \end{bmatrix}}_{n\times n(P+1)}$$
$$+ \sum_{i=1}^n \underbrace{\begin{bmatrix} 0_{(d+1)\times d} & 0_{(d+1)} \\ I_d & 0_d \\ 0_{(n+1)\times d} & 0_{n+1} \end{bmatrix}}_{(2d+n+2)\times(d+1)} \underbrace{X}_{(d+1)\times n} \underbrace{\begin{bmatrix} 0_{n\times(i-1)(P+1)} & f(L_0)e_i^{(i)} & f(L_1)e_i^{(i)} & \cdots & f(L_P)e_i^{(i)} & 0_{n\times(n-i)(P+1)} \end{bmatrix}}_{n\times n(P+1)}$$

$$+ \underbrace{\begin{bmatrix} 0_{(2d+1)\times(P+1)} & \cdots & 0_{(2d+1)\times(P+1)} \\ S & \cdots & S \\ Ce_1^{(n)}1_{1\times(P+1)} & \cdots & Ce_n^{(n)}1_{1\times(P+1)} \end{bmatrix}}_{(2d+n+2)\times n(P+1)}$$

$$= \underbrace{\begin{bmatrix} T_1 & T_2 & \cdots & T_n \end{bmatrix}}_{(2d+n+2)\times n(P+1)},$$

where $\{L_j\}_{j=0}^P$ are the $P+1$ interpolation points (Definition D.3); $e_i^{(n)} \in \mathbb{R}^n$ is the one-hot vector with 1 at index $i$ and 0 elsewhere; $C$ is a constant to be determined later, and

$$1_{1\times(P+1)} := \underbrace{\begin{bmatrix} 1 & 1 & \cdots & 1 \end{bmatrix}}_{1\times(P+1)},$$

$$S := \underbrace{\begin{bmatrix} -L_0^2 & -L_1^2 & \cdots & L_P^2 \end{bmatrix}}_{1\times(P+1)},$$

$$T_i := \underbrace{\begin{bmatrix} 2L_0x_i & 2L_1x_i & \cdots & 2L_Px_i \\ 2L_0y_i & 2L_1y_i & \cdots & 2L_Py_i \\ f(L_0)x_i & f(L_1)x_i & \cdots & f(L_P)x_i \\ -L_0^2 & -L_1^2 & \cdots & -L_P^2 \\ Ce_i^{(n)} & Ce_i^{(n)} & \cdots & Ce_i^{(n)} \end{bmatrix}}_{(2d+n+2)\times(P+1)}.$$

So our $\mathrm{Linear}(Z)$ is:

$$\mathrm{Linear}(Z) = \underbrace{\begin{bmatrix} T_1 & T_2 & \cdots & T_n \\ W_{d\times n} & 0_{d\times nP} \\ -1_{1\times n} & 0_{1\times nP} \\ 0_{d\times n} & 0_{d\times nP} \\ 1_{1\times n} & 0_{1\times nP} \\ I_n & 0_{n\times nP} \end{bmatrix}}_{2(2d+n+2)\times n(P+1)} = \underbrace{\begin{bmatrix} 2L_0x_1 & \cdots & 2L_Px_1 & \cdots & \cdots & 2L_0x_n & \cdots & 2L_Px_n \\ 2L_0y_1 & \cdots & 2L_Py_1 & \cdots & \cdots & 2L_0y_n & \cdots & 2L_Py_n \\ f(L_0)x_1 & \cdots & f(L_P)x_1 & \cdots & \cdots & f(L_0)x_n & \cdots & f(L_P)x_n \\ -L_0^2 & \cdots & -L_P^2 & \cdots & \cdots & -L_0^2 & \cdots & -L_P^2 \\ Ce_1^{(n)} & \cdots & Ce_1^{(n)} & \cdots & \cdots & Ce_n^{(n)} & \cdots & Ce_n^{(n)} \\ & W_{d\times n} & & & & & 0_{d\times nP} \\ & -1_{1\times n} & & & & & 0_{1\times nP} \\ & 0_{d\times n} & & & & & 0_{d\times nP} \\ & 1_{1\times n} & & & & & 0_{1\times nP} \\ & I_n & & & & & 0_{n\times nP} \end{bmatrix}}_{2(2d+n+2)\times n(P+1)}.$$

Now we construct $W_K, W_Q, W_V, W_O$ to be:

$$W_K := \underbrace{\begin{bmatrix} I_{2d+n+2} & 0_{(2d+n+2)\times(2d+n+2)} \end{bmatrix}}_{(2d+n+2)\times 2(2d+n+2)},$$

$$W_Q := \underbrace{\begin{bmatrix} 0_{(2d+n+2)\times(2d+n+2)} & I_{2d+n+2} \end{bmatrix}}_{(2d+n+2)\times 2(2d+n+2)},$$

$$W_V := \underbrace{\begin{bmatrix} 0_{d\times(d+1)} & I_d & 0_{d\times(2d+2n+3)} \end{bmatrix}}_{d\times 2(2d+n+2)},$$

$$W_O := \underbrace{\begin{bmatrix} I_n \\ 0_{nP\times n} \end{bmatrix}}_{n(P+1)\times n}.$$

Thus,

$$W_K \text{Linear}(Z) = \underbrace{[T_1 \quad T_2 \quad \cdots \quad T_n]}_{(2d+n+2) \times n(P+1)}, \qquad \left(W_K \text{ selects the } T_i \text{ blocks in Linear}(Z)\right)$$

$$W_Q \text{Linear}(Z) = \underbrace{\begin{bmatrix} W_{d \times n} & 0_{d \times nP} \\ -1_{1 \times n} & 0_{1 \times nP} \\ 0_{d \times n} & 0_{d \times nP} \\ 1_{1 \times n} & 0_{1 \times nP} \\ I_n & 0_{n \times nP} \end{bmatrix}}_{(2d+n+2) \times n(P+1)}, \quad \left(W_Q \text{ selects the bottom } (2d+n+2) \text{ rows in Linear}(Z)\right)$$

$$W_V \text{Linear}(Z) = \underbrace{[F_1 \quad F_2 \quad \cdots \quad F_n]}_{d \times n(P+1)}, \qquad \left(W_V \text{ selects the } (d+2)\text{-th row in } T_i\right)$$

where we define $F_i$ as:

$$F_i := \underbrace{[f(L_0)x_i \quad f(L_1)x_i \quad \cdots \quad f(L_P)x_i]}_{d \times (P+1)}.$$

Therefore,

$$\text{Attn}_s \circ \text{Linear}(Z)$$
$$= W_V \text{Linear}(Z) \cdot \text{Softmax}_\beta((W_K \text{Linear}(Z))^\top W_Q \text{Linear}(Z)) \cdot W_O$$

$$= \underbrace{[F_1 \quad F_2 \quad \cdots \quad F_n]}_{d \times n(P+1)} \text{Softmax}_\beta(\overbrace{\underbrace{[T_1 \quad T_2 \quad \cdots \quad T_n]^\top}_{n(P+1) \times (2d+n+2)} \underbrace{\begin{bmatrix} W_{d \times n} & 0_{d \times nP} \\ -1_{1 \times n} & 0_{1 \times nP} \\ 0_{d \times n} & 0_{d \times nP} \\ 1_{1 \times n} & 0_{1 \times nP} \\ I_n & 0_{n \times nP} \end{bmatrix}}_{(2d+n+2) \times n(P+1)}}^{n(P+1) \times n(P+1)}) \underbrace{\begin{bmatrix} I_n \\ 0_{nP \times n} \end{bmatrix}}_{n(P+1) \times n}. \quad \text{(D.15)}$$

For simplicity of presentation, we define

$$\widetilde{T} := \underbrace{[T_1 \quad T_2 \quad \cdots \quad T_n]^\top}_{n(P+1) \times (2d+n+2)}.$$

For the $\text{Softmax}_\beta$ part in (D.15), we have:

$$\text{Softmax}_\beta((W_K \text{Linear}(Z))^\top W_Q \text{Linear}(Z)) \cdot W_O$$

$$= \text{Softmax}_\beta(\underbrace{\widetilde{T}}_{n(P+1) \times (2d+n+2)} \underbrace{\begin{bmatrix} W_{d \times n} & 0_{d \times nP} \\ -1_{1 \times n} & 0_{1 \times nP} \\ 0_{d \times n} & 0_{d \times nP} \\ 1_{1 \times n} & 0_{1 \times nP} \\ I_n & 0_{n \times nP} \end{bmatrix}}_{(2d+n+2) \times n(P+1)}) \underbrace{\begin{bmatrix} I_n \\ 0_{nP \times n} \end{bmatrix}}_{n(P+1) \times n} \qquad \left(\text{By the definition of } \widetilde{T}\right)$$

$$= \text{Softmax}_\beta \left( \left[ \widetilde{T} \begin{bmatrix} W_{d\times n} \\ -1_{1\times n} \\ 0_{d\times n} \\ 1_{1\times n} \\ I_n \end{bmatrix} \quad \widetilde{T} \begin{bmatrix} 0_{d\times nP} \\ 0_{1\times nP} \\ 0_{d\times nP} \\ 0_{1\times nP} \\ 0_{n\times nP} \end{bmatrix} \right] \right) \underbrace{\begin{bmatrix} I_n \\ 0_{nP\times n} \end{bmatrix}}_{n(P+1)\times n}$$

$$\underbrace{\phantom{XXXXX}}_{n(P+1)\times n} \quad \underbrace{\phantom{XXXXXX}}_{n(P+1)\times nP}$$

$$\left( \text{By distributivity of matrix multiplication over block concatenation} \right)$$

$$= \left[ \text{Softmax}_\beta(\widetilde{T} \begin{bmatrix} W_{d\times n} \\ -1_{1\times n} \\ 0_{d\times n} \\ 1_{1\times n} \\ I_n \end{bmatrix}) \quad \text{Softmax}_\beta(\widetilde{T} \begin{bmatrix} 0_{d\times nP} \\ 0_{1\times nP} \\ 0_{d\times nP} \\ 0_{1\times nP} \\ 0_{n\times nP} \end{bmatrix}) \right] \begin{bmatrix} I_n \\ 0_{nP\times n} \end{bmatrix}_{n(P+1)\times n}$$

$$\underbrace{\phantom{XXXXX}}_{n(P+1)\times n} \quad \underbrace{\phantom{XXXXXX}}_{n(P+1)\times nP}$$

$$\left( \text{By the column-wise operation nature of } \text{Softmax}_\beta \right)$$

$$= \text{Softmax}_\beta( \underbrace{[T_1 \quad T_2 \quad \cdots \quad T_n]^\top}_{n(P+1)\times(2d+n+2)} \underbrace{\begin{bmatrix} W_{d\times n} \\ -1_{1\times n} \\ 0_{d\times n} \\ 1_{1\times n} \\ I_n \end{bmatrix}}_{(2d+n+2)\times n} ). \qquad \left( \begin{bmatrix} I_n \\ 0_{nP\times n} \end{bmatrix} \text{ selects the first } \text{Softmax}_\beta \text{ block} \right)$$

Since our target is a token-wise approximation, we focus on a single token. We consider the $c$-th column ($c \in [n]$) in the $\text{Softmax}_\beta$ part, and we have

$$(\text{Softmax}_\beta((W_K \text{Linear}(Z))^\top W_Q \text{Linear}(Z)) \cdot W_O)_{:,c} = \text{Softmax}_\beta( \underbrace{\begin{bmatrix} T_1^\top \\ T_2^\top \\ \vdots \\ T_n^\top \end{bmatrix}}_{n(P+1)\times(2d+n+2)} \cdot \underbrace{\begin{bmatrix} w \\ -1 \\ 0_d \\ 1 \\ e_c^{(n)} \end{bmatrix}}_{(2d+n+2)\times 1} )$$

$$= \text{Softmax}_\beta( \underbrace{\begin{bmatrix} M_{1,c} \\ M_{2,c} \\ \vdots \\ M_{n,c} \end{bmatrix}}_{n(P+1)\times 1} ),$$

where each sub-block $M_{i,c} \in \mathbb{R}^{(P+1)\times 1}$ for $i \in [n]$ is

$$M_{i,c} := \underbrace{T_i^\top}_{(P+1)\times(2d+n+2)} \cdot \underbrace{\begin{bmatrix} w \\ -1 \\ 0_d \\ 1 \\ e_c^{(n)} \end{bmatrix}}_{(2d+n+2)\times 1}$$

$$= \underbrace{\begin{bmatrix} 2L_0 x_i^\top & 2L_0 y_i & f(L_0)x_i^\top & -L_0^2 & C(e_i^{(n)})^\top \\ 2L_1 x_i^\top & 2L_1 y_i & f(L_1)x_i^\top & -L_1^2 & C(e_i^{(n)})^\top \\ \vdots & \vdots & \vdots & \vdots & \vdots \\ 2L_P x_i^\top & 2L_P y_i & f(L_P)x_i^\top & -L_P^2 & C(e_i^{(n)})^\top \end{bmatrix}}_{(P+1)\times(2d+n+2)} \cdot \underbrace{\begin{bmatrix} w \\ -1 \\ 0_d \\ 1 \\ e_c^{(n)} \end{bmatrix}}_{(2d+n+2)\times 1} \qquad \left( \text{By transpose of } T_i \right)$$

$$= \underbrace{\begin{bmatrix} 2L_0 x_i^\top w - 2L_0 y_i - L_0^2 + C\mathbb{1}_{i=c} \\ 2L_1 x_i^\top w - 2L_1 y_i - L_1^2 + C\mathbb{1}_{i=c} \\ \vdots \\ 2L_P x_i^\top w - 2L_P y_i - L_P^2 + C\mathbb{1}_{i=c} \end{bmatrix}}_{(P+1)\times 1},$$

where $\mathbb{1}_{i=c}$ denotes the indicator function of $i = c$.

For simplicity, let

$$u_j^{(i)} := 2L_j x_i^\top w - 2L_j y_i - L_j^2, \quad \text{for} \quad j \in \{0, \dots, P\}, \tag{D.16}$$

such that

$$M_{i,c} = \underbrace{\begin{bmatrix} u_0^{(i)} + C\mathbb{1}_{i=c} \\ u_1^{(i)} + C\mathbb{1}_{i=c} \\ \vdots \\ u_P^{(i)} + C\mathbb{1}_{i=c} \end{bmatrix}}_{(P+1)\times 1}.$$

This means that

$$(\text{Softmax}_\beta((W_K \text{Linear}(Z))^\top W_Q \text{Linear}(Z)) \cdot W_O)_{:,c}$$

$$= \text{Softmax}(\beta \underbrace{\begin{bmatrix} M_{1,c} \\ M_{2,c} \\ \vdots \\ M_{n,c} \end{bmatrix}}_{n(P+1)\times 1})$$

$$= \sum_{i=1}^n \sum_{j=0}^P \frac{\exp\left\{\beta(u_j^{(i)} + C\mathbb{1}_{i=c})\right\}}{\sum_{i'=1}^n \sum_{j'=0}^P \exp\left\{\beta(u_{j'}^{(i')} + C\mathbb{1}_{i'=c})\right\}} \underbrace{e_{(i-1)(P+1)+(j+1)}^{(n(P+1))}}_{n(P+1)\times 1}.$$

$$\left(\text{By the definition of Softmax}_\beta\right)$$

Thus we have

$$\text{Attn}_s \circ \text{Linear}(Z)_{:,c}$$

$$= W_V \text{Linear}(Z) \cdot (\text{Softmax}_\beta((W_K \text{Linear}(Z))^\top W_Q \text{Linear}(Z)) \cdot W_O)_{:,c}$$

$$= \underbrace{\begin{bmatrix} F_1 & \cdots & F_n \end{bmatrix}}_{d\times n(P+1)} \cdot \sum_{i=1}^n \sum_{j=0}^P \frac{\exp\left\{\beta(u_j^{(i)} + C\mathbb{1}_{i=c})\right\}}{\sum_{i'=1}^n \sum_{j'=0}^P \exp\left\{\beta(u_{j'}^{(i')} + C\mathbb{1}_{i'=c})\right\}} \underbrace{e_{(i-1)(P+1)+(j+1)}^{(n(P+1))}}_{n(P+1)\times 1}$$

$$= \underbrace{\begin{bmatrix} f(L_0)x_1 & \cdots & f(L_P)x_1 & \cdots & f(L_0)x_n & \cdots & f(L_P)x_n \end{bmatrix}}_{d\times n(P+1)} \cdot \qquad \left(\text{By the definition of } F_i\right)$$

$$\sum_{i=1}^n \sum_{j=0}^P \frac{\exp\left\{\beta(u_j^{(i)} + C\mathbb{1}_{i=c})\right\}}{\sum_{i'=1}^n \sum_{j'=0}^P \exp\left\{\beta(u_{j'}^{(i')} + C\mathbb{1}_{i'=c})\right\}} \underbrace{e_{(i-1)(P+1)+(j+1)}^{(n(P+1))}}_{n(P+1)\times 1}$$

$$= \sum_{i=1}^n \sum_{j=0}^P \frac{\exp\left\{\beta(u_j^{(i)} + C\mathbb{1}_{i=c})\right\}}{\sum_{i'=1}^n \sum_{j'=0}^P \exp\left\{\beta(u_{j'}^{(i')} + C\mathbb{1}_{i'=c})\right\}}$$

$$\underbrace{[f(L_0)x_1 \quad \cdots \quad f(L_P)x_1 \quad \cdots \quad f(L_0)x_n \quad \cdots \quad f(L_P)x_n]}_{d \times n(P+1)} \cdot \underbrace{e^{(n(P+1))}_{(i-1)(P+1)+(j+1)}}_{n(P+1) \times 1}$$

$$\text{(By distributivity of matrix multiplication)}$$

$$= \sum_{i=1}^{n} \sum_{j=0}^{P} \frac{\exp\left\{\beta(u_j^{(i)} + C\mathbb{1}_{i=c})\right\}}{\sum_{i'=1}^{n} \sum_{j'=0}^{P} \exp\left\{\beta(u_{j'}^{(i')} + C\mathbb{1}_{i'=c})\right\}} f(L_j) \underbrace{x_i}_{d \times 1} \cdot$$

$$\text{(The one-hot vector retrieves the } ((i-1)(P+1)+(j+1))\text{-th column)}$$

Again, our goal is to approximate $f(x_c^\top w - y_c)x_c$ with:

$$\text{Attn}_s \circ \text{Linear}(Z)_{:,c} = \sum_{i=1}^{n} \sum_{j=0}^{P} \frac{\exp\left\{\beta(u_j^{(i)} + C\mathbb{1}_{i=c})\right\}}{\sum_{i'=1}^{n} \sum_{j'=0}^{P} \exp\left\{\beta(u_{j'}^{(i')} + C\mathbb{1}_{i'=c})\right\}} f(L_j)x_i. \quad (D.17)$$

We start to analyze the summation of weights $\sum_{j=0}^{P}(\cdots)$ for $i = c$ and $i \neq c$. We use the result of this analysis to bound our target approximation $\|\text{Attn}_s \circ \text{Linear}(Z)_{:,c} - f(x_c^\top w - y_c)x_c\|_\infty$ later.

- For every $i \in [n]$, if $i \neq c$, we have

$$\frac{\sum_{j=0}^{P} \exp\left\{\beta u_j^{(i)}\right\}}{\sum_{i'=1}^{n} \sum_{j'=0}^{P} \exp\left\{\beta(u_{j'}^{(i')} + C\mathbb{1}_{i'=c})\right\}} \qquad \text{(By } \mathbb{1}_{i \neq c} = 0 \text{ for } i \neq c\text{)}$$

$$< \frac{\sum_{j=0}^{P} \exp\left\{\beta u_j^{(i)}\right\}}{\sum_{j'=0}^{P} \exp\left\{\beta(u_{j'}^{(i')} + C)\right\}} \qquad (D.18)$$

$$\leq \epsilon_0, \qquad (D.19)$$

where (D.18) is by taking only the $i' = c$ term, and the last line is by the softmax-shift equality

$$\frac{\sum_{j=0}^{P} e^{u_j}}{\sum_{j'=0}^{P} e^{v_{j'}+C}} = \frac{\sum_{j=0}^{P} e^{u_j}}{e^C \sum_{j'=0}^{P} e^{v_{j'}}},$$

for any constant $C$ and choosing $C := M - \frac{1}{\beta}\ln\epsilon_0 = (\max_j L_j) \cdot (2dB^2 + 2B) - \frac{1}{\beta}\ln\epsilon_0$ with $\epsilon_0 > 0$.[5]

Thus, the weight assigned to $i \neq c$ is tiny.

---

[5]More explicitly, recall (D.16): $u_j^{(i)} := 2L_j x_i^\top w - 2L_j y_i - L_j^2$. Since $\max\{\|X\|_\infty, \|W\|_\infty\} \leq B$, we have

$$\|x_i\|_\infty \leq B, \quad |y_i| \leq B, \quad \|w\|_\infty \leq B,$$

which implies $\|w\|_1 \leq dB$. Let $L_\star := \max_j |L_j|$. For a fixed pair of $i, i'$, we have

$$u_j^{(i)} - u_j^{(i')} = 2L_j \cdot ((x_i - x_{i'})^\top w - (y_i - y_{i'}))$$

$$\leq 2|L_j| \cdot (|(x_i - x_{i'})^\top w| + |(y_i - y_{i'})|) \qquad \text{(By triangle inequality)}$$

$$\leq 2L_\star \cdot (\|x_i - x_{i'}\|_\infty \cdot \|w\|_1 + |(y_i - y_{i'})|) \quad \text{(By } L_\star := \max_j |L_j| \text{ and Hölder's inequality)}$$

$$\leq 2L_\star \cdot ((\|x_i\|_\infty + \|x_{i'}\|_\infty) \cdot \|w\|_1 + |y_i| + |y_{i'}|) \qquad \text{(By triangle inequality)}$$

$$\leq 2L_\star \cdot ((2B) \cdot dB + 2B) \qquad \text{(By } \|x_i\|_\infty \leq B, |y| \leq B \text{ and } \|w\|_1 \leq dB\text{)}$$

$$\leq 2L_\star \cdot (2dB^2 + 2B) =: M.$$

- For $i = c$, we have

$$\frac{\sum_{j=0}^{P} \exp\left\{\beta(u_j^{(c)} + C\right\})}{\sum_{i'=1}^{n} \sum_{j'=0}^{P} \exp\left\{\beta(u_{j'}^{(i')} + C\mathbb{1}_{i'=c})\right\}}$$

$$= 1 - \frac{\sum_{i\neq c}^{n} \sum_{j=0}^{P} \exp\left\{\beta(u_j^{(i)} + C\mathbb{1}_{i=c})\right\}}{\sum_{i'=1}^{n} \sum_{j'=0}^{P} \exp\left\{\beta(u_{j'}^{(i')} + C\mathbb{1}_{i'=c})\right\}}$$

$$\left(\text{By } \sum_{i\neq c}^{n} \sum_{j=0}^{P}(\cdots) + \sum_{i=c}^{n} \sum_{j=0}^{P}(\cdots) = 1\right)$$

$$\geq 1 - (n-1)\epsilon_0, \tag{D.21}$$

where the last inequality follows from (i.e., (D.19))

$$\frac{\sum_{i\neq c}^{n} \sum_{j=0}^{P} \exp\left\{\beta(u_j^{(i)} + C\mathbb{1}_{i=c})\right\}}{\sum_{i'=1}^{n} \sum_{j'=0}^{P} \exp\left\{\beta(u_{j'}^{(i')} + C\mathbb{1}_{i'=c})\right\}} \leq (n-1)\epsilon_0,$$

and setting $0 < \epsilon_0 < 1/(n-1)$. Therefore, the weight concentrates at $i = c$.

From (D.19), (D.21), and our target approximation

$$\|\mathrm{Attn}_s \circ \mathrm{Linear}(Z)_{:,c} - f(x_c^\top w - y_c)x_c\|_\infty$$

$$= \|\sum_{i=1}^{n} \sum_{j=0}^{P} \frac{\exp\left\{\beta(u_j^{(i)} + C\mathbb{1}_{i=c})\right\}}{\sum_{i'=1}^{n} \sum_{j'=0}^{P} \exp\left\{\beta(u_{j'}^{(i')} + C\mathbb{1}_{i'=c})\right\}} f(L_j)x_i - f(x_c^\top w - y_c)x_c\|_\infty, \tag{D.22}$$

we split (D.22) into two terms

$$\|\sum_{i=1}^{n} \sum_{j=0}^{P} \frac{\exp\left\{\beta(u_j^{(i)} + C\mathbb{1}_{i=c})\right\}}{\sum_{i'=1}^{n} \sum_{j'=0}^{P} \exp\left\{\beta(u_{j'}^{(i')} + C\mathbb{1}_{i'=c})\right\}} f(L_j)x_i - f(x_c^\top w - y_c)x_c\|_\infty$$

$$= \|\sum_{i\neq c}^{n} \sum_{j=0}^{P} \frac{\exp\left\{\beta(u_j^{(i)})\right\}}{\sum_{i'=1}^{n} \sum_{j'=0}^{P} \exp\left\{\beta(u_{j'}^{(i')} + C\mathbb{1}_{i'=c})\right\}} f(L_j)x_i$$

$$+ \sum_{j=0}^{P} \frac{\exp\left\{\beta(u_j^{(i)} + C)\right\}}{\sum_{i'=1}^{n} \sum_{j'=0}^{P} \exp\left\{\beta(u_{j'}^{(i')} + C\mathbb{1}_{i'=c})\right\}} f(L_j)x_i - f(x_c^\top w - y_c)x_c\|_\infty$$

$$\left(\text{By splitting the summation over } i \text{ into two parts: } i = c \text{ and } i \neq c\right)$$

---

Hence, we have $u_j^{(i)} \leq u_j^{(i')} + M$, which implies

$$e^{\beta u_j^{(i)}} \leq e^{\beta M} e^{\beta u_j^{(i')}}, \quad \text{for all} \quad j \in \{0, \dots, P\}. \tag{D.20}$$

Then, (D.18) becomes, for any constant $C$,

$$\frac{\sum_{j=0}^{P} \exp\left\{\beta u_j^{(i)}\right\}}{\exp\{\beta C\} \sum_{j'=0}^{P} \exp\left\{\beta u_{j'}^{(i')}\right\}} \leq \frac{e^{\beta M} \sum_{j=0}^{P} \exp\left\{\beta u_j^{(i')}\right\}}{e^{\beta C} \sum_{j'=0}^{P} \exp\left\{\beta u_{j'}^{(i')}\right\}} = e^{\beta(M-C)}.$$

Choosing $C := M - \frac{1}{\beta}\ln\epsilon_0 = (\max_j L_j) \cdot (2dB^2 + 2B) - \frac{1}{\beta}\ln\epsilon_0$, we obtain the desired bound $\epsilon_0$.

$$\leq \|\underbrace{\sum_{i\neq c}^{n}\sum_{j=0}^{P}\frac{\exp\left\{\beta(u_j^{(i)})\right\}}{\sum_{i'=1}^{n}\sum_{j'=0}^{P}\exp\left\{\beta(u_{j'}^{(i')}+C\mathbb{1}_{i'=c})\right\}}f(L_j)x_i}_{(I)}\|_\infty \tag{D.23}$$

$$+ \|\underbrace{\sum_{j=0}^{P}\frac{\exp\left\{\beta(u_j^{(i)}+C)\right\}}{\sum_{i'=1}^{n}\sum_{j'=0}^{P}\exp\left\{\beta(u_{j'}^{(i')}+C\mathbb{1}_{i'=c})\right\}}f(L_j)x_i - f(x_c^\top w - y_c)x_c}_{(II)}\|_\infty,$$

$$\text{(By triangle inequality)}$$

where we are capable of bounding term $(I)$ with (D.19) and term $(II)$ as follows.

For term $(I)$ in (D.23), we have

$$(I)$$

$$= \|\sum_{i\neq c}^{n}\sum_{j=0}^{P}\frac{\exp\left\{\beta(u_j^{(i)})\right\}}{\sum_{i'=1}^{n}\sum_{j'=0}^{P}\exp\left\{\beta(u_{j'}^{(i')}+C\mathbb{1}_{i'=c})\right\}}f(L_j)x_i\|_\infty$$

$$\leq \sum_{i\neq c}^{n}\sum_{j=0}^{P}\|\frac{\exp\left\{\beta(u_j^{(i)})\right\}}{\sum_{i'=1}^{n}\sum_{j'=0}^{P}\exp\left\{\beta(u_{j'}^{(i')}+C\mathbb{1}_{i'=c})\right\}}f(L_j)x_i\|_\infty \qquad \text{(By triangle inequality)}$$

$$= \sum_{i\neq c}^{n}\sum_{j=0}^{P}\frac{\exp\left\{\beta(u_j^{(i)})\right\}}{\sum_{i'=1}^{n}\sum_{j'=0}^{P}\exp\left\{\beta(u_{j'}^{(i')}+C\mathbb{1}_{i'=c})\right\}}\|f(L_j)x_i\|_\infty \quad \text{(By non-negativity of exponential)}$$

$$= \sum_{i\neq c}^{n}\sum_{j=0}^{P}\frac{\exp\left\{\beta(u_j^{(i)})\right\}}{\sum_{i'=1}^{n}\sum_{j'=0}^{P}\exp\left\{\beta(u_{j'}^{(i')}+C\mathbb{1}_{i'=c})\right\}}|f(L_j)|\cdot\|x_i\|_\infty$$

$$\text{(By }\|f(L_j)x_i\|_\infty = |f(L_j)|\cdot\|x_i\|_\infty)$$

$$\leq \frac{\sum_{i\neq c}^{n}\sum_{j=0}^{P}\exp\left\{\beta(u_j^{(i)})\right\}}{\sum_{i'=1}^{n}\sum_{j'=0}^{P}\exp\left\{\beta(u_{j'}^{(i')}+C\mathbb{1}_{i'=c})\right\}}B_f\cdot B$$

$$\text{(By }B_f := \max|f| \text{ and } \max\{\|X\|_\infty, \|W\|_\infty\}\leq B)$$

$$\leq (n-1)\epsilon_0 B_f B, \qquad\qquad\qquad\qquad\qquad\qquad\qquad\qquad\qquad \text{(By (D.19))}$$

where we define $B_f := \max|f|$ as the bound for $f$.

For term $(II)$ in (D.23), we have

$$(II)$$

$$= \|\sum_{j=0}^{P}\frac{\exp\left\{\beta(u_j^{(c)}+C)\right\}}{\sum_{i'=1}^{n}\sum_{j'=0}^{P}\exp\left\{\beta(u_{j'}^{(i')}+C\mathbb{1}_{i'=c})\right\}}f(L_j)\underbrace{x_c}_{d\times 1} - f(x_c^\top w - y_c)\underbrace{x_c}_{d\times 1}\|_\infty$$

$$= \|\sum_{j=0}^{P}\frac{\exp\left\{\beta(u_j^{(c)}+C)\right\}}{\sum_{i'=1}^{n}\sum_{k=0}^{P}\exp\left\{\beta(u_k^{(i')}+C\mathbb{1}_{i'=c})\right\}}\cdot(f(L_j)x_c - f(x_c^\top w - y_c)x_c)$$

$$- (1 - \frac{\sum_{j'=0}^{P}\exp\left\{\beta(u_{j'}^{(c)}+C)\right\}}{\sum_{i'=1}^{n}\sum_{k=0}^{P}\exp\left\{\beta(u_k^{(i')}+C\mathbb{1}_{i'=c})\right\}})\cdot f(x_c^\top w - y_c)x_c\|_\infty$$

$$\text{(By }\sum_j \frac{A_j}{C}B_j - D = \sum_j \frac{A_j}{C}(B_j - D) - (1 - \frac{\sum_{j'}A_{j'}}{C})D)$$

$$
\begin{aligned}
= \| & \sum_{j=0}^{P} \frac{\exp\left\{\beta(u_j^{(c)} + C)\right\}}{\sum_{j'=0}^{P} \exp\left\{\beta(u_{j'}^{(c)} + C)\right\}} \cdot \frac{\sum_{j'=0}^{P} \exp\left\{\beta(u_{j'}^{(c)} + C)\right\}}{\sum_{i'=1}^{n} \sum_{k=0}^{P} \exp\left\{\beta(u_k^{(i')} + C\mathbb{1}_{i'=c})\right\}} \cdot (f(L_j)x_c - f(x_c^\top w - y_c)x_c) \\
& - (1 - \frac{\sum_{j'=0}^{P} \exp\left\{\beta(u_{j'}^{(c)} + C)\right\}}{\sum_{i'=1}^{n} \sum_{k=0}^{P} \exp\left\{\beta(u_k^{(i')} + C\mathbb{1}_{i'=c})\right\}}) \cdot f(x_c^\top w - y_c)x_c \|_\infty \\
& \qquad\qquad \left(\text{By } \sum_j \tfrac{A_j}{C}(B_j - D) - (1 - \tfrac{\sum_{j'} A_{j'}}{C})D = \sum_j \tfrac{A_j}{E} \cdot \tfrac{E}{C}(B_j - D) - (1 - \tfrac{\sum_{j'} A_{j'}}{C})D\right)
\end{aligned}
$$

$$
\begin{aligned}
\leq \| & \sum_{j=0}^{P} \frac{\exp\left\{\beta(u_j^{(c)} + C)\right\}}{\sum_{j'=0}^{P} \exp\left\{\beta(u_{j'}^{(c)} + C)\right\}} \cdot (f(L_j) - f(x_c^\top w - y_c))x_c \\
& - (1 - \frac{\sum_{j'=0}^{P} \exp\left\{\beta(u_{j'}^{(c)} + C)\right\}}{\sum_{i'=1}^{n} \sum_{k=0}^{P} \exp\left\{\beta(u_k^{(i')} + C\mathbb{1}_{i'=c})\right\}}) \cdot f(x_c^\top w - y_c)x_c \|_\infty \qquad\qquad \left(\text{By } \tfrac{E}{C} < 1\right)
\end{aligned}
$$

$$
\begin{aligned}
\leq & \sum_{j=0}^{P} \frac{\exp\left\{\beta(u_j^{(c)} + C)\right\}}{\sum_{j'=0}^{P} \exp\left\{\beta(u_{j'}^{(c)} + C)\right\}} \cdot |f(L_j) - f(x_c^\top w - y_c)| \cdot \|x_c\|_\infty \\
& - (1 - \frac{\sum_{j'=0}^{P} \exp\left\{\beta(u_{j'}^{(c)} + C)\right\}}{\sum_{i'=1}^{n} \sum_{k=0}^{P} \exp\left\{\beta(u_k^{(i')} + C\mathbb{1}_{i'=c})\right\}}) \cdot |f(x_c^\top w - y_c)| \cdot \|x_c\|_\infty \\
& \qquad\qquad \left(\text{By triangle inequality, and } \|av\|_\infty \leq |a| \cdot \|v\|_\infty \text{ where } a \in \mathbb{R} \text{ and } v \in \mathbb{R}^d\right)
\end{aligned}
$$

$$
\begin{aligned}
\leq & \sum_{j=0}^{P} \frac{\exp\left\{\beta(u_j^{(c)} + C)\right\}}{\sum_{j'=0}^{P} \exp\left\{\beta(u_{j'}^{(c)} + C)\right\}} \cdot |f(L_j) - f(x_c^\top w - y_c)| \cdot \|x_c\|_\infty + (n-1)\epsilon_0 B_f \|x_c\|_\infty \\
& \qquad\qquad\qquad\qquad\qquad\qquad \left(\text{By (D.21) and } B_f := \max|f|\right)
\end{aligned}
$$

$$
\begin{aligned}
= & \sum_{j=0}^{P} \frac{\exp\left\{\beta u_j^{(c)}\right\}}{\sum_{j'=0}^{P} \exp\left\{\beta u_{j'}^{(c)}\right\}} \cdot |f(L_j) - f(x_c^\top w - y_c)| \cdot \|x_c\|_\infty + (n-1)\epsilon_0 B_f B \\
& \qquad \left(\text{By } \exp\left\{\beta(u_j^{(c)} + C)\right\} = \exp\left\{\beta u_j^{(c)}\right\} \exp\{\beta C\} \text{ and } \max\{\|X\|_\infty, \|W\|_\infty\} \leq B\right)
\end{aligned}
$$

$$
= \underbrace{\sum_{j=0}^{P} \frac{\exp\{-\beta(x_c^\top w - y_c - L_j)^2\}}{\sum_{j'=0}^{P} \exp\{-\beta(x_c^\top w - y_c - L_{j'})^2\}} \cdot |f(L_j) - f(x_c^\top w - y_c)| \cdot \|x_c\|_\infty}_{:=(II\text{-}1)} + \underbrace{(n-1)\epsilon_0 B_f B}_{:=(II\text{-}2)},
$$

$$(D.24)$$

where the last equality follows from completing the square

$$
u_j^{(c)} = 2L_j x_c^\top w - 2L_j y_c - L_j^2 = -(L_j - (x_c^\top w - y_c))^2 + (x_c^\top w - y_c)^2.
$$

For term $(II\text{-}1)$ in (D.24), we have

$$
\begin{aligned}
& (II\text{-}1) \\
= & \sum_{j=0}^{P} \frac{\exp\{-\beta(x_c^\top w - y_c - L_j)^2\}}{\sum_{j'=0}^{P} \exp\{-\beta(x_c^\top w - y_c - L_{j'})^2\}} \cdot |f(L_j) - f(x_c^\top w - y_c)| \cdot \|x_c\|_\infty \\
= & \sum_{j: |L_j - (x_c^\top w - y_c)| \leq \Delta L} \frac{\exp\{-\beta(x_c^\top w - y_c - L_j)^2\}}{\sum_{j'=0}^{P} \exp\{-\beta(x_c^\top w - y_c - L_{j'})^2\}} \cdot |f(L_j) - f(x_c^\top w - y_c)| \cdot \|x_c\|_\infty
\end{aligned}
$$

$$+ \sum_{j:|L_j-(x_c^\top w - y_c)|>\Delta L} \frac{\exp\{-\beta(x_c^\top w - y_c - L_j)^2\}}{\sum_{j'=0}^{P}\exp\{-\beta(x_c^\top w - y_c - L_{j'})^2\}} \cdot |f(L_j) - f(x_c^\top w - y_c)| \cdot \|x_c\|_\infty,$$

$$(D.25)$$

where we define $\Delta L := (L_{\max} - L_{\min})/P$ and divide the interpolation points into two groups with one group at least $\Delta L$ away from $x_c^\top w - y_c$, and the other within $\Delta L$.

For the first term in (D.25), we set $\Delta L$ to be sufficiently small ($P$ large enough) such that,

$$|f(t) - f(t')| \le \epsilon_1, \quad \forall\, \epsilon_1 > 0,$$

when $|t - t'| \le \Delta L$.

For the second term in (D.25), we set $\beta$ to be sufficiently large such that

$$\sum_{j:|L_j-(x_c^\top w - y_c)|>\Delta L} \frac{\exp\{-\beta(x_c^\top w - y_c - L_j)^2\}}{\sum_{j'=0}^{P}\exp\{-\beta(x_c^\top w - y_c - L_{j'})^2\}} \le \epsilon_2, \qquad (D.26)$$

for any $0 < \epsilon_2 < 1$.

Thus, for term $(II\text{-}1)$, we have

$$(II\text{-}1)$$

$$= \sum_{j:|L_j-(x_c^\top w - y_c)|\le\Delta L} \frac{\exp\{-\beta(x_c^\top w - y_c - L_j)^2\}}{\sum_{j'=0}^{P}\exp\{-\beta(x_c^\top w - y_c - L_{j'})^2\}} \cdot |f(L_j) - f(x_c^\top w - y_c)| \cdot \|x_c\|_\infty$$

$$+ \sum_{j:|L_j-(x_c^\top w - y_c)|>\Delta L} \frac{\exp\{-\beta(x_c^\top w - y_c - L_j)^2\}}{\sum_{j'=0}^{P}\exp\{-\beta(x_c^\top w - y_c - L_{j'})^2\}} \cdot |f(L_j) - f(x_c^\top w - y_c)| \cdot \|x_c\|_\infty$$

$$\le \sum_{j:|L_j-(x_c^\top w - y_c)|\le\Delta L} \frac{\exp\{-\beta(x_c^\top w - y_c - L_j)^2\}}{\sum_{j'=0}^{P}\exp\{-\beta(x_c^\top w - y_c - L_{j'})^2\}} \cdot \epsilon_1 \cdot B$$

$$+ \sum_{j:|L_j-(x_c^\top w - y_c)|>\Delta L} \frac{\exp\{-\beta(x_c^\top w - y_c - L_j)^2\}}{\sum_{j'=0}^{P}\exp\{-\beta(x_c^\top w - y_c - L_{j'})^2\}} \cdot (|f(L_j)| + |f(x_c^\top w - y_c)|) \cdot B$$

$$\qquad (\text{By } |f(L_j) - f(x_c^\top w - y_c)| < \epsilon_1, \max\{\|X\|_\infty, \|W\|_\infty\} \le B, \text{ and triangle inequality})$$

$$\le \sum_{j:|L_j-(x_c^\top w - y_c)|\le\Delta L} \frac{\exp\{-\beta(x_c^\top w - y_c - L_j)^2\}}{\sum_{j'=0}^{P}\exp\{-\beta(x_c^\top w - y_c - L_{j'})^2\}} \cdot \epsilon_1 \cdot B$$

$$+ \sum_{j:|L_j-(x_c^\top w - y_c)|>\Delta L} \frac{\exp\{-\beta(x_c^\top w - y_c - L_j)^2\}}{\sum_{j'=0}^{P}\exp\{-\beta(x_c^\top w - y_c - L_{j'})^2\}} \cdot 2B_f \cdot B \qquad (\text{By } B_f := \max|f|)$$

$$\le \underbrace{\sum_{j:|L_j-(x_c^\top w - y_c)|\le\Delta L} \frac{\exp\{-\beta(x_c^\top w - y_c - L_j)^2\}}{\sum_{j'=0}^{P}\exp\{-\beta(x_c^\top w - y_c - L_{j'})^2\}}}_{:=\kappa} \cdot \epsilon_1 \cdot B + \epsilon_2 \cdot 2B_f \cdot B \qquad (\text{By (D.26)})$$

$$\le \epsilon_1 \cdot B + \epsilon_2 \cdot 2B_f \cdot B. \qquad\qquad (\text{By } \kappa < 1)$$

Combining $(I)$, $(II\text{-}1)$, and $(II\text{-}2)$, we have:

$$\|\text{Attn}_s \circ \text{Linear}(Z)_{:,c} - f(x_c^\top w - y_c)x_c\|_\infty \le \underbrace{(n-1)\epsilon_0 B_f B}_{\text{from } (I)} + \underbrace{\epsilon_1 B + 2\epsilon_2 B_f B}_{\text{from } (II\text{-}1)} + \underbrace{(n-1)\epsilon_0 B_f B}_{\text{from } (II\text{-}2)}.$$

Since $\epsilon_0$, $\epsilon_1$ and $\epsilon_2$ are arbitrarily small, we have

$$\|\mathrm{Attn}_s \circ \mathrm{Linear}(Z)_{:,c} - f(x_c^\top w - y_c)x_c\|_\infty \le \epsilon,$$

for any $\epsilon > 0$.

This completes the proof. $\qquad\square$

## D.2    PROOF OF COROLLARY 3.1.2

**Corollary D.2.1** (In-Context Emulation of a Single GD Step; Corollary 3.1.2 Restate).    Let $\ell : \mathbb{R} \to \mathbb{R}$ be differentiable and define $\widehat{L}_n(w) := \frac{1}{n} \sum_{i=1}^n \ell(w^\top x_i - y_i)$. For any step size $\eta > 0$ and any $\epsilon > 0$, there exist a single-head attention $\mathrm{Attn}_s$ and a linear map $\mathrm{Linear}$ such that, with $Z = [X; W]$ as in (3.1), choosing the readout $u := \frac{1}{n}\mathbf{1}_n$ (equivalently, right-multiply by $W_O = u$ in Definition 2.1), we have

$$\widehat{w}_{\mathrm{GD}} := \underbrace{w + \overbrace{(\mathrm{Attn}_s \circ \mathrm{Linear}(Z))u}^{\widehat{G}u=\widehat{g}} \in \mathbb{R}^d}_{w - \eta\widehat{g}} \quad \text{and} \quad \|\widehat{w}_{\mathrm{GD}} - \underbrace{(w - \eta\nabla\widehat{L}_n(w))}_{w_{\mathrm{GD}}^+}\|_\infty \le \epsilon.$$

*Proof.* From Theorem 3.1 and Corollary 3.1.1, let $f(\cdot) := -\eta\ell'(\cdot)$, and we derive that $\|(\mathrm{Attn}_s \circ \mathrm{Linear})_i - (-\eta\nabla\ell(w^\top x_i - y_i)x_i)\|_\infty \le \epsilon$ for all $i \in [n]$. Let readout $u := \frac{1}{n}\mathbf{1}_n$. Then, we have

$$\begin{aligned}
\mathrm{Attn}_s \circ \mathrm{Linear}(Z))u &= \frac{1}{n}\sum_{i=1}^n (\mathrm{Attn}_s \circ \mathrm{Linear}(Z))_i \\
&= \frac{1}{n}\sum_{i=1}^n -\eta\nabla\ell(w^\top x_i - y_i)x_i + \epsilon' \\
&= -\eta\nabla\widehat{L}_n(w) + \epsilon'
\end{aligned}$$

Therefore,

$$\begin{aligned}
\widehat{w}_{\mathrm{GD}} &= w + \frac{1}{n}\sum_{i=1}^n (\mathrm{Attn}_s \circ \mathrm{Linear}(Z))_i \\
&= w - \eta\nabla\widehat{L}_n(w) + \epsilon' \\
&= w_{\mathrm{GD}}^+ + \epsilon',
\end{aligned}$$

where $\widehat{w}_{\mathrm{GD}}$ is the approximate and $w_{\mathrm{GD}}^+$ is the exact GD iterate. This completes the proof. $\quad\square$

## D.3    PROOF OF COROLLARY 3.1.3

**Theorem D.3** (In-Context Emulation of Linear Regression; Corollary 3.1.3 Restate).    For any dataset $\{(x_i, y_i)\}_{i=1}^n$ with $x_i \in \mathbb{R}^d$, $y_i \in \mathbb{R}$ and any $\epsilon > 0$, there exists a depth-$L$ stack of single-head attention layers $\{\mathrm{Attn}_s^l\}_{l=0}^{L-1}$, linear maps $\{\mathrm{Linear}^l\}_{l=0}^{L-1}$, and a readout $u \in \mathbb{R}^n$ such that the iterates

$$\widehat{w}_{\mathrm{linear}}^{l+1} := \widehat{w}_{\mathrm{linear}}^l + (\mathrm{Attn}_s^l \circ \mathrm{Linear}^l(Z^l))u, \quad \ell = 0, \dots, L-1,$$

with $Z^l := [X; W^l]$ and $W^l := [\widehat{w}_{\mathrm{linear}}^l, \dots, \widehat{w}_{\mathrm{linear}}^l]$ as in (3.1) (for any fixed bounded $w^l$), satisfy

$$\|\widehat{w}_{\mathrm{linear}}^L - w_{\mathrm{linear}}\|_\infty \le \epsilon,$$

where $w_{\mathrm{linear}} := \mathrm{argmin}_{w \in \mathbb{R}^d} \frac{1}{2n}\sum_{i=1}^n (\langle w, x_i \rangle - y_i)^2$.

*Proof.* Define the squared-loss objective

$$\widehat{L}_n(w) := \frac{1}{2n} \sum_{i=1}^{n} (\langle w, x_i \rangle - y_i)^2, \qquad w_{\text{linear}} := \arg\min_{w \in \mathbb{R}^d} \widehat{L}_n(w).$$

Assume $\widehat{L}_n$ is $\alpha$-strongly convex and $\beta$-smooth, and take stepsize $\eta = 1/\beta$.

Let $\{w_{\text{GD}}^l\}_{l \geq 0}$ be the exact gradient descent iterates for minimizing $\widehat{L}_n$.

By Corollary 3.1.2, each attention layer emulates one GD step with per-layer:

$$\|\widehat{w}_{\text{linear}}^{l+1} - w_{\text{GD}}^{l+1}\|_\infty \leq \epsilon/2, \qquad l = 0, \dots, L-1. \tag{D.27}$$

Next, applying Corollary D.1.1 to $L = \widehat{L}_n$ gives

$$\|w_{\text{GD}}^L - w_{\text{linear}}\|_2^2 \leq \exp\left(-\frac{L}{\kappa}\right) \|w_{\text{GD}}^0 - w_{\text{linear}}\|_2^2, \qquad \kappa = \beta/\alpha.$$

Taking square roots and using $\|\cdot\|_\infty \leq \|\cdot\|_2$ yields

$$\|w_{\text{GD}}^L - w_{\text{linear}}\|_\infty \leq \|w_{\text{GD}}^L - w_{\text{linear}}\|_2 \leq \exp\left(-\frac{L}{2\kappa}\right) \|w_{\text{GD}}^0 - w_{\text{linear}}\|_2. \tag{D.28}$$

Choose $L$ large enough so that $\|w_{\text{GD}}^L - w_{\text{linear}}\|_\infty \leq \epsilon/2$.

Finally, by triangle inequality together with (D.27) and (D.28),

$$\|\widehat{w}_{\text{linear}}^L - w_{\text{linear}}\|_\infty \leq \|\widehat{w}_{\text{linear}}^L - w_{\text{GD}}^L\|_\infty + \|w_{\text{GD}}^L - w_{\text{linear}}\|_\infty \leq \epsilon.$$

This completes the proof. $\qquad\square$

## D.4 PROOF OF COROLLARY 3.1.4

**Theorem D.4** (Restate of Corollary 3.1.4: In-Context Emulation of Ridge Regression). For any input-output pair $(x_i, y_i)$, where $x_i \in \mathbb{R}^d, y_i \in \mathbb{R}, i \in [n]$, and any $\epsilon > 0$, there exists a depth-$L$ stack of single-head attention layers $\{\text{Attn}_s^l\}_{l=0}^{L-1}$, linear maps $\{\text{Linear}^l\}_{l=0}^{L-1}$, and a readout $u \in \mathbb{R}^n$ such that the iterates

$$\widehat{w}_{\text{ridge}}^{l+1} := \widehat{w}_{\text{ridge}}^l + (\text{Attn}_s^l \circ \text{Linear}^l(Z^l))u, \quad \ell = 0, \dots, L-1,$$

with $Z^l := [X; W^l]$ and $W^l := [\widehat{w}_{\text{ridge}}^l, \dots, \widehat{w}_{\text{ridge}}^l]$ as in (3.1) (for any fixed bounded $w^l$), satisfy

$$\|\widehat{w}_{\text{ridge}}^L - w_{\text{ridge}}\|_\infty \leq \epsilon,$$

where $w_{\text{ridge}} := \text{argmin}_{w \in \mathbb{R}^d} \frac{1}{2n} \sum_{i=1}^{n} (\langle w, x_i \rangle - y_i)^2 + \frac{\lambda}{2}\|w\|_2^2$ with regularization term $\lambda \geq 0$.

*Proof.* Define the loss objective

$$\widehat{L}_n(w) := \frac{1}{2n} \sum_{i=1}^{n} (\langle w, x_i \rangle - y_i)^2 + \frac{\lambda}{2}\|w\|_2^2, \qquad w_{\text{ridge}} := \arg\min_{w \in \mathbb{R}^d} \widehat{L}_n(w).$$

$\widehat{L}_n$ is $\alpha$-strongly convex and $\beta$-smooth, and take stepsize $\eta = 1/\beta$. Let $\{w_{\text{GD}}^l\}_{l \geq 0}$ be the exact gradient descent iterates for minimizing $\widehat{L}_n$.

By Corollary 3.1.2, each attention layer emulates one GD step with per-layer:

$$\|\widehat{w}_{\text{ridge}}^{l+1} - w_{\text{GD}}^{l+1}\|_\infty \le \epsilon/2, \qquad l = 0, \dots, L-1. \tag{D.29}$$

Next, applying Corollary D.1.1 to $L = \widehat{L}_n$ gives

$$\|w_{\text{GD}}^L - w_{\text{ridge}}\|_2^2 \le \exp\left(-\frac{L}{\kappa}\right) \|w_{\text{GD}}^0 - w_{\text{ridge}}\|_2^2, \qquad \kappa = \beta/\alpha.$$

Taking square roots and using $\|\cdot\|_\infty \le \|\cdot\|_2$ yields

$$\|w_{\text{GD}}^L - w_{\text{ridge}}\|_\infty \le \|w_{\text{GD}}^L - w_{\text{ridge}}\|_2 \le \exp\left(-\frac{L}{2\kappa}\right) \|w_{\text{GD}}^0 - w_{\text{ridge}}\|_2. \tag{D.30}$$

Choose $L$ large enough so that $\|w_{\text{GD}}^L - w_{\text{ridge}}\|_\infty \le \epsilon/2$.

Finally, by triangle inequality together with (D.29) and (D.30),

$$\|\widehat{w}_{\text{ridge}}^L - w_{\text{ridge}}\|_\infty \le \|\widehat{w}_{\text{ridge}}^L - w_{\text{GD}}^L\|_\infty + \|w_{\text{GD}}^L - w_{\text{ridge}}\|_\infty \le \epsilon.$$

This completes the proof. $\qquad\square$

## D.5 PROOF OF THEOREM 4.1

**Theorem D.5** (In-Context Emulation of Attention; Theorem 4.1 Restate). Let $X \in \mathbb{R}^{d \times n}$ be an input sequence, and let $W_K, W_Q, W_V \in \mathbb{R}^{d_h \times d}$ be the weight matrices of the target attention head we wish to emulate in-context. Assume $\|W_K X\|_\infty, \|W_Q X\|_\infty, \|W_V X\|_\infty \le B_{KQV}$ with $B_{KQV} > 0$. Then, for any $\epsilon > 0$, there exists a two-layer attention network — a multi-head attention layer $\text{Attn}_m$ followed by a single-head attention layer $\text{Attn}_s$ — such that

$$\|\underbrace{\text{Attn}_s \circ \text{Attn}_m(X_p)}_{\text{Emulator}} - \underbrace{W_V X \text{Softmax}_\beta((W_K X)^\top W_Q X)}_{\text{Target}}\|_\infty \le \epsilon,$$

where $X_p$ is the prompt defined in Definition 4.2.

*Proof.* We state our high-level proof sketch first.

**Step 1: In-Context Weight Encoding.** We define

$$\underbrace{K}_{d_h \times n} := \underbrace{W_K}_{d_h \times d} \cdot \underbrace{X}_{d \times n}, \qquad \underbrace{Q}_{d_h \times n} := \underbrace{W_Q}_{d_h \times d} \cdot \underbrace{X}_{d \times n}, \qquad \underbrace{V}_{d_h \times n} := \underbrace{W_V}_{d_h \times d} \cdot \underbrace{X}_{d \times n}.$$

We aim to approximate the attention mechanism $V \text{Softmax}_\beta(K^\top Q)$ using a two-layer transformer $\text{Attn}_s \circ \text{Attn}_m$. Therefore, the transformer $\text{Attn}_s \circ \text{Attn}_m$ must have in-context access to the information about $W_K, W_Q$ and $W_V$. This is equivalent to exposing the transformer $\text{Attn}_s \circ \text{Attn}_m$ to the target algorithm's specification.

To that end, we augment the input sequence $X$ with two auxiliary blocks:

1. The weight encoding $W_{\text{in}}$ of the target algorithm. $W_{\text{in}}$ contains the vectorization of the target weights $W_K, W_Q$ and $W_V$.

2. A positional encoding $I_n$. $I_n$ exposes token indices.

Concretely, following Definition 4.2, we form

$$X_p = \underbrace{\begin{bmatrix} X \\ W_{\text{in}} \\ I_n \end{bmatrix}}_{(d+6dd_h+n)\times n} \quad \text{with} \quad W_{\text{in}} = \underbrace{\begin{bmatrix} 0 \cdot w & 1 \cdot w & 2 \cdot w & \cdots & (n-1) \cdot w \\ w & w & w & \cdots & w \end{bmatrix}}_{6dd_h \times n},$$

and

$$w = \underbrace{\begin{bmatrix} W_K \\ \hline W_Q \\ \hline W_V \end{bmatrix}}_{3dd_h \times 1}.$$

**Step 2: Multi-Head Decomposition for In-Context Recovery of** $K, Q, V$**.** In this step, we use the multi-head layer $\text{Attn}_m$ to build approximators of $K, Q$, and $V$ from the prompt $X_p$. We denote these approximators by $K', Q'$, and $V'$, corresponding to $K, Q$, and $V$.

Explicitly, we have

$$\| \underbrace{\overbrace{\begin{bmatrix} K' \\ Q' \\ V' \end{bmatrix}}^{\text{Attn}_m(X_p)}}_{3d_h \times n} - \underbrace{\begin{bmatrix} K \\ Q \\ V \end{bmatrix}}_{3d_h \times n} \|_\infty \leq \epsilon_0.$$

Intuitively, this works as: $X_p$ contains the raw input $X$ and the weight encodings of $W_K, W_Q$ and $W_V$. Then $\text{Attn}_m$ "reads" $X$ and the target weight parameters from $X_p$ within its attention heads to form the desired approximation.

**Step 3: Single-Head Assembly for Emulated Map.** We use the single-head layer $\text{Attn}_s$ to perform the attention computation. From $K', Q', V'$, $\text{Attn}_s$ produces

$$V'\text{Softmax}_\beta((K')^\top Q').$$

For reference, the target computation is $V\text{Softmax}_\beta((K)^\top Q)$. This step aligns the output of $\text{Attn}_s$ with the target attention, using the approximated $K', Q', V'$ as inputs.

**Step 4: Error Bound.** Finally, we bound the gap between the computed and target attention:

$$\|V'\text{Softmax}_\beta((K')^\top Q') - V\text{Softmax}_\beta((K)^\top Q)\|_\infty \leq \epsilon_0 + nB_{KQV}\epsilon_1.$$

Our proof starts here.

**Step 1: In-Context Weight Encoding.** For clarity and simplicity, we define

$$k_i := (W_K^\top)_{:,i} \in \mathbb{R}^d, \tag{D.31}$$

$$q_i := (W_Q^\top)_{:,i} \in \mathbb{R}^d, \tag{D.32}$$

$$v_i := (W_V^\top)_{:,i} \in \mathbb{R}^d, \tag{D.33}$$

such that the vectorized weight matrices $\underline{W}_K, \underline{W}_Q, \underline{W}_V$ in Definition 4.2 become

$$\underline{W}_K = \begin{bmatrix} k_1 \\ k_2 \\ \vdots \\ k_{d_h} \end{bmatrix} \in \mathbb{R}^{dd_h}, \ \underline{W}_Q = \begin{bmatrix} q_1 \\ q_2 \\ \vdots \\ q_{d_h} \end{bmatrix} \in \mathbb{R}^{dd_h}, \ \underline{W}_V = \begin{bmatrix} v_1 \\ v_2 \\ \vdots \\ v_{d_h} \end{bmatrix} \in \mathbb{R}^{dd_h},$$

and $w$ becomes

$$w = \begin{bmatrix} \underline{W}_K \\ \underline{W}_Q \\ \underline{W}_V \end{bmatrix} = \begin{bmatrix} k_1 \\ \vdots \\ k_{d_h} \\ q_1 \\ \vdots \\ q_{d_h} \\ v_1 \\ \vdots \\ v_{d_h} \end{bmatrix}.$$

$W_{\mathrm{in}}$ remains as

$$W_{\mathrm{in}} := \begin{bmatrix} 0 \cdot w & 1 \cdot w & 2 \cdot w & \cdots & (n-1) \cdot w \\ w & w & w & \cdots & w \end{bmatrix} \in \mathbb{R}^{6dd_h \times n}.$$

Then, for the input $X$, we append it with the target weights $W_{\mathrm{in}}$ and the positional encoding $I_n$ as in Definition 4.2. We denote this result with $X_p$ and write it out as

$$X_p := \underbrace{\begin{bmatrix} X \\ W_{\mathrm{in}} \\ I_n \end{bmatrix}}_{(d+6dd_h+n)\times n}. \tag{D.34}$$

**Step 2: Multi-Head Decomposition for In-Context Recovery of** $K, Q, V$**.** In this part, we construct approximators for $K, Q$ and $V$ via $\mathrm{Attn}_m$. We construct the approximators by approximating each row of $K, Q, V$ and then aggregating the results across rows. Each row in $K, Q$ and $V$ has the form: $k_i^\top X$, $q_i^\top X$, and $v_i^\top X$. To approximate these rows in $K, Q, V$, we apply Theorem D.1 to each row separately. Namely, we allocate an $H$-head attention to each row of $K, Q$ and $V$ to carry out the row-wise approximations. Since $K, Q, V \in \mathbb{R}^{d_h \times n}$, each of $K, Q, V$ uses $H \cdot d_h$ heads. We interpret $H$ as the number of heads per row dimension, since each $K$, $Q$, and $V$ has $d_h$ rows. Finally, we define a multi-head attention $\mathrm{Attn}_m$ as the union of these three groups of $H d_h$ heads. Therefore, $\mathrm{Attn}_m$ has $3H d_h$ heads in total.

We label the $3H d_h$ heads in $\mathrm{Attn}_m$ as:

$$\mathrm{Attn}^K_{j,\widetilde{h}}, \quad j \in [d_h], \quad \widetilde{h} \in \{J+1, \ldots, J+H\}; \qquad \text{(Approximates } K)$$
$$\mathrm{Attn}^Q_{j,\widetilde{h}}, \quad j \in [2d_h] \setminus [d_h], \quad \widetilde{h} \in \{J+1, \ldots, J+H\}; \qquad \text{(Approximates } Q)$$
$$\mathrm{Attn}^V_{j,\widetilde{h}}, \quad j \in [3d_h] \setminus [2d_h], \quad \widetilde{h} \in \{J+1, \ldots, J+H\}, \qquad \text{(Approximates } V)$$

where we define $J := (j-1)H$ to simplify our notation. Each $\mathrm{Attn}^K_{j,\widetilde{h}}$, $\mathrm{Attn}^Q_{j,\widetilde{h}}$, and $\mathrm{Attn}^V_{j,\widetilde{h}}$ is a single-head attention. Index $j$ identifies the target row, and index $\widetilde{h}$ identifies the head allocated to that row. Here $j \in [2d_h] \setminus [d_h]$ denotes the set difference. That is, $j \in [2d_h] \setminus [d_h]$ means $j \in \{d_h + 1, \ldots, 2d_h\}$.

Thus, $\text{Attn}_m$ consists of three groups of attention heads:

$$\text{Attn}_m := \underbrace{\sum_{j=1}^{d_h} \sum_{\widetilde{h}=J+1}^{J+H} \text{Attn}_{j,\widetilde{h}}^K}_{\text{Approximates } K} + \underbrace{\sum_{j=d_h+1}^{2d_h} \sum_{\widetilde{h}=J+1}^{J+H} \text{Attn}_{j,\widetilde{h}}^Q}_{\text{Approximates } Q} + \underbrace{\sum_{j=2d_h+1}^{3d_h} \sum_{\widetilde{h}=J+1}^{J+H} \text{Attn}_{j,\widetilde{h}}^V}_{\text{Approximates } V},$$

In the subsequent proof, we provide the constructions of $\text{Attn}_{j,\widetilde{h}}^K$, $\text{Attn}_{j,\widetilde{h}}^Q$, $\text{Attn}_{j,\widetilde{h}}^V$ from Theorem D.1.

To apply Theorem D.1 to construct heads in $\text{Attn}_m$, let $a$ and $b$ denote the minimum and maximum of the inner products $k_i^\top x_m$, $q_i^\top x_m$, and $v_i^\top x_m$, over all $i \in [d_h]$ and $m \in [n]$:

$$a \le \min\{k_i^\top x_m, q_i^\top x_m, v_i^\top x_m\} \quad \text{and} \quad \max\{k_i^\top x_m, q_i^\top x_m, v_i^\top x_m\} \le b.$$

Next, we choose

$$H := \lceil \frac{2(b-a)}{(n-2)\epsilon_0} \rceil,$$

such that the interpolation error in Theorem D.1 is at most $\frac{\epsilon_0}{2}$ for any $\epsilon_0 > 0$.

Third, Theorem D.1 requires a single map $A : \mathbb{R}^{d \times n} \to \mathbb{R}^{(3d+n) \times n}$ shared across all $H$ heads. In our construction, we realize this augmentation by prepending each head of $\text{Attn}^H$ in Theorem D.1 with a head-specific linear map $A_{\widetilde{h}} : \mathbb{R}^{(d+6dd_h+n) \times n} \to \mathbb{R}^{(3d+n) \times n}$. The map $A_{\widetilde{h}}$ maps the input $X_p$ to the desired dimension and picks out the target $k_i, q_i$ or $v_i$ (this is equivalent to the $w_s$ in Theorem D.1.) to let $\text{Attn}^H$ perform the desired linear transformation $k_i^\top X$, $v_i^\top X$ or $q_i^\top X$. Here $\widetilde{h}$ still identifies the single head assigned to a specific row. By Lemma D.3, $\text{Attn}^H \circ A_{\widetilde{h}}$ remains an $H$-head attention. Therefore, we use $\text{Attn}^H \circ A_{\widetilde{h}}$ to build the heads in $\text{Attn}_m$.

We construct $A_{\widetilde{h}}$ as

$$A_{\widetilde{h}} := \underbrace{\begin{bmatrix} I_d & 0_{d \times 3dd_h} & 0_{d \times 3dd_h} & 0_{d \times n} \\ 0_{d \times d} & E_{\widetilde{h}} & 0_{d \times 3dd_h} & 0_{d \times n} \\ 0_{d \times d} & E_{\widetilde{h}} & K_{\widetilde{h}} E_{\widetilde{h}} & 0_{d \times n} \\ 0_{n \times d} & 0_{n \times 3dd_h} & 0_{n \times 3dd_h} & I_n \end{bmatrix}}_{(3d+n) \times (d+6dd_h+n)},$$

where

$$E_{\widetilde{h}} := \underbrace{\begin{bmatrix} 0_{d \times (d[\widetilde{h}/H])} & I_d & 0_{d \times (3dd_h - d[\widetilde{h}/H] - d)} \end{bmatrix}}_{d \times 3dd_h},$$

$$K_{\widetilde{h}} := [(\widetilde{h}\%H - 1)(n-2) - 1].$$

Here $\widetilde{h}\%H$ denotes the remainder of dividing $\widetilde{h}$ by $H$. We define $\%$ such that instead of the common $(kH)\%H = 0$,

$$kH\%H = H, \quad \text{for all} \quad k \in \mathbb{N}^+.$$

Applying $A_{\widetilde{h}}$ to $X_p$ yields

$$
A_{\widetilde{h}} \cdot X_p := \underbrace{\begin{bmatrix} I_d & 0_{d \times 3dd_h} & 0_{d \times 3dd_h} & 0_{d \times n} \\ 0_{d \times d} & E_{\widetilde{h}} & 0_{d \times 3dd_h} & 0_{d \times n} \\ 0_{d \times d} & E_{\widetilde{h}} & K_{\widetilde{h}} E_{\widetilde{h}} & 0_{d \times n} \\ 0_{n \times d} & 0_{n \times 3dd_h} & 0_{n \times 3dd_h} & I_n \end{bmatrix}}_{(3d+n) \times (d+6dd_h+n)} \cdot \underbrace{\begin{bmatrix} X \\ W_{\mathrm{in}} \\ I_n \end{bmatrix}}_{(d+6dd_h+n) \times n} \qquad \left(\text{By the definition of } A_{\widetilde{h}} \text{ and } X_p\right)
$$

$$
= \underbrace{\begin{bmatrix} X \\ \left[ E_{\widetilde{h}} \quad 0_{d \times 3dd_h} \right] \cdot W_{\mathrm{in}} \\ \left[ E_{\widetilde{h}} \quad K_{\widetilde{h}} E_{\widetilde{h}} \right] \cdot W_{\mathrm{in}} \\ I_n \end{bmatrix}}_{(3d+n) \times n},
$$

where $\left[ E_{\widetilde{h}} \quad 0_{d \times 3dd_h} \right] W_{\mathrm{in}}$ expands as

$$
\left[ E_{\widetilde{h}} \quad 0_{d \times 3dd_h} \right] W_{\mathrm{in}}
$$

$$
= \underbrace{\left[ E_{\widetilde{h}} \quad 0_{d \times 3dd_h} \right]}_{d \times 6dd_h} \underbrace{\begin{bmatrix} 0 \cdot w & 1 \cdot w & 2 \cdot w & \cdots & (n-1) \cdot w \\ w & w & w & \cdots & w \end{bmatrix}}_{6dd_h \times n}
$$

$$
= \underbrace{E_{\widetilde{h}}}_{d \times 3dd_h} \cdot \underbrace{\left[ 0 \cdot w \quad 1 \cdot w \quad 2 \cdot w \quad \cdots \quad (n-1) \cdot w \right]}_{3dd_h \times n} + \left[ 0_{d \times 3dd_h} \right] \underbrace{\left[ w \quad w \quad w \quad \cdots \quad w \right]}_{3dd_h \times n}
$$

$$
= \underbrace{\left[ 0 \cdot E_{\widetilde{h}} w \quad 1 \cdot E_{\widetilde{h}} w \quad 2 \cdot E_{\widetilde{h}} w \quad \cdots \quad (n-1) \cdot E_{\widetilde{h}} w \right]}_{d \times n},
$$

and

$$
E_{\widetilde{h}} w = \underbrace{\left[ 0_{d \times (d \lceil \widetilde{h}/H \rceil - d)} \quad I_d \quad 0_{d \times (3dd_h - (d \lceil \widetilde{h}/H \rceil - d) - d)} \right]}_{d \times 3dd_h} \begin{bmatrix} k_1 \\ \vdots \\ k_{d_h} \\ q_1 \\ \vdots \\ q_{d_h} \\ v_1 \\ \vdots \\ v_{d_h} \end{bmatrix} \qquad \left(\text{By the definition of } E_{\widetilde{h}} \text{ and } w\right)
$$

$$
\phantom{E_{\widetilde{h}} w} \underbrace{\phantom{\begin{bmatrix} k_1 \end{bmatrix}}}_{3dd_h \times 1}
$$

$$
= \begin{cases} k_{\lceil \widetilde{h}/H \rceil}, & 1 \leq \widetilde{h} \leq H d_h \\ q_{\lceil \widetilde{h}/H \rceil - d_h}, & H d_h < \widetilde{h} \leq 2H d_h \\ v_{\lceil \widetilde{h}/H \rceil - 2d_h}, & 2H d_h < \widetilde{h} \leq 3H d_h \end{cases}. \tag{D.35}
$$

The equality (D.35) holds since $E_{\widetilde{h}}$ selects the $\lceil \widetilde{h}/H \rceil$-th block in $w$.

Similarly, $\left[ E_{\widetilde{h}} \quad K_{\widetilde{h}} E_{\widetilde{h}} \right] \cdot W_{\mathrm{in}}$ expands as

$$
\left[ E_{\widetilde{h}} \quad K_{\widetilde{h}} E_{\widetilde{h}} \right] W_{\mathrm{in}}
$$

$$
= \underbrace{\left[ E_{\widetilde{h}} \quad K_{\widetilde{h}} E_{\widetilde{h}} \right]}_{d \times 6dd_h} \underbrace{\begin{bmatrix} 0 \cdot w & 1 \cdot w & 2 \cdot w & \cdots & (n-1) \cdot w \\ w & w & w & \cdots & w \end{bmatrix}}_{6dd_h \times n} \qquad \left(\text{By the definition of } W_{\mathrm{in}}\right)
$$

$$= \underbrace{E_{\widetilde{h}}}_{d \times 3dd_h} \cdot \underbrace{\begin{bmatrix} 0 \cdot w & 1 \cdot w & 2 \cdot w & \cdots & (n-1) \cdot w \end{bmatrix}}_{3dd_h \times n} + \underbrace{K_{\widetilde{h}} E_{\widetilde{h}}}_{d \times 3dd_h} \underbrace{\begin{bmatrix} w & w & w & \cdots & w \end{bmatrix}}_{3dd_h \times n}$$

$$\left( \text{By } \begin{bmatrix} A & B \end{bmatrix} \begin{bmatrix} C \\ D \end{bmatrix} = AB + CD \right)$$

$$= \underbrace{\begin{bmatrix} K_{\widetilde{h}} E_{\widetilde{h}} w & (K_{\widetilde{h}} + 1) E_{\widetilde{h}} w & \cdots & (K_{\widetilde{h}} + n - 1) E_{\widetilde{h}} w \end{bmatrix}}_{d \times n},$$

Up to here, we are capable of selecting a target $k_i, q_i$ or $v_i$, and we start to build our heads in $\mathrm{Attn}_m$. When $1 \le \widetilde{h} \le Hd_h$, we compute $A_{\widetilde{h}} \cdot X_p$ as

$$A_{\widetilde{h}} \cdot X_p = \underbrace{\begin{bmatrix} & X & & \\ 0 \cdot k_{\lceil \widetilde{h}/H \rceil} & 1 \cdot k_{\lceil \widetilde{h}/H \rceil} & \cdots & (n-1) \cdot k_{\lceil \widetilde{h}/H \rceil} \\ k_{\lceil \widetilde{h}/H \rceil} & k_{\lceil \widetilde{h}/H \rceil} & \cdots & k_{\lceil \widetilde{h}/H \rceil} \\ & I_n & & \end{bmatrix}}_{(3d+n) \times n} .$$

This means every $\widetilde{h}$ in $\{J + 1, \ldots, J + H\}$ with $j \in [d_h]$ has the same $A_{\widetilde{h}} \cdot X_p$:

$$\underbrace{\begin{bmatrix} & X & & \\ 0 \cdot k_j & 1 \cdot k_j & \cdots & (n-1) \cdot k_j \\ k_j & k_j & \cdots & k_j \\ & I_n & & \end{bmatrix}}_{(3d+n) \times n}$$

For each $j \in [d_h]$, by Theorem D.1, there exists an $H$-head attention $\mathrm{Attn}'_j : \mathbb{R}^{(3d+n) \times n} \to \mathbb{R}^{3d_h \times n}$, such that the output satisfies

$$\| \underbrace{\mathrm{Attn}'_j(\overbrace{A_{\widetilde{h}} \cdot X_p}^{(3d+n) \times n})_{:,i}}_{3d_h \times 1} - (k_j^\top x_i) \underbrace{e_j^{(3d_h)}}_{3d_h \times 1} \|_\infty \le \epsilon_0, \tag{D.36}$$

for every $i \in [n]$ and any $\epsilon_0 > 0$.

From (D.36), we have

$$\| \underbrace{\mathrm{Attn}'_j(\overbrace{A_{\widetilde{h}} \cdot X_p}^{(3d+n) \times n})}_{3d_h \times n} - \underbrace{e_j^{(3d_h)} k_j^\top X}_{3d_h \times n} \|_\infty \le \epsilon_0,$$

where

$$e_j^{(3d_h)} k_j^\top X = \underbrace{\begin{bmatrix} (k_j^\top x_1) e_j^{(3d_h)} & (k_j^\top x_2) e_j^{(3d_h)} & \cdots & (k_j^\top x_n) e_j^{(3d_h)} \end{bmatrix}}_{3d_h \times n} .$$

We use $\mathrm{Attn}'^{(s)}_j$ to label the heads in $\mathrm{Attn}'_j$, and we define $\mathrm{Attn}^K_{j,\widetilde{h}}(Z)$ to be

$$\mathrm{Attn}^K_{j,\widetilde{h}}(Z) := \mathrm{Attn}'^{(\widetilde{h})}_j(A_{\widetilde{h}} \cdot Z), \qquad \left( Z \in \mathbb{R}^{(d+6dd_h+n) \times n} \text{ denotes any input} \right)$$

where $j \in [d_h]$ and $\widetilde{h} \in \{J + 1, \ldots, J + H\}$.

By Lemma D.3, $\text{Attn}^K_{j,\widetilde{h}}(Z)$ is still an attention.

Thus

$$\text{Attn}^K_j(Z) := \sum_{\widetilde{h}=J+1}^{J+H} \text{Attn}^K_{j,\widetilde{h}}(Z),$$

is also an attention.

Thus, we have

$$\| \underbrace{\text{Attn}^K_j(X_p)}_{3d_h \times n} - \underbrace{e^{(3d_h)}_j k^\top_j X}_{3d_h \times n} \|_\infty \leq \epsilon_0.$$

This means that

$$\| \underbrace{\sum_{j=1}^{d_h} \text{Attn}^K_j(X_p)}_{3d_h \times n} - \underbrace{\begin{bmatrix} K \\ 0_{d_h \times n} \\ 0_{d_h \times n} \end{bmatrix}}_{3d_h \times n} \|_\infty \leq \epsilon_0. \tag{D.37}$$

Similarly for $Q$, by (D.35), when $Hd_h < \widetilde{h} \leq 2Hd_h$, we have

$$E_{\widetilde{h}} w = \underbrace{q_{\lceil \widetilde{h}/H \rceil - d_h}}_{d \times 1},$$

and

$$A_{\widetilde{h}} \cdot X_p = \underbrace{\begin{bmatrix} & & X & \\ 0 \cdot q_{j-d_h} & 1 \cdot q_{j-d_h} & \cdots & (n-1) \cdot q_{j-d_h} \\ q_{j-d_h} & q_{j-d_h} & \cdots & q_{j-d_h} \\ & & I_n & \end{bmatrix}}_{(3d+n) \times n},$$

where $j \in [2d_h] \setminus [d_h]$.

For each $j \in [2d_h] \setminus [d_h]$, by Theorem D.1, there exists an $H$-head attention $\text{Attn}''_j : \mathbb{R}^{(3d+n) \times n} \to \mathbb{R}^{3d_h \times n}$, such that

$$\| \underbrace{\text{Attn}''_j(A_{\widetilde{h}} \cdot X_p)}_{3d_h \times n} - \underbrace{e^{(3d_h)}_j q^\top_{j-d_h} X}_{3d_h \times n} \|_\infty \leq \epsilon_0,$$

for any $\epsilon_0 > 0$.

Then we construct $\text{Attn}^Q_{j,\widetilde{h}}$ in a way similar to $\text{Attn}^K_{j,\widetilde{h}}$

$$\text{Attn}^Q_{j,\widetilde{h}}(Z) := \text{Attn}''^{(\widetilde{h})}_j(A_{\widetilde{h}} \cdot Z), \qquad \left( Z \in \mathbb{R}^{(d+6dd_h+n) \times n} \text{ denotes any input} \right)$$

where $j \in [2d_h] \setminus [d_h]$ and $\widetilde{h} \in \{J+1, \ldots, J+H\}$.

By Lemma D.3, $\text{Attn}^Q_{j,\widetilde{h}}(Z)$ is an attention.

Thus

$$\operatorname{Attn}_j^Q(Z) := \sum_{\widetilde{h}=J+1}^{J+H} \operatorname{Attn}_{j,\widetilde{h}}^Q(Z),$$

is also an attention.

Thus, we have

$$\| \underbrace{\operatorname{Attn}_j^Q(X_p)}_{3d_h \times n} - \underbrace{e_j^{(3d_h)} q_{j-d_h}^\top X}_{3d_h \times n} \|_\infty \le \epsilon_0.$$

This means that

$$\| \underbrace{\sum_{j=d_h+1}^{2d_h} \operatorname{Attn}_j^Q(X_p)}_{3d_h \times n} - \underbrace{\begin{bmatrix} 0_{d_h \times n} \\ Q \\ 0_{d_h \times n} \end{bmatrix}}_{3d_h \times n} \|_\infty \le \epsilon_0. \tag{D.38}$$

For $V$, with analogous construction to that of $K$ and $Q$, there exists an $H$-head attention $\operatorname{Attn}_j^V : \mathbb{R}^{(3d+n)\times n} \to \mathbb{R}^{3d_h \times n}$ such that

$$\| \underbrace{\operatorname{Attn}_j^V(X_p)}_{3d_h \times n} - \underbrace{e_j^{(3d_h)} v_{j-2d_h}^\top X}_{3d_h \times n} \|_\infty \le \epsilon_0,$$

for each $j \in [3d_h] \setminus [2d_h]$ and any $\epsilon_0 > 0$.

This means that

$$\| \underbrace{\sum_{j=2d_h+1}^{3d_h} \operatorname{Attn}_j^V(X_p)}_{3d_h \times n} - \underbrace{\begin{bmatrix} 0_{d_h \times n} \\ 0_{d_h \times n} \\ V \end{bmatrix}}_{3d_h \times n} \|_\infty \le \epsilon_0. \tag{D.39}$$

Combining (D.37), (D.38) and (D.39), we have

$$\|\operatorname{Attn}_m(X_p) - \begin{bmatrix} K \\ Q \\ V \end{bmatrix}\|_\infty$$

$$= \| \sum_{j=1}^{d_h} \operatorname{Attn}_j^K(X_p) - \begin{bmatrix} K \\ 0_{d_h \times n} \\ 0_{d_h \times n} \end{bmatrix} + \sum_{j=d_h+1}^{2d_h} \operatorname{Attn}_j^Q(X_p) - \begin{bmatrix} 0_{d_h \times n} \\ Q \\ 0_{d_h \times n} \end{bmatrix} + \sum_{j=2d_h+1}^{3d_h} \operatorname{Attn}_j^V(X_p) - \begin{bmatrix} 0_{d_h \times n} \\ 0_{d_h \times n} \\ V \end{bmatrix} \|_\infty$$

$$\le \epsilon_0. \tag{D.40}$$

We define

$$\underbrace{\begin{bmatrix} K' \\ Q' \\ V' \end{bmatrix}}_{3d_h \times n} := \operatorname{Attn}_m(X_p).$$

Thus, (D.40) becomes

$$\|\begin{bmatrix} K' \\ Q' \\ V' \end{bmatrix} - \begin{bmatrix} K \\ Q \\ V \end{bmatrix}\|_\infty \le \epsilon_0, \tag{D.41}$$
$$\underbrace{\phantom{\begin{bmatrix} K' \\ Q' \\ V' \end{bmatrix}}}_{3d_h \times n} \underbrace{\phantom{\begin{bmatrix} K \\ Q \\ V \end{bmatrix}}}_{3d_h \times n}$$

**Step 3: Single-Head Assembly for Emulated Map.** Our goal in this part is to reconstruct the attention mechanism

$$V'\text{Softmax}_\beta((K')^\top Q'), \quad \text{and} \quad V\text{Softmax}_\beta((K)^\top Q),$$

from $K', Q', V'$ and $K, Q, V$ via $\text{Attn}_s$.

In order to achieve this, we construct $\text{Attn}_s$ to be

$$\text{Attn}_s(Z) := \underbrace{[0_{d_h \times 2d_h} \quad I_{d_h}]}_{:=W_{V,s}} Z \cdot \text{Softmax}_\beta((\underbrace{[I_{d_h} \quad 0_{d_h \times 2d_h}]}_{:=W_{K,s}} Z)^\top \underbrace{[0_{d_h \times d_h} \quad I_{d_h} \quad 0_{d_h \times d_h}]}_{:=W_{Q,s}} Z),$$

where $Z \in \mathbb{R}^{3d_h \times n}$ denotes any input.

Thus, we have

$$\text{Attn}_s(\overbrace{\begin{bmatrix} K \\ Q \\ V \end{bmatrix}}^{d_h \times n}) = \underbrace{V}_{d_h \times n} \text{Softmax}_\beta(\underbrace{(K)^\top Q}_{n \times n}), \tag{D.42}$$
$$\underbrace{\phantom{\begin{bmatrix} K \\ Q \\ V \end{bmatrix}}}_{3d_h \times n}$$

and

$$\text{Attn}_s(\overbrace{\begin{bmatrix} K' \\ Q' \\ V' \end{bmatrix}}^{d_h \times n}) = \underbrace{V'}_{d_h \times n} \text{Softmax}_\beta(\underbrace{(K')^\top Q'}_{n \times n}). \tag{D.43}$$
$$\underbrace{\phantom{\begin{bmatrix} K' \\ Q' \\ V' \end{bmatrix}}}_{3d_h \times n}$$

**Step 4: Error Bound.** From (D.42) and (D.43), we have

$$\text{Attn}_s(\begin{bmatrix} K' \\ Q' \\ V' \end{bmatrix}) - \text{Attn}_s(\begin{bmatrix} K \\ Q \\ V \end{bmatrix})$$

$$= \underbrace{V'}_{d_h \times n} \cdot \underbrace{\text{Softmax}_\beta(K'^\top Q')}_{n \times n} - \underbrace{V}_{d_h \times n} \cdot \underbrace{\text{Softmax}_\beta(K^\top Q)}_{n \times n} \qquad \text{(By (D.43) and (D.42))}$$

$$= V'\text{Softmax}_\beta(K'^\top Q') - V\text{Softmax}_\beta(K'^\top Q') + V\text{Softmax}_\beta(K'^\top Q') - V\text{Softmax}_\beta(K^\top Q)$$

$$= (V' - V)\text{Softmax}_\beta(K'^\top Q') + V(\text{Softmax}_\beta(K'^\top Q') - \text{Softmax}_\beta(K^\top Q)), \tag{D.44}$$

and the last equality follows from the distributivity of matrix multiplication.

Then, (D.44) yields

$$\|\text{Attn}_s(\begin{bmatrix} K' \\ Q' \\ V' \end{bmatrix}) - \text{Attn}_s(\begin{bmatrix} K \\ Q \\ V \end{bmatrix})\|_\infty$$

$$= \|(V' - V)\text{Softmax}_\beta(K'^\top Q') + V(\text{Softmax}_\beta(K'^\top Q') - \text{Softmax}_\beta(K^\top Q))\|_\infty$$

$$\leq \underbrace{\|(V'-V)\mathrm{Softmax}_\beta(K'^\top Q')\|_\infty}_{:=(I)} + \underbrace{\|V(\mathrm{Softmax}_\beta(K'^\top Q') - \mathrm{Softmax}_\beta(K^\top Q))\|_\infty}_{:=(II)}, \quad \text{(D.45)}$$

and the last inequality follows from the triangle inequality.

For term $(I)$ in (D.45), since each column in $\mathrm{Softmax}_\beta(K'^\top Q')$ sums up to 1, then

$$\underbrace{(V'-V)}_{d_h \times n} \underbrace{\mathrm{Softmax}_\beta(K'^\top Q')_{:,j}}_{n \times 1},$$

is a weighted sum of the columns in $(V'-V)$.

Thus we have

$$\|(V'-V)\mathrm{Softmax}_\beta(K'^\top Q')_{:,j}\|_\infty \leq \|V'-V\|_\infty \leq \epsilon_0,$$

and the first inequality holds since the column average of $(V'-V)$ has a maximum entry no greater than the maximum entry among the original columns in $(V'-V)$. The second inequality holds since (D.41).

Then we get

$$(I) \leq \epsilon_0. \quad \text{(D.46)}$$

Term $(II)$ in (D.45) is

$$(II) = \|V(\mathrm{Softmax}_\beta(K'^\top Q') - \mathrm{Softmax}_\beta(K^\top Q))\|_\infty.$$

For simplicity of presentation, we define

$$\Delta S := \mathrm{Softmax}_\beta(K'^\top Q') - \mathrm{Softmax}_\beta(K^\top Q),$$

such that for each entry in $(II)$, we have

$$
\begin{aligned}
|(V\Delta S)_{ij}| &= |\sum_{k=1}^n V_{ik}(\Delta S)_{kj}| && \text{(By the definition of matrix multiplication)} \\
&\leq \sum_{k=1}^n |V_{ik}| \cdot |(\Delta S)_{kj}| && \text{(By triangle inequality and } |ab| = |a| \cdot |b| \text{ for all } a, b \in \mathbb{R}) \\
&\leq \sum_{k=1}^n \|V\|_\infty \cdot \|\Delta S\|_\infty && \text{(By } |V_{ik}| \leq \|V\|_\infty \text{ and } |(\Delta S)_{kj}| \leq \|\Delta S\|_\infty) \\
&= n\|V\|_\infty \cdot \|\Delta S\|_\infty,
\end{aligned}
$$

and this leads to

$$(II) \leq n\|V\|_\infty \cdot \|\Delta S\|_\infty. \quad \text{(D.47)}$$

For each entry in $\Delta S$, we have

$$
\begin{aligned}
&|(\Delta S)_{i,j}| && \text{(D.48)} \\
&= |(\mathrm{Softmax}_\beta(K'^\top Q') - \mathrm{Softmax}_\beta(K^\top Q))_{i,j}| \\
&= |\frac{e^{\beta K'_i \cdot Q'_j}}{\sum_{i'=1}^n e^{\beta K'_{i'} \cdot Q'_j}} - \frac{e^{\beta K_i \cdot Q_j}}{\sum_{i'=1}^n e^{\beta K_{i'} \cdot Q_j}}| && (K'_i, Q'_i, K_i, Q_i \text{ denote the } i\text{-th column in } K', Q', K, Q)
\end{aligned}
$$

$$
\begin{aligned}
&= \left| \frac{e^{\beta K_i' \cdot Q_j'}}{\sum_{i'=1}^n e^{\beta K_{i'}' \cdot Q_j'}} - \frac{e^{\beta K_i \cdot Q_j}}{\sum_{i'=1}^n e^{\beta K_{i'}' \cdot Q_j'}} + \frac{e^{\beta K_i \cdot Q_j}}{\sum_{i'=1}^n e^{\beta K_{i'}' \cdot Q_j'}} - \frac{e^{\beta K_i \cdot Q_j}}{\sum_{i'=1}^n e^{\beta K_{i'} \cdot Q_j}} \right| \\
&\leq \left| \frac{e^{\beta K_i' \cdot Q_j'} - e^{\beta K_i \cdot Q_j}}{\sum_{i'=1}^n e^{\beta K_{i'}' \cdot Q_j'}} \right| + \left| e^{\beta K_i \cdot Q_j} \left( \frac{1}{\sum_{i'=1}^n e^{\beta K_{i'}' \cdot Q_j'}} - \frac{1}{\sum_{i'=1}^n e^{\beta K_{i'} \cdot Q_j}} \right) \right| \quad \text{(By triangle inequality)} \\
&= \frac{e^{\beta K_i' \cdot Q_j'}}{\sum_{i'=1}^n e^{\beta K_{i'}' \cdot Q_j'}} \left| 1 - e^{\beta(K_i \cdot Q_j - K_i' \cdot Q_j')} \right| + \frac{e^{\beta K_i \cdot Q_j}}{\sum_{i'=1}^n e^{\beta K_{i'} \cdot Q_j}} \left| \frac{\sum_{i'=1}^n e^{\beta K_{i'} \cdot Q_j}}{\sum_{i'=1}^n e^{\beta K_{i'}' \cdot Q_j'}} - 1 \right| \\
&\hspace{9cm} \text{(By non-negativity of exponential)} \\
&< \underbrace{\left| 1 - e^{\beta(K_i \cdot Q_j - K_i' \cdot Q_j')} \right|}_{:=(II\text{-}1)} + \underbrace{\left| 1 - \frac{\sum_{i'=1}^n e^{\beta K_{i'} \cdot Q_j}}{\sum_{i'=1}^n e^{\beta K_{i'}' \cdot Q_j'}} \right|}_{:=(II\text{-}2)}, \quad\quad\quad\quad\quad \text{(D.49)}
\end{aligned}
$$

and the last inequality holds since

$$
\frac{e^{\beta K_i' \cdot Q_j'}}{\sum_{i'=1}^n e^{\beta K_{i'}' \cdot Q_j'}} < 1, \quad \frac{e^{\beta K_i \cdot Q_j}}{\sum_{i'=1}^n e^{\beta K_{i'} \cdot Q_j}} < 1.
$$

To bound term $(II\text{-}1)$ in (D.49), we recall

$$
\left\| \underbrace{\begin{bmatrix} K' \\ Q' \\ V' \end{bmatrix}}_{3d_h \times n} - \underbrace{\begin{bmatrix} K \\ Q \\ V \end{bmatrix}}_{3d_h \times n} \right\|_\infty \leq \epsilon_0,
$$

so

$$
\left\| \underbrace{K' - K}_{d_h \times n} \right\|_\infty \leq \epsilon_0,
$$
$$
\left\| \underbrace{Q' - Q}_{d_h \times n} \right\|_\infty \leq \epsilon_0.
$$

Let $K_i'$, $Q_i'$, $K_i$, $Q_i$ denote the $i$-th column in $K'$, $Q'$, $K$, $Q$, then we have

$$
\underbrace{\Delta K_i}_{d_h \times 1} := \underbrace{K_i' - K_i}_{d_h \times 1}, \quad \|\Delta K_i\|_\infty \leq \epsilon_0,
$$
$$
\underbrace{\Delta Q_i}_{d_h \times 1} := \underbrace{Q_i' - Q_i}_{d_h \times 1}, \quad \|\Delta Q_i\|_\infty \leq \epsilon_0.
$$

Thus, for term $(II\text{-}1)$ in (D.49), we have

$$
\begin{aligned}
&(II\text{-}1) \\
&= \left| 1 - \exp\left\{ \beta(K_i \cdot Q_j - K_i' \cdot Q_j') \right\} \right| \\
&= \left| 1 - \exp\left\{ \beta(K_i \cdot Q_j - (K_i + \Delta K_i) \cdot (Q_j + \Delta Q_j)) \right\} \right| \\
&\hspace{4cm} \text{(By } K_i' = K_i + \Delta K_i \text{ and } Q_i' = Q_i + \Delta Q_i) \\
&= \left| 1 - \exp\left\{ -\beta(K_i \cdot \Delta Q_j + Q_j \cdot \Delta K_i + \Delta K_i \cdot \Delta Q_j) \right\} \right|, \\
&\text{(By } K_i \cdot Q_j - (K_i + \Delta K_i) \cdot (Q_i + \Delta Q_i) = -(K_i \cdot \Delta Q_j + Q_j \cdot \Delta K_i + \Delta K_i \cdot \Delta Q_j))
\end{aligned}
$$

and we know

$$
K_i \cdot \Delta Q_j + Q_j \cdot \Delta K_i + \Delta K_i \cdot \Delta Q_j
$$

$$\leq d_h \cdot \|K_i\|_\infty \|\Delta Q_j\|_\infty + d_h \cdot \|Q_j\|_\infty \|\Delta K_i\|_\infty + d_h \cdot \|\Delta K_i\|_\infty \|\Delta Q_j\|_\infty$$
$$\qquad\qquad \text{(By } a \cdot b \leq d_h \|a\|_\infty \|b\|_\infty \text{ for all } a, b \in \mathbb{R}^{d_h})$$
$$\leq 2d_h B_{KQV} \epsilon_0 + d_h \epsilon_0^2. \qquad \text{(By } \|K_i\|_\infty, \|Q_j\|_\infty \leq B_{KQV} \text{ and } \|\Delta K_i\|_\infty, \|\Delta Q_j\|_\infty \leq \epsilon_0)$$

Thus, we have

$$(II\text{-}1) \leq |1 - e^{-\beta d_h (2B_{KQV}\epsilon_0 + \epsilon_0^2)}|. \tag{D.50}$$

For term $(II\text{-}2)$ in (D.49), we have

$$(II\text{-}2)$$
$$= |1 - \frac{\sum_{i'=1}^n e^{\beta K_{i'} \cdot Q_j}}{\sum_{i'=1}^n e^{\beta K'_{i'} \cdot Q'_j}}| \qquad\qquad \text{(By the definition of } (II\text{-}2))$$
$$= |1 - \frac{\sum_{i'=1}^n e^{\beta K_{i'} \cdot Q_j}}{\sum_{i'=1}^n e^{\beta (K_{i'} + \Delta K_{i'}) \cdot (Q_j + \Delta Q_j)}}| \qquad \text{(By } K'_{i'} = K_{i'} + \Delta K_{i'} \text{ and } Q'_i = Q_i + \Delta Q_i)$$
$$= |1 - \frac{\sum_{i'=1}^n e^{\beta K_{i'} \cdot Q_j}}{\sum_{i'=1}^n e^{\beta (K_{i'} \cdot Q_j + K_{i'} \cdot \Delta Q_j + Q_j \cdot \Delta K_{i'} + \Delta K_{i'} \cdot \Delta Q_j)}}|,$$

and for all $i'$ in the denominator, we have

$$K_{i'} \cdot Q_j + K_{i'} \cdot \Delta Q_j + Q_j \cdot \Delta K_{i'} + \Delta K_{i'} \cdot \Delta Q_j$$
$$\leq K_{i'} \cdot Q_j + d_h \cdot \|K_{i'}\|_\infty \|\Delta Q_j\|_\infty + d_h \cdot \|Q_j\|_\infty \|\Delta K_{i'}\|_\infty + d_h \cdot \|\Delta K_{i'}\|_\infty \|\Delta Q_j\|_\infty$$
$$\qquad\qquad \text{(By } a \cdot b \leq d_h \|a\|_\infty \|b\|_\infty \text{ for all } a, b \in \mathbb{R}^{d_h})$$
$$\leq K_{i'} \cdot Q_j + 2d_h B_{KQV} \epsilon_0 + d_h \epsilon_0^2. \quad \text{(By } \|K_{i'}\|_\infty, \|Q_j\|_\infty \leq B_{KQV} \text{ and } \|\Delta K_{i'}\|_\infty, \|\Delta Q_j\|_\infty \leq \epsilon_0)$$

Thus,

$$(II\text{-}2)$$
$$= |1 - \frac{\sum_{i'=1}^n e^{\beta K_{i'} \cdot Q_j}}{\sum_{i'=1}^n e^{\beta (K_{i'} \cdot Q_j + K_{i'} \cdot \Delta Q_j + Q_j \cdot \Delta K_{i'} + \Delta K_{i'} \cdot \Delta Q_j)}}|$$
$$\leq |1 - \frac{\sum_{i'=1}^n e^{\beta K_{i'} \cdot Q_j}}{\sum_{i'=1}^n e^{\beta (K_{i'} \cdot Q_j + 2d_h B_{KQV} \epsilon_0 + d_h \epsilon_0^2)}}|$$
$$= |1 - \frac{\sum_{i'=1}^n e^{\beta K_{i'} \cdot Q_j}}{e^{\beta d_h (2B_{KQV}\epsilon_0 + \epsilon_0^2)} \sum_{i'=1}^n e^{\beta K_{i'} \cdot Q_j}}| \qquad \left(e^{\beta d_h (2B_{KQV}\epsilon_0 + \epsilon_0^2)} \text{ is independent of } i'\right)$$
$$= |1 - e^{-\beta d_h (2B_{KQV}\epsilon_0 + \epsilon_0^2)}|, \tag{D.51}$$

and the last equality holds since the common factor $\sum_{i'=1}^n e^{\beta K_{i'} \cdot Q_j}$ cancels out.

Combining (D.49), (D.50), and (D.51), we have

$$|(\text{Softmax}_\beta (K'^\top Q') - \text{Softmax}_\beta (K^\top Q))_{i,j}|$$
$$< 2|1 - e^{-\beta d_h (2B_{KQV}\epsilon_0 + \epsilon_0^2)}|$$
$$\leq 2|1 - e^{-\beta d_h (2B_{KQV}\epsilon_0 + \epsilon_0)}|. \qquad \text{(By requiring } 0 < \epsilon_0 \leq 1)$$

Thus for any $0 < \epsilon_1 < 2$, when $\epsilon_0$ satisfies

$$0 < \epsilon_0 \leq \min\{1, \frac{-\ln(1 - \frac{\epsilon_1}{2})}{\beta d_h (2B_{KQV} + 1)}\},$$

we have

$$|(\text{Softmax}_\beta(K'^\top Q') - \text{Softmax}_\beta(K^\top Q))_{i,j}| < \epsilon_1. \tag{D.52}$$

From (D.47) and (D.52), we have

$$(II) \leq n\|V\|_\infty\|\Delta S\|_\infty < nB_{KQV}\epsilon_1, \tag{D.53}$$

since $\|V\|_\infty \leq B_{KQV}$ and $\|\Delta S\|_\infty < \epsilon_1$.

Combining (D.45) with (D.46) and (D.53) yields

$$\|\text{Attn}_s(\begin{bmatrix} K' \\ Q' \\ V' \end{bmatrix}) - \text{Attn}_s(\begin{bmatrix} K \\ Q \\ V \end{bmatrix})\|_\infty < \epsilon_0 + nB_{KQV}\epsilon_1.$$

When we take $\epsilon_0$ and $\epsilon_1$ to be infinitely small, the right-hand side tends to $0$.

This completes the proof. $\qquad\square$

### D.6    PROOF OF THEOREM 4.2

**Theorem D.6** (Theorem 4.2 Restate). Let $X \in \mathbb{R}^{d \times n}$ be the input sequence, and let $W_K, W_Q, W_V \in \mathbb{R}^{n \times d}$ be the weight matrices of the target attention. Assume $B = \max\{\|X\|_\infty, \|W_K\|_\infty, \|W_Q\|_\infty, \|W_V\|_\infty\}$ and $\|W_K X\|_\infty, \|W_Q X\|_\infty, \|W_V X\|_\infty \leq B_{KQV}$ for $B_{KQV} \geq 0$. Then, for any $\epsilon > 0$, there exists a single-head attention layer $\text{Attn}_s$ followed by a multi-head attention layer with linear projections such that

$$\|\text{Attn}_s \circ (\sum_{j=1}^{3n} \text{Attn}_j \circ \text{Linear}_j(\begin{bmatrix} X \\ W_K^\top \\ W_Q^\top \\ W_V^\top \end{bmatrix})) - \underbrace{W_V X}_{n \times n} \text{Softmax}_\beta \underbrace{((W_K X)^\top W_Q X)}_{n \times n}\|_\infty \leq \epsilon.$$

*Proof.* Follow our proof sketch in Appendix A.2, our proof consists of four conceptual steps.

**Step 1: Encoding Weights into the Input.**    For clarity and simplicity, we define

$$k_i := (W_K^\top)_{:,i} \in \mathbb{R}^d,$$
$$q_i := (W_Q^\top)_{:,i} \in \mathbb{R}^d,$$
$$v_i := (W_V^\top)_{:,i} \in \mathbb{R}^d,$$

such that $W_K^\top, W_Q^\top, W_V^\top$ writes out as

$$W_K^\top = \underbrace{[k_1 \quad k_2 \quad \cdots \quad k_n]}_{d \times n}, \quad W_Q^\top = \underbrace{[q_1 \quad q_2 \quad \cdots \quad q_n]}_{d \times n}, \quad W_V^\top = \underbrace{[v_1 \quad v_2 \quad \cdots \quad v_n]}_{d \times n}.$$

Then, we express the input as

$$\begin{bmatrix} X \\ W_K^\top \\ W_Q^\top \\ W_V^\top \end{bmatrix} = \underbrace{\begin{bmatrix} x_1 & x_2 & \cdots & x_n \\ k_1 & k_2 & \cdots & k_n \\ q_1 & q_2 & \cdots & q_n \\ v_1 & v_2 & \cdots & v_n \end{bmatrix}}_{4d \times n} \tag{D.54}$$

where $x_i, k_i, q_i$ and $v_i$ are all $d$ dimensional vectors for $i \in [n]$.

**Step 2: Multi-Head Approximation of $K, Q, V$.** For the simplicity of presentation, we define

$$K := \underbrace{W_K}_{n \times d} \underbrace{X}_{d \times n}, \quad Q := \underbrace{W_Q}_{n \times d} \underbrace{X}_{d \times n}, \quad V := \underbrace{W_V}_{n \times d} \underbrace{X}_{d \times n}.$$

Writing $W_K$, $W_Q$, and $W_V$ row-wise as

$$W_K = \underbrace{\begin{bmatrix} k_1^\top \\ k_2^\top \\ \vdots \\ k_n^\top \end{bmatrix}}_{n \times d}, \quad W_Q = \underbrace{\begin{bmatrix} q_1^\top \\ q_2^\top \\ \vdots \\ q_n^\top \end{bmatrix}}_{n \times d}, \quad W_V = \underbrace{\begin{bmatrix} v_1^\top \\ v_2^\top \\ \vdots \\ v_n^\top \end{bmatrix}}_{n \times d},$$

and $X = [x_1 \ \cdots \ x_n]$, we express $K$, $Q$, and $V$ entry-wise as

$$K = \begin{bmatrix} k_1^\top x_1 & k_1^\top x_2 & \cdots & k_1^\top x_n \\ k_2^\top x_1 & k_2^\top x_2 & \cdots & k_2^\top x_n \\ \vdots & \vdots & \vdots & \vdots \\ k_n^\top x_1 & k_n^\top x_2 & \cdots & k_n^\top x_n \end{bmatrix},$$

$$Q = \begin{bmatrix} q_1^\top x_1 & q_1^\top x_2 & \cdots & q_1^\top x_n \\ q_2^\top x_1 & q_2^\top x_2 & \cdots & q_2^\top x_n \\ \vdots & \vdots & \vdots & \vdots \\ q_n^\top x_1 & q_n^\top x_2 & \cdots & q_n^\top x_n \end{bmatrix},$$

$$V = \begin{bmatrix} v_1^\top x_1 & v_1^\top x_2 & \cdots & v_1^\top x_n \\ v_2^\top x_1 & v_2^\top x_2 & \cdots & v_2^\top x_n \\ \vdots & \vdots & \vdots & \vdots \\ v_n^\top x_1 & v_n^\top x_2 & \cdots & v_n^\top x_n \end{bmatrix}.$$

Here $k_i^\top$, $q_i^\top$, and $v_i^\top$ identify the $i$-th row of $K, Q$ and $V$, while $x_j$ identifies the $j$-th column.

In this section, our goal is to approximate $K$, $Q$, and $V$. Our strategy is to approximate $K$, $Q$, and $V$ row by row, and within each row, entry by entry. More precisely, for each $i \in [n]$, we approximate

$$k_i^\top X, \quad q_i^\top X, \quad v_i^\top X,$$

by approximating the scalar products

$$k_i^\top x_j, \quad q_i^\top x_j, \quad v_i^\top x_j, \quad \text{for all } j \in [n],$$

and then collecting these approximations to form approximations of the full matrices $K$, $Q$, and $V$.

To approximate each scalar $k_i^\top x_j$, $q_i^\top x_j$, and $v_i^\top x_j$, we first determine their joint range over all $i, j \in [n]$. Within this joint range, we construct a set of uniform-space grid points. Then, we approximate each target entry $k_i^\top x_j$, $q_i^\top x_j$, or $v_i^\top x_j$ by an entry-specific weighted sum of these grid points, where grid points closer to the target entry value receive larger weights. In this way, we represent every entry by its own set of interpolation weights, while all approximations share the same global grid.

We introduce our notation for the uniform grid points used in our interpolation scheme.

**Interpolations.** We recall

$$B = \max(\|X\|_\infty, \|W_K\|_\infty, \|W_Q\|_\infty, \|W_V\|_\infty).$$

Thus, for all $i, j \in [n]$,

$$-d \cdot B^2 \leq k_i^\top x_j, q_i^\top x_j, v_i^\top x_j \leq d \cdot B^2. \tag{D.55}$$

Namely, $[-dB^2, dB^2]$ contains all entries of $K$, $Q$, and $V$.

Then, we take $L_0 := -dB^2$ and $L_P := dB^2$ as the two endpoints of our interpolation and define for $i \in \{0\} \cup [P]$

$$L_i := \frac{iL_P + (P - i)L_0}{P}, \tag{D.56}$$

where $P$ is the number of interpolation steps (the number of equal divisions of $[L_0, L_P]$). The points $\{L_i\}_{i=0}^{P}$ form our uniform grid over the target range.

We use $\Delta L$ to denote the length of the interval between two neighboring grid points. We have

$$\Delta L := \frac{L_P - L_0}{P} = \frac{2dB^2}{P}. \tag{D.57}$$

Now we have all the ingredients needed to approximate each entry using a weighted sum. However, the input,

$$\begin{bmatrix} X \\ W_K^\top \\ W_Q^\top \\ W_V^\top \end{bmatrix} = \underbrace{\begin{bmatrix} x_1 & x_2 & \cdots & x_n \\ k_1 & k_2 & \cdots & k_n \\ q_1 & q_2 & \cdots & q_n \\ v_1 & v_2 & \cdots & v_n \end{bmatrix}}_{4d \times n},$$

contains information from all rows in the target $K$, $Q$, and $V$, but does not contain the grid points. We need a mechanism to select a specific $k_i$, $q_i$, or $v_i$ (corresponding to one row of $K$, $Q$, or $V$) and to include the grid points for us.

To address this, we introduce row-specific linear transformations $\text{Linear}_j$, where $j \in [3n]$, since we have $n$ rows for each of $K$, $Q$, and $V$. Each $\text{Linear}_j$ serves two purposes: it incorporates the input and the uniform grid points, and selects the $k_i$, $q_i$, or $v_i$ associated with index $i$ (corresponding to one row of $K$, $Q$, or $V$).

For the clarity of presentation, we relabel these $3n$ linear transformations according to whether they are responsible for $K$, $Q$, or $V$

$$\begin{aligned} \text{Linear}_j^K &:= \text{Linear}_j, \quad j \in [n], &&\text{(Responsible for } K) \\ \text{Linear}_j^Q &:= \text{Linear}_{n+j}, \quad j \in [n], &&\text{(Responsible for } Q) \\ \text{Linear}_j^V &:= \text{Linear}_{2n+j}, \quad j \in [n]. && \end{aligned} \tag{D.58}$$

Later in the proof, we specify the explicit form of these $\text{Linear}_j$.

So far, $\text{Linear}_j$ allows us to combine the input with the uniform grid points and to select the desired $k_i$, $q_i$, or $v_i$. The next step is to *implement* the entry-specific weighted sums to approximate the entries of $K$, $Q$, and $V$.

For this, we use a row-specific single-head attention: for each $i \in [n]$, we assign one head to approximate $k_i^\top X$ using the weighted sum, one head to approximate $q_i^\top X$ in the same manner, and one head to approximate $v_i^\top X$ in the same manner. Each such head operates token-wise: given its designated row $i$, the head approximates all scalars $k_i^\top x_j$, $q_i^\top x_j$, or $v_i^\top x_j$ across $j \in [n]$.

Since each of $K$, $Q$, and $V$ has $n$ rows and we use a single-head for each row, we use a total of $3n$ heads to approximate $K$, $Q$, and $V$. We use $\text{Attn}_j$ to label these $3n$ heads and $j \in [3n]$.

Again, for the clarity of presentation, we provide another equivalent way, as $\text{Attn}_j$, to label these $3n$ heads

$$
\begin{aligned}
\text{Attn}_j^K &:= \text{Attn}_j, \quad j \in [n], \\
\text{Attn}_j^Q &:= \text{Attn}_{n+j}, \quad j \in [n], \\
\text{Attn}_j^V &:= \text{Attn}_{2n+j}, \quad j \in [n].
\end{aligned}
\tag{D.59}
$$

Later in our proof, we provide the construction of these $\text{Attn}_j$ explicitly.

Now we are ready to approximate each of $K, Q$, and $V$. We approximate $K$ first to demonstrate our procedure and deal with $Q$ and $V$ in a similar manner later.

**In-Context Calculation of $K$.** First, we define the linear transformation $\text{Linear}_j^K : \mathbb{R}^{4d \times n} \to \mathbb{R}^{(2d+3) \times (P+1)}$ attached before $\text{Attn}_j^K$ as:

$$
\text{Linear}_j^K(Z) := \underbrace{\begin{bmatrix} 0_{d \times d} & 0_{d \times d} & 0_{d \times 2d} \\ 0_{d \times d} & I_d & 0_{d \times 2d} \\ 0_{3 \times d} & 0_{3 \times d} & 0_{3 \times 2d} \end{bmatrix}}_{(2d+3) \times 4d} \underbrace{Z}_{4d \times n} \underbrace{\begin{bmatrix} 2L_0 e_j^{(n)} & 2L_1 e_j^{(n)} & \cdots & 2L_P e_j^{(n)} \end{bmatrix}}_{n \times (P+1)} +
$$

$$
\left( Z \in \mathbb{R}^{4d \times n} \text{ denotes any input} \right)
$$

$$
\underbrace{\begin{bmatrix} I_d & 0_{d \times d} & 0_{d \times 2d} \\ 0_{d \times d} & 0_{d \times d} & 0_{d \times 2d} \\ 0_{3 \times d} & 0_{3 \times d} & 0_{3 \times 2d} \end{bmatrix}}_{(2d+3) \times 4d} \underbrace{Z}_{4d \times n} \underbrace{\begin{bmatrix} I_n & 0_{n \times (P+1-n)} \end{bmatrix}}_{n \times (P+1)} + \underbrace{\begin{bmatrix} 0_{2d \times (P+1)} \\ M_1 \\ M_L \end{bmatrix}}_{(2d+3) \times (P+1)},
$$

where $M_1, M_L$ are

$$
M_1 := \underbrace{\begin{bmatrix} 1_{1 \times n} & 0_{1 \times (P+1-n)} \end{bmatrix}}_{1 \times (P+1)},
\tag{D.60}
$$

$$
M_L := \underbrace{\begin{bmatrix} L_0 & L_1 & \cdots & L_P \\ -L_0^2 & -L_1^2 & \cdots & -L_P^2 \end{bmatrix}}_{2 \times (P+1)}.
\tag{D.61}
$$

The $\text{Linear}_j^K$ layer takes the input $\begin{bmatrix} X^\top & W_K & W_Q & W_V \end{bmatrix}^\top$ and outputs in the following way:

$$
\text{Linear}_j^K \left( \begin{bmatrix} X \\ W_K^\top \\ W_Q^\top \\ W_V^\top \end{bmatrix} \right)
$$

$$
= \underbrace{\begin{bmatrix} 0_{d \times d} & 0_{d \times d} & 0_{d \times 2d} \\ 0_{d \times d} & I_d & 0_{d \times 2d} \\ 0_{3 \times d} & 0_{3 \times d} & 0_{3 \times 2d} \end{bmatrix}}_{(2d+3) \times 4d} \underbrace{\begin{bmatrix} x_1 & x_2 & \cdots & x_n \\ k_1 & k_2 & \cdots & k_n \\ q_1 & q_2 & \cdots & q_n \\ v_1 & v_2 & \cdots & v_n \end{bmatrix}}_{4d \times n} \underbrace{\begin{bmatrix} 2L_0 e_j^{(n)} & 2L_1 e_j^{(n)} & \cdots & 2L_P e_j^{(n)} \end{bmatrix}}_{n \times (P+1)} +
$$

$$
\underbrace{\begin{bmatrix} I_d & 0_{d \times d} & 0_{d \times 2d} \\ 0_{d \times d} & 0_{d \times d} & 0_{d \times 2d} \\ 0_{3 \times d} & 0_{3 \times d} & 0_{3 \times 2d} \end{bmatrix}}_{(2d+3) \times 4d} \underbrace{\begin{bmatrix} x_1 & x_2 & \cdots & x_n \\ k_1 & k_2 & \cdots & k_n \\ q_1 & q_2 & \cdots & q_n \\ v_1 & v_2 & \cdots & v_n \end{bmatrix}}_{4d \times n} \underbrace{\begin{bmatrix} I_n & 0_{n \times (P+1-n)} \end{bmatrix}}_{n \times (P+1)} + \underbrace{\begin{bmatrix} 0_{2d \times (P+1)} \\ M_1 \\ M_L \end{bmatrix}}_{(2d+3) \times (P+1)} \quad (\text{By (D.54)})
$$

$$
= \underbrace{\begin{bmatrix} 0_{d\times 1} & 0_{d\times 1} & \cdots & 0_{d\times 1} \\ k_1 & k_2 & \cdots & k_n \\ 0_{3\times 1} & 0_{3\times 1} & \cdots & 0_{3\times 1} \end{bmatrix}}_{(2d+3)\times n} \underbrace{\begin{bmatrix} 2L_0 e_j^{(n)} & 2L_1 e_j^{(n)} & \cdots & 2L_P e_j^{(n)} \end{bmatrix}}_{n\times(P+1)} +
$$

$$
\underbrace{\begin{bmatrix} x_1 & x_2 & \cdots & x_n \\ 0_{d\times 1} & 0_{d\times 1} & \cdots & 0_{d\times 1} \\ 0_{3\times 1} & 0_{3\times 1} & \cdots & 0_{3\times 1} \end{bmatrix}}_{(2d+3)\times n} \underbrace{\begin{bmatrix} I_n & 0_{n\times(P+1-n)} \end{bmatrix}}_{n\times(P+1)} + \underbrace{\begin{bmatrix} 0_{2d\times(P+1)} \\ M_1 \\ M_L \end{bmatrix}}_{(2d+3)\times(P+1)}
$$

$$
\left( \text{By selecting } k_i \text{ and } x_i \text{ with } I_d \text{ for all } i \in [n] \right)
$$

$$
= \underbrace{\begin{bmatrix} 0_{d\times 1} & 0_{d\times 1} & \cdots & 0_{d\times 1} \\ 2L_0 k_j & 2L_1 k_j & \cdots & 2L_P k_j \\ 0_{3\times 1} & 0_{3\times 1} & \cdots & 0_{3\times 1} \end{bmatrix}}_{(2d+3)\times(P+1)} + \underbrace{\begin{bmatrix} x_1 & x_2 & \cdots & x_n & 0_{d\times 1} & \cdots & 0_{d\times 1} \\ 0_{d\times 1} & 0_{d\times 1} & \cdots & 0_{d\times 1} & 0_{d\times 1} & \cdots & 0_{d\times 1} \\ 0_{3\times 1} & 0_{3\times 1} & \cdots & 0_{3\times 1} & 0_{3\times 1} & \cdots & 0_{3\times 1} \end{bmatrix}}_{(2d+3)\times(P+1)} +
$$

$$
\left( \text{By selecting } k_j \text{ with } e_j^{(n)} \right)
$$

$$
\underbrace{\begin{bmatrix} 0_{2d\times 1} & 0_{2d\times 1} & \cdots & 0_{2d\times 1} & 0_{2d\times 1} & \cdots & 0_{2d\times 1} \\ 1 & 1 & \cdots & 1 & 0 & \cdots & 0 \\ L_0 & L_1 & \cdots & L_{n-1} & L_n & \cdots & L_P \\ -L_0^2 & -L_1^2 & \cdots & -L_{n-1}^2 & -L_n^2 & \cdots & -L_P^2 \end{bmatrix}}_{(2d+3)\times(P+1)}
$$

$$
\left( \text{By the definition of } M_1 \text{ and } M_L; \text{ i.e., (D.60) and (D.61)} \right)
$$

$$
= \underbrace{\begin{bmatrix} x_1 & x_2 & \cdots & x_n & 0_d & \cdots & 0_d \\ 2L_0 k_j & 2L_1 k_j & \cdots & 2L_{n-1} k_j & 2L_n k_j & \cdots & 2L_P k_j \\ 1 & 1 & \cdots & 1 & 0 & \cdots & 0 \\ L_0 & L_1 & \cdots & L_{n-1} & L_n & \cdots & L_P \\ -L_0^2 & -L_1^2 & \cdots & -L_{n-1}^2 & -L_n^2 & \cdots & -L_P^2 \end{bmatrix}}_{(2d+3)\times(P+1)}. \tag{D.62}
$$

Next, we construct $\mathrm{Attn}_j^K : \mathbb{R}^{(2d+3)\times(P+1)} \to \mathbb{R}^{3n\times n}$ to be

$$
\mathrm{Attn}_j^K(D) := \underbrace{W_{\widehat{V}}^{K;j} D}_{3n\times(P+1)} \cdot \underbrace{\mathrm{Softmax}_\beta((W_{\widehat{K}}^{K;j} D)^\top W_{\widehat{Q}}^{K;j} D)}_{(P+1)\times(P+1)} \cdot \underbrace{W_{\widehat{O}}^{K;j}}_{(P+1)\times n},
$$

where $D \in \mathbb{R}^{(2d+3)\times(P+1)}$ denotes any input, and

$$
W_{\widehat{K}}^{K;j} := \underbrace{\begin{bmatrix} 0_{d\times d} & I_d & 0_{d\times 1} & 0_{d\times 1} & 0_{d\times 1} \\ 0_{1\times d} & 0_{1\times d} & 0 & 0 & 1 \end{bmatrix}}_{(d+1)\times(2d+3)}, \tag{D.63}
$$

$$
W_{\widehat{Q}}^{K;j} := \underbrace{\begin{bmatrix} I_d & 0_{d\times d} & 0_{d\times 1} & 0_{d\times 1} & 0_{d\times 1} \\ 0_{1\times d} & 0_{1\times d} & 1 & 0 & 0 \end{bmatrix}}_{(d+1)\times(2d+3)}, \tag{D.64}
$$

$$
W_{\widehat{V}}^{K;j} := \underbrace{e_j^{(3n)}}_{3n\times 1} \underbrace{\begin{bmatrix} 0_{1\times(2d+1)} & 1 & 0 \end{bmatrix}}_{1\times(2d+3)}, \tag{D.65}
$$

$$
W_{\widehat{O}}^{K;j} := \underbrace{\begin{bmatrix} I_n \\ 0_{(P+1-n)\times n} \end{bmatrix}}_{(P+1)\times n}. \tag{D.66}
$$

We define the $\widehat{K}_j^K$ of $\mathrm{Attn}_j^K$ to be:

$$
\widehat{K}_j^K := W_{\widehat{K}}^{K;j} \cdot \mathrm{Linear}_j^K \left( \begin{bmatrix} X \\ W_K^\top \\ W_Q^\top \\ W_V^\top \end{bmatrix} \right)
$$

$$
= \underbrace{\begin{bmatrix} 0_{d\times d} & I_d & 0_{d\times 1} & 0_{d\times 1} & 0_{d\times 1} \\ 0_{1\times d} & 0_{1\times d} & 0 & 0 & 1 \end{bmatrix}}_{(d+1)\times(2d+3)} \cdot \underbrace{\begin{bmatrix} x_1 & x_2 & \cdots & x_n & \cdots & 0_d \\ 2L_0 k_j & 2L_1 k_j & \cdots & 2L_{n-1}k_j & \cdots & 2L_P k_j \\ 1 & 1 & \cdots & 1 & \cdots & 0 \\ L_0 & L_1 & \cdots & L_{n-1} & \cdots & L_P \\ -L_0^2 & -L_1^2 & \cdots & -L_{n-1}^2 & \cdots & -L_P^2 \end{bmatrix}}_{(2d+3)\times(P+1)}
$$

$$
\textcolor{gray}{(\text{By (D.63) and (D.62)})}
$$

$$
= \underbrace{\begin{bmatrix} 2L_0 k_j & 2L_1 k_j & \cdots & 2L_{n-1}k_j & \cdots & 2L_P k_j \\ -L_0^2 & -L_1^2 & \cdots & -L_{n-1}^2 & \cdots & -L_P^2 \end{bmatrix}}_{(d+1)\times(P+1)}, \tag{D.67}
$$

and the last equality holds since $I_d$ selects the $2L_i k_j$ row, and $1$ selects the $-L_i^2$ row where $i \in \{0\} \cup [P]$.

We define the $\widehat{Q}_j^K$ of $\mathrm{Attn}_j^K$ to be:

$$
\widehat{Q}_j^K := W_{\widehat{Q}}^{K;j} \cdot \mathrm{Linear}_j^K \left( \begin{bmatrix} X \\ W_K^\top \\ W_Q^\top \\ W_V^\top \end{bmatrix} \right)
$$

$$
= \underbrace{\begin{bmatrix} I_d & 0_{d\times d} & 0_{d\times 1} & 0_{d\times 1} & 0_{d\times 1} \\ 0_{1\times d} & 0_{1\times d} & 1 & 0 & 0 \end{bmatrix}}_{(d+1)\times(2d+3)} \cdot \underbrace{\begin{bmatrix} x_1 & x_2 & \cdots & x_n & \cdots & 0_d \\ 2L_0 k_j & 2L_1 k_j & \cdots & 2L_{n-1}k_j & \cdots & 2L_P k_j \\ 1 & 1 & \cdots & 1 & \cdots & 0 \\ L_0 & L_1 & \cdots & L_{n-1} & \cdots & L_P \\ -L_0^2 & -L_1^2 & \cdots & -L_{n-1}^2 & \cdots & -L_P^2 \end{bmatrix}}_{(2d+3)\times(P+1)}
$$

$$
\textcolor{gray}{(\text{By (D.64) and (D.62)})}
$$

$$
= \underbrace{\begin{bmatrix} x_1 & x_2 & \cdots & x_n & 0_{d\times(P+1-n)} \\ 1 & 1 & \cdots & 1 & 0_{1\times(P+1-n)} \end{bmatrix}}_{(d+1)\times(P+1)}, \tag{D.68}
$$

and the last equality holds since $I_d$ selects the $x_i$ row where $i \in [n]$, and $1$ selects the $1_\mathrm{s}$ row.

We define the $\widehat{V}_j^K$ of $\mathrm{Attn}_j^K$ to be:

$$
\widehat{V}_j^K := W_{\widehat{V}}^{K;j} \cdot \mathrm{Linear}_j^K \left( \begin{bmatrix} X \\ W_K^\top \\ W_Q^\top \\ W_V^\top \end{bmatrix} \right)
$$

$$
= \underbrace{e_j^{(3n)}}_{3n\times 1} \underbrace{\begin{bmatrix} 0_{1\times(2d+1)} & 1 & 0 \end{bmatrix}}_{1\times(2d+3)} \cdot \underbrace{\begin{bmatrix} x_1 & x_2 & \cdots & x_n & 0_d & \cdots & 0_d \\ 2L_0 k_j & 2L_1 k_j & \cdots & 2L_{n-1}k_j & 2L_n k_j & \cdots & 2L_P k_j \\ 1 & 1 & \cdots & 1 & 0 & \cdots & 0 \\ L_0 & L_1 & \cdots & L_{n-1} & L_n & \cdots & L_P \\ -L_0^2 & -L_1^2 & \cdots & -L_{n-1}^2 & -L_n^2 & \cdots & -L_P^2 \end{bmatrix}}_{(2d+3)\times(P+1)}
$$

$$
\textcolor{gray}{(\text{By (D.65) and (D.62)})}
$$

$$= \underbrace{e_j^{(3n)}}_{3n \times 1} \underbrace{\begin{bmatrix} L_0 & L_1 & \cdots & L_{n-1} & L_n & \cdots & L_P \end{bmatrix}}_{1 \times (P+1)}, \tag{D.69}$$

and the last equality holds since the $1$ selects the $L_i$ row where $i \in \{0\} \cup [P]$.

Combining the results of $\widehat{K}_j^K$ and $\widehat{Q}_j^K$, we calculate the $\text{Softmax}_\beta((\widehat{K}_j^K)^\top \widehat{Q}_j^K)$ in $\text{Attn}_j^K$ as

$$\text{Softmax}_\beta((\widehat{K}_j^K)^\top \widehat{Q}_j^K)$$

$$= \text{Softmax}_\beta(\underbrace{\begin{bmatrix} 2L_0 k_j & 2L_1 k_j & \cdots & 2L_{n-1} k_j & \cdots & 2L_P k_j \\ -L_0^2 & -L_1^2 & \cdots & -L_{n-1}^2 & \cdots & -L_P^2 \end{bmatrix}^\top}_{(P+1)\times(d+1)} \underbrace{\begin{bmatrix} x_1 & x_2 & \cdots & x_n & 0_{d\times(P+1-n)} \\ 1 & 1 & \cdots & 1 & 0_{1\times(P+1-n)} \end{bmatrix}}_{(d+1)\times(P+1)})$$

$$\text{(By the definition of } \widehat{K}_j^K \text{ and } \widehat{Q}_j^K; \text{ i.e. (D.67) and (D.68))}$$

$$= \text{Softmax}_\beta(\underbrace{\begin{bmatrix} 2L_0 k_j^\top & -L_0^2 \\ 2L_1 k_j^\top & -L_1^2 \\ \vdots & \vdots \\ 2L_P k_j^\top & -L_P^2 \end{bmatrix}}_{(P+1)\times(d+1)} \underbrace{\begin{bmatrix} x_1 & x_2 & \cdots & x_n & 0_{d\times(P+1-n)} \\ 1 & 1 & \cdots & 1 & 0_{1\times(P+1-n)} \end{bmatrix}}_{(d+1)\times(P+1)}) \quad \text{(By the transpose of } \widehat{K}_j^K)$$

$$= \text{Softmax}_\beta(\underbrace{\begin{bmatrix} 2L_0 k_j^\top x_1 - L_0^2 & 2L_0 k_j^\top x_2 - L_0^2 & \cdots & 2L_0 k_j^\top x_n - L_0^2 & 0_{d\times(P+1-n)} \\ 2L_1 k_j^\top x_1 - L_1^2 & 2L_1 k_j^\top x_2 - L_1^2 & \cdots & 2L_1 k_j^\top x_n - L_1^2 & 0_{d\times(P+1-n)} \\ \vdots & \vdots & & \vdots & \vdots \\ 2L_P k_j^\top x_1 - L_P^2 & 2L_P k_j^\top x_2 - L_P^2 & \cdots & 2L_P k_j^\top x_n - L_P^2 & 0_{d\times(P+1-n)} \end{bmatrix}}_{(P+1)\times(P+1)})$$

$$\text{(By matrix multiplication)}$$

$$= \text{Softmax}_\beta(\underbrace{\begin{bmatrix} -(k_j^\top x_1 - L_0)^2 + (k_j^\top x_1)^2 & \cdots & -(k_j^\top x_n - L_0)^2 + (k_j^\top x_n)^2 & 0_{d\times(P+1-n)} \\ -(k_j^\top x_1 - L_1)^2 + (k_j^\top x_1)^2 & \cdots & -(k_j^\top x_n - L_1)^2 + (k_j^\top x_n)^2 & 0_{d\times(P+1-n)} \\ \vdots & & \vdots & \vdots \\ -(k_j^\top x_1 - L_P)^2 + (k_j^\top x_1)^2 & \cdots & -(k_j^\top x_n - L_P)^2 + (k_j^\top x_n)^2 & 0_{d\times(P+1-n)} \end{bmatrix}}_{(P+1)\times(P+1)})$$

$$\left(2L_i k_j^\top x_m - L_i^2 = -(k_j^\top x_m - L_i)^2 + (k_j^\top x_m)^2 \text{ where } i \in \{0\} \cup [P] \text{ and } m \in [n]\right)$$

$$= \text{Softmax}_\beta(\underbrace{\begin{bmatrix} -(k_j^\top x_1 - L_0)^2 & \cdots & -(k_j^\top x_n - L_0)^2 & 0_{d\times(P+1-n)} \\ -(k_j^\top x_1 - L_1)^2 & \cdots & -(k_j^\top x_n - L_1)^2 & 0_{d\times(P+1-n)} \\ \vdots & & \vdots & \vdots \\ -(k_j^\top x_1 - L_P)^2 & \cdots & -(k_j^\top x_n - L_P)^2 & 0_{d\times(P+1-n)} \end{bmatrix}}_{(P+1)\times(P+1)}), \tag{D.70}$$

and the last line holds since the following property of $\text{Softmax}_\beta$

$$\text{Softmax}_\beta(v) = \text{Softmax}_\beta(v + C \cdot 1_{(P+1)\times 1}),$$

for any vector $v \in \mathbb{R}^{P+1}$ and $C \in \mathbb{R}$.

From (D.70), we have

$$\text{Softmax}_\beta((\widehat{K}_j^K)^\top \widehat{Q}_j^K) \cdot W_{\widehat{O}}^{K;j}$$

$$= \text{Softmax}_\beta \left( \underbrace{\begin{bmatrix} -(k_j^\top x_1 - L_0)^2 & \cdots & -(k_j^\top x_n - L_0)^2 & 0_{d\times(P-n+1)} \\ -(k_j^\top x_1 - L_1)^2 & \cdots & -(k_j^\top x_n - L_1)^2 & 0_{d\times(P-n+1)} \\ \vdots & & \vdots & \vdots \\ -(k_j^\top x_1 - L_P)^2 & \cdots & -(k_j^\top x_n - L_P)^2 & 0_{d\times(P-n+1)} \end{bmatrix}}_{(P+1)\times(P+1)} \right) \underbrace{\begin{bmatrix} I_n \\ 0_{(P+1-n)\times n} \end{bmatrix}}_{(P+1)\times n}$$

$$= \text{Softmax}_\beta \left( \underbrace{\begin{bmatrix} -(k_j^\top x_1 - L_0)^2 & \cdots & -(k_j^\top x_n - L_0)^2 \\ -(k_j^\top x_1 - L_1)^2 & \cdots & -(k_j^\top x_n - L_1)^2 \\ \vdots & & \vdots \\ -(k_j^\top x_1 - L_P)^2 & \cdots & -(k_j^\top x_n - L_P)^2 \end{bmatrix}}_{(P+1)\times n} \right), \tag{D.71}$$

where the last line follows from the column-wise nature of the $\text{Softmax}_\beta()$ function.

From (D.71), we have

$$(\text{Softmax}_\beta((\widehat{K}_j^K)^\top \widehat{Q}_j^K) \cdot W_{\widehat{O}}^{K;j})_{r,c} = \frac{e^{-\beta(L_r - k_j^\top x_c)^2}}{\sum_{s=0}^{P} e^{-\beta(L_s - k_j^\top x_c)^2}},$$

for every $r \in \{0\} \cup [P]$ and $c \in [n]$.

Thus, for each column in (D.71), we have

$$(\text{Softmax}_\beta((\widehat{K}_j^K)^\top \widehat{Q}_j^K) \cdot W_{\widehat{O}}^{K;j})_{:,c} = \underbrace{\sum_{r=0}^{P} \frac{e^{-\beta(L_r - k_j^\top x_c)^2}}{\sum_{s=0}^{P} e^{-\beta(L_s - k_j^\top x_c)^2}} e_{r+1}^{(P+1)}}_{(P+1)\times 1}. \tag{D.72}$$

Combining $\widehat{V}_j^K$ and $(\text{Softmax}_\beta((\widehat{K}_j^K)^\top \widehat{Q}_j^K) \cdot W_{\widehat{O}}^{K;j})_{:,c}$, we obtain

$$\widehat{V}_j^K \cdot (\text{Softmax}_\beta((\widehat{K}_j^K)^\top \widehat{Q}_j^K) \cdot W_{\widehat{O}}^{K;j})_{:,c}$$

$$= \underbrace{e_j^{(3n)}}_{3n\times 1} \underbrace{\begin{bmatrix} L_0 & L_1 & \cdots & L_{n-1} & L_n & \cdots & L_P \end{bmatrix}}_{1\times(P+1)} \underbrace{\sum_{r=0}^{P} \frac{e^{-\beta(L_r - k_j^\top x_c)^2}}{\sum_{s=0}^{P} e^{-\beta(L_s - k_j^\top x_c)^2}} e_{r+1}^{(P+1)}}_{(P+1)\times 1}$$
$$\left(\text{By (D.69) and (D.72)}\right)$$

$$= \underbrace{e_j^{(3n)}}_{3n\times 1} \sum_{r=0}^{P} \frac{e^{-\beta(L_r - k_j^\top x_c)^2}}{\sum_{s=0}^{P} e^{-\beta(L_s - k_j^\top x_c)^2}} \underbrace{\begin{bmatrix} L_0 & L_1 & \cdots & L_{n-1} & L_n & \cdots & L_P \end{bmatrix}}_{1\times(P+1)} \underbrace{e_{r+1}^{(P+1)}}_{(P+1)\times 1}$$
$$\left(\text{By the distributivity of matrix multiplication}\right)$$

$$= \underbrace{e_j^{(3n)}}_{3n\times 1} \underbrace{\sum_{r=0}^{P} \frac{e^{-\beta(L_r - k_j^\top x_c)^2}}{\sum_{s=0}^{P} e^{-\beta(L_s - k_j^\top x_c)^2}} L_r}_{\text{scalar}} \qquad \left(e_{r+1}^{(P+1)} \text{ selects } L_r \text{ for every } r \in \{0\} \cup [P]\right)$$

$$= \sum_{r=0}^{P} \frac{e^{-\beta(L_r - k_j^\top x_c)^2}}{\sum_{s=0}^{P} e^{-\beta(L_s - k_j^\top x_c)^2}} L_r e_j^{(3n)}, \tag{D.73}$$

for every $c \in [n]$.

Hence,

$$\widehat{V}_j^K \cdot (\text{Softmax}_\beta((\widehat{K}_j^K)^\top \widehat{Q}_j^K) \cdot W_{\widehat{O}}^{K;j})_{:,c},$$

is a weighted average of the vectors $L_r e_j^{(3n)}$, with weights depending on $\beta$ and the distance between $L_r$ and $k_j^\top x_c$.

We recall: $\widehat{V}_j^K \cdot (\text{Softmax}_\beta((\widehat{K}_j^K)^\top \widehat{Q}_j^K) \cdot W_{\widehat{O}}^{K;j})_{:,c}$ gives the $c$-th column of $\text{Attn}_j^K$. Therefore, each column of $\text{Attn}_j^K$ stores a weighted sum as an approximator for each entry in $k_j^\top X$.

We show that (D.73) is close to $k_j^\top x_c$:

$$\|\widehat{V}_j^K \cdot (\text{Softmax}_\beta((\widehat{K}_j^K)^\top \widehat{Q}_j^K) \cdot W_{\widehat{O}}^{K;j})_{:,c} - k_j^\top x_c \cdot e_j^{(3n)}\|_\infty$$

$$= \|\underbrace{\sum_{r=0}^{P} \frac{e^{-\beta(L_r - k_j^\top x_c)^2}}{\sum_{s=0}^{P} e^{-\beta(L_s - k_j^\top x_c)^2}} L_r}_{\text{scalar}} \cdot e_j^{(3n)} - \underbrace{k_j^\top x_c}_{\text{scalar}} \cdot e_j^{(3n)}\|_\infty \qquad \text{(By (D.73))}$$

$$= \|(\sum_{r=0}^{P} \frac{e^{-\beta(L_r - k_j^\top x_c)^2}}{\sum_{s=0}^{P} e^{-\beta(L_s - k_j^\top x_c)^2}} L_r - k_j^\top x_c) \cdot e_j^{(3n)}\|_\infty$$

$$= |\sum_{r=0}^{P} \frac{e^{-\beta(L_r - k_j^\top x_c)^2}}{\sum_{s=0}^{P} e^{-\beta(L_s - k_j^\top x_c)^2}} L_r - k_j^\top x_c| \qquad \text{(We have one non-zero entry in } e_j^{(3n)}\text{)}$$

$$= |\sum_{r=0}^{P} \frac{e^{-\beta(L_r - k_j^\top x_c)^2}}{\sum_{s=0}^{P} e^{-\beta(L_s - k_j^\top x_c)^2}} L_r - \frac{\sum_{r=0}^{P} e^{-\beta(L_r - k_j^\top x_c)^2}}{\sum_{s=0}^{P} e^{-\beta(L_s - k_j^\top x_c)^2}} k_j^\top x_c|$$

$$\text{(By } (\sum_{r=0}^{P} e^{-\beta(L_r - k_j^\top x_c)^2})/(\sum_{s=0}^{P} e^{-\beta(L_s - k_j^\top x_c)^2}) = 1)$$

$$= |\sum_{r=0}^{P} \frac{e^{-\beta(L_r - k_j^\top x_c)^2}}{\sum_{s=0}^{P} e^{-\beta(L_s - k_j^\top x_c)^2}} (L_r - k_j^\top x_c)|$$

$$= |\sum_{r:|L_r - k_j^\top x_c| < \Delta L} \frac{e^{-\beta(L_r - k_j^\top x_c)^2}}{\sum_{s=0}^{P} e^{-\beta(L_s - k_j^\top x_c)^2}} (L_r - k_j^\top x_c) + \sum_{r:|L_r - k_j^\top x_c| \geq \Delta L} \frac{e^{-\beta(L_r - k_j^\top x_c)^2}}{\sum_{s=0}^{P} e^{-\beta(L_s - k_j^\top x_c)^2}} (L_r - k_j^\top x_c)|$$

$$\text{(By dividing the } L_r \text{ into two groups: one within } \Delta L \text{ away from } k_j^\top x_c, \text{ one at least } \Delta L \text{ away from } k_j^\top x_c\text{)}$$

$$\leq |\sum_{r:|L_r - k_j^\top x_c| < \Delta L} \frac{e^{-\beta(L_r - k_j^\top x_c)^2}}{\sum_{s=0}^{P} e^{-\beta(L_s - k_j^\top x_c)^2}} (L_r - k_j^\top x_c)| + |\sum_{r:|L_r - k_j^\top x_c| \geq \Delta L} \frac{e^{-\beta(L_r - k_j^\top x_c)^2}}{\sum_{s=0}^{P} e^{-\beta(L_s - k_j^\top x_c)^2}} (L_r - k_j^\top x_c)|$$

$$\text{(By triangle inequality)}$$

$$\leq \underbrace{\sum_{r:|L_r - k_j^\top x_c| < \Delta L} \frac{e^{-\beta(L_r - k_j^\top x_c)^2}}{\sum_{s=0}^{P} e^{-\beta(L_s - k_j^\top x_c)^2}} |L_r - k_j^\top x_c|}_{:=(I)} + \underbrace{\sum_{r:|L_r - k_j^\top x_c| \geq \Delta L} \frac{e^{-\beta(L_r - k_j^\top x_c)^2}}{\sum_{s=0}^{P} e^{-\beta(L_s - k_j^\top x_c)^2}} |L_r - k_j^\top x_c|}_{:=(II)},$$

$$\text{(D.74)}$$

and the last inequality holds due to the triangle inequality and the non-negativity of the exponential function.

For term $(I)$ in (D.74), we have

$$(I)$$

$$= \sum_{r:|L_r - k_j^\top x_c| < \Delta L} \frac{e^{-\beta(L_r - k_j^\top x_c)^2}}{\sum_{s=0}^{P} e^{-\beta(L_s - k_j^\top x_c)^2}} |L_r - k_j^\top x_c| \qquad \text{(By the definition of term } (I) \text{ in (D.74))}$$

$$< \sum_{r:|L_r - k_j^\top x_c| < \Delta L} \frac{e^{-\beta(L_r - k_j^\top x_c)^2}}{\sum_{s=0}^{P} e^{-\beta(L_s - k_j^\top x_c)^2}} \Delta L \qquad \text{(In this group of } L_r, |L_r - k_j^\top x_c| < \Delta L\text{)}$$

$$\leq \Delta L, \qquad \text{(D.75)}$$

and the last inequality holds since

$$\frac{\sum_{r:|L_r - k_j^\top x_c| < \Delta L} e^{-\beta(L_r - k_j^\top x_c)^2}}{\sum_{s=0}^{P} e^{-\beta(L_s - k_j^\top x_c)^2}} \leq 1. \qquad \text{(The numerator is part of the denominator)}$$

For term $(II)$ in (D.74), we have

$$(II)$$

$$= \sum_{r:|L_r - k_j^\top x_c| \geq \Delta L} \frac{e^{-\beta(L_r - k_j^\top x_c)^2}}{\sum_{s=0}^{P} e^{-\beta(L_s - k_j^\top x_c)^2}} |L_r - k_j^\top x_c| \qquad \text{(By the definition of term $(II)$ in (D.74))}$$

$$\leq \sum_{r:|L_r - k_j^\top x_c| \geq \Delta L} \frac{e^{-\beta(L_r - k_j^\top x_c)^2}}{\sum_{s=0}^{P} e^{-\beta(L_s - k_j^\top x_c)^2}} 2dB^2$$

$$\text{(By (D.55) and (D.56), we have $|L_r - k_j^\top x_c| \leq 2dB^2$)}$$

$$\leq \sum_{r:|L_r - k_j^\top x_c| \geq \Delta L} \frac{e^{-\beta\Delta L^2}}{\sum_{s=0}^{P} e^{-\beta(L_s - k_j^\top x_c)^2}} 2dB^2$$

$$\text{(By $|L_r - k_j^\top x_c| \geq \Delta L$, we have $e^{-\beta(L_r - k_j^\top x_c)^2} \leq e^{-\beta\Delta L^2}$)}$$

$$\leq \sum_{r:|L_r - k_j^\top x_c| \geq \Delta L} \frac{e^{-\beta\Delta L^2}}{\max_s \{e^{-\beta(L_s - k_j^\top x_c)^2}\}} 2dB^2$$

$$\text{(We only keep the contribution from the nearest $L_s$ to $k_j^\top x_c$)}$$

$$\leq \sum_{r:|L_r - k_j^\top x_c| \geq \Delta L} \frac{e^{-\beta\Delta L^2}}{e^{-\beta\frac{\Delta L^2}{4}}} 2dB^2, \qquad (D.76)$$

and the last inequality holds since, by our construction of $L_s$ in (D.56), the distance from $k_j^\top x_c$ to the nearest $L_s$ is at most $\frac{\Delta L}{2}$. That is,

$$|L_{s_0} - k_j^\top x_c| \leq \frac{\Delta L}{2}, \quad \text{for} \quad s_0 = \underset{s}{\arg\min} |L_s - k_j^\top x_c|.$$

From (D.76), we have

$$\sum_{|L_r - k_j^\top x_c| \geq \Delta L} e^{-\frac{3}{4}\beta\Delta L^2} 2dB^2 \leq P e^{-\frac{3}{4}\beta\Delta L^2} 2dB^2, \qquad (D.77)$$

and the last inequality holds since, by our construction of $L_r$ in (D.56), at most $P$ points satisfy $|L_r - k_j^\top x_c| \geq \Delta L$. This $P$-point scenario occurs when the value of $k_j^\top x_c$ equals one of the $L_r$ grid points.

Combining (D.74), (D.75), and (D.77), we have:

$$\|\widehat{V}_j^K \cdot (\text{Softmax}_\beta((\widehat{K}_j^K)^\top \widehat{Q}_j^K) \cdot W_{\widehat{O}}^{K;j})_{:,c} - k_j^\top x_c \cdot e_j^{(3n)}\|_\infty \leq \underbrace{\Delta L}_{:=(a)} + \underbrace{P e^{-\frac{3}{4}\beta\Delta L^2} 2dB^2}_{:=(b)}. \qquad (D.78)$$

For term $(a)$ in (D.78), we recall

$$\Delta L = \frac{2dB^2}{P}. \qquad \text{(By the definition of $\Delta L$. i.e., (D.57))}$$

To bound $\Delta L$, we choose

$$P \geq \frac{4dB^2}{\epsilon_1},$$

for any $\epsilon_1 > 0$, such that

$$\Delta L \leq \frac{\epsilon_1}{2}.$$

For term $(b)$ in (D.78), we set

$$\beta \geq \frac{4}{3} \frac{1}{(\Delta L)^2} \ln\left(\frac{4dB^2 P}{\epsilon_1}\right),$$

such that

$$Pe^{-\frac{3}{4}\beta\Delta L^2} 2dB^2 \leq \frac{\epsilon_1}{2}.$$

Thus, from (D.78), we have

$$\|\widehat{V}_j^K \cdot (\mathrm{Softmax}_\beta((\widehat{K}_j^K)^\top \widehat{Q}_j^K) \cdot W_{\widehat{O}}^{K;j})_{:,c} - \underbrace{k_j^\top x_c}_{\text{scalar}} \cdot \underbrace{e_j^{(3n)}}_{3n \times 1}\|_\infty \leq \Delta L + Pe^{-\frac{3}{4}\beta\Delta L^2} 2dB^2$$

$$\leq \frac{\epsilon_1}{2} + \frac{\epsilon_1}{2}$$

$$= \epsilon_1,$$

and this leads to

$$\|\widehat{V}_j^K \cdot \mathrm{Softmax}_\beta((\widehat{K}_j^K)^\top \widehat{Q}_j^K) \cdot W_{\widehat{O}}^{K;j} - \underbrace{e_j^{(3n)}}_{3n \times 1} \cdot \underbrace{k_j^\top X}_{1 \times n}\|_\infty \leq \epsilon_1. \tag{D.79}$$

We recall

$$\widehat{V}_j^K \cdot \mathrm{Softmax}_\beta((\widehat{K}_j^K)^\top \widehat{Q}_j^K) \cdot W_{\widehat{O}}^{K;j}$$

$$= W_{\widehat{V}}^{K;j}\mathrm{Linear}_j^K(\begin{bmatrix} X \\ W_K^\top \\ W_Q^\top \\ W_V^\top \end{bmatrix}) \cdot \mathrm{Softmax}_\beta((W_{\widehat{K}}^{K;j}\mathrm{Linear}_j^K(\begin{bmatrix} X \\ W_K^\top \\ W_Q^\top \\ W_V^\top \end{bmatrix}))^\top W_{\widehat{Q}}^{K;j}\mathrm{Linear}_j^K(\begin{bmatrix} X \\ W_K^\top \\ W_Q^\top \\ W_V^\top \end{bmatrix})) \cdot W_{\widehat{O}}^{K;j}$$

$$\qquad\qquad\qquad\qquad\qquad\qquad\qquad\qquad\qquad (\text{By the definition of } \widehat{K}_j^K, \widehat{Q}_j^K, \text{ and } \widehat{V}_j^K)$$

$$= \mathrm{Attn}_j^K \circ \mathrm{Linear}_j^K(\begin{bmatrix} X \\ W_K^\top \\ W_Q^\top \\ W_V^\top \end{bmatrix}). \qquad\qquad\qquad\qquad\qquad (\text{By the definition of } \mathrm{Attn}_j^K)$$

Thus, we write (D.79) as

$$\|\mathrm{Attn}_j^K \circ \mathrm{Linear}_j^K(\begin{bmatrix} X \\ W_K^\top \\ W_Q^\top \\ W_V^\top \end{bmatrix}) - \underbrace{e_j^{(3n)}}_{3n \times 1} \cdot \underbrace{k_j^\top X}_{1 \times n}\|_\infty \leq \epsilon_1,$$

and we sum over the index $j$ to obtain the approximation across rows

$$\|\sum_{j=1}^{n} \text{Attn}_j^K \circ \text{Linear}_j^K (\begin{bmatrix} X \\ W_K^\top \\ W_Q^\top \\ W_V^\top \end{bmatrix}) - \begin{bmatrix} K \\ 0_{n \times n} \\ 0_{n \times n} \end{bmatrix} \|_\infty \leq \epsilon_1, \tag{D.80}$$

for any $\epsilon_1 > 0$.

**In-Context Calculation of $Q$ and $V$.** We approximate $Q$ and $V$ using the same procedure as that of $K$.

We start with $Q$.

Again, we define $\text{Linear}_j^Q$ preceding $\text{Attn}_j^Q$ first. We construct $\text{Linear}_j^Q$ similarly to $\text{Linear}_j^K$. The only difference is the position of the identity $I_d$ in the first term. Explicitly,

$$\text{Linear}_j^Q(Z) := \underbrace{\begin{bmatrix} 0_{d \times d} & 0_{d \times d} & 0_{d \times d} & 0_{d \times d} \\ 0_{d \times d} & 0_{d \times d} & I_d & 0_{d \times d} \\ 0_{3 \times d} & 0_{3 \times d} & 0_{3 \times d} & 0_{3 \times d} \end{bmatrix}}_{(2d+3) \times 4d} \underbrace{Z}_{4d \times n} \underbrace{\begin{bmatrix} 2L_0 e_j^{(n)} & 2L_1 e_j^{(n)} & \cdots & 2L_P e_j^{(n)} \end{bmatrix}}_{n \times (P+1)} +$$

$$\underbrace{\begin{bmatrix} I_d & 0_{d \times d} & 0_{d \times 2d} \\ 0_{d \times d} & 0_{d \times d} & 0_{d \times 2d} \\ 0_{3 \times d} & 0_{3 \times d} & 0_{3 \times 2d} \end{bmatrix}}_{(2d+3) \times 4d} \underbrace{Z}_{4d \times n} \underbrace{\begin{bmatrix} I_n & 0_{n \times (P+1-n)} \end{bmatrix}}_{n \times (P+1)} + \underbrace{\begin{bmatrix} 0_{2d \times (P+1)} \\ M_1 \\ M_L \end{bmatrix}}_{(2d+3) \times (P+1)}.$$

$\text{Linear}_j^Q$ takes $\begin{bmatrix} X^\top & W_K & W_Q & W_V \end{bmatrix}^\top$ as input and outputs:

$$\text{Linear}_j^Q(\begin{bmatrix} X \\ W_K^\top \\ W_Q^\top \\ W_V^\top \end{bmatrix}) = \begin{bmatrix} x_1 & x_2 & \cdots & x_n & 0_d & \cdots & 0_d \\ 2L_0 q_j & 2L_1 q_j & \cdots & 2L_{n-1} q_j & 2L_n q_j & \cdots & 2L_P q_j \\ 1 & 1 & \cdots & 1 & 0 & \cdots & 0 \\ L_0 & L_1 & \cdots & L_{n-1} & L_n & \cdots & L_P \\ -L_0^2 & -L_1^2 & \cdots & -L_{n-1}^2 & -L_n^2 & \cdots & -L_P^2 \end{bmatrix}.$$

Next, we construct $\text{Attn}_j^Q : \mathbb{R}^{(2d+3) \times (P+1)} \rightarrow \mathbb{R}^{3n \times n}$ to be

$$\text{Attn}_j^Q(D) := \underbrace{W_{\widehat{V}}^{Q;j} D}_{3n \times (P+1)} \cdot \underbrace{\text{Softmax}_\beta((W_{\widehat{K}}^{Q;j} D)^\top W_{\widehat{Q}}^{Q;j} D)}_{(P+1) \times (P+1)} \cdot \underbrace{W_{\widehat{O}}^{Q;j}}_{(P+1) \times n},$$

where $D \in \mathbb{R}^{(2d+3) \times (P+1)}$ denotes any input, and

$$W_{\widehat{K}}^{Q;j} := W_{\widehat{K}}^{K;j} = \underbrace{\begin{bmatrix} 0_{d \times d} & I_d & 0_{d \times 1} & 0_{d \times 1} & 0_{d \times 1} \\ 0_{1 \times d} & 0_{1 \times d} & 0 & 0 & 1 \end{bmatrix}}_{(d+1) \times (2d+3)},$$

$$W_{\widehat{Q}}^{Q;j} := W_{\widehat{Q}}^{K;j} = \underbrace{\begin{bmatrix} I_d & 0_{d \times d} & 0_{d \times 1} & 0_{d \times 1} & 0_{d \times 1} \\ 0_{1 \times d} & 0_{1 \times d} & 1 & 0 & 0 \end{bmatrix}}_{(d+1) \times (2d+3)},$$

$$W_{\widehat{V}}^{Q;j} := \underbrace{e_{n+j}^{(3n)}}_{3n \times 1} \underbrace{\begin{bmatrix} 0_{1 \times d} & 0_{1 \times d} & 0 & 1 & 0 \end{bmatrix}}_{1 \times (2d+3)},$$

$$W_{\widehat{O}}^{Q;j} := W_{\widehat{O}}^{K;j} = \underbrace{\begin{bmatrix} I_n \\ 0_{(P+1-n)\times n} \end{bmatrix}}_{(P+1)\times n}.$$

We define the $\widehat{K}_j^Q$ of $\mathrm{Attn}_j^Q$ to be

$$\widehat{K}_j^Q := W_{\widehat{K}}^{Q;j} \cdot \mathrm{Linear}_j^Q(\begin{bmatrix} X \\ W_K^\top \\ W_Q^\top \\ W_V^\top \end{bmatrix}) = \underbrace{\begin{bmatrix} 2L_0 q_j & 2L_1 q_j & \cdots & 2L_{n-1} q_j & \cdots & 2L_P q_j \\ -L_0^2 & -L_1^2 & \cdots & -L_{n-1}^2 & \cdots & -L_P^2 \end{bmatrix}}_{(d+1)\times(P+1)}. \qquad \text{(D.81)}$$

We define the $\widehat{Q}_j^Q$ of $\mathrm{Attn}_j^Q$ to be

$$\widehat{Q}_j^Q := W_{\widehat{Q}}^{Q;j} \cdot \mathrm{Linear}_j^Q(\begin{bmatrix} X \\ W_K^\top \\ W_Q^\top \\ W_V^\top \end{bmatrix}) = \underbrace{\begin{bmatrix} x_1 & x_2 & \cdots & x_n & 0_{d\times(P+1-n)} \\ 1 & 1 & \cdots & 1 & 0_{1\times(P+1-n)} \end{bmatrix}}_{(d+1)\times(P+1)}. \qquad \text{(D.82)}$$

We define the $\widehat{V}_j^Q$ of $\mathrm{Attn}_j^Q$ to be

$$\widehat{V}_j^Q := W_{\widehat{V}}^{Q;j} \cdot \mathrm{Linear}_j^Q(\begin{bmatrix} X \\ W_K^\top \\ W_Q^\top \\ W_V^\top \end{bmatrix}) = \underbrace{e_{n+j}^{(3n)}}_{3n\times 1} \underbrace{\begin{bmatrix} L_0 & L_1 & \cdots & L_{n-1} & L_n & \cdots & L_P \end{bmatrix}}_{1\times(P+1)}. \qquad \text{(D.83)}$$

Then, by going through the same calculations as those of $K$, we have

$$\| \sum_{j=1}^n \mathrm{Attn}_j^Q \circ \mathrm{Linear}_j^Q(\begin{bmatrix} X \\ W_K^\top \\ W_Q^\top \\ W_V^\top \end{bmatrix}) - \begin{bmatrix} 0_{n\times n} \\ Q \\ 0_{n\times n} \end{bmatrix} \|_\infty \le \epsilon_1. \qquad \text{(D.84)}$$

To approximate $V$, we define

$$\mathrm{Linear}_j^V(Z) := \underbrace{\begin{bmatrix} 0_{d\times d} & 0_{d\times d} & 0_{d\times d} & 0_{d\times d} \\ 0_{d\times d} & 0_{d\times d} & 0_{d\times d} & I_d \\ 0_{3\times d} & 0_{3\times d} & 0_{3\times d} & 0_{3\times d} \end{bmatrix}}_{(2d+3)\times 4d} \underbrace{Z}_{4d\times n} \underbrace{\begin{bmatrix} 2L_0 e_j^{(n)} & 2L_1 e_j^{(n)} & \cdots & 2L_P e_j^{(n)} \end{bmatrix}}_{n\times(P+1)} +$$

$$\underbrace{\begin{bmatrix} I_d & 0_{d\times d} & 0_{d\times 2d} \\ 0_{d\times d} & 0_{d\times d} & 0_{d\times 2d} \\ 0_{3\times d} & 0_{3\times d} & 0_{3\times 2d} \end{bmatrix}}_{(2d+3)\times 4d} \underbrace{Z}_{4d\times n} \underbrace{\begin{bmatrix} I_n & 0_{n\times(P+1-n)} \end{bmatrix}}_{n\times(P+1)} + \underbrace{\begin{bmatrix} 0_{2d\times(P+1)} \\ M_1 \\ M_L \end{bmatrix}}_{(2d+3)\times(P+1)}.$$

$\mathrm{Linear}_j^V$ outputs in a similar manner as $\mathrm{Linear}_j^K$:

$$\mathrm{Linear}_j^V(Z) = \begin{bmatrix} x_1 & x_2 & \cdots & x_n & 0_d & \cdots & 0_d \\ 2L_0 v_j & 2L_1 v_j & \cdots & 2L_{n-1} v_j & 2L_n v_j & \cdots & 2L_P v_j \\ 1 & 1 & \cdots & 1 & 0 & \cdots & 0 \\ L_0 & L_1 & \cdots & L_{n-1} & L_n & \cdots & L_P \\ -L_0^2 & -L_1^2 & \cdots & -L_{n-1}^2 & -L_n^2 & \cdots & -L_P^2 \end{bmatrix}.$$

Next, we construct $\text{Attn}_j^V : \mathbb{R}^{(2d+3)\times(P+1)} \to \mathbb{R}^{3n\times n}$ to be

$$\text{Attn}_j^V := \underbrace{W_{\widehat{V}}^{V;j} D}_{3n\times(P+1)} \cdot \underbrace{\text{Softmax}_\beta((W_{\widehat{K}}^{V;j} D)^\top W_{\widehat{Q}}^{V;j} D)}_{(P+1)\times(P+1)} \underbrace{W_{\widehat{O}}^{V;j}}_{(P+1)\times n},$$

where $D \in \mathbb{R}^{(2d+3)\times(P+1)}$ denotes any input, and

$$W_{\widehat{K}}^{V;j} := W_{\widehat{K}}^{K;j} = \underbrace{\begin{bmatrix} 0_{d\times d} & I_d & 0_{d\times 1} & 0_{d\times 1} & 0_{d\times 1} \\ 0_{1\times d} & 0_{1\times d} & 0 & 0 & 1 \end{bmatrix}}_{(d+1)\times(2d+3)},$$

$$W_{\widehat{Q}}^{V;j} := W_{\widehat{Q}}^{K;j} = \underbrace{\begin{bmatrix} I_d & 0_{d\times d} & 0_{d\times 1} & 0_{d\times 1} & 0_{d\times 1} \\ 0_{1\times d} & 0_{1\times d} & 1 & 0 & 0 \end{bmatrix}}_{(d+1)\times(2d+3)},$$

$$W_{\widehat{V}}^{V;j} := \underbrace{e_{2n+j}^{(3n)}}_{3n\times 1} \underbrace{[0_{1\times d} \quad 0_{1\times d} \quad 0 \quad 1 \quad 0]}_{1\times(2d+3)},$$

$$W_{\widehat{O}}^{V;j} := W_{\widehat{O}}^{K;j} = \underbrace{\begin{bmatrix} I_n \\ 0_{(P+1-n)\times n} \end{bmatrix}}_{(P+1)\times n}.$$

We define the $\widehat{K}_j^V$ to be

$$\widehat{K}_j^V := W_{\widehat{K}}^{V;j} \cdot \text{Linear}_j^V \left( \begin{bmatrix} X \\ W_K^\top \\ W_Q^\top \\ W_V^\top \end{bmatrix} \right) = \underbrace{\begin{bmatrix} 2L_0 v_j & 2L_1 v_j & \cdots & 2L_{n-1} v_j & \cdots & 2L_P v_j \\ -L_0^2 & -L_1^2 & \cdots & -L_{n-1}^2 & \cdots & -L_P^2 \end{bmatrix}}_{(d+1)\times(P+1)}. \quad \text{(D.85)}$$

We define the $\widehat{Q}_j^V$ to be

$$\widehat{Q}_j^V := W_{\widehat{Q}}^{V;j} \cdot \text{Linear}_j^V \left( \begin{bmatrix} X \\ W_K^\top \\ W_Q^\top \\ W_V^\top \end{bmatrix} \right) = \underbrace{\begin{bmatrix} x_1 & x_2 & \cdots & x_n & 0_{d\times(P+1-n)} \\ 1 & 1 & \cdots & 1 & 0_{1\times(P+1-n)} \end{bmatrix}}_{(d+1)\times(P+1)}. \quad \text{(D.86)}$$

We define the $\widehat{V}_j^V$ to be

$$\widehat{V}_j^V := W_{\widehat{V}}^{V;j} \cdot \text{Linear}_j^V \left( \begin{bmatrix} X \\ W_K^\top \\ W_Q^\top \\ W_V^\top \end{bmatrix} \right) = \underbrace{e_{2n+j}^{(3n)}}_{3n\times 1} \underbrace{[L_0 \quad L_1 \quad \cdots \quad L_{n-1} \quad L_n \quad \cdots \quad L_P]}_{1\times(P+1)}. \quad \text{(D.87)}$$

Similarly, by going through the same calculations as those of $K$, we have

$$\| \sum_{j=1}^n \text{Attn}_j^V \circ \text{Linear}_j^V \left( \begin{bmatrix} X \\ W_K^\top \\ W_Q^\top \\ W_V^\top \end{bmatrix} \right) - \begin{bmatrix} 0_{n\times n} \\ 0_{n\times n} \\ V \end{bmatrix} \|_\infty \le \epsilon_1. \quad \text{(D.88)}$$

Then, by combining (D.80), (D.84), and (D.88), we have

$$
\| \sum_{j=1}^{n} \text{Attn}_j^K \circ \text{Linear}_j^K \left( \begin{bmatrix} X \\ W_K^\top \\ W_Q^\top \\ W_V^\top \end{bmatrix} \right) - \begin{bmatrix} K \\ 0_{n \times n} \\ 0_{n \times n} \end{bmatrix} + \sum_{j=1}^{n} \text{Attn}_j^Q \circ \text{Linear}_j^Q \left( \begin{bmatrix} X \\ W_K^\top \\ W_Q^\top \\ W_V^\top \end{bmatrix} \right) - \begin{bmatrix} 0_{n \times n} \\ Q \\ 0_{n \times n} \end{bmatrix} +
$$

$$
\sum_{j=1}^{n} \text{Attn}_j^V \circ \text{Linear}_j^V \left( \begin{bmatrix} X \\ W_K^\top \\ W_Q^\top \\ W_V^\top \end{bmatrix} \right) - \begin{bmatrix} 0_{n \times n} \\ 0_{n \times n} \\ V \end{bmatrix} \|_\infty \le \epsilon_1.
$$

As previously stated in (D.58), $\text{Linear}_j^K, \text{Linear}_j^Q$ and $\text{Linear}_j^V$ denote $\text{Linear}_j, \text{Linear}_{n+j}$ and $\text{Linear}_{2n+j}$ respectively. Also, as in (D.59), $\text{Attn}_j^K, \text{Attn}_j^Q$ and $\text{Attn}_j^V$ denote $\text{Attn}_j, \text{Attn}_{n+j}$ and $\text{Attn}_{2n+j}$.

Thus, we have

$$
\| \sum_{j=1}^{3n} \text{Attn}_j \circ \text{Linear}_j \left( \begin{bmatrix} X \\ W_K^\top \\ W_Q^\top \\ W_V^\top \end{bmatrix} \right) - \underbrace{\begin{bmatrix} K \\ Q \\ V \end{bmatrix}}_{3n \times n} \|_\infty \le \epsilon_1. \tag{D.89}
$$

We define

$$
\begin{bmatrix} K' \\ Q' \\ V' \end{bmatrix} := \sum_{j=1}^{3n} \text{Attn}_j \circ \text{Linear}_j \left( \begin{bmatrix} X \\ W_K^\top \\ W_Q^\top \\ W_V^\top \end{bmatrix} \right),
$$

such that (D.89) becomes

$$
\| \begin{bmatrix} K' \\ Q' \\ V' \end{bmatrix} - \begin{bmatrix} K \\ Q \\ V \end{bmatrix} \|_\infty \le \epsilon_1. \tag{D.90}
$$

**Step 3: Single-Head Assembly of the Attention Output.** Our goal in this part is to reconstruct the attention mechanism

$$
V' \text{Softmax}_\beta((K')^\top Q'), \quad \text{and} \quad V \text{Softmax}_\beta((K)^\top Q),
$$

from $K', Q', V'$ and $K, Q, V$ via $\text{Attn}_s$.

To achieve the reconstruction of attention mechanisms, we build $\text{Attn}_s$ as

$$
\text{Attn}_s(Z) := [0_{n \times 2n} \quad I_n] Z \cdot \text{Softmax}_\beta(([I_n \quad 0_{n \times 2n}] Z)^\top [0_{n \times n} \quad I_n \quad 0_{n \times n}] Z),
$$

where $Z \in \mathbb{R}^{3n \times n}$ denotes any input.

Then, we have

$$
\text{Attn}_s( \underbrace{\overbrace{\begin{bmatrix} K \\ Q \\ V \end{bmatrix}}^{n \times n}}_{3n \times n} ) = \underbrace{V}_{n \times n} \text{Softmax}_\beta( \underbrace{(K)^\top Q}_{n \times n} ),
$$

and

$$\text{Attn}_s(\overbrace{\begin{bmatrix} K' \\ Q' \\ V' \end{bmatrix}}^{n \times n}) = \underbrace{V'}_{n \times n} \text{Softmax}_\beta(\underbrace{(K')^\top Q'}_{n \times n}).$$

$$\underbrace{\phantom{\begin{bmatrix} K' \\ Q' \\ V' \end{bmatrix}}}_{3n \times n}$$

**Step 4: Error Bound** From the results of **Step 3**, we have

$$\text{Attn}_s(\begin{bmatrix} K' \\ Q' \\ V' \end{bmatrix}) - \text{Attn}_s(\begin{bmatrix} K \\ Q \\ V \end{bmatrix})$$

$$= V'\text{Softmax}_\beta(K'^\top Q') - V\text{Softmax}_\beta(K^\top Q)$$

$$= V'\text{Softmax}_\beta(K'^\top Q') - V\text{Softmax}_\beta(K'^\top Q') + V\text{Softmax}_\beta(K'^\top Q') - V\text{Softmax}_\beta(K^\top Q)$$

$$= (V' - V)\text{Softmax}_\beta(K'^\top Q') + V(\text{Softmax}_\beta(K'^\top Q') - \text{Softmax}_\beta(K^\top Q)).$$

Thus, we have

$$\|\text{Attn}_s(\begin{bmatrix} K' \\ Q' \\ V' \end{bmatrix}) - \text{Attn}_s(\begin{bmatrix} K \\ Q \\ V \end{bmatrix})\|_\infty$$

$$= \|(V' - V)\text{Softmax}_\beta(K'^\top Q') + V(\text{Softmax}_\beta(K'^\top Q') - \text{Softmax}_\beta(K^\top Q))\|_\infty$$

$$\leq \underbrace{\|(V' - V)\text{Softmax}_\beta(K'^\top Q')\|_\infty}_{:=(A)} + \underbrace{\|V(\text{Softmax}_\beta(K'^\top Q') - \text{Softmax}_\beta(K^\top Q))\|_\infty}_{:=(B)}, \quad \text{(D.91)}$$

and the last inequality follows from the triangle inequality.

For term $(A)$ in (D.91), since each column in $\text{Softmax}_\beta(K'^\top Q')$ sums up to 1, then for each column of $(A)$,

$$\underbrace{(V' - V)}_{n \times n} \underbrace{\text{Softmax}_\beta(K'^\top Q')_{:,j}}_{n \times 1},$$

is a weighted sum of the columns from $(V' - V)$.

Then, we have

$$\|(V' - V)\text{Softmax}_\beta(K'^\top Q')_{:,j}\|_\infty \leq \|V' - V\|_\infty \leq \epsilon_1,$$

and the first inequality holds since the column average of $(V' - V)$ has a maximum entry no greater than the maximum entry among the original columns in $(V' - V)$. The second inequality holds since (D.90). This conclusion holds for every column in term $(A)$, so we obtain

$$(A) \leq \epsilon_1. \quad \text{(D.92)}$$

Term $(B)$ in (D.91) is

$$(B) = \|V(\text{Softmax}_\beta(K'^\top Q') - \text{Softmax}_\beta(K^\top Q))\|_\infty.$$

For the simplicity of presentation, we define

$$\Delta S := \text{Softmax}_\beta(K'^\top Q') - \text{Softmax}_\beta(K^\top Q),$$

such that for each entry in $(B)$, we have

$$
\begin{aligned}
|(V\Delta S)_{ij}| = |\sum_{k=1}^{n} V_{ik}(\Delta S)_{kj}| &&& \text{(By the definition of matrix multiplication)} \\
&\leq \sum_{k=1}^{n} |V_{ik}| \cdot |(\Delta S)_{kj}| &&& \text{(By triangle inequality and } |ab| = |a| \cdot |b| \text{ for all } a, b \in \mathbb{R}) \\
&\leq \sum_{k=1}^{n} \|V\|_{\infty} \cdot \|\Delta S\|_{\infty} &&& \text{(By } |V_{ik}| \leq \|V\|_{\infty} \text{ and } |(\Delta S)_{kj}| \leq \|\Delta S\|_{\infty}) \\
&= n\|V\|_{\infty} \cdot \|\Delta S\|_{\infty},
\end{aligned}
$$

and this leads to

$$
(B) \leq n\|V\|_{\infty} \cdot \|\Delta S\|_{\infty}. \tag{D.93}
$$

For each entry in $\Delta S$, we have

$$
\begin{aligned}
&|(\Delta S)_{i,j}| \\
&= |(\text{Softmax}_{\beta}(K'^{\top}Q') - \text{Softmax}_{\beta}(K^{\top}Q))_{i,j}| \\
&= |\frac{e^{\beta K_i' \cdot Q_j'}}{\sum_{i'=1}^{n} e^{\beta K_{i'}' \cdot Q_j'}} - \frac{e^{\beta K_i \cdot Q_j}}{\sum_{i'=1}^{n} e^{\beta K_{i'} \cdot Q_j}}| && (K_i', Q_i', K_i, Q_i \text{ denote the } i\text{-th column in } K', Q', K, Q) \\
&= |\frac{e^{\beta K_i' \cdot Q_j'}}{\sum_{i'=1}^{n} e^{\beta K_{i'}' \cdot Q_j'}} - \frac{e^{\beta K_i \cdot Q_j}}{\sum_{i'=1}^{n} e^{\beta K_{i'}' \cdot Q_j'}} + \frac{e^{\beta K_i \cdot Q_j}}{\sum_{i'=1}^{n} e^{\beta K_{i'}' \cdot Q_j'}} - \frac{e^{\beta K_i \cdot Q_j}}{\sum_{i'=1}^{n} e^{\beta K_{i'} \cdot Q_j}}| \\
&\leq |\frac{e^{\beta K_i' \cdot Q_j'} - e^{\beta K_i \cdot Q_j}}{\sum_{i'=1}^{n} e^{\beta K_{i'}' \cdot Q_j'}}| + |e^{\beta K_i \cdot Q_j}(\frac{1}{\sum_{i'=1}^{n} e^{\beta K_{i'}' \cdot Q_j'}} - \frac{1}{\sum_{i'=1}^{n} e^{\beta K_{i'} \cdot Q_j}})| && \text{(By triangle inequality)} \\
&= \frac{e^{\beta K_i' \cdot Q_j'}}{\sum_{i'=1}^{n} e^{\beta K_{i'}' \cdot Q_j'}} |1 - e^{\beta(K_i \cdot Q_j - K_i' \cdot Q_j')}| + \frac{e^{\beta K_i \cdot Q_j}}{\sum_{i'=1}^{n} e^{\beta K_{i'} \cdot Q_j}} |\frac{\sum_{i'=1}^{n} e^{\beta K_{i'} \cdot Q_j}}{\sum_{i'=1}^{n} e^{\beta K_{i'}' \cdot Q_j'}} - 1| \\
&&& \text{(By non-negativity of exponential)} \\
&< \underbrace{|1 - e^{\beta(K_i \cdot Q_j - K_i' \cdot Q_j')}|}_{:=(B\text{-}1)} + \underbrace{|1 - \frac{\sum_{i'=1}^{n} e^{\beta K_{i'} \cdot Q_j}}{\sum_{i'=1}^{n} e^{\beta K_{i'}' \cdot Q_j'}}|}_{:=(B\text{-}2)}, \tag{D.94}
\end{aligned}
$$

and the last inequality holds since

$$
\frac{e^{\beta K_i' \cdot Q_j'}}{\sum_{i'=1}^{n} e^{\beta K_{i'}' \cdot Q_j'}} < 1, \quad \frac{e^{\beta K_i \cdot Q_j}}{\sum_{i'=1}^{n} e^{\beta K_{i'} \cdot Q_j}} < 1.
$$

To bound term $(B\text{-}1)$ in (D.94), we recall

$$
\|\underbrace{\begin{bmatrix} K' \\ Q' \\ V' \end{bmatrix}}_{3n \times n} - \underbrace{\begin{bmatrix} K \\ Q \\ V \end{bmatrix}}_{3n \times n}\|_{\infty} \leq \epsilon_1,
$$

so

$$
\|\underbrace{K' - K}_{n \times n}\|_{\infty} \leq \epsilon_1,
$$

$$\|\underbrace{Q' - Q}_{n \times n}\|_\infty \le \epsilon_1.$$

Let $K_i'$, $Q_i'$, $K_i$, $Q_i$ denote the $i$-th column in $K'$, $Q'$, $K$, $Q$, then we have

$$\underbrace{\Delta K_i}_{n \times 1} \coloneqq \underbrace{K_i' - K_i}_{n \times 1}, \quad \|\Delta K_i\|_\infty \le \epsilon_1,$$

$$\underbrace{\Delta Q_i}_{n \times 1} \coloneqq \underbrace{Q_i' - Q_i}_{n \times 1}, \quad \|\Delta Q_i\|_\infty \le \epsilon_1.$$

Thus, for term $(B\text{-}1)$ in (D.94), we have

$$
\begin{aligned}
& (B\text{-}1) \\
&= |1 - \exp\{\beta(K_i \cdot Q_j - K_i' \cdot Q_j')\}| \\
&= |1 - \exp\{\beta(K_i \cdot Q_j - (K_i + \Delta K_i) \cdot (Q_j + \Delta Q_j))\}| \\
&\qquad\qquad\qquad\qquad \text{(By } K_i' = K_i + \Delta K_i \text{ and } Q_i' = Q_i + \Delta Q_i) \\
&= |1 - \exp\{-\beta(K_i \cdot \Delta Q_j + Q_j \cdot \Delta K_i + \Delta K_i \cdot \Delta Q_j)\}|, \\
&\text{(By } K_i \cdot Q_j - (K_i + \Delta K_i) \cdot (Q_i + \Delta Q_i) = -(K_i \cdot \Delta Q_j + Q_j \cdot \Delta K_i + \Delta K_i \cdot \Delta Q_j))
\end{aligned}
$$

and we know

$$
\begin{aligned}
& K_i \cdot \Delta Q_j + Q_j \cdot \Delta K_i + \Delta K_i \cdot \Delta Q_j \\
& \le n \cdot \|K_i\|_\infty \|\Delta Q_j\|_\infty + n \cdot \|Q_j\|_\infty \|\Delta K_i\|_\infty + n \cdot \|\Delta K_i\|_\infty \|\Delta Q_j\|_\infty \\
&\qquad\qquad\qquad\qquad\qquad \text{(By } a \cdot b \le n\|a\|_\infty \|b\|_\infty \text{ for all } a, b \in \mathbb{R}^n) \\
& \le 2nB_{KQV}\epsilon_1 + n\epsilon_1^2. \qquad \text{(By } \|K_i\|_\infty, \|Q_j\|_\infty \le B_{KQV} \text{ and } \|\Delta K_i\|_\infty, \|\Delta Q_j\|_\infty \le \epsilon_1)
\end{aligned}
$$

Thus, we have

$$(B\text{-}1) \le |1 - e^{-\beta n(2B_{KQV}\epsilon_1 + \epsilon_1^2)}|. \tag{D.95}$$

For term $(B\text{-}2)$ in (D.94), we have

$$
\begin{aligned}
& (B\text{-}2) \\
&= |1 - \frac{\sum_{i'=1}^n e^{\beta K_{i'} \cdot Q_j}}{\sum_{i'=1}^n e^{\beta K_{i'}' \cdot Q_j'}}| \qquad\qquad\qquad\qquad\qquad \text{(By the definition of } (B\text{-}2)) \\
&= |1 - \frac{\sum_{i'=1}^n e^{\beta K_{i'} \cdot Q_j}}{\sum_{i'=1}^n e^{\beta(K_{i'} + \Delta K_{i'}) \cdot (Q_j + \Delta Q_j)}}| \qquad \text{(By } K_{i'}' = K_{i'} + \Delta K_{i'} \text{ and } Q_i' = Q_i + \Delta Q_i) \\
&= |1 - \frac{\sum_{i'=1}^n e^{\beta K_{i'} \cdot Q_j}}{\sum_{i'=1}^n e^{\beta(K_{i'} \cdot Q_j + K_{i'} \cdot \Delta Q_j + Q_j \cdot \Delta K_{i'} + \Delta K_{i'} \cdot \Delta Q_j)}}|,
\end{aligned}
$$

and for all $i'$ in the denominator, we have

$$
\begin{aligned}
& K_{i'} \cdot Q_j + K_{i'} \cdot \Delta Q_j + Q_j \cdot \Delta K_{i'} + \Delta K_{i'} \cdot \Delta Q_j \\
& \le K_{i'} \cdot Q_j + n \cdot \|K_{i'}\|_\infty \|\Delta Q_j\|_\infty + n \cdot \|Q_j\|_\infty \|\Delta K_{i'}\|_\infty + n \cdot \|\Delta K_{i'}\|_\infty \|\Delta Q_j\|_\infty \\
&\qquad\qquad\qquad\qquad\qquad \text{(By } a \cdot b \le n\|a\|_\infty \|b\|_\infty \text{ for all } a, b \in \mathbb{R}^n) \\
& \le K_{i'} \cdot Q_j + 2nB_{KQV}\epsilon_1 + n\epsilon_1^2. \quad \text{(By } \|K_{i'}\|_\infty, \|Q_j\|_\infty \le B_{KQV} \text{ and } \|\Delta K_{i'}\|_\infty, \|\Delta Q_j\|_\infty \le \epsilon_1)
\end{aligned}
$$

Thus,

$$(B\text{-}2)$$

$$= \left| 1 - \frac{\sum_{i'=1}^{n} e^{\beta K_{i'} \cdot Q_j}}{\sum_{i'=1}^{n} e^{\beta (K_{i'} \cdot Q_j + K_{i'} \cdot \Delta Q_j + Q_j \cdot \Delta K_{i'} + \Delta K_{i'} \cdot \Delta Q_j)}} \right|$$

$$\leq \left| 1 - \frac{\sum_{i'=1}^{n} e^{\beta K_{i'} \cdot Q_j}}{\sum_{i'=1}^{n} e^{\beta (K_{i'} \cdot Q_j + 2nB_{KQV}\epsilon_1 + n\epsilon_1^2)}} \right|$$

$$= \left| 1 - \frac{\sum_{i'=1}^{n} e^{\beta K_{i'} \cdot Q_j}}{e^{\beta n (2B_{KQV}\epsilon_1 + \epsilon_1^2)} \sum_{i'=1}^{n} e^{\beta K_{i'} \cdot Q_j}} \right| \qquad \left( e^{\beta n (2B_{KQV}\epsilon_1 + \epsilon_1^2)} \text{ is independent of } i' \right)$$

$$= \left| 1 - e^{-\beta n (2B_{KQV}\epsilon_1 + \epsilon_1^2)} \right|, \tag{D.96}$$

and the last equality holds since the common factor $\sum_{i'=1}^{n} e^{\beta K_{i'} \cdot Q_j}$ cancels out.

Combining (D.94), (D.95), and (D.96), we have

$$|(\text{Softmax}_\beta(K'^\top Q') - \text{Softmax}_\beta(K^\top Q))_{i,j}|$$
$$< 2|1 - e^{-\beta n (2B_{KQV}\epsilon_1 + \epsilon_1^2)}|$$
$$\leq 2|1 - e^{-\beta n (2B_{KQV}\epsilon_1 + \epsilon_1)}|. \qquad (\text{By requiring } 0 < \epsilon_1 \leq 1)$$

Thus, for any $0 < \epsilon_0 < 2$, when $\epsilon_1$ satisfies

$$0 < \epsilon_1 \leq \min\left\{1, \frac{-\ln\left(1 - \frac{\epsilon_0}{2}\right)}{\beta n (2B_{KQV} + 1)}\right\},$$

we have

$$|(\text{Softmax}_\beta(K'^\top Q') - \text{Softmax}_\beta(K^\top Q))_{i,j}| < \epsilon_0. \tag{D.97}$$

From (D.93) and (D.97), we have

$$(B) \leq n\|V\|_\infty \|\Delta S\|_\infty < nB_{KQV}\epsilon_0, \tag{D.98}$$

since $\|V\|_\infty \leq B_{KQV}$ and $\|\Delta S\|_\infty < \epsilon_0$.

Combining (D.91), (D.92) and (D.98) yields

$$\left\| \text{Attn}_s\left( \begin{bmatrix} K' \\ Q' \\ V' \end{bmatrix} \right) - \text{Attn}_s\left( \begin{bmatrix} K \\ Q \\ V \end{bmatrix} \right) \right\|_\infty < \epsilon_1 + nB_{KQV}\epsilon_0.$$

When we take $\epsilon_0$ and $\epsilon_1$ to be infinitely small, the right-hand side tends to $0$.

This completes the proof. $\qquad \square$

### D.7 PROOF OF COROLLARY 4.2.1

**Theorem D.7** (Restate of Corollary 4.2.1: In-Context Emulation of Statistical Methods). Let $\mathcal{A}$ denote the set of all the in-context algorithms that a single-layer attention is able to approximate. For an $a \in \mathcal{A}$ (that is, a specific algorithm), let $W_K^a, W_Q^a, W_V^a$ denote the weights of the attention that implements this algorithm. For any $\epsilon > 0$ and any finite set $\mathcal{A}_0 \in \mathcal{A}$, there exists a 2-layer attention $\text{Attn} \circ \text{Attn}_m$ such that

$$\left\| \sum_{j=1}^{3n} \text{Attn}_s \circ \text{Attn}_j \circ \text{Linear}_j \left( \begin{bmatrix} X \\ W^a \end{bmatrix} \right) - a(X) \right\|_\infty \leq \epsilon, \quad a \in \mathcal{A}_0,$$

where $W^a$ is the $W$ defined as Definition 4.2 using $W_K^a, W_Q^a, W_V^a$.

*Proof.* Without loss of generality, assume all $W_K^a, W_Q^a, W_V^a$ to be of the same hidden dimension since we are always able to pad them to the same size. According to Theorem 4.2, there exists a network $\sum_{j=1}^n \text{Attn}_s \circ \text{Attn}_j \circ \text{Linear}_j$ that approximate $a(X)$ with an error no larger than $\epsilon > 0$ when given input of the form:

$$\begin{bmatrix} X \\ W_K^{a\top} \\ W_Q^{a\top} \\ W_V^{a\top} \end{bmatrix}.$$

Then for a set of $a \in \mathcal{A}_0$, define $P_m := \max_{a \in \mathcal{A}_0} P_\epsilon^{(a)}$.

By Theorem 4.2, there exists a network consisting of a self-attention followed by a multi-head attention with a linear layer and parameter $P$ equals to $P_m$, such that for any $a \in \mathcal{A}_0$, we have

$$\| \sum_{j=1}^{3n} \text{Attn}_s \circ \text{Attn}_j \circ \text{Linear}_j \left( \begin{bmatrix} X \\ W_K^{a\top} \\ W_Q^{a\top} \\ W_V^{a\top} \end{bmatrix} \right) - a(X) \|_\infty \leq \epsilon, \quad a \in \mathcal{A}_0.$$

This completes the proof. $\qquad\square$

# E  IN-CONTEXT APPLICATION OF STATISTICAL METHODS BY MODERN HOPFIELD NETWORK

**Definition E.1** (Modern Hopfield Network). Define $Y = (y_1, \cdots, y_N)^\top \in \mathbb{R}^{d_y \times N}$ as the raw stored pattern, $R = (r_1, \cdots, r_S)^\top \in \mathbb{R}^{R_r \times S}$ as the raw state pattern, and $W_Q \in \mathbb{R}^{d \times d_r}$, $W_K \in \mathbb{R}^{d \times d_y}$, $W_V \in \mathbb{R}^{d_v \times d}$ as the projection matrices. A Hopfield layer Hopfield is defined as:

$$\text{Hopfield}(R; Y, W_Q, W_K, W_V) := \underbrace{W_V}_{d_v \times d} \overbrace{W_K Y}^{d \times N} \text{Softmax}(\underbrace{\beta (W_K Y)^\top W_Q R}_{N \times S}) \in \mathbb{R}^{d_v \times S}, \quad \text{(E.1)}$$

where $\beta$ is a temperature parameter.
With $K \in \mathbb{R}^{d \times N}$ denoting $W_K Y$, $Q \in \mathbb{R}^{d \times S}$ denoting $W_Q R$ and $V \in \mathbb{R}^{d_v \times N}$ denoting $W_V W_K Y$, (E.1) writes out as:

$$\text{Hopfield}(R; Y, W_Q, W_K, W_V) := V \text{Softmax}(\beta \cdot K^\top Q) \in \mathbb{R}^{d_v \times S}.$$

**Theorem E.1.** Let $Z = [z_1, z_2, \cdots, z_n] \in \mathbb{R}^{d \times n}$ denote the input from a compact input domain. For any linear transformation $l(z) = a^\top z + b : \mathbb{R}^d \to \mathbb{R}$, and any continuous function $f : \mathbb{R} \to \mathbb{R}^o$ where $o$ is the output dimension, there exists a Hopfield network Hopfield such that

$$\| \text{Hopfield}(Z) - [f(l(z_1)) \quad f(l(z_2)) \quad \cdots \quad f(l(z_n))] \|_\infty \leq \epsilon,$$

for any $\epsilon > 0$.

*Proof.* We first perform a simple token-wise linear transformation on the input:

$$\text{Linear}(Z) := \begin{bmatrix} I_{d \times d} \\ 0_{1 \times d} \end{bmatrix} Z + \begin{bmatrix} 0_{d \times n} \\ 1_{1 \times n} \end{bmatrix} = \begin{bmatrix} Z \\ 1_{1 \times n} \end{bmatrix} \in \mathbb{R}^{(d+1) \times n}.$$

We then construct $W_Q$ to be:

$$W_Q := I_{(d+1)},$$

which is an identity matrix of dimension $\mathbb{R}^{(d+1)\times(d+1)}$.

This yields that

$$Q := W_Q \text{Linear}(Z) = \begin{bmatrix} Z \\ 1_{1\times n} \end{bmatrix} \in \mathbb{R}^{(d+1)\times n}.$$

Following the definition of *Interpolations* in Appendix D.6, $K, V$ are constructed as (here we omit $Y$ since it's not the input):

$$K := \begin{bmatrix} 2L_0 a & 2L_1 a & \cdots & 2L_P a \\ 2L_0 b - L_0^2 & 2L_1 b - L_1^2 & \cdots & 2L_P b - L_P^2 \end{bmatrix},$$
$$V := [f(L_0) \quad f(L_1) \quad \cdots \quad f(L_P)].$$

By Definition E.1, we have

$\text{Hopfield}(Z)$

$$= [f(L_0) \quad f(L_1) \quad \cdots \quad f(L_P)] \text{Softmax}\left(\beta \begin{bmatrix} 2l(z_1)L_0 - L_0^2 & 2l(z_2)L_0 - L_0^2 & \cdots & 2l(z_n)L_0 - L_0^2 \\ 2l(z_1)L_1 - L_1^2 & 2l(z_2)L_1 - L_1^2 & \cdots & 2l(z_n)L_1 - L_1^2 \\ \vdots & \vdots & & \vdots \\ 2l(z_1)L_P - L_P^2 & 2l(z_2)L_P - L_P^2 & \cdots & 2l(z_n)L_P - L_P^2 \end{bmatrix}\right).$$

This is equivalent to:

$\text{Hopfield}(Z)$

$$= [f(L_0) \quad f(L_1) \quad \cdots \quad f(L_P)] \text{Softmax}\left(-\beta \begin{bmatrix} (l(z_1)-L_0)^2 & (l(z_2)-L_0)^2 & \cdots & (l(z_n)-L_0)^2 \\ (l(z_1)-L_1)^2 & (l(z_2)-L_1)^2 & \cdots & (l(z_n)-L_1)^2 \\ \vdots & \vdots & & \vdots \\ (l(z_1)-L_P)^2 & (l(z_2)-L_P)^2 & \cdots & (l(z_n)-L_P)^2 \end{bmatrix}\right).$$

For any column $c \in [n]$ in $\text{Hopfield}(Z)$, we have

$$\text{Hopfield}(Z)_{:,c} = [f(L_0) \quad f(L_1) \quad \cdots \quad f(L_P)] \text{Softmax}\left(-\beta \begin{bmatrix} (l(z_c)-L_0)^2 \\ (l(z_c)-L_1)^2 \\ \vdots \\ (l(z_c)-L_P)^2 \end{bmatrix}\right)$$

$$= \sum_{r=1}^{P} \frac{e^{-\beta(l(z_c)-L_r)^2}}{\sum_{r'=1}^{P} e^{-\beta(l(z_c)-L_{r'})^2}} f(L_r).$$

When $\beta$ is large enough, we have

$$\sum_{(l(z_c)-L_r)^2 \geq \Delta L} \frac{e^{-\beta(l(z_c)-L_r)^2}}{\sum_{r'=1}^{P} e^{-\beta(l(z_c)-L_{r'})^2}} \leq \sum_{(l(z_i)-L_r)^2 \geq \Delta L} \frac{e^{-\beta\Delta L}}{e^{-\beta\frac{\Delta L}{2}}} \leq Pe^{-\frac{\beta\Delta L}{2}} \leq \epsilon_1,$$

for any $\epsilon_1 > 0$.

This means that the proportion of the $f(L_r)$ in $\text{Hopfield}(Z)_{:,c}$ that deviates from $l(z_c)$ is no larger than $\epsilon_1$.

Since $f$ and $l$ are continuous, and $Z$ comes from a compact domain, $l(z_i)$ comes from a compact domain for all $i \in [n]$. Thus $f$ is uniformly continuous on its input domain. This means that for any $\epsilon_2 > 0$, there exists a $\delta > 0$ such that when $(x - y)^2 \leq \delta$, $\|f(x) - f(y)\|_\infty \leq \epsilon_2$.

Configuring $\Delta L \leq \delta$ yields:

$$\|\text{Hopfield}(Z)_{:,c} - f(l(z_c))\|_\infty$$

$$\leq \sum_{r=1}^{P} \frac{e^{-\beta(l(z_c)-L_r)^2}}{\sum_{r'=1}^{P} e^{-\beta(l(z_c)-L_{r'})^2}} \|f(L_r) - f(l(z_c))\|_\infty$$

$$= \sum_{(l(z_c)-L_r)^2 \geq \Delta L} \frac{e^{-\beta(l(z_c)-L_r)^2}}{\sum_{r'=1}^{P} e^{-\beta(l(z_c)-L_{r'})^2}} \|f(L_r) - f(l(z_c))\|_\infty$$

$$+ \sum_{(l(z_c)-L_r)^2 \leq \Delta L} \frac{e^{-\beta(l(z_c)-L_r)^2}}{\sum_{r'=1}^{P} e^{-\beta(l(z_c)-L_{r'})^2}} \|f(L_r) - f(l(z_c))\|_\infty$$

$$\leq \epsilon_1 \cdot 2B + (1 - \epsilon_1)\epsilon_2,$$

where $B := \|f\|_{L_\infty}$ is the bound of $f$ in infinite norm.

We set $\epsilon_2 \leq \epsilon/2$, $\epsilon_1 \leq \epsilon/(4B)$. This yields:

$$\|\text{Hopfield}(Z)_{:,c} - f(l(z_c))\|_\infty \leq \epsilon_1 \cdot 2B + (1 - \epsilon_1)\epsilon_2$$

$$\leq \frac{\epsilon}{4B} \cdot 2B + 1 \cdot \frac{\epsilon}{2} = \epsilon.$$

This completes the proof. $\qquad\qquad\qquad\qquad\qquad\qquad\qquad\qquad\qquad\qquad\qquad\qquad\qquad\square$

**Theorem E.2.** Define

$$X := \begin{bmatrix} x_1 & x_2 & \cdots & x_n \\ y_1 & y_2 & \cdots & y_n \end{bmatrix} \in \mathbb{R}^{(d+1)\times n} \quad \text{and} \quad W := \begin{bmatrix} w & w & \cdots & w \end{bmatrix} \in \mathbb{R}^{d \times n},$$

where $x_i \in \mathbb{R}^d$ and $y_i \in \mathbb{R}$ are the input-output pairs. $w \in \mathbb{R}^d$ is the linear coefficient to optimize. Suppose $x_i, y_i$ and $w$ are bounded by $B$ in infinite norm.
For any continuous function $f : \mathbb{R} \to \mathbb{R}$, there exists a Hopfield layer $\text{Hopfield}$ with linear connections such that

$$\|\text{Hopfield}(W; X) - \begin{bmatrix} f(w^\top x_1 - y_1)x_1 & f(w^\top x_2 - y_2)x_2 & \cdots & f(w^\top x_n - y_n)x_n \end{bmatrix}\|_\infty \leq \epsilon,$$

for any $\epsilon > 0$.

*Proof.* Before plugging input $W$ to the Hopfield layer, we pass it through a linear transformation $\text{Linear}_w$:

$$\text{Linear}_w(W) := \begin{bmatrix} I_d \\ 0_{(d+n+2)\times d} \end{bmatrix} W + \begin{bmatrix} 0_{d\times n} \\ -1_{1\times n} \\ 0_{d\times n} \\ -1_{1\times n} \\ I_n \end{bmatrix} = \begin{bmatrix} W \\ -1_{1\times n} \\ 0_{d\times n} \\ -1_{1\times n} \\ I_n \end{bmatrix} \in \mathbb{R}^{(2d+n+2)\times n}.$$

We also pass $X$ through a linear transformation $\text{Linear}_x$:

$$\text{Linear}_x(X)$$

$$:= \sum_{i=1}^{n} \underbrace{\begin{bmatrix} I_{d+1} \\ 0_{(d+1+n)\times(d+1)} \end{bmatrix}}_{(2d+n+2)\times(d+1)} \underbrace{X}_{(d+1)\times n} \underbrace{\begin{bmatrix} 0_{n\times(i-1)(P+1)} & 2L_0 e_i^{(n)} & 2L_1 e_i^{(n)} & \cdots & 2L_P e_i^{(n)} & 0_{n\times(n-i)(P+1)} \end{bmatrix}}_{n\times n(P+1)}$$

$$+ \sum_{i=1}^{n} \underbrace{\begin{bmatrix} 0_{(d+1)\times d} & 0_{(d+1)} \\ I_d & 0_d \\ 0_{(n+1)\times d} & 0_{n+1} \end{bmatrix}}_{(2d+n+2)\times(d+1)} X \begin{bmatrix} 0_{n\times(i-1)(P+1)} & f(L_0)e_i^{(i)} & f(L_1)e_i^{(i)} & \cdots & f(L_P)e_i^{(i)} & 0_{n\times(n-i)(P+1)} \end{bmatrix}$$

$$+ \underbrace{\begin{bmatrix} 0_{(2d+1)\times(P+1)} & \cdots & 0_{(2d+1)\times(P+1)} \\ S & \cdots & S \\ (2dB^2 + B - \ln\epsilon_0)e_1^{(n)}1_{1\times(P+1)} & \cdots & (2dB^2 + B - \ln\epsilon_0)e_n^{(n)}1_{1\times(P+1)} \end{bmatrix}}_{(2d+n+2)\times n(P+1)}$$

$$= \begin{bmatrix} T_1 & T_2 & \cdots & T_n \end{bmatrix},$$

where

$$1_{1\times(P+1)} := \begin{bmatrix} 1 & 1 & \cdots & 1 \end{bmatrix} \in \mathbb{R}^{1\times(P+1)},$$
$$S := \begin{bmatrix} -L_0^2 & -L_1^2 & \cdots & L_P^2 \end{bmatrix} \in \mathbb{R}^{1\times(P+1)},$$
$$T_i := \begin{bmatrix} 2L_0 x_i & 2L_1 x_i & \cdots & 2L_P x_i \\ 2L_0 y_i & 2L_1 y_i & \cdots & 2L_P y_i \\ f(L_0)x_i & f(L_1)x_i & \cdots & f(L_P)x_i \\ -L_0^2 & -L_1^2 & \cdots & -L_P^2 \\ (2dB^2 + B - \ln\epsilon_0)e_i^{(n)} & (2dB^2 + B - \ln\epsilon_0)e_i^{(n)} & \cdots & (2dB^2 + B - \ln\epsilon_0)e_i^{(n)} \end{bmatrix} \in \mathbb{R}^{(2d+n+2)\times(P+1)}.$$

Here $\epsilon_0$ is a parameter that we will designate later according to $\epsilon$.

Now construct $W_K, W_Q, W_V$ to be:

$$W_Q := I_{2d+n+2},$$
$$W_K := I_{2d+n+2},$$
$$W_V^\top := \begin{bmatrix} 0_{d\times(d+1)} & I_d & 0_{d\times(n+1)} \end{bmatrix} \in \mathbb{R}^{d\times(2d+n+2)}.$$

Therefore, by Definition E.1, the output becomes:

$$\text{Hopfield}(\text{Linear}_w(W); \text{Linear}_x(X)) = W_V \text{Linear}_x(X)\text{Softmax}(\beta \text{Linear}_x(X)^\top \text{Linear}_w(W)),$$

where

$$\text{Softmax}(\text{Linear}_x(X)^\top \text{Linear}_w(W)) = \text{Softmax}(\beta \begin{bmatrix} T_1 & T_2 & \cdots & T_n \end{bmatrix}^\top \begin{bmatrix} W \\ -1_{1\times n} \\ 0_{d\times n} \\ -1_{1\times n} \\ I_n \end{bmatrix}).$$

This is equivalent to:

$$(\text{Linear}_x(X)^\top \text{Linear}_w(W))_{:,c} = \begin{bmatrix} T_1^\top \\ T_2^\top \\ \vdots \\ T_n^\top \end{bmatrix} \cdot \begin{bmatrix} w \\ -1 \\ 0_d \\ -1 \\ e_c^{(n)} \end{bmatrix}$$

$$= \begin{bmatrix} M_{1,c} \\ M_{2,c} \\ \vdots \\ M_{n,c} \end{bmatrix},$$

where

$$M_{i,c} := T_i^\top \cdot \begin{bmatrix} w \\ -1 \\ 0_d \\ -1 \\ e_i^{(n)} \end{bmatrix}$$

$$= \begin{bmatrix} 2L_0 x_i^\top w - 2L_0 y_i - L_0^2 + (2dB^2 + B - \ln \epsilon_0) \mathbb{1}_{i=c} \\ 2L_1 x_i^\top w - 2L_1 y_i - L_1^2 + (2dB^2 + B - \ln \epsilon_0) \mathbb{1}_{i=c} \\ \cdots \\ 2L_P x_i^\top w - 2L_P y_i - L_P^2 + (2dB^2 + B - \ln \epsilon_0) \mathbb{1}_{i=c} \end{bmatrix},$$

where $i \in [n]$ and $c \in [n]$, and $\mathbb{1}_{i=c}$ represents the indicator function of $i = c$.

This means that

$$\mathrm{Softmax}(\beta \mathrm{Linear}_x(X)^\top \mathrm{Linear}_w(W))_{:,c}$$

$$= \mathrm{Softmax}(\beta \begin{bmatrix} M_{1,c} \\ M_{2,c} \\ \vdots \\ M_{n,c} \end{bmatrix})$$

$$= \beta \sum_{i=1}^n \sum_{j=1}^P \frac{\exp\{2L_j x_i^\top w - 2L_j y_i - L_0^2 + (2dB^2 + B - \ln \epsilon_0) \mathbb{1}_{i=c}\}}{\sum_{i'=1}^n \sum_{j'=0}^P \exp\{u_{j'}^{(i')} + (2dB^2 + B - \ln \epsilon_0) \mathbb{1}_{i=c}\}} e_{(i-1)P+j}^{(nP)}.$$

Thus we have (without loss of generality, we ignore the $\beta$ parameter in Softmax):

$$\mathrm{Hopfield}(\mathrm{Linear}_w(W); \mathrm{Linear}_x(X))_{:,c}$$
$$= W_V \mathrm{Linear}_x(X) \mathrm{Softmax}(\mathrm{Linear}_x(X)^\top \mathrm{Linear}_w(W))_{:,c}$$
$$\quad (W_v \text{ only retrieves the } (d+2)\text{-th row in } T_i)$$

$$= [F_1 \quad \cdots \quad F_n] \sum_{i=1}^n \sum_{j=1}^P \frac{\exp\{2L_j x_i^\top w - 2L_j y_i - L_j^2 + (2dB^2 + B - \ln \epsilon_0) \mathbb{1}_{i=c}\}}{\sum_{i'=1}^n \sum_{j'=0}^P \exp\{2L_{j'} x_{i'}^\top w - 2L_{j'} y_{i'} - L_{j'}^2 + (2dB^2 + B - \ln \epsilon_0) \mathbb{1}_{i=c}\}} e_{(i-1)P+j}^{(nP)}$$

$$= \sum_{i=1}^n \sum_{j=0}^P \frac{\exp\{2L_j x_i^\top w - 2L_j y_i - L_j^2 + (2dB^2 + B - \ln \epsilon_0) \mathbb{1}_{i=c}\}}{\sum_{i'=1}^n \sum_{j'=0}^P \exp\{2L_{j'} x_{i'}^\top w - 2L_{j'} y_{i'} - L_{j'}^2 + (2dB^2 + B - \ln \epsilon_0) \mathbb{1}_{i=c}\}} f(L_j) x_i,$$

where $F$ is:

$$F_i := [f(L_0) x_i \quad f(L_1) x_i \quad \cdots \quad f(L_P) x_i].$$

For every $i \in [n]$, if $i \neq c$, we have

$$\sum_{j=0}^P \frac{\exp\{2L_j x_i^\top w - 2L_j y_i - L_j^2 + (2dB^2 + B - \ln \epsilon_0) \mathbb{1}_{i=c}\}}{\sum_{i'=1}^n \sum_{j'=0}^P \exp\{2L_{j'} x_{i'}^\top w - 2L_{j'} y_{i'} - L_{j'}^2 + (2dB^2 + B - \ln \epsilon_0) \mathbb{1}_{i=c}\}}$$

$$= \sum_{j=0}^{P} \frac{\exp\{2L_j x_i^\top w - 2L_j y_i - L_j^2\}}{\sum_{i'=1}^{n} \sum_{j'=0}^{P} \exp\{2L_{j'} x_{i'}^\top w - 2L_{j'} y_{i'} - L_{j'}^2 + (2dB^2 + B - \ln \epsilon_0)\mathbb{1}_{i=c}\}}$$

$$< \sum_{j=0}^{P} \frac{\exp\{2L_j x_i^\top w - 2L_j y_i - L_j^2\}}{\sum_{j'=0}^{P} \exp\{2L_{j'} x_{i'}^\top w - 2L_{j'} y_{i'} - L_{j'}^2 + (2dB^2 + B - \ln \epsilon_0)\}}$$

(only taking the $i' = c$ part)

$$< \sum_{j=0}^{P} \frac{\exp\{2dB^2 + B\}}{P \exp(2dB^2 + B - \ln \epsilon_0)} = \epsilon_0.$$

For $i = c$, since

$$\sum_{i \neq c}^{n} \sum_{j=0}^{P} \frac{\exp\{2L_j x_i^\top w - 2L_j y_i - L_j^2 + (2dB^2 + B - \ln \epsilon_0)\mathbb{1}_{i=c}\}}{\sum_{i'=1}^{n} \sum_{j'=0}^{P} \exp\{2L_{j'} x_{i'}^\top w - 2L_{j'} y_{i'} - L_{j'}^2 + (2dB^2 + B - \ln \epsilon_0)\mathbb{1}_{i=c}\}} \leq (n-1)\epsilon_0,$$

we have

$$\frac{\sum_{j=0}^{P} \exp\{u_j^{(c)} + (2dB^2 + B - \ln \epsilon_0)\}}{\sum_{i'=1}^{n} \sum_{j'=0}^{P} \exp\{2L_{j'} x_{i'}^\top w - 2L_{j'} y_{i'} - L_{j'}^2 + (2dB^2 + B - \ln \epsilon_0)\mathbb{1}_{i=c}\}}$$

$$= \sum_{j=0}^{P} \frac{\exp\{u_j^{(c)} + (2dB^2 + B - \ln \epsilon_0)\}}{\sum_{i'=1}^{n} \sum_{j'=0}^{P} \exp\{2L_{j'} x_{i'}^\top w - 2L_{j'} y_{i'} - L_{j'}^2 + (2dB^2 + B - \ln \epsilon_0)\mathbb{1}_{i=c}\}}$$

$$\geq 1 - (n-1)\epsilon_0.$$

Thus for the parts in the weighted sum output that corresponds to rows in $M_{:,c}$ in the attention score matrix, we have

$$\| \sum_{j=0}^{P} \frac{\exp\{u_j^{(c)} + (2dB^2 + B - \ln \epsilon_0)\}}{\sum_{i'=1}^{n} \sum_{j'=0}^{P} \exp\{2L_{j'} x_{i'}^\top w - 2L_{j'} y_{i'} - L_{j'}^2 + (2dB^2 + B - \ln \epsilon_0)\mathbb{1}_{i=c}\}} f(L_j) x_c - f(x_c^\top w - y_c) x_c \|_\infty$$

$$= \| \sum_{j=0}^{P} \frac{\exp\{u_j^{(c)} + (2dB^2 + B - \ln \epsilon_0)\}}{\sum_{j'=0}^{P} \exp\{u_{j'}^{(c)} + (2dB^2 + B - \ln \epsilon_0)\}} (f(L_j) x_c - f(x_c^\top w - y_c) x_c)$$

$$\cdot \frac{\sum_{j'=0}^{P} \exp\{u_{j'}^{(c)} + (2dB^2 + B - \ln \epsilon_0)\}}{\sum_{i'=1}^{n} \sum_{k=0}^{P} \exp\{u_k^{(i')} + (2dB^2 + B - \ln \epsilon_0)\mathbb{1}_{i=c}\}}$$

$$- (1 - \frac{\sum_{j'=0}^{P} \exp\{u_{j'}^{(c)} + (2dB^2 + B - \ln \epsilon_0)\}}{\sum_{i'=1}^{n} \sum_{k=0}^{P} \exp\{u_k^{(i')} + (2dB^2 + B - \ln \epsilon_0)\mathbb{1}_{i=c}\}}) f(x_c^\top w - y_c) x_c \|_\infty$$

$$\leq \sum_{j=0}^{P} \frac{\exp\{u_j^{(c)} + (2dB^2 + B - \ln \epsilon_0)\}}{\sum_{j'=0}^{P} \exp\{u_{j'}^{(c)} + (2dB^2 + B - \ln \epsilon_0)\}} |f(L_j) - f(x_c^\top w - y_c)| \cdot d \|x_c\|_\infty$$

$$- (1 - \frac{\sum_{j'=0}^{P} \exp\{u_{j'}^{(c)} + (2dB^2 + B - \ln \epsilon_0)\}}{\sum_{i'=1}^{n} \sum_{k=0}^{P} \exp\{u_k^{(i')} + (2dB^2 + B - \ln \epsilon_0)\mathbb{1}_{i=c}\}}) |f(x_c^\top w - y_c)| \|x_c\|_\infty$$

$$\leq \sum_{j=0}^{P} \frac{\exp\left\{u_j^{(c)} + (2dB^2 + B - \ln \epsilon_0)\right\}}{\sum_{j'=0}^{P} \exp\left\{u_{j'}^{(c)} + (2dB^2 + B - \ln \epsilon_0)\right\}} |f(x_c^\top w - y_c)| \|x_c\|_\infty$$
$$+ (n-1)\epsilon_0 B_f \|x_c\|_\infty$$

$$= \sum_{j=0}^{P} \frac{\exp\left\{u_j^{(c)}\right\}}{\sum_{j'=0}^{P} \exp\left\{u_{j'}^{(c)}\right\}} |f(L_j) - f(x_c^\top w - y_c)| \|x_c\|_\infty + (n-1)\epsilon_0 B_f \|x_c\|_\infty$$

$$= \sum_{j=0}^{P} \frac{\exp\{-\beta(x_c^\top w - y_c - L_j)^2\}}{\sum_{j'=0}^{P} \exp\{-\beta(x_c^\top w - y_c - L_{j'})^2\}} |f(L_j) - f(x_c^\top w - y_c)| \|x_c\|_\infty + (n-1)\epsilon_0 B_f \|x_c\|_\infty,$$

where we define $B_f := |f|$ as the bound for $f$.

For any $\epsilon_1 > 0$, set $\Delta L$ to be sufficiently small such that

$$|f(x) - f(y)| \leq \epsilon_1,$$

when $|x - y| \leq \Delta L$.

Then when $\beta$ is sufficiently large, we have

$$\sum_{|L_i - (x_c^\top w - y_c)| > \Delta L} \frac{\exp\{-\beta(x_c^\top w - y_c - L_j)^2\}}{\sum_{j'=0}^{P} \exp\{-\beta(x_c^\top w - y_c - L_{j'})^2\}} \leq \epsilon_2,$$

for any $\epsilon_2 > 0$.

Thus

$$\sum_{j=0}^{P} \frac{\exp\{-\beta(x_c^\top w - y_c - L_j)^2\}}{\sum_{j'=0}^{P} \exp\{-\beta(x_c^\top w - y_c - L_{j'})^2\}} |f(L_j) - f(x_c^\top w - y_c)|$$

$$= \sum_{|L_i - (x_c^\top w - y_c)| > \Delta L} \frac{\exp\{-\beta(x_c^\top w - y_c - L_j)^2\}}{\sum_{j'=0}^{P} \exp\{-\beta(x_c^\top w - y_c - L_{j'})^2\}} |f(L_j) - f(x_c^\top w - y_c)|$$

$$+ \sum_{|L_i - (x_c^\top w - y_c)| \leq \Delta L} \frac{\exp\{-\beta(x_c^\top w - y_c - L_j)^2\}}{\sum_{j'=0}^{P} \exp\{-\beta(x_c^\top w - y_c - L_{j'})^2\}} |f(L_j) - f(x_c^\top w - y_c)|$$

$$\leq \epsilon_2 \cdot 2B_f + \epsilon_1.$$

This completes the proof. $\qquad\square$

**Corollary E.2.1** (In-Context GD of Hopfield Layer). Define

$$X := \begin{bmatrix} x_1 & x_2 & \cdots & x_n \\ y_1 & y_2 & \cdots & y_n \end{bmatrix} \in \mathbb{R}^{(d+1)\times n} \quad \text{and} \quad W := \begin{bmatrix} w & w & \cdots & w \end{bmatrix} \in \mathbb{R}^{d\times n},$$

where $x_i \in \mathbb{R}^d$ and $y_i \in \mathbb{R}$ are the input-output pairs. $w \in \mathbb{R}^d$ is the linear coefficient we aim to optimize. For any differentiable loss function $\ell : \mathbb{R} \to \mathbb{R}$, There exists a Hopfield layer Hopfield with linear connections such that

$$\|\text{Hopfield}(W; X) - \begin{bmatrix} \nabla\ell(w^\top x_1 - y_1)x_1 & \nabla\ell(w^\top x_2 - y_2)x_2 & \cdots & \nabla\ell(w^\top x_n - y_n)x_n \end{bmatrix} \|_\infty \leq \epsilon,$$

for any $\epsilon > 0$.

*Proof.* Replacing the continuous function $f$ in Theorem E.2 with $\nabla\ell$ completes the proof. $\qquad\square$

