# OpenReview forum: "In-Context Algorithm Emulation in Fixed-Weight Transformers"
_ICLR.cc/2026/Conference — ICLR 2026 Poster_

### Official Review · Reviewer_wRby · 2025-10-28

**Soundness:** 3
**Presentation:** 4
**Contribution:** 3
**Rating:** 8
**Confidence:** 4

**Summary:**

This work provides a theoretical framework that **frozen softmax attention modules** within minimal Transformer architectures can emulate a wide class of algorithms purely through **prompt programming**. The authors formalize two emulation modes — *task-specific* and *prompt-programmable*. In the **task-specific mode**, they show that a single-head softmax attention layer with a linear map can approximate functions of the general form $f(w^\top x - y)x$ for any continuous $f:\mathbb{R}\to\mathbb{R}$ and arbitrary precision $\epsilon>0$. This includes key algorithmic operations such as per-sample gradient computation, single-step and multi-step gradient descent, linear regression, and ridge regression. The main theoretical statement, **Theorem 3.1**, asserts that for any bounded input $X\in\mathbb{R}^{(d+1)\times n}$ and coefficient vector $w\in\mathbb{R}^d$, there exists a single-head attention $Attn_s$ and linear map $Linear$ to achieve the desired precision.

From this, they derive **Corollary 3.1.1**, which shows that with $f(t) = \ell'(t)$ for differentiable loss $\ell$, the layer emulates per-sample gradients $\ell'(w^\top x_i - y_i)x_i$, and **Corollary 3.1.2** extends this to one-step gradient descent: for $\eta>0$, there exists an attention map producing the respective approximation of the $w^{+}_{GD}$.

Multi layer extensions show that stacking $(L+1)$ single-head attention layers yields $L$ gradient descent steps with bounded cumulative error. Similarly, **Corollaries 3.1.3** and **3.1.4** construct attention-based emulations for linear and ridge regression respectively.

In the **prompt-programmable mode**, the paper’s main result (**Theorem 4.1**) proves that a two-layer softmax attention module with fixed weights can emulate any single attention head by embedding its weights $(W_K, W_Q, W_V)$ into the prompt. Specifically, defining the prompt as
$$X_p = \begin{bmatrix} X \ W_{\text{in}} \ I_n \end{bmatrix}, \quad W_{\text{in}} = \begin{bmatrix} 0\cdot w & 1\cdot w & \cdots & (n-1)\cdot w \ w & w & \cdots & w \end{bmatrix},$$
where $w = [\text{vec}(W_K);\text{vec}(W_Q);\text{vec}(W_V)]$, they prove
$$|Attn_s \circ Attn_m(X_p) - W_V X \operatorname{Softmax}((W_KX)^\top W_QX)|_\infty \leq \epsilon.$$
Thus, a **two-layer frozen Transformer** is *universal* for all algorithms implementable by single-layer attention. **Theorem 4.2** provides an equivalent formulation, showing that a single-head attention followed by a multi-head layer with linear projections approximates the same target attention mapping, implying that prompt-programmed softmax attention retains permutation equivariance and can simulate any bounded attention mechanism. **Corollary 4.2.1** extends this to a library $\mathcal{A}_0 = {a_1,\ldots,a_k}$ of algorithms, proving that one fixed two-layer module can emulate every $a\in\mathcal{A}_0$

Finally, the authors generalize this to arbitrary linear networks, showing that any trainable linear map $f(x)=\Theta x$ can be emulated by encoding $\Theta$ in the prompt — turning static networks into in-context learners that execute dynamically programmable linear mappings.

Empirical studies (Figures 1–2, Table 1) confirm that softmax attention layers approximate $f(w^\top x - y)x$ and even emulate attention heads with mean squared error decreasing roughly as $O(1/H)$ with the number of heads $H$. The findings demonstrate that fixed-weight softmax attention networks can serve as **prompt-programmable algorithmic evaluators**.

**Strengths:**

1. The extension of previous works to propose a framework adaptable to a broader class of algorithms via the prompt-programmability with the fixed attention module transformer is a strong contribution to the community, especially to the line of research focused on guarantees for prompting based approaches.

2. I believe the emphasis in lines 361-363 about - “our results are constructive, providing concrete emulation examples in contrast to prior prompting expressivity (Wang et al., 2023; Furuya et al., 2024) or Turing-completeness results ...” - is a strong contribution as well.

**Weaknesses:**

1. I have a concern about the final bound discussed in lines 1704 to 1727. Specifically, for the value of $\epsilon$ that will be attained by plugging the values of $B_f$ and $B$, the bounds for $\epsilon_1$ and $\epsilon_2$ need to be quite tight. Can the authors discuss this in a bit more detail? I think this is an important characterization.

2. I have  a serious question about experimental section 5.1 - since the data has been generated synthetically and the parameters ‘W’ as well as the labels ‘y’ are known, I understand that the task should be use the construction in the proof of theorem 3.1 to adjust the input as well as the attention weight matrices appropriately to observe the approximation error. Why was there a need to use an optimizer to perform explicit training? And even if the training has been performed using the optimizer, a direct check is to compute the difference between the learned attention layer weights and the weights generated explicitly in lines 1578-1585 of the proof – although I understand this will likely fail due to issues in optimization and convergence to the other minima. Open to any other suggestions from the authors and looking forward to the clarification in case I mis-understood something here.

**Questions:**

1. The use of ‘L’ seems overloaded at times. What exactly are ‘L’ and ‘P’ in the proof of theorem D.1 in appendix, initiated at lines 1553-54 of appendix in the definition of $\textbf{X}$ ? Do they follow the same initialization as in lines 2040-2050 on page 38 in the proof of theorem D.6?

2. I am curious if the authors have thought about the extensions to the cases with positional encoding (some form of absolute/relative positional encodings), instead of the Identify pos encoding used in the proofs, along with the default input $\textbf{X}$. Please note that this question DOES NOT reflect my score, and I am just curious, as the authors have put in much effort to characterize the in-context setup, whether there are any thoughts on the positional encoding case. I understand some forms of positional encoding will break equivariance as well, but irrespective of this, what is your opinion on whether positional encodings will induce any extra complications in the proof sketch if any?

---

> ### Author Response · Authors · 2025-11-13
> **Rebuttal 1: Responses to Strengths and Weaknesses**
>
> ### We thank the reviewer for the detailed review. In response, we have addressed all concerns and questions in the following replies and revisions in the updated draft. All changes from the originally submitted version are highlighted in blue in the revised PDF.
>
> ---
> > **Strength 2.** I believe the emphasis in lines 361-363 about - “our results are constructive, providing concrete emulation examples in contrast to prior prompting expressivity (Wang et al., 2023; Furuya et al., 2024) or Turing-completeness results ...” - is a strong contribution as well.
>
> **Response:**
>
> Thanks for the encouragement.
>
> To clarify, our notion of “in-context algorithmic universality” remains restricted. Our results (Thm. 4.1 and 4.2) apply only to **attention-implementable** algorithms. This is still a general class, but it covers only a subset of the full range of algorithms considered in prompt-universality and Turing-completeness results. Achieving parity with those results would require extending Thm. 3.1 to allow arbitrary target functions $f$. However, we remark that our results still complement these results with constructive examples.
>
> We are currently working on this extension by relaxing the approximation target in Thm. 4.1 and 4.2 from a single attention layer to broader function classes.
>
> ---
> > **Weakness 1.** I have a concern about the final bound discussed in lines 1704 to 1727. Specifically, for the value of $\epsilon$ that will be attained by plugging the values of $B_f$ and $B$, the bounds for $\epsilon_1$ and $\epsilon_2$ need to be quite tight. Can the authors discuss this in a bit more detail? I think this is an important characterization.
>
> **Response:**
>
> Thanks for pointing this out. We agree that the proof clarity in the original submission was suboptimal. We have revised accordingly. In our latest revision, all main results’ proofs are refined for better structure and clarity.
>
> The concerned bound in original `line 1704-1727` is now `line 2199-2201` , with each term explicitly bounded and all intermediate steps included. The corresponding derivations and references have been corrected. We also added an explicit decomposition of the target approximation in D.22, D.23, D.24 (`line 2021-2140` of the latest revision) to make the dependence on $B_f$, $B$, $\epsilon_1$, and $\epsilon_2$ transparent.
>
> ---
>
> > **Weakness 2.**  I have a serious question about experimental section 5.1 - since the data has been generated synthetically and the parameters ‘W’ as well as the labels ‘y’ are known, I understand that the task should be use the construction in the proof of theorem 3.1 to adjust the input as well as the attention weight matrices appropriately to observe the approximation error. Why was there a need to use an optimizer to perform explicit training? And even if the training has been performed using the optimizer, a direct check is to compute the difference between the learned attention layer weights and the weights generated explicitly in lines 1578-1585 of the proof – although I understand this will likely fail due to issues in optimization and convergence to the other minima. Open to any other suggestions from the authors and looking forward to the clarification in case I mis-understood something here.
>
> **Response:**
>
> Thanks for the question. You are correct that Section 6.1 does not implement the explicit construction in Theorem 3.1. The goal of the experiment is different: to test whether the same functional behavior can be recovered through standard training, without any handcrafted initialization.
>
> The construction in Theorem 3.1 is a fully explicit existence proof (see Appendix B.1). It requires carefully structured tokens and fixed weights with identity and indicator submatrices. Implementing this exactly would demand precise alignment between input format and attention weights, which is not optimizer-friendly. Instead, we train the attention layer from scratch using Adam, and observe that it successfully learns to approximate $f(w^\top x - y)x$ with near-zero error.
>
> This choice is consistent with prior work. Bai et al. (2023) adopt the same strategy: they validate their theoretical constructions by training standard transformers on synthetic tasks rather than instantiating the hand-crafted theoretical weights.
>
> We agree that comparing the learned weights to the constructed ones is in principle possible, but likely uninformative due to symmetry and non-uniqueness of solutions. Our aim is to demonstrate that in-context emulation is achievable under standard training, without assuming access to analytic construction.

---

> ### Author Response · Authors · 2025-11-13
> **Rebuttal 2: Responses to Questions**
>
> > **Question 1.** The use of ‘L’ seems overloaded at times. What exactly are ‘L’ and ‘P’ in the proof of theorem D.1 in appendix, initiated at lines 1553-54 of appendix in the definition of $\textbf{X}$ ? Do they follow the same initialization as in lines 2040-2050 on page 38 in the proof of theorem D.6?
>
> **Response:**
> Thanks for your detailed proofreading. You are absolutely correct. We did not state the definition of L explicitly enough in the original submission. L, P are the interpolation points and precision parameter following [Hu et al., 2025].
>
> We have revised accordingly. In our latest revision pdf `line 1710-1715` (right before the proof of Thm 3), we define them explicitly. We quote them below for your reference:
> >> **Definition D.3 (Interpolation Points).**
> Define $P+1$ interpolation points of the effective domain of $f$ (the range of $w^\top x - y$) as
> $L\_j = L\_{\min} + \frac{j}{P}(L\_{\max} - L\_{\min})$ for $j = 0,1,\ldots,P$,
> where $[L\_{\min}, L\_{\max}]$ is a bounded interval containing all values of $w^\top x - y$.
>
> ---
> > **Question 2.** I am curious if the authors have thought about the extensions to the cases with positional encoding (some form of absolute/relative positional encodings), instead of the Identify pos encoding used in the proofs, along with the default input $\textbf{X}$. Please note that this question DOES NOT reflect my score, and I am just curious, as the authors have put in much effort to characterize the in-context setup, whether there are any thoughts on the positional encoding case. I understand some forms of positional encoding will break equivariance as well, but irrespective of this, what is your opinion on whether positional encodings will induce any extra complications in the proof sketch if any?
>
>  **Response:**
>
> Thanks for the question. We appreciate your attention to detail, as this question is on point with our current research.
>
> Yes, we have considered positional encodings. We currently have ongoing work extending our setting to sequential algorithms (e.g., MCMC, Bayesian updates, time-series procedures), where positional information becomes essential. So far, we have only verified that absolute positional encodings are compatible with our proof technique. More complex positional encodings remain unexplored.
>
> The main complication is that additional structure in the positional encoding can interfere with the clean token-routing behavior we rely on. In such cases, extra operations (e.g., linear preprocessing or an additional attention layer) may be needed to recover the required routing guarantees. Further systematic analysis is left for future work.
>
> ---
>
> We have addressed all your comments and questions. Thank you again for your detailed review. Please do not hesitate to let us know if there are any other aspects of our work that you would like us to clarify :)

---

> ### Author Response · Authors · 2025-11-14
> **Additional exps in Sec C.3 with hardcoded weights**
>
> Dear reviewer wRby,
>
> We have added a new Sec. C.3 for additional numerical validation of Thm. 4.2, where all weights and prompts are hard-wired as you suggested.
>
> We summarize as follows for your convenience.
>
> **Experiment setup.**
> We construct a synthetic dataset by sampling $X \in \mathbb{R}^{n \times d}$ i.i.d. from the uniform distribution on $[-1,1]$. For each sample, we draw three weight matrices $W_K, W_Q, W_V \in \mathbb{R}^{n \times d}$ from $N(0,1)$ and form
> $$
> K = W_K X^\top,\quad Q = W_Q X^\top,\quad V = W_V X^\top \in \mathbb{R}^{n \times n}.
> $$
> The “ground-truth’’ attention output is
> $$
> Y = V \operatorname{Softmax}(K^\top Q) \in \mathbb{R}^{n \times n}.
> $$
>
> **Handcrafted emulator.**
> Following the construction in Appendix D.6, we hard-wire (i) the linear-layer weights, (ii) the attention weights, and (iii) the interpolation points of the two-layer softmax attention module so that it implements the frozen emulator prescribed by Thm. 4.2. The model runs in a zero-shot, one-pass setting with *no* training or parameter updates.
>
> **Results.**
> We compare the MSE between the handcrafted emulator output and the target attention $Y$. We fix the number of data points $n$, input dimension $d$, softmax temperature $\beta$, and number of test samples, and vary only the number of interpolation points $P$. As shown in Fig. 5 and Table 4 below, the approximation error rapidly decreases as $P$ increases, and the variance across samples also shrinks, quantitatively confirming that the handcrafted frozen attention closely approximates softmax attention in practice.
>
> Table 4: **Sensitivity to the number of interpolation points $P$.** We report mean $\pm$ std of the MSE over 4 random samples with $n = 12$, $d = 4$, and $\beta = 2$.
>
> | $P$      | 20              | 30              | 40              | 60              | 80              |
> |---------:|----------------:|----------------:|----------------:|----------------:|----------------:|
> | Mean MSE | $4.002 \times 10^{-1}$ | $2.442 \times 10^{-2}$ | $5.852 \times 10^{-4}$ | $5.770 \times 10^{-9}$ | $5.037 \times 10^{-14}$ |
> | Std      | $2.393 \times 10^{-1}$ | $1.451 \times 10^{-2}$ | $8.538 \times 10^{-5}$ | $1.994 \times 10^{-9}$ | $1.620 \times 10^{-14}$ |
>
> (See new Fig. 5 in Sec. C.3 for a visualization of MSE $\pm$ std as a function of $P$.)
>
> [Figure 5: Sensitivity of handcrafted attention emulator]( https://imgur.com/a/GkEmmyb)
>
> **Caption of Fig 5.** We report MSE loss (mean$\pm$ std) between outputs of handcrafted frozen attention and target attention varying number of interpolation points $P$ over $4$ samples. We choose $n=12, d=4, \beta=2, \text{samples}=4$ for evaluation.
>
> Code to reproduce this part is also available in the supplementary material.
>
> ---
>
> We hope this address your concerns. Looking forward to further discussions!

---

> > ### Comment · Reviewer_wRby · 2025-11-16
> > **Response to the rebuttal**
> >
> > Hi, thanks for the details you have mentioned here. Especially given the empirical validation of the hard-wired weights, along with the clarification of the terms $B_f$, $B$, $\epsilon_1$ and $\epsilon_2$ (while I still believe the bound dependence is not quite tight, but I see it kind of harder to make this dependence tight - so I can't stress it too much at this point since I can't think much better dependence myself at this point)
> > I am increasing the score to 10 and am willing to support my decision should there be a consensus vote from the ACs and SACs. Good luck with the rest of the rebuttal.

---

> ### Author Response · Authors · 2025-11-16
> **thank you note**
>
> Thank you so much for your careful reading and endorsement (and speedy response!). We are very happy that our responses resolved your concerns. Wishing you all the best as well!

---

### Official Review · Reviewer_BBMQ · 2025-11-01

**Soundness:** 3
**Presentation:** 3
**Contribution:** 3
**Rating:** 6
**Confidence:** 4

**Summary:**

This paper studies the expressive power of the transformer architecture (especially a single attention layer) in the in-context learning (ICL) setting. It shows that (i) a single attention layer can implement the map $[x; y; w] \mapsto f(x^\top w - y)x$, extending much of the existing work on the ICL expressivity of transformers; and (ii) Transformers can express the functorial map $(X, A) \mapsto A(X)$, where $A$ is an attention layer specified by its weights, i.e., there exists a fixed-weight transformer that, given a prompt containing data X and a "program" A, executes the program and outputs A(X), provided A is represented in terms of attention weights.

**Strengths:**

The paper is well written, and the intuition is explained in detail. The theoretical results are strong and extend several prior papers on the ICL capabilities of Transformer architectures.

**Weaknesses:**

(1) The innovation over Hu et al. (2025) should be discussed more comprehensively. For the proof of Theorem 3.1, the ideas appear to follow Hu et al. (2025), albeit with substantial technical development. It would be beneficial to clarify which ideas/techniques are inherited from prior work and what are new here.

(2) The dimension of the linear layer is not stated explicitly in the theorems. Providing an explicit bound on this dimension (e.g., in Theorem 3.1, in terms of regularity conditions on $f$) would help readers understand the expressivity at bounded width and would be useful for establishing generalization bounds.

I will adjust my score provided the issues above are addressed.

**Questions:**

(1) Section 4.1 shows how transformer can express $(X, A) \mapsto A(X)$ when $A$ is described by attention weights. Could a stronger result show that a transformer can express $(X, A) \mapsto A(X)$ for circuits/Turing machines A subject to structural constraints (e.g., bounded depth)? Does the construction extend to this setting?

(2) Corollary 3.1.3 and 3.1.4 are stated for single-layer attention, but their proof seems to be based on stacking attention attention layers that are implementing GD.

---

> ### Author Response · Authors · 2025-11-12
> **Rebuttal 1: Responses to Weaknesses and Questions**
>
> ### We thank the reviewer for the detailed review. In response, we have addressed all concerns and questions in the following replies and revisions in the updated draft. All changes from the originally submitted version are highlighted in blue in the revised PDF.
>
>
> ---
>
> > **Weaknesses 1.** The innovation over Hu et al. (2025) should be discussed more comprehensively. For the proof of Theorem 3.1, the ideas appear to follow Hu et al. (2025), albeit with substantial technical development. It would be beneficial to clarify which ideas/techniques are inherited from prior work and what are new here.
>
>
> **Response:**
>
> **We use Hu et al. (2025) solely as a technical lemma (now Lemma B.1) to approximate hard attention using finite-temperature softmax.** It is a tool to control softmax behavior and ensure token selection behaves as intended.
>
> **Beyond that, our constructions and proofs are new.** We explicitly **design the prompt to encode algorithm parameters into the input sequence** (e.g., Eq. (3.1), prompt matrix $Z = [X; W]$), and **construct fixed emulator weights to compute the algorithm update.** The prompt format uses indicator patterns and interpolation points tailored to the function class, and our architecture design induces large query-key gaps that guide attention deterministically. **None of these appear in Hu et al. (2025).**
>
> **Our goal is not just expressivity, but constructive in-context emulation.** The Transformer is fixed. Only the prompt changes across tasks. **This prompt-programmable emulation and algorithm-swapping mechanism are not studied in prior work.**
>
> ---
>
> > **Weaknesses 2.** The dimension of the linear layer is not stated explicitly in the theorems. Providing an explicit bound on this dimension (e.g., in Theorem 3.1, in terms of regularity conditions on
> ) would help readers understand the expressivity at bounded width and would be useful for establishing generalization bounds.
>
> **Response:**
> Thanks for pointing this out. We have made revisions accordingly.
>
> **The revised version had to make all dimensions explicit whenever possible.** We have revised the proofs of main theorems (Thm 3.1, 4.1, 4.2) and revised our proof sketches. The revision includes typos and errors fixes, and clarity improvements.
>
> ---
>
> > **Question 1.** Section 4.1 shows how transformer can express $(X,A)\to A$ when $A$ is described by attention weights. **Could a stronger result show that a transformer can express $(X,A)\to A$ for circuits/Turing machines $A$ subject to structural constraints (e.g., bounded depth)?** Does the construction extend to this setting?
>
> **Response:**
>
> It’s possible. We have ongoing projects working in this direction. Here is a brief sketch: first we extend the Thm 4.1 result to a more general in-context universal approximation form. So that the emulation target is relaxed from attention form. Then, we proof function composition emulation based on this “in-context UAP”. Then, we will have enough tools to construct the mentioned circuits or structural emulation targets.
>
> ---
>
> > **Question 2.** Corollary 3.1.3 and 3.1.4 are stated for single-layer attention, but their proof seems to be based on stacking attention attention layers that are implementing GD.
>
> **Response:**
> Thank you for pointing this out. To clarify, both Corollaries 3.1.3 and 3.1.4 build on Corollary 3.1.2, which shows that a single-layer softmax attention emulates one GD step. Our proofs unroll this single-step primitive (by stacking or looping) as an analysis device with standard GD convergence. The final construction remains a single-head, single-layer attention with a linear readout that approximates the linear/ridge minimizer to arbitrary accuracy in one forward pass.
>
> If desired, one can realize multi-step behavior either by stacking identical layers or by looping a single layer with prompt updates $w\leftarrow\hat w_{\text{GD}}$ in $Z=[X;W]$ (feeding $\hat w_{\text{GD}}$ back). The corollaries themselves state the single-layer case.
>
> ---
>
> We hope that the revisions and clarifications provided in this response address the reviewer's concerns and make the value of our work clear. We look forward to further feedback and discussions.

---

### Official Review · Reviewer_6rGC · 2025-11-01

**Soundness:** 3
**Presentation:** 3
**Contribution:** 2
**Rating:** 6
**Confidence:** 4

**Summary:**

This paper investigates the capability of fixed-weight Transformer architectures to emulate a broad class of algorithms through in-context prompting, addressing a core question in Transformer research: how frozen models execute diverse tasks via context alone. The authors formalize two modes of in-context algorithm emulation:

1. Task-Specific Mode: A single-head softmax attention layer (with a linear map) can universally approximate functions of the form $f(w^\mathrm{T}x-y)$ (for any continuous $f$) to arbitrary precision. This template includes key machine learning algorithms like gradient descent (GD), linear regression, and ridge regression.
2. Prompt-Programmable Mode: A single fixed-weight two-layer softmax attention module achieves universality—emulating all task-specific algorithms via prompt design alone. The key mechanism is encoding algorithm parameters into prompts to create sharp dot-product gaps, forcing softmax attention to follow intended computations without feed-forward layers or parameter updates.
The authors validate their theory numerically: synthetic experiments confirm accurate approximation of continuous functions, with error decreasing as the number of attention heads increases; real-world tests on the Ames Housing dataset show the frozen module matches performance of task-specific baselines.

In conclusion, this paper show a minimalist transformer architecture serve as a general-purpose algorithm emulator in context through prompt design.

**Strengths:**

- Foundational Theoretical Value: The formalization of two emulation modes and universal approximation results establish a rigorous basis for viewing ICL as "in-context algorithm emulation," addressing open questions about fixed-weight Transformer flexibility.

- Interpretability: Unlike black-box ICL studies, the paper provides a clear mechanism (prompt-encoded parameters + softmax routing) for how frozen models execute algorithms—enabling future work on principled prompt engineering.

- Broad Algorithmic Coverage: The task-specific template unifies diverse algorithms (GD, linear/ridge regression), showing that a single attention layer can capture core ML workflows.

**Weaknesses:**

- Prompt Scalability: The prompt length grows linearly with the weight dimension of the target algorithm (Section 6, Limitations). This limits practicality for high-dimensional algorithms (e.g., deep neural network training), as prompts could become prohibitively long.

- Lack of Comparison to Prompt-Tuning: The paper focuses on hand-crafted prompts but does not compare to learned prompt-tuning methods (e.g., Lester et al., 2021). It is unclear how hand-crafted prompts perform relative to learned prompts for complex algorithms.

- Limited Algorithm Diversity: The empirical validation focuses on regression and GD—algorithms with clear linear/gradient-based structures. It is unknown if the framework extends to non-linear algorithms (e.g., decision trees, k-means) or sequential tasks (e.g., sorting).

- No Language/Vision Extension: The paper tests only tabular/structured data. It is unclear if the prompt-programmable framework applies to language (e.g., few-shot text classification) or vision (e.g., in-context image segmentation)—domains where ICL is widely used.

**Questions:**

- Prompt Compression: Given that prompt length scales with weight dimension, have you explored methods to compress prompt-encoded parameters (e.g., low-rank approximation, quantization)? Would compression preserve the sharp dot-product gaps needed for accurate emulation?

- Non-Linear Algorithms: Your framework excels at linear/gradient-based algorithms, but how would it handle non-linear algorithms (e.g., logistic regression with sigmoid loss, or k-means clustering)? Do you need to modify the prompt structure to capture non-linearities, or does the continuous function $f$ in Theorem 3.1 suffice?

- Feed-Forward Layers: You omit feed-forward layers in your constructions. Would adding feed-forward layers improve emulation accuracy for complex algorithms, or does this break the "minimal architecture" claim? Do feed-forward layers introduce interference with prompt-encoded parameters?

---

> ### Author Response · Authors · 2025-11-12
> **Rebuttal 1: Responses to Weaknesses Part 1**
>
> ### We thank the reviewer for the detailed review. In response, we have addressed all concerns and questions in the following replies and revisions in the updated draft. All changes from the originally submitted version are highlighted in blue in the revised PDF.
>
> ---
>
> > **Weaknesses 1** Prompt Scalability: The prompt length grows linearly with the weight dimension of the target algorithm (Section 6, Limitations). This limits practicality for high-dimensional algorithms (e.g., deep neural network training), as prompts could become prohibitively long.
>
> **Response:**
>
> The linear scaling of prompt length is an inherent consequence of our generality. **We respectfully remind the reviewer that such trade-off (complexity vs generality) is well-understood in expressiveness results in ML.** Achieving broad generality often comes at the cost of large resource overhead. Moreover, **such complexity only reflects a theoretical upper-bound (worst-case) scenario rather than a bottleneck for typical use cases.**
>
> Our focus was on conceptual universality rather than immediate practicality. It is a proof-of-concept for what a Transformer could do in principle. So we accepted a longer prompt as the price for covering arbitrary algorithms. Importantly, this is not a new algorithm or system we expect to deploy at scale.
>
> ---
>
> > **Weaknesses 2** Lack of Comparison to Prompt-Tuning: The paper focuses on hand-crafted prompts but does not compare to learned prompt-tuning methods (e.g., Lester et al., 2021). It is unclear how hand-crafted prompts perform relative to learned prompts for complex algorithms.
>
> **Response:**
>
> Thanks for pointing this out, but we believe there might be some potential oversights. **This omission is intentional, as our work has a different scope.**
>
> We are not proposing a new practical prompting technique to compete with existing prompt tuning. Rather, **we provide a theoretical constructive example (toy model) of prompt-programmable in-context algorithm emulation (*one* fixed model for *many* tasks/algos).**
>
> **A direct empirical comparison to learned prompt tuning methods would be *apple-to-orange.*** Those methods focus on optimizing prompts for specific tasks to improve performance, whereas our goal is to explain how a fixed Transformer can implement algorithms when given the right prompt.
>
> Instead of comparing empirical performance, we relate to prompt-tuning at a conceptual level. As discussed in Remark 4.4, **our results complement rather than compete with prompt-tuning: existing prompt-tuning “universality” findings are largely existential (showing that a suitable prompt exists), whereas our construction is explicit and constructive.** We show how such prompts can be built to realize algorithmic behavior.
>
> We apologize for any confusion caused. We will clarify this distinction in the final version to avoid any confusion: our contribution is explanatory (understanding capabilities) rather than a new method needing head-to-head comparison with prompt tuning.
>
> ---
>
> > **Weaknesses 3** Limited Algorithm Diversity: The empirical validation focuses on regression and GD—algorithms with clear linear/gradient-based structures. It is unknown if the framework extends to non-linear algorithms (e.g., decision trees, k-means) or sequential tasks (e.g., sorting).
>
> **Response:**
>
> We selected regression and gradient descent as proof-of-concept cases because they have well-defined algorithmic structures and allow clear positioning within existing literature. However, we stress that **our main result is not limited to these examples.** In fact, Corollary 4.2.1 shows that a fixed two-layer attention module can emulate any algorithm implementable by a single-layer attention network through appropriate prompts.
>
> This means **our prompt-programmable framework is in principle very general. It covers all “attention-implementable” algorithms, not just gradient-based ones.** While we did not implement those explicitly, the generality of our framework suggests that such extensions are possible.
>
> The current paper’s experiments were limited in scope simply due to practicality and clarity of exposition. We agree it would strengthen the work to showcase a wider variety of tasks (in both theory and experiemnts). Yet, given the length and scope of this current draft, we leave them for future work.

---

> ### Author Response · Authors · 2025-11-12
> **Rebuttal 2: Responses to Weaknesses Part 2**
>
> > **Weaknesses 4** No Language/Vision Extension: The paper tests only tabular/structured data. It is unclear if the prompt-programmable framework applies to language (e.g., few-shot text classification) or vision (e.g., in-context image segmentation)—domains where ICL is widely used.
>
> **Response:**
>
> We appreciate the comment. However, **this concern is outside the scope of our paper.** Our contribution is purely formal. **We analyze a frozen attention-only module as a minimal toy model to explain how prompts can program in-context algorithm execution and, by extension, the general-purpose ability of GPT-style Transformer models.**
>
> As stated in the introduction, **our setup is a stylized theoretical problem designed to isolate the core mechanism of in-context computation.** We intentionally validate the framework on **tightly controlled, structured tasks (e.g., synthetic regression) where correctness of in-context algorithm execution can be verified against ground truth.** Extending to language or vision would introduce domain-specific complexities and confounding factors that are orthogonal to our theoretical focus and would defeat the purpose of this study.

---

> ### Author Response · Authors · 2025-11-12
> **Rebuttal 3: Responses to Questions**
>
> > **Q1.** Prompt Compression: Given that prompt length scales with weight dimension, have you explored methods to compress prompt-encoded parameters (e.g., low-rank approximation, quantization)? Would compression preserve the sharp dot-product gaps needed for accurate emulation?
>
> **Response:**
>
> Thanks for the question. In response, we respectfully remind the reviewer that, the considerations of this work are purely formal.
> No new method or model is proposed. Hence, testing those empirical methods/techniques is beyond the scope of this paper.
>
> For the 2nd question, from a theoretical perspective, yes, it is possible the compression techniques preserve the sharp dot-product gaps needed for accurate emulation. Possibly with a bounded additional error. One just needs to apply known results from the theory community to facilitate the proofs [a,b].
>
> [a] Computational Limits of Low-Rank Adaptation (LoRA) Fine-Tuning for Transformer Models (ICLR 2024) https://arxiv.org/abs/2406.03136
>
> [b] Fast Attention Requires Bounded Entries (NeurIPS 2023) https://arxiv.org/abs/2302.13214
>
>
> ---
>
> > **Q2.** Non-Linear Algorithms: Your framework excels at linear/gradient-based algorithms, but how would it handle non-linear algorithms (e.g., logistic regression with sigmoid loss, or k-means clustering)? Do you need to modify the prompt structure to capture non-linearities, or does the continuous function $f$ in Theorem 3.1 suffice?
>
> **Response:**
>
> Thanks for the question. Yes, it's possible to extend or include nonlinear algorithms as emulation targets.
>
> Our main results (thm 4.1, 4.2) hold for all "attention-implementable" algorithms. While we did not explicitly show them in our paper (given the scope and the length of the current draft.), our constructions extend to them in principle. We have some ongoing projects along this direction.
>
> To be precise, here we provide a brief sketch of how to extend to nonlinear cases.
> - There are many prior studies showing nonlinear algorithms can be implemented by attention in-context (e.g., Bayesian/spectral methods, randan walks...etc.)
> - Find them and identify the "task-specific attention layer" used to implement such algorithms in a constructive fashion
> - Apply our techniques outlined in Sec 4 to approximate these "task-specific attention layer" in-context and hence emulate the target nonlinear algorithms.
>
> Once an algorithm can be implemented by a one-layer attention module, our two-layer emulator can approximate it through prompting. The prompt structure does not need major changes. We only need the attention representation of the target nonlinear algorithm (i.e., other "attention-implementable" examples similar to the ones in our Sec 3).
>
> ---
>
> > **Q3.** Feed-Forward Layers: You omit feed-forward layers in your constructions. Would adding feed-forward layers improve emulation accuracy for complex algorithms, or does this break the "minimal architecture" claim? Do feed-forward layers introduce interference with prompt-encoded parameters?
>
> **Response:**
>
> Thank you for the question. The feed-forward layer (FFN) is not needed in our constructions and is redundant for our purpose.
>
> Yes, adding FFNs would break the minimal architecture claim. FFNs are universal approximators at the token level. Relying on their expressiveness would blur the contribution of attention and make it unclear which component performs the computation.
>
> Moreover, a fixed FFN would transform the token embeddings that carry prompt-encoded parameters, disrupting the designed dot-product gaps and attention routing. This would complicate both the construction and analysis (note: "complicate" but still "doable" in principle). One can modify our proof by using the |attn - fnn| < eps bounding techniques outlined in [Sec 3.4, Hu et al., 2025] (https://arxiv.org/abs/2504.15956). It should be a trivial chained reduction.
>
> **We therefore omit FFNs to isolate the role of softmax attention, the core mechanism of our study. Our introduction has emphasized the consideration of this intended stylized setup.** Extending the analysis to attention + FFN settings is an interesting future direction but beyond our current scope.
>
> ---
>
> Thank you for your comments and questions on our work. We have addressed all your concerns in above responses. Should any issues remain, we are more than happy to provide additional clarifications.

---

### Official Review · Reviewer_YySt · 2025-11-03

**Soundness:** 3
**Presentation:** 3
**Contribution:** 2
**Rating:** 2
**Confidence:** 4

**Summary:**

think this paper makes only a marginal contribution over prior work (e.g., [1], [2]). The authors show how a single attention layer can emulate, in context, a broad class of algorithms (with an additional linear layer on the input) — specifically, any function of the form $f(w^Tx-y)x$. As special cases, they derive linear regression, (stochastic) gradient descent, and ridge regression. However, all these results have already been demonstrated in previous works [1], [2], [3], [4] etc.
They further show that attention itself can be emulated in context using two attention layers. One question that arises is why one would want to emulate attention in context when direct access to the attention mechanism is already available. This part mainly illustrates that any attention layer can, in principle, be emulated in context by a fixed-weight transformer by defining appropriate prompts(prompt-specific mode).

[1]: Bai, Yu, et al. "Transformers as statisticians: Provable in-context learning with in-context algorithm selection." Advances in neural information processing systems 36 (2023): 57125-57211.

[2]: Giannou, A., Rajput, S., Sohn, J., Lee, K., Lee, J.D. &amp; Papailiopoulos, D.. (2023). Looped Transformers as Programmable Computers.

[3]: Von Oswald, Johannes, et al. "Transformers learn in-context by gradient descent." International Conference on Machine Learning. PMLR, 2023.

[4]: Giannou, Angeliki, et al. "How Well Can Transformers Emulate In-context Newton's Method?." arXiv preprint arXiv:2403.03183 (2024).

**Strengths:**

1. The distinction between task-specific and prompt-programmable emulation is a useful conceptual framing.

2. The coverage of residual update forms $f(w^Tx-y)x$ is broad and subsumes many standard learning procedures.

3. The extension from emulating a single-layer attention mechanism to emulating full (linear) networks is conceptually compelling.

**Weaknesses:**

1. Most (if not all) of the results presented in this paper have already been proved in previous works, possibly using deeper—but still fixed—architectures. Results on linear regression have appeared in [1], [2], [3], and those on ridge regression in [1], [4] (using softmax attention instead of linear attention introduces only a small approximation error).

In-context algorithm selection was also demonstrated in [1] and [4]—not explicitly, but implicitly through the design of pointer mechanisms that determine which function is executed in context.

Overall, the contribution of this paper is limited.

[1]: Bai, Yu, et al. "Transformers as statisticians: Provable in-context learning with in-context algorithm selection." Advances in neural information processing systems 36 (2023): 57125-57211.

[2]:Von Oswald, Johannes, et al. "Transformers learn in-context by gradient descent." International Conference on Machine Learning. PMLR, 2023.

[3]: Giannou, Angeliki, et al. "How Well Can Transformers Emulate In-context Newton's Method?." arXiv preprint arXiv:2403.03183 (2024).

[4]:  Giannou, A., Rajput, S., Sohn, J., Lee, K., Lee, J.D. &amp; Papailiopoulos, D.. (2023). Looped Transformers as Programmable Computers.

**Questions:**

1. Could the authors clarify the motivation or potential use cases for emulating attention using attention itself? For example, are there theoretical insights or architectural benefits (e.g., modularity, composability) that justify this construction?

2. Can the authors clarify whether their constructions differ in expressive power or generality from prior results (e.g., in [1], [2])? Are there cases where their framework captures algorithms that earlier works could not? (Even if the layers are less -- it is important to have a comparison of constant vs logarithmic for example).

3. How does the proposed notion of “prompt-programmable emulation” differ technically from “in-context algorithm selection” in previous work? Is there a formal distinction or new capability demonstrated here?

4. If the authors believe the contribution lies in unifying or simplifying prior results, could they argue this explicitly and explain how this framework might enable new theoretical or practical insights?

---

> ### Author Response · Authors · 2025-11-12
> **Rebuttal 1: Responses to Weakness Part 1**
>
> ### We thank the reviewer for the detailed review. In response, we have addressed all concerns and questions in the following replies and revisions in the updated draft. Happy to discuss more!
>
> ---
>
> > **Weaknesses:** ***Most (if not all) of the results presented in this paper have already been proved in previous works***... Overall, the contribution of this paper is limited.
>
> **Response:**
>
> Thanks for the opportunity to clarify. We acknowledge that some aspects may have been unclear and apologize for any confusion caused. Below, we provide further clarifications highlighting how our work differs from prior studies.
>
> ### **1. “One fixed‑weight model for many tasks.”**
>
> Thanks for raising this point. Prior works have indeed shown in-context emulation of **individual** algorithms.
> - Works such as [1–3] require **task-specific attention heads** (one weight set per algorithm), and most use linear/ReLU attention rather than softmax.
> - Looped-Transformer “programmable computer’’ constructions [4] use **recurrent, multi-layer** Transformer blocks to run arbitrary programs in a weakly constructive manner. **They are “there exist” type results**. Their “Transformer that executes arbitrary program” is a conceptual (Turing) machine, not a fully specified numerical model.
>
> Our contribution differs in both scope and mechanism. We prove that **one fixed-weight softmax attention module** (1–2 layers, no FFNs, no recurrence) can, **solely via prompt changes**, emulate a **finite library of algorithms**, with **no per-algorithm weights**. Everything is constructive in our theory, including both the fixed model and the prompt program. To our knowledge, such **prompt-programmable algorithm universality for softmax attention** is new.
>
> Formally, Thm 4.1/4.2 show in-context emulation of **any** softmax attention head, and Cor. 4.2.1 gives **finite algorithm-library programmability** within a single frozen architecture. This differs fundamentally from both task-specific constructions and looped execution models.
>
> In our original submissions, we highlighted these distinctions at the end of Sec. 2 and in Remark 4.4. In our revision, we further expands these points and adds a short discussion after Thm 4.1/4.2 to improve clarity.
>
>
> ### **2. Softmax attention (practice‑relevant) vs. linear/ReLU surrogates.**
> Our results are **softmax-native** and do not assume any prior reduction from attention surrogates. As the softmax transformer has been the consensus go-to choice for foundation models, this makes the obtained insights and implications more closely tied to practice. The submitted draft discussed this in Remark 4.4.
>
>
> ###  **3. On the claim “softmax attention = ReLU/linear attention adds only a small approximation error.”**
>
> To our knowledge (as stated in above 1st point), **there is no prior formal result that systematically ports in‑context algorithm‑emulation proofs from linear/ReLU attention to *softmax attention* with explicit, uniform error control.**
>
> Our paper fills this gap by constructively controlling softmax via large dot‑product margins and giving end‑to‑end approximation guarantees for the emulated algorithms and attention heads. (Mechanism and guarantees: Sec. 3–4.)
>
> If the reviewer is aware of a published theorem providing such a reduction in the *in‑context algorithm emulation* setting, we would gladly cite it. Absent that, we respectfully disagree with the “only a small error” assertion.
>
> ### **4. Algorithm selection: implicit pointers vs. explicit prompt‑programmability.**
> [1] indeed demonstrates in‑context algorithm selection, but selection among pre‑installed, task‑specific attention heads is NOT the same as **a single frozen softmax module that changes its algorithm purely via the prompt**. Our Corollary 4.2.1 formalizes this distinction: **the model weights remain fixed, the prompt encodes the algorithm**, and we obtain **uniform approximation guarantees across any finite algorithm library**. This is new relative to [1] and to looped‑transformer programmability in [4].
>
> To be concrete, the paper does NOT merely state “small approximation error”. It gives existence/approximation results “for any $\epsilon>0$” with explicit constructions (Theorem 3.1, Theorem 4.1, 4.2), which is stronger than assuming a surrogate and arguing the gap is small.
>
> We also provide a more explicit comparison in the Q3 response below.
>
> ### **5. Architectural minimality and training‑free nature.**
> Our task‑specific constructions (sec 3) use single‑layer, single‑head softmax with a linear readout. The programmable construction uses two attention layers. All results are training‑free and constructive (explicit prompts/weights), not meta‑learned. This level of **minimality under softmax** and **explicit prompt‑level programmability** is, to our knowledge, not present in [1]–[4].

---

> ### Author Response · Authors · 2025-11-12
> **Rebuttal 2: Responses to Weakness Part 2**
>
> ####  **6. Scope overlap (linear/ridge regression).**
> We agree there is task overlap (linear/ridge/GD appear in prior work). However, **our claim is *not* that these tasks are new, but that the *way* they are realized is new** (i.e., via softmax‑native, fixed‑weight, prompt‑programmable constructions with explicit guarantees.)
>
> This addresses the at least practical 2 gaps:
>
> 1. the gap between surrogate attentions and the softmax used in GPT‑style models.
> 2. the gap between "one fixed-weight model for one task/algo (Sec 3 and [Bai et al, 2023])" and "one fixed-weight model for many tasks/algos (Sec4)"
>
> ---
>
> We hope these points address the reviewer’s concerns and clarify the distinct contributions of our paper. Please let us know if any questions remain. We appreciate the opportunity for further discussion.

---

> ### Author Response · Authors · 2025-11-12
> **Rebuttal 3: Responses to Questions Part 1**
>
> > **Question 1.** Could the authors clarify the motivation or potential use cases for emulating attention using attention itself? For example, are there theoretical insights or architectural benefits (e.g., modularity, composability) that justify this construction?
>
> **Response:**
>
> #### **Motivation.**
>
> 1. Goal: a theory of the general-purpose ability of fixed-weight (pretrained) transformer models
> 2. Motivation 1: successes of many "task-specific ICL" studies in literature. Yet, not so much for general-purpose ability of a fixed model.
> 3. Motivation 2: scientific theory (predictable, controllable and interpretable like physics theories) is hard for pretrained foundation models. We hope to get a clean toy model as the testbed for future computational and statistical theories.
>
> ***Since Section 3 (and prior works such as Bai et al., 2023) have established that many algorithms can themselves be implemented by attention layers, emulating a target attention head immediately enables the emulation of those algorithms*** (as illustrated in Section 4.2 and Corollary 4.2.1). Theorems 4.1-4.2 show that a two-layer frozen softmax-attention module can approximate any target attention head via purely prompting.
>
> This makes “weights-as-data” explicit and separates **algorithm parameters** (in the prompt) from **model parameters** (frozen), which is the core mechanism enabling prompt‑programmability. It yields a minimal, modular interface where swapping the prompt swaps the algorithm without any retraining. Moreover, we now have a clean toy model.
>
> In our latest revision, we added a short discussion paragraph after Thm 4.1 and 4.2 to highlight these.
>
>
> #### **Architectural benefit.**
> Yes. Every algorithm realized by a single‑layer attention in Sec 3 (e.g., one‑step GD, linear, ridge) is an instance of Thms. 4.1-4.2 (emulation of a target attention head). By Cor. 4.2.1, one frozen two‑layer softmax module then emulates any finite subset of these by changing only the prompt. Functionally, these motivate potential use case:
> - A concrete template for prompts that encode parameters: such template can motivate new pretraining or evaluation protocols/objectives to utilize such functionalities ("input as programs" and "transformer/attention as interpreter".).
> - A modular lens: parts of a system can be “hot‑swapped” by prompt rather than edited in weights.
>
>
> #### **Practical Implication.**
> The construction is constructive: it explains how to encode algorithm weights and create large query-key margins so the softmax routes the intended computation. This is useful as a mechanistic template for designing prompts or pretraining curricula that “install” reusable procedures.
>
> ---
>
> > **Question 2.** Can the authors clarify whether their constructions differ in expressive power or generality from prior results (e.g., in [1], [2])?
>
> **Response:**
>
> To clarify, as discussed in above weaknesses, **only our Sec 3 are relevant or comparable to those of [1,2] (and other per-task ICL studies). Sec 3 is “one fixed-weight model for one task/algo”, and Sec 4 is “one fixed-weight model for many tasks/algos.”**
>
> #### **What is strictly new in scope?**
>
> * **Softmax setting.**
>   All main constructions (Theorem 3.1, Cor. 3.1.3–3.1.4) are proved for *softmax* attention, not linear/ReLU proxies. The submission explicitly contrasts this with Bai ’23 and Von Oswald ’23 in Remark 4.4. Thus, the linear and ridge regression..etc. emulations are now obtained *directly* under softmax.
>
> * **Prompt-programmable universality (finite library).**
>   Given Thm 4.1 and Thm 4.2, Cor. 4.2.1 shows a *two-layer* frozen attention module can emulate an *arbitrary finite set* of in-context algorithms (each implementable by a single-layer attention) purely by changing the prompt. Namely, **one fixed-weight model, many tasks**. This goes beyond “one model per algorithm” designs. Please also see the above “Architectural benefit” response.
>
> #### **Depth / constants.**
>
> * **Constant depth for the library.**
>
>     Two attention layers for prompt-programmable emulation, independent of the number of algorithms in the library (finite). Yet, as remarked in the draft, our results require the target algorithms to be (softmax-)attention-implementable.
>
> * **Iterative algorithms.**
>
>     When emulating *L* steps of gradient descent explicitly, the draft uses *(L + 1)* layers (Example 2). No logarithmic-depth claim is made in our submission. This part follows Bai et al 2023.
>
> #### **Beyond earlier tasks?**
>
> * The paper does **not** claim new task families outside those implementable by single-layer attention. **Its novelty is the *softmax-based, prompt-programmable* mechanism.** Sec 4.2 discussed the generality of such prompt-programmability. Sec 4.3 discussed that linear layers in standard networks can be replaced by prompt-programmable attention (turning linear networks into in-context learners). These are different generalizations than earlier task-specific constructions [1-4].

---

> ### Author Response · Authors · 2025-11-12
> **Rebuttal 4: Responses to Questions Part 2**
>
> > **Question 3.** Are there cases where their framework captures algorithms that earlier works could not? (Even if the layers are less -- it is important to have a comparison of constant vs logarithmic for example). ***How does the proposed notion of “prompt-programmable emulation” differ technically from “in-context algorithm selection” in previous work?***
>
> **Response:**
>
> Thanks for bringing this up. We’d like to highlight that, **the concept of “in-context algorithm selection” is an extension (or special case) of the *Task-specific in-context emulation* introduced in [Bai et al. 2023] and discussed in our Sec 3.** As we already discussed the latter in `line 175-179`, `line 288-293` and Remark 4.4 of the submitted draft, we believe the distinctions are fairly contrasted.
>
> We acknowledge that our clarity might not be optimal and apologize for any confusion caused. We have revised the draft accordingly (please see the new discussion below Thm 4.1, 4.2 and our extended Remark 4.4 with more explicit comparison).
>
> Below, we provide additional clarification to further highlight the differences.
>
> We start with a short recap of the “in-context algorithm selection” from [Bai et al. 2023].
>
>
>
> #### **Definition (in-context algorithm selection [Bai et al. 2023]).**
> >> `Def.` A model contains several pre-installed, task-specific modules (for example, attention heads) $\{ M\_{\Theta\_a} \}$ for $a \in \mathcal{A}$. Here $\mathcal{A}$ is some finite set.  Given a prompt $P$ and input $X$, the model internally chooses an index $a^\*(P, X)$ and outputs $M\_{\Theta\_{a^\*}}(P, X)$.   The choice is implicit. No explicit algorithm tag is supplied, and all $\Theta\_a$ are stored in the weights.
>
>
>
>
> #### **What our draft defines (Sec 2, 3, 4).**
> - (1) **Task-specific in-context emulation (Sec 3).**
> A single softmax head with frozen weights computes one specific algorithm $a$ when given its prompt $P_a$.
> This is the softmax version of Bai et al. (2023)’s task-specific constructions.
>
> - (2) **Prompt-programmable in-context emulation (Sec 4).**
> A single frozen two-layer softmax module $T\_\psi$ emulates any target head by reading its weights from the prompt.
> For any finite library $\mathcal{A}\_0$, there exist prompts $\{P\_a\}$ with ${a\in\mathcal{A}\_0}$ such that
> $\|T\_\psi(P\_a, X) - a(X)\|\_\infty \le \epsilon$ for all $a \in \mathcal{A}\_0$.
> Here, the weights are shared and fixed, and the prompt itself carries the algorithms (Cor 4.2.1).
>
> #### **How "in-context algorithm selection" of [Bai et al. 2023] relates to (1).**
> Algorithm selection can be viewed as an extension of (1): the model installs multiple task-specific emulators in its weights and adds a selector that picks one in context.
>
> #### **Technical differences (selection vs. prompt-programmable).**
> - **Where parameters live:** In selection, they are inside the weights as $\{\Theta_a\}$. In prompt-programmable emulation, they are in the prompt.
> - **Model count:** Selection uses multiple per-algorithm blocks. Prompt-programmable emulation uses a single module for a finite library.
> - **Guarantee:** Selection demonstrates implicit routing among installed modules. Prompt-programmable emulation provides a finite-library universality guarantee using one frozen softmax module.
> - **Softmax fidelity:** Our results are softmax-native, while prior selection studies often used linear or ReLU surrogates.
> - **Practical implication:** Selection requires pre-installing each algorithm. Prompt-programmable emulation only requires changing the prompt.

---

> ### Author Response · Authors · 2025-11-12
> **Rebuttal 5: Responses to Questions Part 3**
>
> > **Question 4.** Is there a **formal distinction or new capability** demonstrated here? If the authors believe the contribution lies in **unifying or simplifying prior results**, could they argue this explicitly and explain how this framework might enable **new theoretical or practical insights**?
>
> **Response:**
>
> #### **Formal distinction.**
>
> Following the discussion in the submitted draft (`the end of Sec 2`):
>
> * Prior studies are **task-specific in-context emulations** (one fixed-weight model for one task/algo).
> * Ours are **prompt-programmable in-context emulations** (one fixed-weight model for many tasks/algos).
>
> Formally, Thms. 4.1-4.2 show that any single‑layer attention head can be emulated in‑context by a fixed softmax module. Cor. 4.2.1 then yields a **finite‑library emulation** result: ***one frozen two‑layer softmax network emulates any finite subset of such algorithms by changing only the prompt.***
>
> #### **New capability.**
> 1. **Single‑model, many‑algorithm emulation (softmax‑native):** one fixed two‑layer softmax module, no per‑algorithm heads or retraining.
> 2. **Constant depth for the library:** depth $=2$, independent of library size (finite).
> 3. **Constructive $\epsilon$‑approximation:** explicit prompts (weights‑as‑data) and dot‑product margin control yield guarantees $|T_\psi(P_a,X)-a(X)|_\infty\le \varepsilon$.
> 4. **Separation of concerns:** model weights are fixed. Algorithm parameters live in the prompt. This makes the algorithmic parts isolated and more controllable.
>
> #### **Unifying/simplifying contribution.**
>
> Section 3 gives illustrative softmax constructions for the template $x\mapsto f(\langle w,x\rangle-y)x$ (covering one‑step GD, linear, ridge). Thms. 4.1-4.2 are not limited to this template: **they apply to *any attention‑implementable algorithm* by treating its attention head as the target to emulate.** **Cor. 4.2.1 lifts these per‑algorithm emulators into a single prompt‑programmable library**. These providing a common mechanism and analysis (same architecture, same error budgeting).
>
> In particular, to the best of our knowledge, **this provides the first constructive toy model of fixed-weight transformer exhibiting *general-purpose ability* (i.e., *one* fixed‑weight model for *many* tasks).** Moreover, the construction is explicit, interpretable, and softmax-native. These provide a analytic toy model for studying the theoretical properties of GPT-style model in a scientific fashion (constructive, interpretable, controllable ...etc).
>
> #### **How this enables new insights.**
>
> * **Theory:** a uniform emulator for attention‑implementable algorithms enables library‑level capacity/error analyses and a clean study of prompt‑encoded parameters vs. fixed weights.
> * **Practice:** a minimal softmax mechanism for algorithm swapping by prompt can motivate new evaluation protocols (detecting parameter‑tokens, margin patterns) and training objectives that encourage reusable, prompt‑addressable procedures.
>
> ---
>
> We thank the reviewer again for your time and efforts. We hope these clarifications and revisions address the concerns raised and further improve the quality of our paper. Please let us know if any additional clarification would help strengthen our work. We are eager to engage in further discussions :)

---

> ### Author Response · Authors · 2025-11-13
> **Response to the motivation question raised in "Summary"**
>
> In the `Summary` section of the review, the reviewer asked:
>
> > ***One question that arises is why one would want to emulate attention in context when direct access to the attention mechanism is already available.***
>
> We believe addressing this point helps clarify our motivation and scope, so we hope to take the opportunity to respond here in detail. Please pardon the length of our rebuttal.
>
> ---
>
> **Response.**
>
> This question addresses exactly the distinction we study. Prior works (like [Bai et al. 2023] and many similar task-specific ICL studies) assume direct, per-task access to the attention weights: **each algorithm uses its own attention head or module**. In contrast, large pretrained transformer models operate in a different regime: **weights are frozen, and all adaptation happens through the prompt**.
>
> Our goal is to formalize this regime, as emphasized in our intro and Sec 2. We show that **one fixed softmax attention module can emulate many target attentions in context, with no per-algorithm weights.** This captures the programmability of GPT-style models and provides a clean analytic setting for studying prompt-driven algorithm selection, which is not addressed by prior task-specific constructions.
>
> We hope this clarifies our scope and contributions. We look forward to discussing more.

---

### Author Response · Authors · 2025-11-25
**Follow-Up on Rebuttal Clarifications**

Dear Reviewers,

Thanks for the time and effort you have invested in our paper.

We have addressed all your concerns and questions in our rebuttal with utmost care.

As the discussion period reaches its midpoint, please let us know whether our latest responses resolve your concerns, or if any points still require further clarification. We will be more than happy to engage further discussions :)

Sincerely,

Authors

---

### Author Response · Authors · 2025-11-29
**Global Rebuttal Response**

Dear Reviewers and Program Chairs (and Future Readers),

The unfortunate recent openreview drama has brought a state of uncertainty to all of us. It has made even relatively small things such as rebuttal communications challenging. However, I would like to make one final remark to all of you regarding that. I have, myself, a very certain confidence that I acted with the highest fairness and transparency I could.

We thank all reviewers for your helpful comments and detailed reviews. We have addressed all questions and concerns in our rebuttal exchanges and have revised the paper accordingly. All changes with respect to the original submission are highlighted in blue in the updated draft.

In response to the reviewers’ suggestions, we have revised the paper to improve clarity and readability. These changes include correcting typos, restructuring and making several proofs more explicit and modular, adding additional explanations, remarks, and paragraphs, and updating figure captions to better illustrate our prompt-programmable scheme. **Most importantly, in the new Appendix C.2 we added an experiment with hard-coded weights and prompts that directly instantiates the theoretical emulator. It quantitatively confirms the accuracy of Theorem 4.2 in practice.**

Below, we first summarize the main contributions of the work with clear contrasts to the literature, and then list the concrete revisions made in response to the reviews.

---

> ### Author Response · Authors · 2025-11-29
> **Summary of Contributions**
>
> ## **Summary of Contributions**
>
> > ### `TL;DR` We study prompts as programs (algorithm instructions + data) and a fixed-weight softmax Transformer as their interpreter. This is, to our knowledge, the first **softmax-native** and **fully constructive** theory of "**general-purpose behavior**" of fixed-weight/pretrained Transformers. We show that a minimal frozen Transformer can act as a prompt-programmable interpreter for a general class of algorithms, with explicit weights, explicit prompt constructions, and quantitative error bounds.
>
> - **Two modes of in-context algorithm emulation.**
>
>     We present a conceptual framing of the "in-context algorithm emulation" phenomena in foundation models (`line 176-190` of Sec 2):
>     - **(A) Task-specific emulation (Sec 3):** one fixed-weight model per algorithm.
>     - **(B) Prompt-programmable emulation (Sec 4):** one fixed-weight model for many algorithms, with the algorithm swapped purely by changing the prompt.
>
>     This terminology and separation are new in this literature. They give a clean lens on how in-context learning can be seen as algorithm emulation, and they directly match how frozen large models are actually used.
>
> - **(A): Softmax-native, constructive "task-specific" theory (Sec 3).**
>     - The majority of prior studies on in-context algorithm emulation focuses on the "task-specific" mode (A) and uses ReLU/Linear surrogate attention with per-task weights (e.g., Bai et al. 2023; Von Oswald et al. 2023; Giannou et al. 2024 etc, just to name a few).
>     - In contrast, Sec 3 takes the “linear” algorithms in Bai et al. (2023) as concrete task-specific examples and shows how to extend them to softmax-native Transformer models using new softmax attention approximation techniques.
>     - Our constructions are fully explicit and serve as softmax-native task-specific building blocks, that Sec 4 later reuses as examples in the prompt-programmable, multi-algorithm setting (see `line 304-308` and `line 377-385` for discussions).
>
>
>
> - **(B): Prompt-programmable universality (Sec. 4).**
>
>     - Our main novelty is a minimal two-layer, softmax-based, prompt-programmable emulator (Thm 4.1 and 4.2).
>     - We show that **one frozen softmax module can emulate *any* single-layer attention head and hence *any* finite library of such heads**, with the *algorithms and required data encoded in the prompt*.
>     - This stands in sharp contrast to (A) and prior “(task-specific) in-context algorithm selection” results, which rely on multiple pre-installed, task-specific heads (*algorithms stored in the weights*). Please see Remark 4.4.
>     - Corollary 4.2.1 formalizes this as a **prompt-programmable CPU/interpreter**: one fixed module can emulate any finite library of such algorithms by changing only the prompt.
>     - This formalizes an empirically prevalent phenomenon ("one fixed model for many tasks") using a minimal yet still practically relevant toy model (a softmax Transformer) with explicit constructions.
>     - Moreover, our results complement prior beautiful prompting universality and Turing-completeness-style **existence results** with concrete emulation examples, e.g., Furuya et al. 2025 and Giannou et al. (2023) ...etc
>
> Please also see our discussions in `line 377-385` and Remark 4.4.
>
> ---
>
> Bai, Yu, et al. "Transformers as statisticians: Provable in-context learning with in-context algorithm selection." NeurIPS 2023
>
> Von Oswald, Johannes, et al. "Transformers learn in-context by gradient descent." ICML 2023
>
> Giannou, Angeliki, et al. 2024 "How Well Can Transformers Emulate In-context Newton's Method?." AISTATS 2025
>
> Furuya, Takashi, Maarten V. de Hoop, and Gabriel Peyré. "Transformers are universal in-context learners." ICLR 2025.
>
> Giannou, A., Rajput, S., Sohn, J., Lee, K., Lee, J.D. & Papailiopoulos, D. "Looped Transformers as Programmable Computers" ICML 2023

---

> ### Author Response · Authors · 2025-11-29
> **Reviewer-specific clarifications and revision details**
>
> ## **Separation from Prior Work and Reviewer-Specific Clarifications**
>
> - To Reviewer `YySt`, we clarify that the main novelty lies in prompt-programmable emulation (mode B) and softmax-native constructions, not just another task-specific ICL example. We also explain why softmax is not a trivial perturbation of linear attention and why prior linear/ReLU results do not directly imply our theorems.
>
> - To Reviewer `BBMQ`, we emphasize that Hu et al. (2025) is used only as a lemma for approximating hardmax. All prompt designs, emulator architecture, and algorithm-swapping mechanisms are new.
>
> - To Reviewer `6rGC`, we discuss prompt length, potential compression, extensibility to non-linear algorithms, and why we omit FFN layers to preserve minimality and avoid interference with prompt-encoded parameters.
>
> - To Reviewer `wRby`, we add a new experiment with hard-coded weights and prompts that directly instantiates the theoretical emulator. This confirms the quantitative accuracy of Thm 4.2 in practice.
>
> ---
>
> ## **Revision Details**
>
> ### Major revisions include:
>
> - **Clarified taxonomy and positioning of our contributions.**
>   - `line 63-60` Updated Fig 1 caption to make the distinction between *task-specific* and *prompt-programmable* in-context emulation more prominent, and explicitly tied Section 3 to mode (A) and Section 4 to mode (B).
>   - `line 373-385` Expanded the discussion following Thm 4.1 and 4.2
>   - `line 397-412` Expanded Remark 4.4 to compare our results more explicitly with Bai et al. (2023) for task-specific mode and Giannou et al. (2023) for prompt-programmable mode.
>
>
> - **Proof and notation refinements for Thm 3.1 and Sec 3.**
>   - `line 1710, Def D.3` Introduced a clear definition of interpolation points and precision parameters before Thm 3.1 / Thm D.1
>   - `line 2021-2140` Made the dependence of the approximation error on $B$, $B_f$, $\epsilon_1,\epsilon_2,\epsilon_3$ explicit.
>   - `line 1708-2206, Sec D.1` Reorganized the proof of Thm 3.1 to separate the contributions of task-specific approximation, softmax stability, and interpolation error, addressing Reviewer `wRby`’s concerns about the tightness and readability of the final bounds.
>
> - **Proof and notation refinements for Thm 4.1 and 4.2.**
>   - `Sec A, D5, D6` Rewrote the proof sketches in Sec A and detailed proofs in Sec D.5-D.6 to clarify the three-step structure: (i) encoding target weights into the prompt, (ii) multi-head recovery of $K, Q, V$, and (iii) single-head assembly and joint error analysis.
>   - `Sec C, D` Made dimensions of all linear layers and attention weights explicit, as suggested by Reviewer `BBMQ`, so that bounded-width considerations and complexity comparisons are more transparent.
>
> - **New experiment with hard-coded emulator weights (Sec C.3).**
>   - `Sec C.3` Added a new proof-of-concept experiment that instantiates the frozen two-layer attention emulator prescribed by Thm 4.2 using handcrafted weights and prompts, without any training.
>   - These new numerical results also align well with our theory.
>
> ### Minor revisions include:
>
> - `line 142-155` Updated descriptions of "Linear Layer" to be precise.
> - `lne 321-323` Added vectorization definition (Def 4.1) to make Sec 4 more precise
> - Conducted 3 more rounds of proofreading and fixed typos and notation errors flagged by the reviewers
> - Standardized notation across the main text and appendix, especially between Proof Sketches and Proofs.
>
> ----
>
> To reiterate, we have addressed all questions and concerns raised in the reviews and rebuttal exchanges with as much care, respect, and fairness as we could, and have revised the paper accordingly.
>
> We hope these changes resolve the outstanding issues and improve the clarity and overall quality of the work. We thank all reviewers and chairs again for their time, effort and understanding during this challenging review cycle!

---

### Meta-Review · Area_Chair_wnfV · 2025-12-30

**Summary:**

The authors investigate the approximation power of transformers, with softmax activations and with one or two layers, focusing on the ability to approximate models that look like single index models (i.e., of the form f(w^T x - y) for some function f.)  They show that if one embeds the  weights w into the embedding matrix in addition to the (x_i ,y_i) samples, the transformer can approximate a wide family of functions of the form f(w^T x-y).  Their approach for doing this is, to my knowledge, novel: the construction allows for the transformer to determine a set of interpolation points of the residual w^T x - y and store the values of f at these points, with the softmax operator selecting which interpolation points are relevant.

Three of the four reviewers had fairly favorable views of the paper, and one was strongly critical.  I wasn't convinced by the negative reviewer's claims. From what I can tell, the proof technique is novel (and unique), and quite different from the prior works -- the discretization/interpolation/softmax-to-select-interpolation-point approach is, to my knowledge, novel and distinct, and also central to allowing for the fixed weights to work over distinct functions f.

The main concern of mine was that it feels very unnatural to embed the single index model's weights w inside the embedding matrix.  None of the reviewers raised this concern, but to me it is the most significant one, and is to me the strongest argument against accepting this paper.  But I think the proof technique is quite interesting, and the headline result's interestingness ("Fixed-weight two layer transformers can emulate a large class of distinct learning algorithms with significant prompt optimization") outweighs the drawbacks of this, so I recommend acceptance.

**Reviewer Concerns:**

I believe YySt's concerns were largely addressed by the rebuttal - I don't see much evidence that the proof techniques are related to prior ones, or how these results can be implied by earlier ones.  From what I know, the core technique of interpolation/discretization/softmax-to-select-interpolation-points is novel here.

The other reviewers did not have significant concerns that needed to be addressed.

**Reviewer Scores:**

wRby said they would raise their score to 10

BBMQ might have raised to an 8

6rGC would have kept at a 6, I think.

YySt would have likely kept at 2 or maybe raised to a 4.

---

### Decision · Program_Chairs · 2026-01-26

Accept (Poster)